# Dynamics of immune memory and learning in bacterial communities

**Madeleine Bonsma-Fisher[1], Sidhartha Goyal[1,2]***

[1]Department of Physics, University of Toronto, Toronto, Canada; [2]Institute of Biomaterials & Biomedical Engineering, University of Toronto, Toronto, Canada

**Abstract** From bacteria to humans, adaptive immune systems provide learned memories of past infections. Despite their vast biological differences, adaptive immunity shares features from microbes to vertebrates such as emergent immune diversity, long-term coexistence of hosts and pathogens, and fitness pressures from evolving pathogens and adapting hosts, yet there is no conceptual model that addresses all of these together. To this end, we propose and solve a simple phenomenological model of CRISPR-based adaptive immunity in microbes. We show that in coexisting phage and bacteria populations, immune diversity in both populations is coupled and emerges spontaneously, that bacteria track phage evolution with a context-dependent lag, and that high levels of diversity are paradoxically linked to low overall CRISPR immunity. We define average immunity, an important summary parameter predicted by our model, and use it to perform synthetic time-shift analyses on available experimental data to reveal different modalities of coevolution. Finally, immune cross-reactivity in our model leads to qualitatively different states of evolutionary dynamics, including an influenza-like traveling wave regime that resembles a similar state in models of vertebrate adaptive immunity. Our results show that CRISPR immunity provides a tractable model, both theoretically and experimentally, to understand general features of adaptive immunity.

## Editor's evaluation

In this important work, the authors develop a theory for the coevolutionary dynamics of bacteria and phages, where the major evolutionary pressure comes from CRISPR-Cas adaptive immunity in bacteria. Through extensive stochastic numerical simulations and analytical calculations, the article presents a compelling analysis of the emergent properties of immune interactions, in the regime of a single proto-spacer and a single spacer. Some of the trends highlighted by the model are recovered from experimental data. The main results concern how diversity in both phage and bacteria population is linked and is shaped by immunity, and should be of broad interest in immunology.

**\*For correspondence:**
goyal@physics.utoronto.ca

**Competing interest:** The authors declare that no competing interests exist.

## Introduction

Adaptive immunity equips organisms to survive changing pathogen attacks across their lifetime. Many diverse organisms from bacteria to humans possess adaptive immune systems, and their presence shapes the survival, diversity, and evolution of both hosts and pathogens. How adaptive immunity changes the landscape of host-pathogen coexistence, how immune diversity emerges and evolves, and how the pressures of evolving pathogens and adaptive immunity are coupled to produce unique evolutionary outcomes: all of these factors are of fundamental importance to understanding the role of adaptive immunity in populations.

These questions have naturally been explored in the vertebrate adaptive immune system, which protects humans and other vertebrates from evolving pathogens. In these organisms, a diverse repertoire of T cell and B cell receptors can rapidly recognize and respond to a wide range of threats.

Immune specificity is determined by the unique genetic sequence of each cell's receptor, and individuals may harbour millions to billions of unique sequences distributed across four or more orders of magnitude of abundance (*Desponds et al., 2016*; *Mora and Walczak, 2019*; *de Greef et al., 2020*). Quantitative frameworks to model immune diversity and clone abundance have revealed that simple low-level interactions can give rise to complex outcomes including broad distributions of clone abundance (*Desponds et al., 2016*; *Mora and Walczak, 2019*; *de Greef et al., 2020*; *Mayer et al., 2015*; *Gaimann et al., 2020*; *Dessalles et al., 2022*), long-lived biologically realistic transient states (*Yan et al., 2019*; *Gaimann et al., 2020*), and clonal restructuring following immune challenges (*Childs et al., 2015*; *Puelma Touzel et al., 2020*; *Sachdeva et al., 2020*; *Molari et al., 2020*; *Gaimann et al., 2020*). Phenomenological models of pathogen coevolution with the immune system have accelerated our understanding of how the fitness landscape generated by the immune system constrains pathogen evolution (*Luksza and Lässig, 2014*; *Marchi et al., 2019*; *Yan et al., 2019*; *Schnaack and Nourmohammad, 2021*; *Chardès et al., 2022*), how the adaptive immune system responds to rapid pathogen evolution (*Wang et al., 2015*; *Nourmohammad et al., 2019*; *Schnaack and Nourmohammad, 2021*; *Chardès et al., 2022*), and what drives pathogen extinction (*Yan et al., 2019*; *Marchi et al., 2019* or the extinction of particular clonal cell lineages *Nourmohammad et al., 2019*; *Sachdeva et al., 2020*). These models have also explored trade-offs such as between immune receptor specificity and cross-reactivity (*Mayer et al., 2015*; *Nourmohammad et al., 2016*), between the specificity of host-pathogen discrimination and sensitivity to pathogens (*Childs et al., 2015*; *Downie et al., 2021*; *Metcalf et al., 2017*), between the speed of an immune response and the efficiency of that response (*Schnaack and Nourmohammad, 2021*), or between metabolic resource use and immune coverage (*Chardès et al., 2022*). All of these models have shown rich dynamics and qualitatively different states of diversity and evolution arising from simple rules. However, experiments in vertebrates are difficult: vertebrate immunity depends on a complex interplay of many cell types and experiments are time-consuming because of long generation times (*Altan-Bonnet et al., 2020*).

Adaptive immunity in microbes is realized through the CRISPR system, conceptually related to the vertebrate adaptive immune system. The CRISPR system is functionally simple, yet it is incredibly powerful, as indicated by its widespread presence in many diverse bacteria and archaea *Koonin and Makarova, 2019* and its experimentally demonstrated ability to provide strong immunity against phages (*Paez-Espino et al., 2013*; *Paez-Espino et al., 2015*; *van Houte et al., 2016*; Bondy-Denomy et al., 2013). Attacking phages expose their DNA to bacteria, and bacteria with a CRISPR immune system acquire small segments of phage DNA, called spacers. They store spacers in their genome and use them to recognize and destroy matching phage sequences in future infections: spacers are transcribed into RNA and guide DNA-cleaving CRISPR-associated proteins to recognize and cut re-infecting phages. Spacers provide a highly specific immune memory of infecting phages, preventing recognized phages from reproducing. In turn, phages can acquire mutations in the protospacer regions of their genome that are targeted by spacers. These features of the CRISPR immune system mean that (a) phage genetic evolution occurs by selection for escape mutants, and (b) the network of CRISPR immune interactions between bacteria and phages can be inferred by sequencing the genomes of co-living bacteria and phages. Spacer acquisition and phage mutation are rare random events, and many such events must be observed in order to understand their impact on populations. Bacteria and phages have short life cycles and can reach large population size, making it possible to build a statistical picture of the impacts of adaptive immunity.

The kinetics and interactions of phages and bacteria with CRISPR systems have been the subject of numerous experiments (*van Houte et al., 2016*; *Common et al., 2019*; *Common et al., 2020*; *Chabas et al., 2021*; *Dimitriu et al., 2022*; *Guillemet et al., 2021*). Some themes have emerged from experimental studies of CRISPR immunity: (a) high spacer diversity relative to phage diversity increases the likelihood of phage extinction (*van Houte et al., 2016*; *Common et al., 2020*; *Guillemet et al., 2021*), (b) bacteria become more immune to phages over time (*Laanto et al., 2017*; *Morley et al., 2017*; *Common et al., 2019*; *Pyenson and Marraffini, 2020*), and (c) phages readily gain mutations (*Weinberger et al., 2012a*; *Paez-Espino et al., 2013*; *Levin et al., 2013*; *Pyenson et al., 2017*; *Watson et al., 2019*; *Pyenson and Marraffini, 2020*; *Guillemet et al., 2021*; *Guerrero et al., 2021a*) and sometimes genome rearrangements (*Paez-Espino et al., 2015*) to escape CRISPR targeting. Explorations of CRISPR immunity in natural environments have also documented ongoing spacer acquisition and phage escape mutations (*Weinberger et al., 2012a*; *Guerrero et al., 2021a*).

Likewise, previous theoretical work has addressed the impact of parameters such as spacer acquisition rate and phage mutation rate on spacer diversity (*Childs et al., 2012*; *Han et al., 2013*; *Han and Deem, 2017*) and population survival and extinction (*Weinberger et al., 2012b*), how costs of CRISPR immunity impact bacteria-phage coexistence (*Skanata and Kussell, 2021*) and the maintenance of CRISPR immunity (*Levin, 2010*; *Weinberger et al., 2012b*; *Westra et al., 2015*; *Gurney et al., 2019*), how spacer diversity impacts population outcomes (*He and Deem, 2010*; *Weinberger et al., 2012a*; *Childs et al., 2012*; *Haerter and Sneppen, 2012*; *Han et al., 2013*; *Childs et al., 2014*; *Bradde et al., 2017*; *Han and Deem, 2017*), and how stochasticity and initial conditions impact population survival (*Bradde et al., 2019*; *Chabas et al., 2018*). Notably, foundational work by *Childs et al., 2014*; *Childs et al., 2012* and *Weinberger et al., 2012a*; *Weinberger et al., 2012b* found through simulations that spacer diversity readily emerges in a population of CRISPR-competent bacteria interacting with mutating phages.

However, the majority of both experiments and theory are based on observations and models of transient phenomena and short-term dynamics, while it is at long timescales that natural microbial communities experience bacteria-phage coexistence. Some notable experiments have measured long-term coexistence (*Paez-Espino et al., 2015*; *Wei et al., 2011*), and long-term sequential sequencing data from natural populations is becoming more available (*Gómez and Buckling, 2011*; *Burstein et al., 2016*), but appropriate theories to understand steady-state coexistence, sequence evolution and turnover, and immune memory in microbial populations remain rare. Because the processes of growth, death, and immune interaction are inherently random, understanding population establishment and extinction requires a fully stochastic analysis, and theoretical models that explore long-term coexistence have been partially deterministic to date (*Weinberger et al., 2012a*; *Weinberger et al., 2012b*; *Childs et al., 2012*; *Levin et al., 2013*; *Childs et al., 2014*; *Santos et al., 2014*; *Weissman et al., 2018*; *Gurney et al., 2019*). These models do not accurately capture rare stochastic events, in particular mutation, establishment, and extinction. Notable fully stochastic simulations of CRISPR immunity, on the other hand, have lacked rigorous analytic results (*Han et al., 2013*; *Han and Deem, 2017*).

To understand the emergent properties of immune memory and diversity in microbial populations and how phages and bacteria coexist long-term, we developed a simple theoretical model of bacteria and phages interacting with adaptive immunity. We model a population of bacteria with CRISPR immune systems interacting with phages that can mutate to escape CRISPR targeting, building on our previous work that assumed a clonal population of phages with multiple protospacers in each phage (*Bonsma-Fisher et al., 2018*). We model phages with single protospacers in this work to efficiently track mutations in large populations over long timescales. We stochastically simulate thousands of bacteria-phage populations across a range of population sizes, spacer acquisition rates, spacer effectiveness rates, and phage mutation rates, and derive analytic expressions for the probability of establishment for new phage mutants, the time to extinction for phage and bacterial clones, and the dependence of bacterial spacer diversity on spacer acquisition rate, effectiveness, and phage mutation rate. Our simulations are fully stochastic and run for many thousands of generations to accurately capture the dynamics of establishment and extinction, yet the underlying model is simple enough to solve analytically. We show that even with the simplest assumptions of uniform spacer acquisition and effectiveness, complex dynamics and a wide range of outcomes of diversity and population structure are possible. We recover and reinterpret experimentally observed feaures: (a) we find that high diversity is not beneficial for bacteria when phage and bacterial diversity is strongly coupled, (b) we show that bacterial immunity can either track new phage mutations rapidly or keep a memory for a long time, but not both, and (c) we find emergent diversity resulting from selection for phage mutations that evade CRISPR targeting, linking diversity to the dynamical quantities of establishment and extinction. We compute bacterial average immunity in our simulations and in available experimental data and show that our model predicts qualitative trends that are visible in data. Finally, we show that adding immune cross-reactivity leads to qualitatively different states of evolutionary dynamics: (a) a traveling wave regime that resembles a similar state in models of vertebrate adaptive immunity (*Yan et al., 2019*; *Marchi et al., 2019*; *Marchi et al., 2021*) emerges when high cross-reactivity creates a fitness gradient for phage evolution, and (2) a regime of low turnover protected from new establishment by the reduced fitness of new phage mutants.

## Results

### Bacteria and phages dynamically coexist and coevolve

We model bacteria and phage interacting and coevolving in a well-mixed system (*Figure 1A* and 'Model'). Bacteria divide by consuming nutrients and phages reproduce by creating a burst of $B$ new phages after successfully infecting a bacterium. Bacteria can contain a single CRISPR spacer that confers immunity against phages with a matching protospacer. Phages are labelled with a single protospacer type, a binary sequence of length $L = 30$ that can mutate to a new type during a burst with probability $\mu L$, where $\mu$ is the per-base mutation rate per generation. All simulations begin with a single clonal phage population unless otherwise specified.

Coexistence occurs across a wide range of parameters but is not guaranteed: below a certain success probability $p_V^0 = \frac{1}{B}\left(\frac{gf}{(1-f)\alpha} + 1\right)$, phages are not able to reproduce often enough to overcome their base death rate due to outflow and adsorption and are driven extinct (grey area of *Figure 1D*; *Bonsma-Fisher et al., 2018*). In this expression, $g$ is the bacterial growth rate, $B$ is the phage burst size, $\alpha$ is the phage adsorption rate, $f = F/(gC_0)$ is a normalized outflow rate, and $C_0$ is the inflow nutrient concentration. This is the same extinction threshold reported by *Payne et al., 2018* as the cutoff for achieving herd immunity in a well-mixed bacterial population. To a first approximation, phages must successfully infect every $1/B$ bacteria they encounter, but if bacteria are growing quickly, then phages must do better to overcome bacterial growth, leading to the extra terms in this expression (see 'Phage extinction threshold'). We write this extinction threshold as $A = \frac{(Bp_V-1)(1-f)\alpha}{fg}$. Above the phage extinction threshold ($A > 1$), the phage population size increases with increasing $p_V$ but eventually decreases again as bacterial numbers are driven too low to support a large phage population (*Bonsma-Fisher et al., 2018*). A similar non-monotonicity as a function of the probability of naive bacterial resistance ($1 - p_V$) was described in theoretical work by *Weinberger et al., 2012b*. In our model the position of the peak in phage population size as a function of infection success probability is determined by $e$, the effectiveness of CRISPR spacers against phage; increasing $e$ pushes the peak to higher $p_V$ (*Figure 1D*). While $e$ is a constant parameter that determines the outcome of pairwise interactions between bacteria and phages, the bacterial population as a whole possesses an average immunity to phages that is a weighted average of all the possible pairwise interactions (*Figure 1A* inset). It is the overall average immunity that determines population outcomes, which we describe in detail in 'Pathogen and host diversity must be considered together'.

To focus on regimes where bacteria and phages coexist, we select parameters within the deterministic coexistence regime to explore bacteria-phage coevolution. Even in this regime, stochastic extinction will eventually come for one or both populations in simulations (*Figure 1E*), though the timescale of extinction may be extremely long for large population sizes (*Badali and Zilman, 2020*). Phages are more susceptible to stochastic extinction than bacteria because of their large burst size $B$ which increases their overall population fluctuations (Appendix 3). The length of coexistence before stochastic extinction depends on population size as well as simulation parameters and initial conditions (see 'Stochastic population extinction'): phage populations can be rescued from extinction by high mutation rate (*Figure 1—figure supplement 3*) or high initial protospacer diversity (*Figure 1—figure supplement 6*), but are more likely to go extinct if spacer effectiveness is high (*Figure 1—figure supplement 1*). Conversely, bacteria are more likely to go extinct if spacer effectiveness or spacer acquisition rate are low (*Figure 1—figure supplement 4*). Population survival and persistence in natural populations is impacted by additional factors we do not address in our model, including immigration (*Volkov et al., 2003*; *Chabas et al., 2016*, niche partitioning *Simek et al., 2010*; *Weitz et al., 2013*; *Mills et al., 2013*; *Badali and Zilman, 2020*; *Voigt et al., 2021*), environmental fluctuations (*Abreu et al., 2020*; *Voigt et al., 2021*), and spatial structure (*Haerter et al., 2011*; *Haerter and Sneppen, 2012*; *Heilmann et al., 2010*; *Heilmann et al., 2012*; *Simmons et al., 2018*; *Skanata and Kussell, 2021*).

Across a wide range of coexistence parameters, our simulations show continual phage evolution and bacterial CRISPR adaptation in response (*Figure 1B*). New phage protospacer clones arise from a single founding clone by mutation, and a small fraction of new mutants grow to a large size and become established. Once phage clones become large, bacteria acquire matching spacers and an immune bacterial subpopulation becomes established. The specific protospacer and spacer types present in the population continually change as old types go extinct and new types are created by

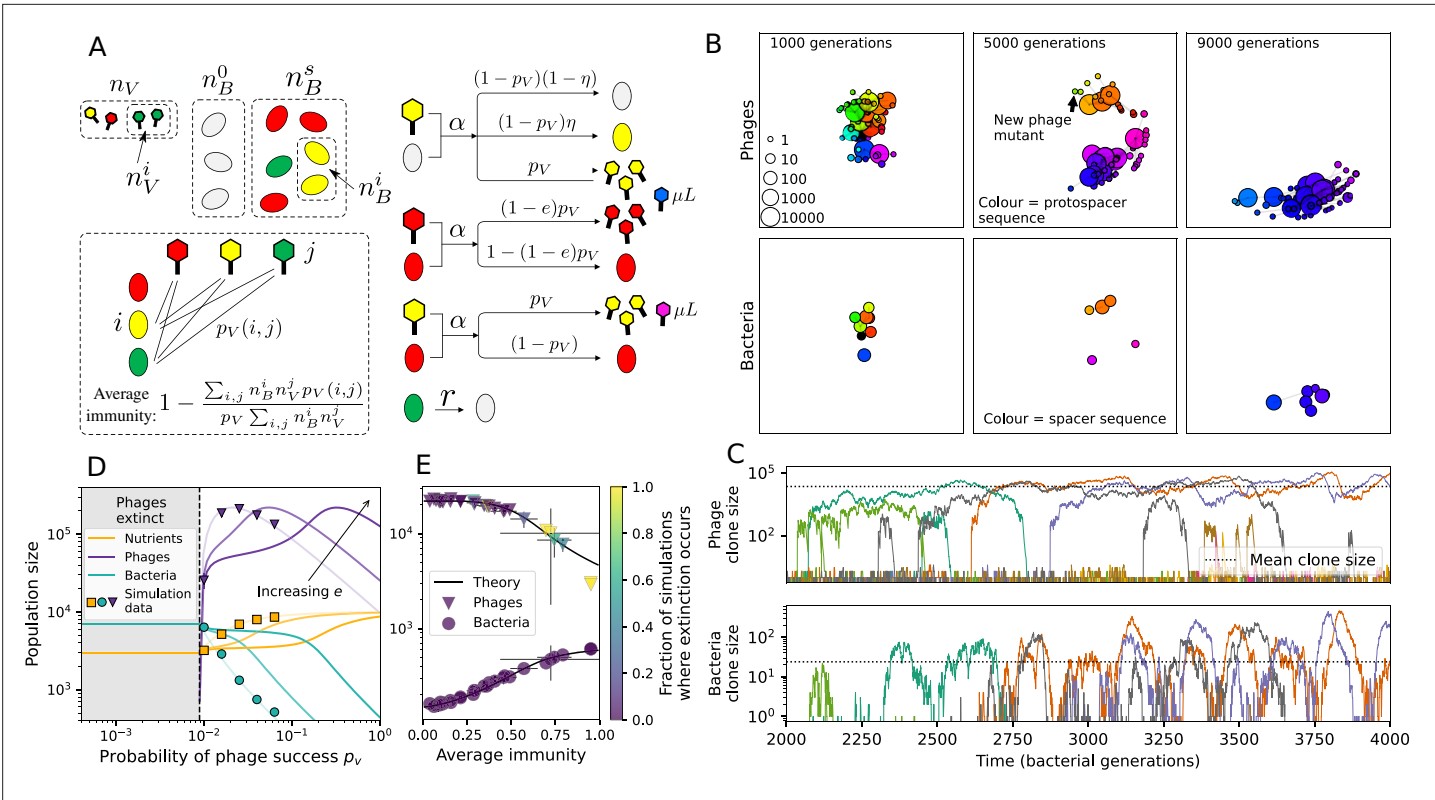

**Figure 1.** Model description.

(**A**) We model bacteria and phages interacting in a well-mixed vessel. We track nutrient concentration, phage population size ($n_V$), and bacteria population size ($n_B$). Bacteria can either have no spacer ($n_B^0$) or a spacer of type $i$ ($n_B^i$, $\sum_i n_B^i = n_B^s$), and phages can have a single protospacer of type $j$ ($n_V^j$). With rate $\alpha$, a phage interacts with a bacterium. If the bacterium does not have a matching spacer, the phage kills with probability $p_V$ and produces a burst of $B$ new phages, while for bacteria with a matching spacer that probability is reduced to $p_V^s = p_V(1 - e)$, $0 \leq e \leq 1$. Bacteria without spacers that survive an attack have a chance to acquire a spacer with probability $\eta$, and bacteria with spacers lose them at rate $r$. Lower inset: average immunity is the weighted average pairwise immunity between spacer-containing bacteria and phages, given by $1 - \frac{\sum_{i,j} n_B^i n_V^j p_V(i,j)}{p_V \sum_{i,j} n_B^i n_V^j}$. The probability of a phage with protospacer $j$ successfully infecting a bacterium with spacer $i$ is $p_V(i,j)$. (**B**) Three time points in a typical simulation with $C_0 = 10^4$, $e = 0.95$, $\eta = 10^{-4}$, and $\mu = 10^{-5}$. Coloured circles represent unique protospacer or spacer sequences; shared sequences are shown with the same colour. The size of each circle is proportional to clone size, and new mutants are shown radially more distant from the centre. (**C**) Ten individual clone trajectories vs simulation time for phages (top) and bacteria (bottom). The mean clone size is shown with a horizontal dashed line. (**D**) Total phage, bacteria, and nutrient concentration as a function of phage success probability $p_V$. Markers show an average over five independent simulations for different values of $p_V$ with $C_0 = 10^4$, $\eta = 10^{-3}$, $e = 0.95$, and $\mu = 10^{-7}$. Solid lines show theoretical predictions for different constant values of effective $e$. As $p_V$ decreases, phages go extinct at a critical value given by $A = 1$, where $A = \frac{(Bp_V-1)(1-f)\alpha}{fg}$. (**E**) Total phage and bacteria population size as a function of average bacterial immunity to phages. Colours indicate the fraction of simulations in which phage or bacteria go extinct before a set endpoint. Solid lines show the mean-field prediction. Error bars are the standard deviation across three or more independent simulations.

The online version of this article includes the following video and figure supplement(s) for figure 1:

**Figure supplement 1.** Probability of stochastic extinction at low spacer acquisition.

**Figure supplement 2.** Probability of stochastic extinction at high spacer acquisition.

**Figure supplement 3.** Time to extinction for phages vs. mutation rate.

**Figure supplement 4.** Time to extinction for bacteria vs. mutation rate.

**Figure supplement 5.** Time to extinction for phages for different initial diversity and low spacer acquisition.

**Figure supplement 6.** Time to extinction for phages for different initial diversity and high spacer acquisition.

**Figure 1—video 1.** An animation of a typical simulation of bacteria and phages interacting with CRISPR immunity.
https://elifesciences.org/articles/81692/figures#fig1video1

phage mutation, but the average total diversity and average overlap between bacteria and phage remains constant at steady state (Figure 8). Both bacteria and phage clones stochastically go extinct, completing the life cycle of a clonal population (*Figure 1C*).

## Phages drive stable emergent sequence diversity

New phage protospacer clones continually arise and go extinct in our simulations, generating turnover in clone identity in the population. Despite constant turnover, however, the total number of clones remains fixed at steady state. We use the mean number of bacterial clones at steady state, designated $m$, as a measure of system diversity. This choice of diversity measurement is equivalent to the Hartley entropy of the clone size distribution, a special case of the Rényi entropy (*Mora and Walczak, 2016*; *Altan-Bonnet et al., 2020*). This definition weights all clones equally regardless of their abundance; such a measurement is not appropriate when clone size distributions are very broad and small clones may be unsampled, but is reasonable when clone size distributions are relatively narrow and all clones are sampled (*Mora and Walczak, 2016*). In our simulation results, both bacteria and phage populations exhibit relatively narrow clone size distributions across a range of parameters, with the exception of low values of spacer acquisition $\eta$ (*Figure 2A*, *Figure 2—figure supplement 3*, *Figure 2—figure supplement 4*). Even at low $\eta$, however, clone size distributions are approximately exponential, indicating that they are not scale-invariant and that the mean clone size still captures important information about the full clone size distribution.

What determines clonal diversity? Many factors that correlate with transient diversity have been experimentally identified, such as phage extinction and slower phage evolution at high bacterial spacer diversity (*van Houte et al., 2016*; *Common et al., 2020*) and maintenance of a diverse bacterial population when exposed to diverse phages (*Paez-Espino et al., 2015*; *Common et al., 2019*; *Guillemet et al., 2021*; *Lopatina et al., 2019*), but a conceptual framework to understand emergent diversity has remained elusive. For instance, while initial high spacer diversity puts low-diversity phage populations under intense pressure to the point of driving them extinct (*van Houte et al., 2016*; *Common et al., 2020*), is the same true for emergent bacterial diversity after an extended period of coexistence? Is observed high bacterial spacer diversity indicative of successful bacterial escape from phage predation or an indicator of increased phage pressure? In our model, phage and bacterial diversity is tightly coupled: the number of large phage clones is approximately the same as the number of bacterial clones (Figure 22). This is also the case in experimental coevolution data from *Paez-Espino et al., 2015*: the number of phage protospacer types is on the same order of magnitude as the number of bacterial spacer types across most similarity thresholds (Figure 72). There is evidence that this coupling of diversity may also occur in the wild: a recent longitudinal study of *Gordonia* bacteria interacting with phage in a wastewater treatment plant identified 14 high-coverage phage genotypes and 11 high-coverage bacterial variants based on CRISPR spacer sequence (*Guerrero et al., 2021a*).

Using the tight correspondence between bacterial diversity and phage diversity in our model, we calculate the overall steady-state diversity by balancing the effective phage clone mutation rate $\bar{\mu} = \frac{1}{gC_0}\alpha B p_V (1 - e^{-\mu L})(1 - \frac{e\nu}{m})n_V n_B$, phage clone establishment probability $P_{est}$, and the time to extinction for large phage clones $T_{ext}$ (details in 'Measuring diversity'):

$$m = P_{est}\bar{\mu}T_{ext} \tag{1}$$

*Equation 1* arises from the simple statement that the number of large clones must be equal to their establishment rate ($P_{est}\bar{\mu}$) multiplied by their average time to extinction ($T_{ext}$). This relationship successfully predicts the number of bacterial clones at steady state across a wide range of parameters and a wide range of diversity values (*Figure 2B* inset, *Figure 2—figure supplement 1*, and *Figure 2—figure supplement 2*). At the lowest value of $\eta$ the prediction tends to overestimate the number of clones – in this regime, low acquisition means that phage clones go extinct because of clonal interference before bacteria are able to acquire spacers (Figure 16, 'Measuring diversity').

Through approximations ('Analytic approximations for diversity'), we find that diversity depends on a single combined parameter to the power 1/3 (*Figure 2B*, *Equation 2*), and this parameter is proportional to spacer effectiveness $e$, the probability of bacterial survival followed by spacer acquisition $(1 - p_V)\eta$, and the phage mutation rate $\mu L$.

$$m \approx \left( \frac{4e\eta(1 - p_V)\mu L(gC_0(1 - f))^3}{B^2\alpha^2 p_V^3 r} \right)^{\frac{1}{3}} \tag{2}$$

Each of these parameters intuitively increases diversity (e.g., a higher phage mutation rate means that phage diversity increases and bacterial diversity follows suit). What is surprising is that their combined

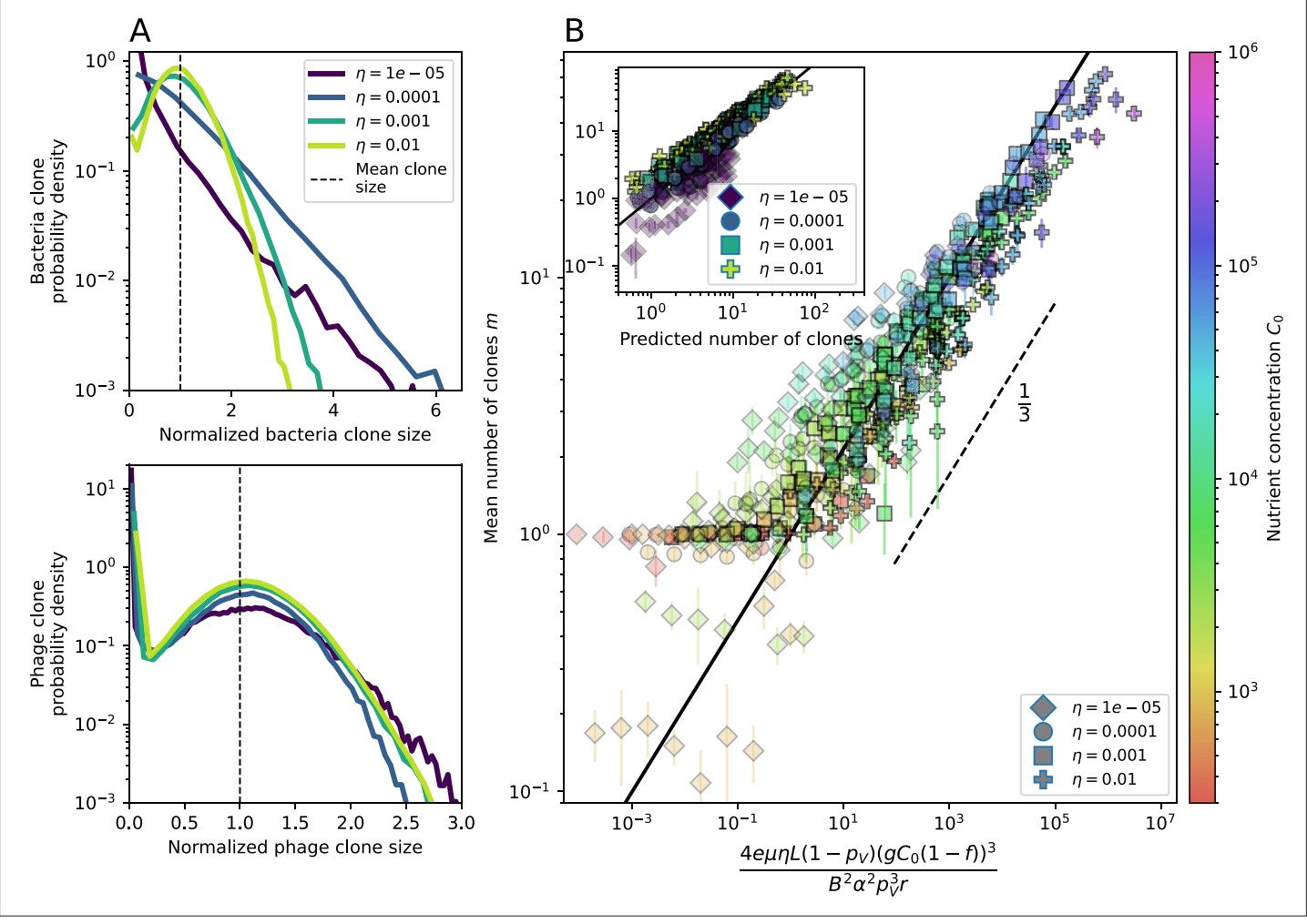

**Figure 2.** Diversity depends sub-linearly on parameters. (**A**) Bacteria and phage clone size distributions normalized to the measured mean clone size for $C_0 = 10^5$, $\mu = 3 \times 10^{-7}$, and $e = 0.95$. As $\eta$ increases, both clone size distributions become more sharply peaked. (**B**) The mean number of bacterial clones depends only on a combined parameter in the limit of small average immunity (generally coinciding with high $C_0$). (Inset) The mean number of bacterial clones can be predicted by numerically solving **Equation 1** for $m$. The two lowest values of $\eta$ are shown with lighter shading. Error bars are the standard deviation across three or more independent simulations.

The online version of this article includes the following figure supplement(s) for figure 2:

**Figure supplement 1.** Predicted diversity grouped by spacer effectiveness.

**Figure supplement 2.** Predicted diversity grouped by mutation rate.

**Figure supplement 3.** Clone size histograms by spacer effectiveness.

**Figure supplement 4.** Clone size histograms by phage mutation rate.

effect on diversity is to a power much less than 1: this 1/3 exponent means that if the mutation rate increased tenfold, the diversity would only increase by about a factor of two (**Appendix 2—figure 1**). In contrast, a simple neutral model of cell division with mutations gives a linearly proportional increase in diversity for the same increase in mutation rate (**Appendix 2—figure 2**).

To understand where the dependence of diversity on these key parameters come from, we look more closely at each component expression. The effective phage mutation rate $\bar{\mu}$ depends linearly on the parameter $\mu : \bar{\mu} \approx \frac{gC_0(1-f)f\mu L}{\alpha P_V}$, while both the probability of establishment and the time to extinction depend inversely on diversity $m : p_{est} \approx \frac{1}{m}\frac{ce\eta(1-P_V)gC_0(1-f)}{B_{P_V}r}$ (**Equation 5**) and $T_{ext} \approx \frac{1}{m}\frac{2gC_0(1-f)}{f\alpha Bp_V}$ (**Equation 173**, Appendix 3). By comparison with **Equation 1**, we find that $m^3$ depends approximately linearly on mutation rate, resulting in the weak $m \propto \mu^{\frac{1}{3}}$ dependence on mutation rate.

The dependence of diversity on both $e$ and $\eta$ comes from the probability of phage establishment since $\bar{\mu}$ depends only very weakly on these parameters through its dependence on total population sizes and $T_{ext}$ depends explicitly on $m$ alone, not $e$ or $\eta$. The phage probability of establishment is proportional to $\frac{e\eta(1-p_V)}{m}$ (*Equation 5*), and as before, this gives $m^3 \propto e\eta(1-p_V)$. Bacteria are more successful at high $\eta(1-p_V)e$, which increases the phage establishment probability. Previous theoretical work has predicted that diversity increases as spacer acquisition rate increases (*Childs et al., 2012*); here, we provide a quantitative prediction for this dependence. In the following sections, we explore phage establishment in more detail.

## What determines the fitness and establishment of new mutants?

We find that diversity emerges in our model from the balance of phage clone establishment and extinction. However, only some phage mutants escape initial stochastic extinction and survive long enough to become established. What determines the fate of a new phage mutant? In our model, a single phage mutation in a protospacer can completely overcome CRISPR targeting, which means that new phage mutants can infect all bacteria equally well and their initial growth rate $s_0$ is independent of CRISPR: $s_0 \approx \alpha n_B(Bp_V - 1) - F$, where $F$ is the chemostat flow rate (a shared death rate for phages and bacteria). Surprisingly, however, even once bacteria start to acquire matching spacers, the probability of establishment for new phage mutants is still well-described by theory in which CRISPR targeting only influences total average population sizes (*Figure 3C*); that is, the specific interaction between a phage and its matching clone can be ignored. Intuitively, this is because phage clones must grow to a certain size before bacteria encounter them enough to begin to acquire spacers, and this size turns out to be large enough to avoid stochastic extinction (Figure 15). The probability of phage establishment is $\frac{2s_0}{B(s_0+\delta_0)}$, where $\delta_0 = F + \alpha n_B(1 - p_V)$ is the initial phage mutant death rate. Importantly, these rates are independent of the population size of *matching* CRISPR bacteria clones; the only dependence on bacteria is on the total bacterial population size $n_B$ (*Appendix 3—figure 1*), which is fairly stable at steady state.

Even though CRISPR targeting does not explicitly affect the establishment probability for new phage mutants, the probability of phage establishment increases as the average bacterial immunity increases (*Figure 3C*). Average immunity is a measure of the overall effectiveness of CRISPR immunity for the entire bacterial population; it is the average of all pairwise immunities between phage clones and bacterial clones weighted by their population sizes (*Figure 1A* inset). Because higher average immunity is beneficial for bacteria and leads to larger bacterial population sizes (*Figure 3C*), higher average immunity also means there is stronger selective advantage for new phage mutants that can escape CRISPR targeting (*Figure 3C* and Figure 5A).

For insight into the nature of this dependence, we approximated the probability of establishment at two extremes of average immunity (see 'Dominant balance approximations for $\nu$'). The probability of establishment can also be written

$$P_{est} = \frac{2e\nu}{m(B-1)}$$

(3)

where $\nu$ is the fraction of bacteria that have spacers and $m$ is the average number of bacterial clones at steady state. Low average immunity occurs when $e/m$ is small: in this limit, the fraction of bacteria with spacers is

$$\nu \approx \frac{1}{1 + \frac{r}{\eta(1-p_V)\alpha\tilde{n}_V} - \frac{e}{m}\left(\frac{p_V}{\eta(1-p_V)} - \frac{ABp_V}{(A-1)(Bp_V-1)}\right)}$$

(4)

where $A = \frac{(Bp_V-1)(1-f)\alpha}{fg}$ is the extinction threshold for phages ($A > 1$ for phage survival), and $\tilde{n}_V$ is the phage population size calculated with $e = 0$ (*Equation 52*); this is the extreme limit of low average immunity where total population sizes approach their values in the absence of CRISPR immunity (Figure 26). The quantity $\frac{r}{\eta(1-p_V)\alpha\tilde{n}_V}$ can be understood as the ratio of the rate of spacer loss to the rate of spacer acquisition. If spacer loss is high and acquisition is low, the fraction of bacteria with spacers ($\nu$) decreases. This in turn correlates with a lower probability of phage establishment (*Equation 3*) since phages are less likely to establish when there is less pressure from bacteria defending against them with CRISPR. *Equation 4* is plotted in *Figure 3C* in blue. At very high average immunity, on the other hand, $\nu$ becomes close to 1 regardless of $\eta$ (*Figure 3—figure supplement 2*), which is why the curves in *Figure 3C* overlap at high average immunity ($\nu = 1$ is plotted in green).

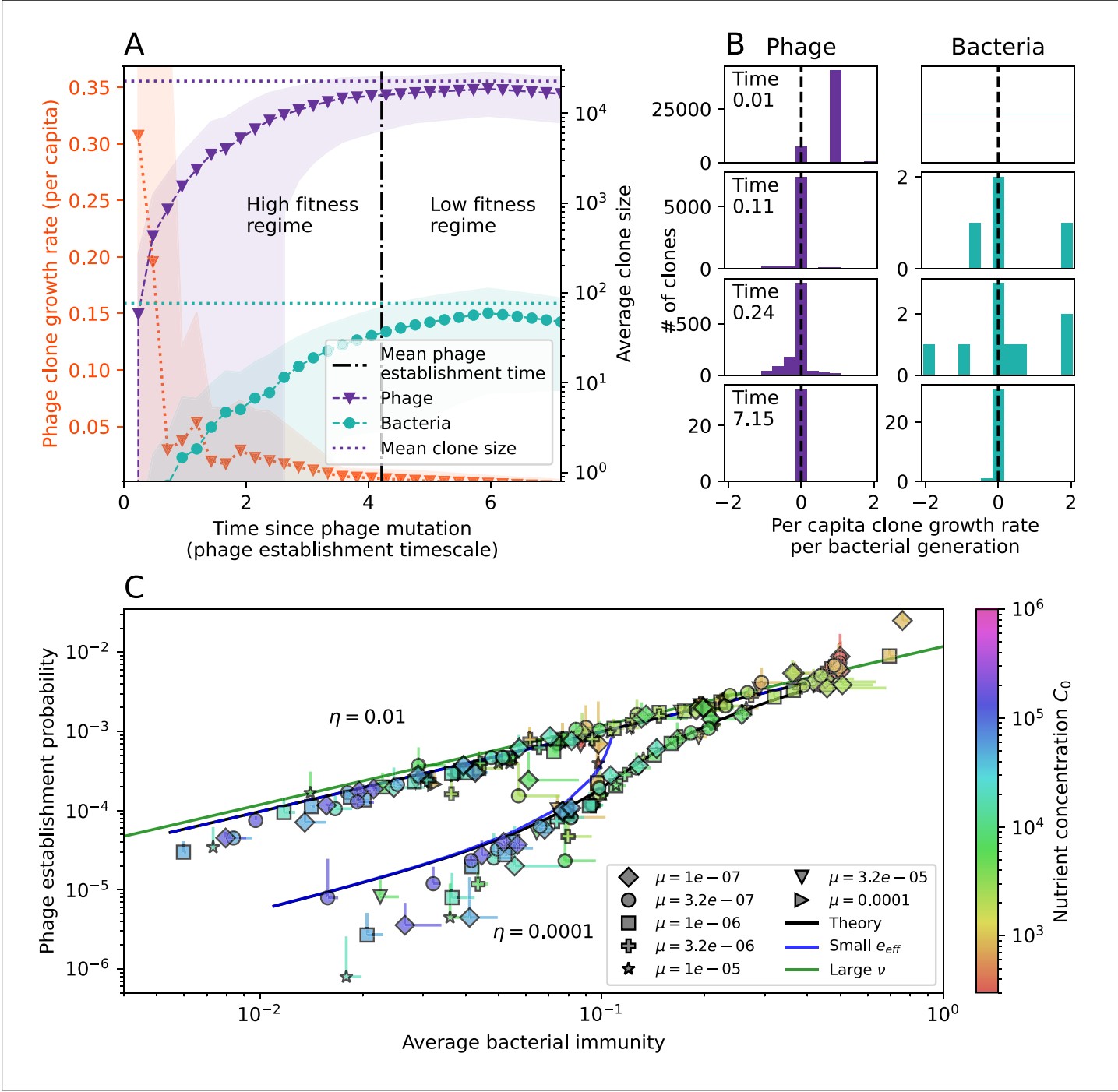

**Figure 3.** The fate of individual clones. (**A**) Phage and bacteria coevolve in two timescale-separated regimes characterized by phage clone fitness. Average phage and bacteria clone size vs. time since phage mutation (right axis), and average clone growth rate vs. time since phage mutation (left axis). Markers show the average over all clone trajectories after steady state from six simulations with the same parameters. (**B**) Histograms of individual clone fitness grouped by time since phage mutation. Phage clones initially have fitness $gt_0$, but rapidly most clones reach neutral growth (fitness $\approx 0$). Bacteria clones also follow suit, initially having fitness $gt_0$ and rapidly reaching 0 fitness on average. Because spacer acquisition for a clone only happens after that clone is created by phage mutation, the top-right panel of (**B**) is empty at the earliest time point following phage mutation. Individual clone trajectories are highly variable. (**C**) Probability of phage clone establishment vs. average immunity. Clones are considered established in simulations when they reach the mean clone size. **Equation 3** with $\nu = 1$ is shown in green and with $\nu$ given by **Equation 4** in blue. In (**A**, **B**), $C_0 = 10^4$, $e = 0.95$, $\eta = 10^{-3}$, and $\mu = 3 \times 10^{-6}$. Error bars are the standard deviation across three or more independent simulations.

The online version of this article includes the following figure supplement(s) for figure 3:

*Figure 3 continued on next page*

*Figure 3 continued*

**Figure supplement 1.** Average clone sizes and fitness over time in a simulation.

**Figure supplement 2.** Approximations for phage establishment probability.

**Figure supplement 3.** Theoretical phage establishment probability with approximations.

For more insight into the parameter dependence of $P_{est}$, we apply the same approximation as for $m$, expanding $P_{est}$ in $1/B$ and $\eta$ ('Approximation for $m$'):

$$P_{est} \approx \frac{1}{m} \frac{2e\eta(1 - p_V)gC_0(1 - f)}{Bp_Vr} \approx \left[ \frac{2(\eta(1 - p_V)e\alpha)^2}{\mu LBr^2} \right]^{\frac{1}{3}} \tag{5}$$

The probability of establishment increases with the probability of bacterial escape and spacer acquisition $\eta(1 - p_V)$ and with spacer effectiveness $e$, but decreases with increasing mutation rate $\mu L$. This is consistent with the intuition that more successful bacteria increase the strength of selection for phage mutants (higher $\eta$ and $e$), but that higher mutation rate reduces the probability of establishment for any particular mutant.

## Cross-reactivity leads to dynamically unique evolutionary states

We next asked how cross-reactivity between spacer and protospacer types impacts population dynamics and outcomes. Adding cross-reactivity was motivated by several experimental observations in CRISPR immunity: (a) In type I and type II CRISPR systems, single mutations in the PAM or protospacer seed regions (approximately 8 nucleotides at the start of a protospacer) can facilitate phage escape, whereas mutations elsewhere in the protospacer are tolerated by the CRISPR system (*Deveau et al., 2008*; *Pyenson and Marraffini, 2020*). Even when a phage manages to escape direct targeting, in type I and II systems an imperfect spacer match can facilitate priming: when Cas machinery binds to a protospacer match, even if unable to cleave the target, the likelihood of acquiring a nearby spacer is increased (*Westra et al., 2015*; *Rao et al., 2017*; *Pyenson and Marraffini, 2020*; *Weissman et al., 2020*). (b) Type III CRISPR targeting in *Staphylococcus epidermidis* has been shown to be completely tolerant to all single and even double mutations, meaning that these systems are naturally cross-reactive (*Pyenson et al., 2017*).

We simulated two types of cross-reactivity: one in which phages experience an exponential decrease in CRISPR effectiveness with mutational distance (*Equation 7*), and a step-function cross-reactivity where phages require additional $1 \leq \theta \leq 3$ mutations to perfectly escape CRISPR targeting (*Equation 8*). Phage success probability without cross-reactivity is given by *Equation 6*, a special case of 7 and 8 with $\theta = 0$.

$$p_V(i,j) = \begin{cases} p_V(1 - e) & \text{if } i = j \\ p_V & \text{if } i \neq j \end{cases} \tag{6}$$

$$p_V(i,j) = p_V \left( 1 - e \exp\left[ -\frac{d_{ij}}{\theta} \right] \right) \tag{7}$$

$$p_V(i,j) = \begin{cases} p_V(1 - e) & \text{if } d_{ij} \leq \theta \\ p_V & \text{if } d_{ij} > \theta \end{cases} \tag{8}$$

The mutational distance $d_{ij} = \sum |(V_i - V_j)|$ is the number of mutations between a protospacer and spacer, and $\theta$ scales the radius of cross-reactivity, with larger $\theta$ meaning more mutations are required to achieve the same immune escape (*Yan et al., 2019*).

Exponential cross-reactivity has been modelled extensively in vertebrate adaptive immunity (Chao et al., 2005; *Mayer et al., 2015*; *Marchi et al., 2019*; *Yan et al., 2019*; *Schnaack and Nourmohammad, 2021*; *Chardès et al., 2022*), and our definition of exponential cross-reactivity as a function of mutational distance is the same as in *Yan et al., 2019*. Step-function cross-reactivity, on the other hand, is reminiscent of the type III CRISPR system in which multiple point mutations are required to escape CRISPR targeting (*Pyenson et al., 2017*) and was also modelled theoretically by *Han et al., 2013*.

We find that cross-reactivity results in strikingly different dynamics of clone establishment and persistence (*Figure 4* and 'Cross-reactivity'), including a travelling wave regime in which genetically

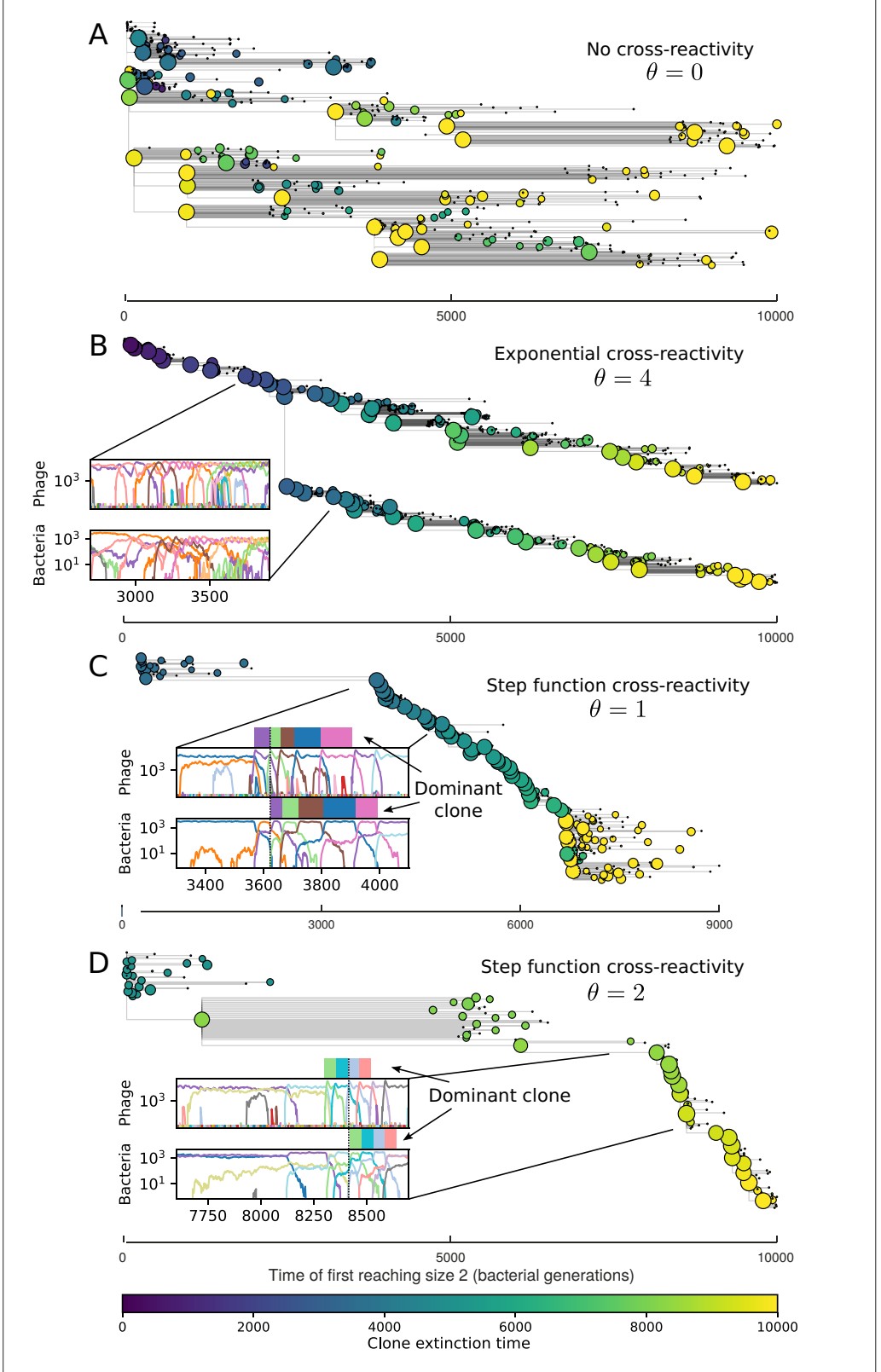

**Figure 4.** Cross-reactivity leads to 'spindly' phylogenies and regime switching. Phage clone phylogenies for four simulations with different cross-reactivities: no cross-reactivity (**A**), exponential cross-reactivity with $\theta = 4$ (**B**), and step-function cross-reactivity with $\theta = 1$ (**C**) and $\theta = 2$ (**D**). All simulations share all other parameters: $C_0 = 10^4, \eta = 10^{-4}, \mu = 10^{-6}, e = 0.95$. Phage clones are plotted at the first time they pass a population size of

*Figure 4 continued on next page*

*Figure 4 continued*

2 to remove clutter from many new mutations destined for extinction, and the size of each circle is logarithmically proportional to the maximum size reached by that clone. Colours indicate the time of extinction of each clone. For each simulation with cross-reactivity, the left inset shows phage (top) and bacteria (bottom) clone sizes over time; colours indicate unique clone identities. Coloured rectangles above insets in (**C**) and (**D**) correspond to the dominant clone at each time. Dominant clone identities are offset by $\theta$ (vertical dashed line for visual aid).

The online version of this article includes the following video(s) for figure 4:

**Figure 4—video 1.** Animation of simulation without cross-reactivity for *Figure 4A*.
https://elifesciences.org/articles/81692/figures#fig4video1

**Figure 4—video 2.** Simulation animation with exponential cross-reactivity for *Figure 4B*.
https://elifesciences.org/articles/81692/figures#fig4video2

**Figure 4—video 3.** Simulation animation with step-function cross-reactivity for *Figure 4C*.
https://elifesciences.org/articles/81692/figures#fig4video3

**Figure 4—video 4.** Simulation animation with step-function cross-reactivity for *Figure 4D*.
https://elifesciences.org/articles/81692/figures#fig4video4

neighbouring clones 'pull' each other along (*Figure 4B–D*), giving way to a regime in which new mutants have very low establishment rates in the case of step-function cross-reactivity (*Figure 4C*). When cross-reactivity is high and phages require multiple mutations to escape targeting, we expect new phage mutants to have a lower fitness on average because they may already be within the immunity range of existing bacterial clones. However, unlike without cross-reactivity, not all new mutant fitnesses are the same because fitness now depends on the distribution of matching spacers in the bacterial population. Cross-reactivity adds a 'direction' in the genetic landscape and a fitness gradient for new phage mutants, leading to a series of rapid establishments and a travelling-wave regime. We believe these rapid establishments also suppress instantaneous diversity; diversity emerges longitudinally instead of concurrently in this regime (*Figure 4* and Figure 61). This is true for both types of cross-reactivity we studied, but we also observed striking qualitative differences between exponential and step-function cross-reactivity. In the former, all phage mutants are guaranteed a slight fitness advantage, and the travelling wave appears right at the start of simulations, with occasional lineage-splitting events (*Figure 4B*). In contrast, step-function cross-reactivity means that mutants within the cross-reactivity radius have no fitness advantage at all. In this case, there is some variable initial length of time required to establish the travelling wave pattern: at the start of a simulation, all mutants are within the cross-reactivity radius and evolve purely neutrally. At least $d = \theta$ new mutants must establish through neutral dynamics before the next mutant can escape CRISPR targeting and grow with high fitness; this leads to the travelling-wave regime appearing later for $\theta = 2$ than for $\theta = 1$, and sometimes never appearing at all. Once the traveling wave appears, the dominant phage and bacteria clone types are exactly offset by $\theta$: the most abundant phage clone will in general be $\theta$ mutations away from the most abundant concurrent bacteria clone. This is directly visible in the clone trajectories in *Figure 4C–D* inset and Figure 54: matching-colour bacteria and phage clones are offset in time. Another regime is also possible in the case of step-function cross-reactivity: if multiple large clones establish that are each outside of all the others' cross-reactivity radii (Figure 63), then new mutants all have zero fitness and the travelling wave comes to an abrupt halt (around 7000 generations in *Figure 4C*). Establishment of new clones is once again extremely rare, and the existing large clones persist for a long time (Figures 53 and 62).

Changes to the fitness landscape of individual clones caused by cross-reactivity also influence population-level outcomes such as diversity. In general, we find that for high cross-reactivity, average diversity is lower than predicted for simulations without cross-reactivity because the probability of phage establishment per mutant decreases (*Figure 5A*). A decrease in phage mutant fitness as the strength of cross-reactivity increases is in line with the dependence of fitness on cross-reactivity reported by *Yan et al., 2019* and *Rouzine and Rozhnova, 2018*. Even with more complicated underlying dynamics, though, measuring average immunity alone is enough to reproduce total population sizes using our simple deterministic equations developed in the limit of no cross-reactivity (*Figure 5C*, 'Total population size'). This is because average immunity completely captures the population-level

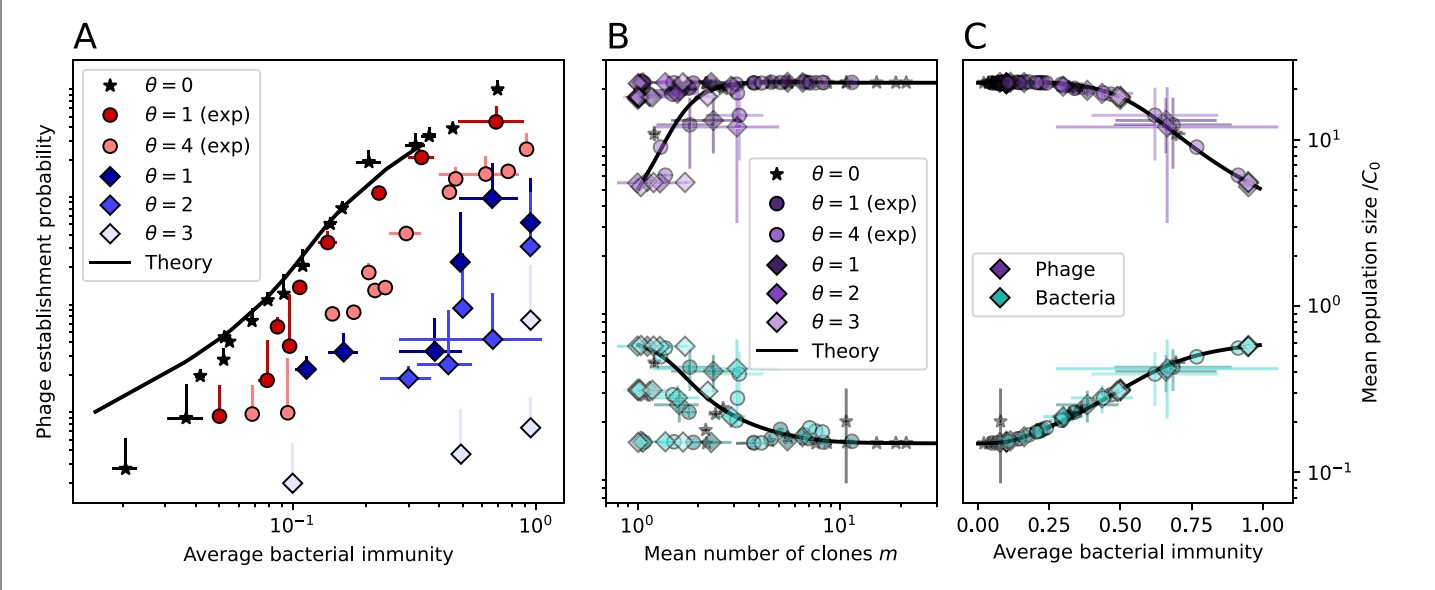

**Figure 5.** Average immunity underlies population outcomes. (**A**) Probability of phage clone establishment vs. average immunity for different amounts and types of cross-reactivity. No cross-reactivity ($\theta = 0$) is shown as black stars, exponential cross-reactivity in red, and step-function cross-reactivity in blue. Simulation averages are shown for $\eta = 10^{-4}$ and $\mu = 10^{-6}$. Error bars are the standard deviation across three or more independent simulations and are shown in both x directions and the positive y direction. (**B, C**) Total phage (purple) and total bacteria (teal) average population sizes vs. the mean number of bacterial clones $m$ (**B**) and vs. average bacterial immunity (**C**) for $\eta = 10^{-4}$. Each point is an average at steady state over three or more independent simulations with the same parameters; error bars are standard deviation. Total sizes are scaled by the initial nutrient concentration $C_0$. Lighter colours indicate stronger cross-reactivity, marker shapes match legends in (**A**) and (**B**). Solid lines are the predicted total population size given by solving *Equations 13*–*17* and using the approximation effective $e \approx e/m$ in (**B**) and the measured average immunity for effective $e$ in (**C**).

The online version of this article includes the following figure supplement(s) for figure 5:

**Figure supplement 1.** Total population sizes predicted by average immunity in a simulation with cross-reactivity.

impact of CRISPR immunity, and we can replace the CRISPR effectiveness parameter $e$ with measured average immunity in *Equations 13*–*17* to get very good agreement between measured and predicted population sizes even away from steady state (*Figure 5—figure supplement 1*). In contrast, inferring average immunity from the number of clones using the approximation that effective $e \approx e/m$ does not give good agreement with simulation results for simulations with cross-reactivity (*Figure 5B*).

*Figure 5B* makes an additional subtle point: higher immune diversity promotes larger population sizes for phages but not for bacteria. This is counterintuitive and appears to conflict with several experimental results that show that more bacterial diversity increases the likelihood of phage extinction (*van Houte et al., 2016*; *Common et al., 2020*) and decreases the ability of phages to adapt (*Morley et al., 2017*). This discrepancy appears at first glance to be a consequence of limiting bacteria and phages to a single spacer and protospacer in our model, but we propose that it is actually because bacteria and phage diversity is decoupled in prior experiments. In the following section, we introduce a toy model to understand the conceptual impact of multiple protospacers and spacers when overall phage and bacterial diversity is coupled.

## Pathogen and host diversity must be considered together

Our model restricts bacteria to a single spacer and phages to a single protospacer. This single-spacer assumption is a reasonable approximation for the experimental systems we consider: many liquid culture experiments find that most bacteria acquire a single spacer when exposed to phages, (*Heler et al., 2015*; *Heler et al., 2019*; *Pyenson and Marraffini, 2020*) even after up to 2 weeks (*Paez-Espino et al., 2013*; *Common et al., 2019*). Additionally, metagenomic results show that most spacer matches to viral sequences are located near the leader end of the CRISPR array *Weinberger et al., 2012a*. These results, combined with theory positing that recently acquired spacer provide more immune benefit than older spacers (*Childs et al., 2012*; *Han et al., 2013*), suggest that dynamics may be dominated by a very small number of spacers per bacterium.

Nevertheless, many bacteria possess tens to hundreds of spacers (*Pavlova et al., 2021*) and the question of multiple protospacers remains. Can we make inferences about how a more realistic multiple spacer or multiple protospacer scenario will change the relationship between diversity and average immunity in our model? First, we note that our model with exponential cross-reactivity is a good approximation for a realistic scenario where bacteria are limited to a single spacer but phages can have multiple protospacers. In this situation, multiple bacteria with unique spacers may target different protospacers in the same phage. A mutation in one protospacer will increase phage fitness but will not provide perfect immune escape, just as in the case of exponential cross-reactivity (*Figure 4B*). Even in this scenario, our qualitative results remain the same: bacteria and phage diversity is coupled and bacteria have higher average immunity at low diversity (*Figure 5B* and Figure 50).

We explored the consequences of both multiple spacers and multiple protospacers with a toy model. We generated synthetic sets of protospacers and spacers and varied the total diversity by changing the size of their pool and distributing their abundances exponentially to qualitatively match data (*Bonsma-Fisher et al., 2018*). We randomly assigned protospacers and spacers to individual phages and bacteria, then calculated average immunity. Importantly, we constrain phage and bacterial diversity to be coupled – overall spacer and protospacer diversity is the kept the same in all scenarios.

We find that across all combinations of array sizes and for two of three scenarios of protospacer diversity, average immunity is negatively correlated with diversity (Figure 69). Only when we limited phage diversity to a single mutating protospacer with all other protospacers conserved were bacteria with multiple spacers reliably able to target conserved protospacers regardless of the level of diversity in the variable protospacer position (Figure 69D). We compared our toy model to the experimental setup from *Common et al., 2020*. In their experiments, bacterial CRISPR spacer diversity was increased while phage diversity was kept constant with a single phage strain able to infect only one of the bacterial clones. This translated to an increasing initial average immunity as bacterial diversity increased and phages were able to infect a smaller proportion of the bacterial population (Figure 69A), which we think is not a realistic situation in a coevolving population of phages and bacteria.

Based on these results, we predict that in many realistic situations average immunity does not increase with diversity if phage and bacterial diversity are coupled. The intuition is this: if diversity of both bacteria and phages increase beyond the functional length of the CRISPR array, bacteria cannot be immune to all phage strains at once and average immunity must go down as diversity increases. The actual benefit of bacterial diversity depends very strongly on the characteristics of *phage* diversity (Figure 69) and manipulations of diversity should be understood in terms of their impact on average immunity. This has implications for understanding the so-called 'dilution effect' which is typically described as a decrease in the fraction of susceptible hosts as host diversity increases (*Chabas et al., 2018*; *Common et al., 2020*). Our model and analysis show that a dilution effect can only occur if host diversity increases out of proportion to pathogen diversity as in *Common et al., 2020* (Figure 69A), whereas if both host and pathogen diversity increase in tandem, the dilution effect actually changes direction, decreasing the fraction of pathogens that a host may be immune to as pathogen diversity increases (Figure 69B–D). In the same vein, previous theoretical work has found that CRISPR with a fitness cost would be selected against when viral diversity is high for this reason (*Weinberger et al., 2012b*; *Iranzo et al., 2013*), notably with models that allowed for multiple spacers and protospacers. Our results, building on existing theory, show that phage and bacterial diversity must be explored in tandem and that CRISPR immunity may provide less benefit when phage diversity is high.

## Dynamics are determined by diversity

How quickly does the phage population evolve? With explicitly modelled spacer and protospacer sequences of length $L = 30$, we quantify the speed of evolution as the 'mutational distance' per generation (*Rouzine and Rozhnova, 2018*): between two time points, how far in genome space has the phage population travelled, or how many mutations have occurred on average (Speed of evolution)?

The speed of evolution is both stable and repeatable between simulations at steady state and highly correlated between bacteria and phage (Figure 38). We find that the speed of evolution is inversely proportional to the time to extinction for large phage clones (*Figure 6B*, 'Measuring speed of evolution'). Intuitively, if phage clones turn over more quickly (small time to extinction), the population is able to move more quickly to a different genetic state, while if the time to extinction is large,

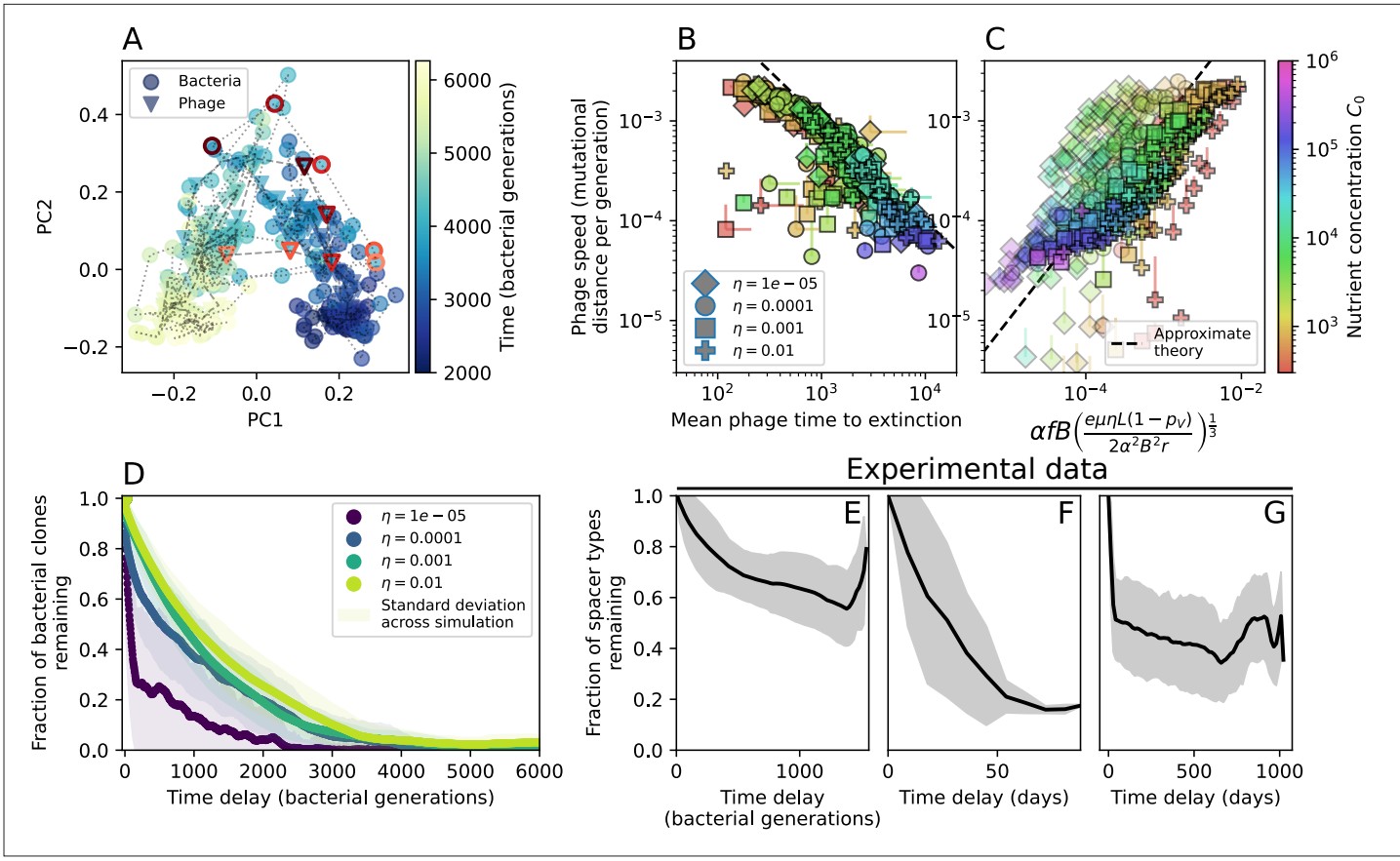

**Figure 6.** Phage evolution and spacer turnover. (**A**) Principal Component Analysis (PCA) decomposition of phage and bacteria clone abundances for a simulation with $C_0 = 10^4$, $e = 0.95$, $\eta = 10^{-4}$, and $\mu = 10^{-5}$. Clone abundances are normalized at each time point, then PCA is performed for the entire phage time series over $\approx 4000$ generations (four times the mean extinction time for phage clones). Bacteria and phage clone abundances are transformed into the PCA coordinates; colours indicate simulation time. Five time points are highlighted in progressively lighter shades of red for emphasis. (**B**) Phage genomic speed of evolution vs. mean large phage clone time to extinction. The phage speed is the weighted average genomic distance between the phage population at the end of the simulation and the phage population at an earlier time, divided by the time interval. The dashed line is $y = \frac{1}{x}$. (**C**) The speed of evolution increases as spacer effectiveness $e$, spacer acquisition probability $\eta$, and phage mutation rate $\mu$ increase. The dashed line shows an approximate theoretical calculation (assuming speed = 1/time to extinction) which captures the trend across a wide range of parameters. Error bars in (**B**) and (**C**) are the standard deviation across three or more independent simulations and are shown in the positive direction only. (**D**) Spacer turnover as a function of time delay for four simulations with $C_0 = 10^4$, $e = 0.95$, and $\mu = 10^{-5}$. The fraction of bacterial clones remaining is the fraction of clones that were present at time $t$ that are still present at time $t+$ delay. Solid lines are an average across steady-state for each value of the time delay; shaded regions are the standard deviation. (**E–G**) Spacer-type turnover calculated as in (**D**) using experimental data from *Paez-Espino et al., 2015* (**E**), metagenomic data sampled from groundwater from *Burstein et al., 2016* (**F**), and metagenomic data sampled from a wastewater treatment plant from *Guerrero et al., 2021a* (**G**). Experimental time points are interpolated to the minimum sampling interval to allow averaging across the experiment.

The online version of this article includes the following video and figure supplement(s) for figure 6:

**Figure 6—video 1.** PCA decomposition of phage and bacteria protospacer and spacer clone abundances for a simulation of bacteria and phages interacting with CRISPR immunity.

https://elifesciences.org/articles/81692/figures#fig6video1

**Figure supplement 1.** Clone size PCA for a simulation with exponential cross-reactivity.

**Figure supplement 2.** Genetic distance vs. diversity and time to extinction.

**Figure supplement 3.** Spacer turnover by spacer effectiveness.

**Figure supplement 4.** Spacer turnover by phage mutation rate.

new mutants are seeded close to parent populations that persist for a long time, limiting the mutational distance. Since the phage time to extinction has a simple relationship to bacterial diversity, we can also relate speed to diversity, and we find that the speed of evolution is proportional to diversity

and proportional to the same parameter combination to the power 1/3 (*Figure 6C*). As in the case of diversity, increasing phage mutation rate $\mu$ increases the speed of evolution sublinearly, and like diversity, the prediction diverges from simulation results at low $\eta$ as described in 'Measuring diversity.

Our simulations show continual spacer turnover at steady state (*Figure 6D*, *Figure 6—figure supplement 3*, *Figure 6—figure supplement 4*), a feature that we might also expect to see in actively evolving laboratory or natural populations of bacteria and phages. We analysed data from a long-term in vitro coevolution experiment with *Streptococcus thermophilus* bacteria and phage (*Paez-Espino et al., 2015*), data from a time-series sampling of a natural aquifer community of bacteria (*Burstein et al., 2016*), and data from a time-series sampling of a wastewater treatment plant (*Guerrero et al., 2021a*) (see 'Materials and methods') and calculated spacer turnover over time (*Figure 6E–G*). We found that spacer sequences experienced turnover in all cases, indicating ongoing change in the spacer content in these populations, though no experimental system sho wed complete turnover of all spacer types. Even if spacer loss is happening for non-selective reasons (e.g., mixing or flow in the aquifer system), turnover indicates that new spacers are being acquired as well and that CRISPR systems are active. Moreover, the timescale of early spacer turnover in the *S. thermophilus* experiment is similar to the timescale in our simulations — after 1000 generations, most simulations have between 40 and 60% of clones remaining (*Figure 6D*) and about 60% of clones remain after 1000 generations in the experimental data (*Figure 6E*). We modelled all simulation parameters on known parameters for *S. thermophilus* where possible.

## Time-shifted average immunity calculated from data reveals distinctive patterns of turnover

Bacteria acquire spacers in response to phage clones becoming large. The spacer composition of the bacterial population tracks the phage protospacer composition with a lag: *Figure 6A* shows the first two components of a PCA decomposition of bacteria and phage abundances over time in a simulation, a visual illustration of bacterial tracking. Without cross-reactivity, trajectories in this lower-dimensional space do not travel in a straight line; they are reminiscent of the diffusive coevolutionary trajectories in antigenic space described in a theoretical model of vertebrate virus-host coevolution (*Marchi et al., 2019*). In the travelling wave regime with cross-reactivity, trajectories are much more ballistic (*Figure 6—figure supplement 1*). Can we quantify how well and how quickly can bacteria track the evolving phage population with CRISPR?

Average immunity is a simple metric that quantifies the overlap between bacteria and phages. In experiments with both microbes and vertebrates, time-shift infectivity analyses between host and pathogen populations typically show that hosts are more immune to pathogens from the past and less immune to pathogens from the future (*Richman et al., 2003*; *Frost et al., 2005*; *Moore et al., 2009*; *Hall et al., 2011*; *Koskella, 2014*; *Betts et al., 2018*; *Common et al., 2019*; *Dewald-Wang et al., 2022*). We conduct a time-shift analysis on our simulation data (*Figure 7A and B*, *Figure 7—figure supplement 11*, *Figure 7—figure supplement 12*) and find that the same pattern holds true, but only for a limited time window: bacteria are indeed more immune to phages from the past, but this past immunity has a peak and then decays as we look further into the past, eventually reaching 0 immunity (*Figure 7B*). The presence of a peak in past immunity reflects the timescale of spacer turnover: once the bacterial population has lost all spacers from a previous time point, it is no longer immune to contemporaneous phages and time-shifted average immunity falls to 0. In simulations, the position of this peak is dependent on parameters such as $\eta$ and $\mu$ (*Figure 7E and F*). As $\mu$ increases, the peak occurs further in the past since phage are now moving more quickly away from bacteria tracking, while as $\eta$ increases, the peak moves closer to the present since bacteria are responding more quickly to changes in the phage population. This suggests that there is a tradeoff between immune memory durability and responding quickly to immune threats: bacteria must choose between tracking the phage population closely or keeping past immunity for a long time.

Time-shift analyses are usually performed explicitly by directly combining stored samples from different time points (*Betts et al., 2018*; *Common et al., 2019*; *Dewald-Wang et al., 2022*). However, by sequencing CRISPR spacers and phage genomes, a pseudo-analysis can be done without any time-shifted competition experiments by calculating the overlap between bacterial spacers and phage protospacers at different times delays. We performed such an analysis for two published datasets: a long-term laboratory coevolution experiment with *S. thermophilus* and phage

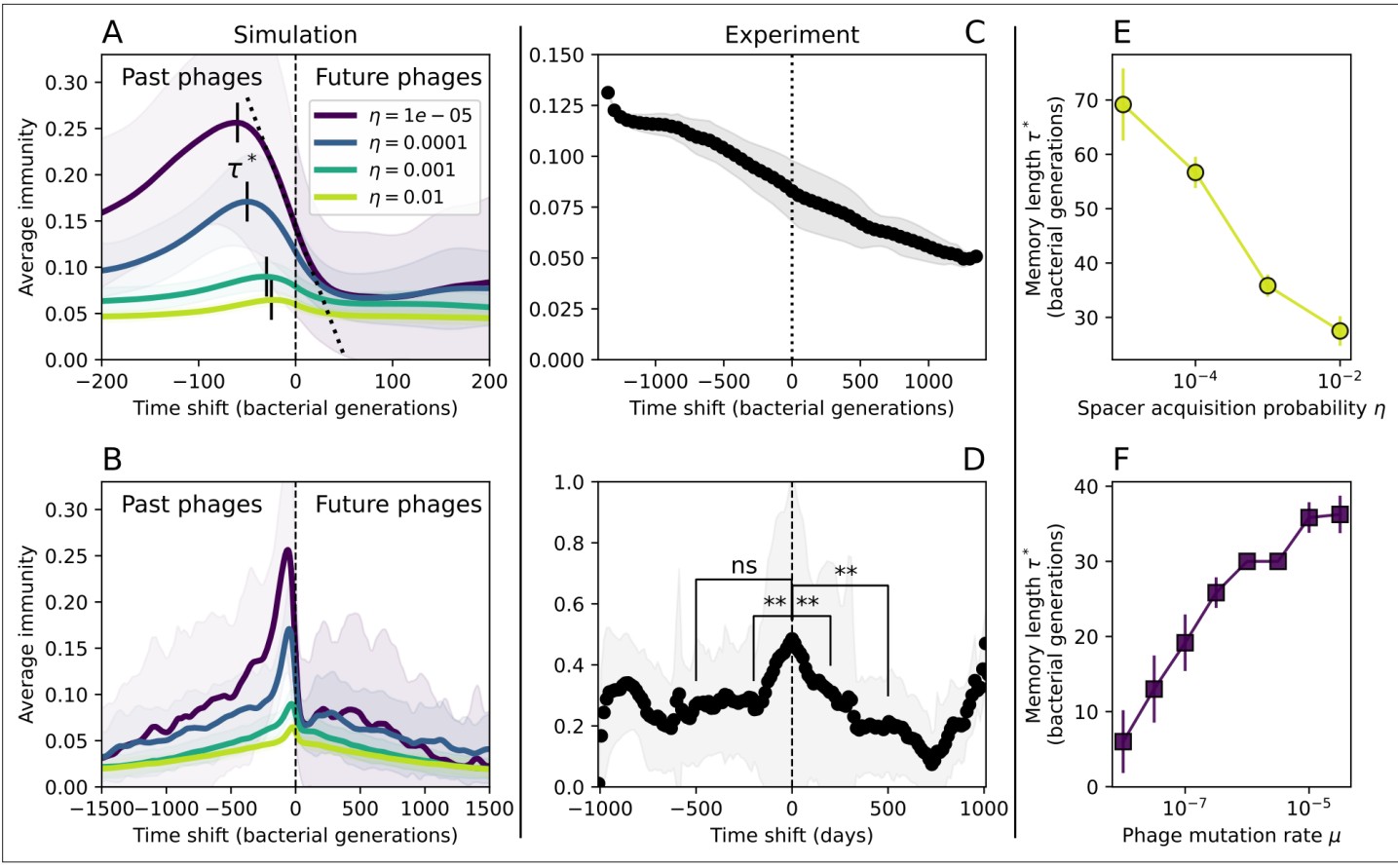

**Figure 7.** Quantifying immune memory in data. (**A, B**) Average immunity of bacteria against phage for four simulations with different values of $\eta$ as a function of time shift. Solid lines are an average across steady state for each value of the time shift; shaded regions are the standard deviation. Average immunity peaks in the recent past (**A**, indicated by $\tau^*$) with a negative slope through zero delay (**A**, black dashed line) and decays to zero at long delays in the past or future (**B**). For all simulations $C_0 = 10^4$, $\mu = 10^{-5}$, and $e = 0.95$. (**C, D**) Average overlap between bacterial spacer and phage protospacer types using data from a lab experiment with *S. thermophilus* and phage from *Paez-Espino et al., 2015* (**C**) and data from a wastewater treatment plant sampled over 3 years from *Guerrero et al., 2021a* (**D**). Spacer types are grouped by 85% similarity, and shaded region is standard deviation across averaged data. Base average immunity values were multiplied by the average number of protospacers corresponding to the *S. thermophilus* CRISPR system (**C**) and the *Gordonia* CRISPR systems (**D**) to account for multiple potential protospacer targets per phage. In (**D**), we compared two time shifts with zero delay average immunity using a Wilcoxon signed-rank test: $p = 0.27$ for lower past immunity at 500 days, $p = 0.008$ for lower past immunity at 200 days, $p = 0.001$ for lower future immunity at 500 days, and $p = 0.003$ for lower future immunity at 200 days. (**E, F**) The position of the peak in past immunity for simulated data vs. spacer acquisition probability $\eta$ (**E**) and phage mutation rate $\mu$ (**F**). The peak position is the time shift value for which the curves in (**A**) are largest, indicated by $\tau^*$. Error bars are the standard deviation across three or more independent simulations.

The online version of this article includes the following figure supplement(s) for figure 7:

**Figure supplement 1.** Time-shifted average immunity for experimental coevolution data.

**Figure supplement 2.** Time-shifted average immunity for experimental coevolution data with partial PAM matches included.

**Figure supplement 3.** Time-shifted average immunity for experimental coevolution data with all sequences regardless of PAM.

**Figure supplement 4.** Time-shifted average immunity for experimental coevolution data with data trimmed from start and end.

**Figure supplement 5.** Time-shifted average immunity for wastewater with data trimmed from start and end.

**Figure supplement 6.** Bootstrapped average immunity with all points randomly shuffled.

**Figure supplement 7.** Bootstrapped average immunity with pairs of bacteria and phage clone sizes shuffled.

**Figure supplement 8.** Time-shifted average immunity for wastewater with data trimmed from the end.

**Figure supplement 9.** Time-shifted average immunity for wastewater with data trimmed from the start.

**Figure supplement 10.** Time-shifted average immunity for wastewater with data trimmed from the start and end.

**Figure supplement 11.** Time-shifted average immunity by spacer effectiveness.

**Figure supplement 12.** Time-shifted average immunity by phage mutation rate.

(*Paez-Espino et al., 2015*) and time series of metagenomic samples over 3 years from a wastewater treatment plant (*Guerrero et al., 2021a*). In both experiments, whole-genome shotgun sequencing was performed on bacteria and phage DNA, and CRISPR spacers and protospacers were recovered and reported. We re-analysed these datasets to detect CRISPR spacers by finding sequences adjacent to known CRISPR repeats in raw reads. We also detected protospacers by finding matches to our detected CRISPR spacers in reads that did not match the CRISPR repeats or the bacterial reference genome(s) (see 'Laboratory coevolution experimental data'; 'Experimental data from wastewater treatment plant').

We grouped spacers with an 85% similarity threshold and interpolated counts between sequenced time points. We calculated the average overlap assuming $e = 1$ (perfect immunity from matching spacers) as a function of time delay, averaging over all combinations of interpolated data with the same time delay (*Figure 7C and D*). We found two qualitatively different trends as a function of time delay. In the laboratory coevolution experiment, we found that bacteria are more immune to past phages and less immune to future phages (*Figure 7C*, *Figure 7—figure supplement 1*), consistent with what we see at short time delay in our model (*Figure 7A*, black dashed line). In contrast, in the wastewater treatment plant data, we found a peak in average immunity that is roughly centred at zero time delay: bacteria are most immune to phages from their same time point, and immunity rapidly decays in both the past and the future (*Figure 7D*), qualitatively similar to our simulation results on very long timescales (*Figure 7B*). These different trends may reflect different regimes of CRISPR immune memory: in the laboratory experiment data, we see no evidence of the decay of immune memory, while in the wastewater treatment data we see a suggestion of decay on the timescale of weeks. Bacteria appear to track the phage population closely in the wastewater data, while in the laboratory data bacteria lag behind the phage population.

Because the variability in average immunity between time points was very high in the wastewater treatment plant data, we performed a Wilcoxon signed-rank test between the average immunity at zero time delay and the average immunity at a time delay ±200 and ±500 (see 'Calculating average immunity for details'). We found that past immunity after 200 days is significantly lower than present (Wilcoxon statistic $Z = 1241$ p-value $p = 0.008$), but that immunity after 500 days is not significantly lower than present ($Z = 392, p = 0.27$). For both time delays, we found that future immunity is significantly lower than present ($Z = 580, p = 0.0012$ for 500 days and $Z = 1293, p = 0.003$ for 200 days). Interestingly, these significance values are not symmetric if we pose the question from the perspective of phage: the overlap between phage and future bacteria at 500 days is lower than the overlap for present phages ($Z = 507, p = 0.024$), and the overlap between phage and past bacteria is lower than the overlap at present ($Z = 388, p = 0.027$, Figure 88). This asymmetry, that bacteria are generally more immune to all past phages while phage are not more infective against all past bacteria, is qualitatively the same as that reported by Dewald-Wang et al. in an explicit time-shift study of bacteria and phage immunity and infectivity in chestnut trees (*Dewald-Wang et al., 2022*).

## Discussion

Many bacteria and archaea possess CRISPR systems, and a significant fraction of these systems are likely to provide immunity against phages (*Brodt et al., 2011*; *Shmakov et al., 2017*; *Pourcel et al., 2020*). Given that bacteria and phages coexist in natural environments over extremely long timescales, the impact of CRISPR immunity in these steady-state conditions has remained underexplored. We constructed a phenomenological model of CRISPR immunity in a bacterial population interacting with phages to explore the impacts of adaptive immunity on population survival, fitness, and diversity. We found that both phage and bacterial genetic diversity emerged spontaneously with a minimal set of interactions, and we derived approximate analytic predictions for population outcomes. These rigorous analyses of our simple model lay a foundation for theoretical analysis of adaptive immunity in host-pathogen systems.

Our model is mechanistically simple, and we left out many known biological features and interactions by choice in order to gain a deep understanding of the factors that influenced our results. We modelled uniform spacer effectiveness, uniform spacer acquisition probability, and a constant phage mutation rate, all in a well-mixed system. This constitutes a null model of CRISPR immunity that provides a useful comparison point for both more accurate mechanistic models and experimental data. CRISPR is one of many other bacterial antiviral defence systems (*Bernheim and Sorek, 2020*), and within the

CRISPR world, CRISPR systems are highly evolutionarily diverse and there are many known differences in function and effect between CRISPR systems of different bacterial species (*Koonin et al., 2017*; *Hille et al., 2018*; *Makarova et al., 2020*; *Koonin and Makarova, 2022*). Experiments typically study a particular CRISPR system, and it is unclear which revealed mechanisms are specific to that system or are a more general property of CRISPR systems. For example, the spacer acquisition rate has been demonstrated to vary across particular phage sequences in several experiments (*Heler et al., 2019*; *Modell et al., 2017*; *Paez-Espino et al., 2013*) and is the source of differences in spacer abundance in some experiments (*Heler et al., 2019*), but whether this is a general principle that causes broad abundance distributions is not known. Many theoretical works also include mechanistic details such as a lag between infection and burst (*Levin et al., 2013*; *Santos et al., 2014*), multiple protospacers and spacers (typically with a fixed upper bound) (*Weinberger et al., 2012b*; *Iranzo et al., 2013*; *Bonsma-Fisher et al., 2018*; *Childs et al., 2014*; *Childs et al., 2012*; *Weissman et al., 2018*, spatial structure *Payne et al., 2018*; *Haerter et al., 2011*, autoimmunity *Weissman et al., 2018*; *Chabas et al., 2021*, and fitness costs of immunity and/or escape mutations *Weinberger et al., 2012b*; *Weissman et al., 2021*). Many phages also contain anti-CRISPRs which impact phage evolution and CRISPR immunity (*Bondy-Denomy et al., 2013*; *Hwang and Maxwell, 2019*). These details are all biologically important, but stripping them away as we do provides great insight into which population features depend on these details and which may be more general properties of adaptive immunity.

## Diversity and average immunity

Experimental manipulations of CRISPR diversity have focused on changing the number of bacterial spacers present in the population (*van Houte et al., 2016*; *Morley et al., 2017*; *Common et al., 2020*) while keeping phage diversity fixed (and low), with the exception of *Guillemet et al., 2021* who also explored high vs. low phage diversity. In contrast, emergent phage and bacterial diversity are tightly coupled in our model (*Figure 2B*). At high diversity, bacteria must 'choose' which of many phage clones to gain immunity against, meaning that they are then immune to a smaller fraction of the total phage population compared to when diversity is low. This is not a result of limiting bacteria and phage to a single spacer or protospacer; this trend also holds when there are multiple spacers or protospacers provided phage and bacteria diversity are correlated, as we showed in 'Pathogen and host diversity must be considered together'.

Our toy model described in 'Pathogen and host diversity must be considered together' shows that the framework of average immunity provides a conceptually intuitive way to understand the impacts of different types of diversity and modes of evolution. Future work to explicitly model the population dynamics of multiple spacers and protospacers with this lens will allow us to understand how and when these different evolutionary modes arise. For instance, it remains unknown how the linkage of different spacers and protospacers within genomes would affect individual clone dynamics. Our work effectively assumes that all spacers and protospacers evolve independently and are uncoupled, but in real genomes evolving with CRISPR, spacers and protospacers may hitchhike to prominence through selection acting on a different sequence in the same genome. Models of CRISPR locus spacer addition have shown that trailer-end clonality emerges from selective sweeps of newly added spacers (*Weinberger et al., 2012a*; *Han et al., 2013*), and hitchhiking will affect interpretations that can be drawn from the abundance of individual spacer clones. Exploring correlations between spacer abundances at different locus positions can be done in experimental data (such as *Paez-Espino et al., 2013*), and extending our simulation to explicitly include multiple protospacers and spacers per genome will provide further data.

Our results suggest that immune diversity itself may not be the most informative observable feature but that average immunity is what actually determines population outcomes. Indeed, when we introduced cross-reactivity between spacer sequences, we found that diversity decreased and that other processes such as establishment were impacted, but that average immunity still correctly predicted population sizes (*Figure 5C*). Average immunity completely summarizes the effect of CRISPR immunity and accurately predicts population outcomes. Experiments that manipulate bacterial diversity only change average immunity if they include a changing proportion of sensitive bacteria as in *Common et al., 2020* (Figure 69A).

Our results showed complex outcomes and a wide range of overall spacer diversity despite limiting bacteria and phage to a single protospacer and spacer. We also observed rapid loss of

previous spacer types as bacteria returned to the naive state before acquiring new spacers. In reality, bacteria can store multiple spacers in their CRISPR arrays (*Pavlova et al., 2021*; *Pourcel et al., 2020*) and phages can have several hundred to several thousand protospacer sequences which are determined by a particular protospacer-adjacent motif (PAM) in type I and II CRISPR systems (*Leenay et al., 2016*). In *S. thermophilus* phage 2972, for example, there are about 230 protospacers that can be acquired into the CRISPR1 locus of *S. thermophilus* (*Paez-Espino et al., 2013*) and 465 that can be acquired into the CRISPR3 locus. Similarly, many bacteria can store tens to hundreds of spacers in their CRISPR arrays (*Pavlova et al., 2021*; *Chen et al., 2022*; *Bradde et al., 2020*; *Pourcel et al., 2020*). Our single-spacer assumption is reasonable for the experimental systems we consider. In liquid culture experiments with staphylococci, most bacteria acquire a single spacer (*Heler et al., 2015*; *Heler et al., 2019*; *Pyenson and Marraffini, 2020*), and in short-term experiments with *S. thermophilus*, most bacteria acquired just one CRISPR spacer over 2 weeks (*Paez-Espino et al., 2013* or one to three spacers over 9 days *Common et al., 2019*). In the experimental data we analysed here (a 200+ day experiment), the average number of newly acquired spacers hovered around 5 throughout the experiment (*Paez-Espino et al., 2015*). In a long-term study of metagenomic data, most spacer matches to extant viral sequences were located near the leader end of CRISPR loci; conserved trailer-end spacers had very few matches to sampled viruses (*Weinberger et al., 2012a*). All these results suggest that dynamics may be dominated by just a handful of spacers per bacterium. In addition, theoretical work has found that the most recently acquired spacer provides more immune benefit than older spacers (*Childs et al., 2012*; *Han et al., 2013*). Our single-protospacer assumption for phages had a more extreme impact, however: in our model, we assumed in the non-cross-reactive case that a single point mutation allowed phages to escape from all CRISPR targeting, when in reality a phage with hundreds of protospacers may need mutations in many different protospacers to fully escape from the bacterial population. Previous work on the impact of CRISPR diversity takes place in this context: that having more diverse bacterial spacers means phages need mutations in multiple protospacers to escape, and that a single mutation only escapes a single bacterial spacer at most, leaving other bacteria still resistant (*Chabas et al., 2018*; *van Houte et al., 2016*; *Common et al., 2020*). We approximated this scenario of gradually increasing phage fitness per protospacer mutation by adding cross-reactivity to our model and showed in 'Cross-reactivity leads to dynamically unique evolutionary states' that the relationship between diversity and average immunity does not qualitatively change. We also observed the same qualitative trend in a toy model exploring the effects of both multiple spacers and multiple protospacers ('Pathogen and host diversity must be considered together').

Our model makes quantitative predictions about the relationship of population size and population outcomes to average immunity that can be tested in data. We directly calculated average immunity in experimental data without the need to assemble phage genomes (*Figure 7*), and we found that average immunity is anticorrelated with phage population size in data in one experimental dataset, as predicted by our model (Figure 77). However, we did not find a correlation between average immunity and read count (a proxy for population size) in wastewater treatment plant data (Figure 90). When calculating average immunity using experimental data, one must make assumptions about the immune benefit of spacers. In our work, we assumed that spacer and protospacer sequences that belong to the same similarity group provide perfect immunity. Other more complex assumptions are possible, and there has been a great deal of work quantifying the efficiency of spacers based on the position of SNPs in the protospacer in vitro (see *Sternberg et al., 2015* for an example) and in vivo (*Soto-Perez et al., 2019*). Additionally, phages regularly acquire mutations in PAMs, and these mutations strongly decrease the targeting efficiency of bacterial spacers (*Leenay et al., 2016*; *Sun et al., 2013*).

In certain limits, average immunity can be related to the Morisita–Horn similarity index $\Psi$ *Wolda, 1981* (see Appendix 1), where instead of comparing two populations from different timepoints or different regions, we compared bacteria and phage populations scaled by the immune benefit of CRISPR. By comparing our definition of average immunity to the Morisita–Horn index, we find that the Morisita overlap is constant across all parameters we study, which implies that the population diversity evolves to attain the highest possible overlap. Average immunity is also theoretically similar to the concept of distributed immunity introduced by *Childs et al., 2014*; *Childs et al., 2012*, the 'immunity' quantity calculated by *Han et al., 2013*, and the probability of an immune encounter used

by *Iranzo et al., 2013* to derive mean-field population predictions. The mathematical definition of average immunity is slightly different than these quantities, but the effect on populations is strikingly similar, and it is significant that several different models and different metrics find that a quantification of immunity is the important predictor for population-level outcomes.

## In silico time-shift experiments yield qualitative trends

We calculated average immunity in two experimental datasets and performed synthetic time-shift experiments by calculating average immunity between time-shifted populations of bacteria and phage. This effectively replicates explicit time-shift experiments (*Richman et al., 2003*; *Frost et al., 2005*; *Moore et al., 2009*; *Blanquart and Gandon, 2013*; *Koskella, 2014*; *Chabas et al., 2016*; *Pyenson and Marraffini, 2020*; *Laanto et al., 2017*; *Common et al., 2019*; *Dewald-Wang et al., 2022*). A similar time-shift calculation using sequence data was performed by *Guillemet et al., 2021*; they approached the question from the phage perspective and found that phages are most infectious against bacteria from the past and least infectious against bacteria from the future. Our results show that powerful insights into the dynamical immune state of a population can be obtained from sequencing data without explicitly performing time-shift experiments. We applied the same simple method to two very different experimental populations, suggesting that this analysis may be feasible in other time-series datasets, including from other natural microbial populations.

Our model predicts that an immune-evolving population in true steady state must eventually experience declining past immunity. CRISPR arrays are not infinite, and spacers must eventually be lost, even though this may happen on very long timescales. This pattern of declining past immunity is also in line with the expectations of fluctuating selection dynamics in which more rare genotypes are more fit (*Gaba and Ebert, 2009*; *Hall et al., 2011*; *Blanquart and Gandon, 2013*; *Dewald-Wang et al., 2022*). Previous theoretical work applied to vertebrate adaptive immunity also supports the existence of a peak in adaptive benefit in the recent past (*Blanquart and Gandon, 2013*; *Nourmohammad et al., 2016*), and some experimental time-shift studies have reported a peak in adaptation for bacteria-phage interactions (*Hall et al., 2011*; *Koskella, 2014*; *Dewald-Wang et al., 2022* and for HIV-immune system interactions *Blanquart and Gandon, 2013*; *Nourmohammad et al., 2016*). We saw two qualitatively different time-shift curves in the experimental data we analysed: in the long-term laboratory coevolution data, bacteria were more immune to all past phages and less immune to all future phages (*Figure 7C*), while in the wastewater treatment plant data, bacteria were most immune to phages in their present context and less immune to phages in both the past and the future (*Figure 7D*). The time-shift pattern of the laboratory coevolution data was consistent with experimental time-shift data that is typically said to be indicative of arms-race dynamics (*Richman et al., 2003*; *Frost et al., 2005*; *Moore et al., 2009*; *Blanquart and Gandon, 2013*; *Betts et al., 2018*; *Common et al., 2019*), while the wastewater data may be consistent with a 'zoomed-out' view of the eventual decline in immunity in both the distant past and distant future (*Nourmohammad et al., 2016*; *Guillemet et al., 2021*). Indeed, the distinction between arms-race dynamics and fluctuating selection dynamics in time-shift experiments may be a question of the timescale being investigated (*Gaba and Ebert, 2009*; *Blanquart and Gandon, 2013*; *Dewald-Wang et al., 2022*). Our simulation results may qualitatively match both of these regimes depending on the timescale observed, which suggests that measuring very long-term coexistence is an area for further work in host-parasite coexistence experiments. In general, we expect that the timescale of memory length is related to the size of CRISPR arrays: if overall rates of spacer gain and loss are balanced, then we expect an individual spacer's lifetime in the population to be proportional to array length. The factors that affect the timescale of memory length is an interesting area for further work.

One reason why past immunity did not unambiguously decline in our analysis of the laboratory coevolution data may be that spacer loss was not observed in the original experiment (*Paez-Espino et al., 2015*. This is also true of several other similar experiments with *S. thermophilus Paez-Espino et al., 2013*; *Weissman et al., 2018*), though spacer loss was directly observed coincident with spacer acquisition in *Deveau et al., 2008*. Spacer loss has also been observed in other experimental systems (*Jiang et al., 2013*; *Rao et al., 2017*; *Deecker and Ensminger, 2020*; *Garrett, 2021*), and indirect evidence of spacer loss through sequence comparisons has been reported for *S. thermophilus* (*Horvath et al., 2008* and metagenomic data *Held et al., 2010*; *Weinberger et al., 2012a*). Spacers may also be functionally lost despite remaining in the genome, either from stochastic decoupling

of RNA polymerase (*Zoephel and Randau, 2013*; *Martynov et al., 2017*; *Soto-Perez et al., 2019* or from a dilution effect of competition for limited Cas protein complexes *Martynov et al., 2017*; *Bradde et al., 2020*; *Garrett, 2021*). The 'loss' rate in our model could also be taken to be gain and loss of the entire CRISPR system, which is known to happen (*Delaney et al., 2012*; *Rollie et al., 2020*), although this would likely happen on much slower timescales than the acquisition and loss we model. The qualitative shape and timescale of time-shift curves like the ones we presented may indirectly contain information about the timescale of spacer loss.

We found in our simulations that memory length decreased as spacer acquisition probability increased and as phage mutation rate decreased: if bacteria followed phages more closely (higher $\eta$) or if phages evolved more slowly (lower $\mu$), the peak in past immunity was closer to the present and memory was more quickly lost (*Figure 7A, E, and F*). A similar correspondence was derived in a theoretical model of optimal immune memory formation in vertebrates: more frequent sampling of the pathogen landscape (corresponding to higher spacer acquisition $\eta$ in our model) led to faster memory loss (*Mayer et al., 2019*). Notably, that result is the optimal update scheme for an immune system wishing to minimize the costs of infection; there is no explicit optimization in our model, yet we find a pattern consistent with a potentially optimal solution.

## Vertebrate adaptive immunity

CRISPR adaptive immunity is mechanistically very different from the vertebrate adaptive immune system, yet there are several striking conceptual similarities between them and in the way past theoretical works have modelled vertebrate immunity. For example, phenomenological models of binding affinity between antigens and the immune system have used the degree of similarity between strings as a metric for interaction strength (*Detours and Perelson, 1999*; *Detours and Perelson, 2000*; *Chao et al., 2005*; *Wang et al., 2015*; *Nourmohammad et al., 2016*; *Sachdeva et al., 2020*), and more abstract models of the change in immunity after a virus mutation have also used string similarity to track the strength of immune coverage (*Luksza and Lässig, 2014*; *Rouzine and Rozhnova, 2018*; *Yan et al., 2019*). This is a simplification of a complex protein-protein interaction in the case of the vertebrate immune system (*Altan-Bonnet et al., 2020*), but it is nearly identical to the actual mechanism of CRISPR immunity, suggesting that lessons from models of CRISPR immunity may in turn be applicable to other forms of adaptive immunity.

Pathogen evolution has been extensively explored in models of vertebrate immunity, and here too we can draw conceptual parallels. We found that the speed of phage evolution depended sublinearly on phage population size in our model (*Figure 6C*), analogous to the weak positive dependence on population size described by *Marchi et al., 2021*. Adding cross-reactivity between spacer types in our model changed overall population outcomes as well as population dynamics. We observed a traveling wave regime in which new phage mutants 'pull' existing bacterial clones along, even if their matching clone is small or extinct (Figure 54). This travelling wave regime is qualitatively similar to that observed in theoretical models of 'ballistic' pathogen evolution in effective low-dimensional antigenic space (*Yan et al., 2019*; *Marchi et al., 2019*; *Marchi et al., 2021*). Similar clone population dynamics were observed in a model of B cell activation and memory generation in which existing memory B cells are reactivated in response to infection with a mutated virus that remains within its cross-reactivity radius (*Chardès et al., 2022*). We also observed splitting of lineages into divergent subtypes in our model, a phenomenon that has been studied in vertebrate pathogen evolution as well (*Marchi et al., 2019*; *Marchi et al., 2021*). We found that the average number of distinct clans was proportional to overall diversity but that increasing cross-reactivity decreased the average clan size ('Number and size of clone clans'). These qualitative similarities highlight the usefulness of CRISPR adaptive immunity to understand population dynamics that also occur in the vertebrate immune system.

Selection in the vertebrate immune system happens at many distinct stages within individuals, including T cell receptor selection in the thymus (*Camaglia et al., 2022* and affinity maturation of B cells following an immune challenge *Chardès et al., 2022*; *Nourmohammad et al., 2019*; *Nourmohammad et al., 2016*; *Molari et al., 2020*). Selection also happens at the population level as pathogens and individuals coevolve over time (*Marchi et al., 2019*; *Yan et al., 2019*, for instance, in the evolution of influenza *Luksza and Lässig, 2014*; *Yan et al., 2019*). In contrast, within-host dynamics are much simpler in bacteria, and the possibilities for within-host coevolution are especially limited in the case of virulent phages that kill their hosts after a single infection. Similarly, in the vertebrate

immune system, extremely high immune diversity is possible in a single individual, whereas in CRISPR immunity, total diversity per individual is many orders of magnitude lower: most bacteria contain tens of spacers with a very few examples of a few hundred spacers (*Pavlova et al., 2021*). This reintroduces the question of how to think about immunity in microbial populations, whether as playing out on the level of individuals or on the level of the entire population. Finally, unlike the vertebrate adaptive immune system, CRISPR immunity is heritable, and so the optimal immune strategy for bacteria takes place on a timescale potentially much longer than an individual's lifetime as well as on the level of the entire population *Mayer et al., 2016*. We have highlighted several analogies between CRISPR adaptive immunity and vertebrate adaptive immunity. Overall, the mechanistic simplicity and experimental tractability of CRISPR immunity mean that abstract ideas from vertebrate immunity can be linked to measurable features of CRISPR immunity, and insights from CRISPR immunity may be useful in understanding vertebrate immunity as well.

## Primed acquisition

In some CRISPR systems, an imprecise match between a spacer and protospacer results in a much higher spacer acquisition rate, a phenomenon called priming (*Rao et al., 2017*). Including cross-reactivity in our model is a way to explore the effect of primed acquisition. However, we modelled cross-reactivity as a change in phage infection success probability, whereas in primed acquisition it is the spacer acquisition rate that changes. Because bacteria must lose their spacer before gaining a new spacer in our model, including primed acquisition directly would require expanding bacterial arrays to at least two spacers. The acquisition probability $\eta$ could then become a function of protospacer-spacer similarity to directly model primed acquisition.

## Selection patterns in CRISPR immunity

The mechanism of selection for immune variants in CRISPR adaptive immunity is a matter of debate. We observed several features that suggest the presence of negative frequency-dependent selection in our model: (a) decreased clone growth rates as clone size increased, (b) cyclic clone size dynamics as immune pairs oscillate in size, and (c) continual turnover in spacer types over time. This is consistent with recent experimental observations of negative frequency-dependent selection in host-pathogen systems (*Betts et al., 2018*; *Guillemet et al., 2021*). Our model also contains the base assumption that phages that can successfully infect dominant bacterial strains will experience positive selection, generating 'kill-the-winner' dynamics that lead to negative frequency-dependent selection for bacteria (*Weinberger et al., 2012a*). Our model does not capture true host-pathogen coevolution because bacteria are not able to evolve beyond 'following' the mutational landscape of phages by acquiring spacers. In true coevolution, bacterial genomic mutations, apart from the ability to acquire spacers, are also drivers of evolution and sources of selection pressure on phages.

## Phase transitions in immunity

We observed large population size fluctuations when average immunity reached a particular value of approximately 10% for the parameters we used (Figure 31). This critical value is determined by a balance of the rate of spacer acquisition and spacer loss that leads to exactly equal fitness for both bacteria with spacers and bacteria without spacers (see 'Dominant balance approximations for $\nu'$'). Despite the presence of spacers with non-zero effectiveness, near this critical point the population appears to behave as if there is no CRISPR immunity at all. This has two interesting results: first, the probability of phage establishment drops to zero for some simulations near this critical point (*Figure 3—figure supplement 2*), since if CRISPR has no influence then mutant phages do not experience strong positive selection. Second, population sizes become unstable near the transition: outcomes with quite different total bacteria population size and fraction of bacteria with spacers are possible (Figures 30 and 31). This critical point will be interesting to explore further as a possible phase transition.

## Experimentally accessible parameters

We found that diversity in our model depended on spacer acquisition rate, spacer effectiveness, and spacer mutation rate. These parameters are all known to vary in natural systems, and some may also be experimentally manipulated. CRISPR systems display a range of spacer acquisition rates even within

the same system (*Heler et al., 2019*), and both spacer acquisition rate and spacer effectiveness may be controlled by the level of expression of Cas proteins, which has been found to be connected to the quorum sensing pathway (*Høyland-Kroghsbo et al., 2017*). Additionally, recent work has found that spacer acquisition is more efficient at slow bacterial growth rate because phages also grow more slowly, giving bacteria more time to acquire spacers (*Høyland-Kroghsbo et al., 2018*; *Dimitriu et al., 2022*). This change in bacterial growth rate can be controlled by nutrient concentration (*Dimitriu et al., 2022*; *Payne et al., 2018*, temperature *Høyland-Kroghsbo et al., 2018*, or by bacteriostatic antibiotics which slow bacterial growth without killing *Dimitriu et al., 2022*). Phage mutations rates also vary and may be experimentally modified (*Kysela and Turner, 2007*). These findings all represent possible experimental manipulations of parameters relevant for CRISPR immunity; for example, given these findings we predict that a bacteria-phage population exposed to bacteriostatic antibiotics would evolve higher CRISPR spacer diversity than one without antibiotics.

## Methods
### Model
In this section, we describe the details of our phenomenological model of bacteria and phages, providing more detail for *Figure 1A*. Sections 'Reactions' and 'Master equation' list the stochastic interactions we simulate, and 'Clone size master equation' lists master equations that are used to derive mean-field equations for this dynamical system (Total population size) and clone size distributions for bacteria and phages (Appendix 2).

We model bacteria and phages interacting in a chemostat. The populations we track are nutrient concentration ($C$), phages ($n_V$), and bacteria ($n_B$). Bacteria can either have no spacer ($n_B^0$) or a spacer of type $i$ ($n_B^i$), and phages can have a single protospacer of type $j$ ($n_V^j$). Nutrients flow in at concentration $C_0$ with rate $F$, and all species flow out with rate $F$. The total number of bacteria with a spacer is $n_B^s$ and the total number of bacteria is $n_B$. Bacteria grow at rate $gC$. With rate $\alpha$, a phage interacts with a bacterium. With probability $p_V$, the phage will kill bacteria without a matching spacer and produce a burst of new phages with size $B$, while for bacteria with a matching spacer that probability is reduced to $p_V^s = (1-e)p_V$ ($0 \leq e \leq 1$). Bacteria without spacers that survive an attack have a chance to acquire a spacer with probability $\eta$. Bacteria with a spacer lose their spacer at rate $r$. Phages are limited to a single protospacer, and phages can mutate to a new protospacer type with probability $\mu L$, where $\mu$ is the per-base mutation rate and $L$ is the protospacer length ($L = 30$ in all simulations). Parameter descriptions and default values are shown in Table 3. See refs *Weissman et al., 2021*; *Gurney et al., 2019* for recent examples of other chemostat-based models.

### Reactions
*Table 1* lists all the interactions present in our model between individual bacteria ($b$), phages ($V$), and nutrients ($C$).

In *Table 1*, $p_V(i,j)$ is the probability of a phage with protospacer type $j$ killing a bacteria with spacer type $i$.

$P_n$ in *Table 1* is the probability of $n$ mutant phages in a burst, given by a binomial distribution:

$$P_n = P(n; B, 1-P_0) = \binom{B}{n}(1-P_0)^n (P_0)^{B-n} \tag{9}$$

where $P_0 = e^{-\mu L}$ is the probability of no mutations for an individual phage. $L$ is the protospacer length and $\mu$ is the mutation probability per base per generation.

In this formulation of the mutation term, new mutants are assumed to be a new, not previously existent, phage type which increases $m$ by $n$, the number of mutations ($+ \sum_{k=1}^{n} V^{m+k}$). In our simulations, phage mutations can happen to existing types, but these are rare events and we neglect them in our theoretical analysis.

### Master equation
The reactions in *Table 1* can be formulated as a master equation describing the probability of observing $n_B^0$ bacteria without spacers, the set of $n_B^i$ bacteria with spacers of type $i$, the set of $n_V^j$ phages with protospacers of type $j$, and a nutrient concentration of $C$ at time $t$ (*Equation 10*).

**Table 1.** Model reactions.

| | |
|---|---|
| $b^{0,i} + C \xrightarrow{g} 2b^{0,i}$ | Bacterium divides |
| $b^{0,i} \xrightarrow{F} \emptyset$ | Bacterium flows out |
| $V^j \xrightarrow{F} \emptyset$ | Phage flows out |
| $\emptyset \xrightarrow{FC_0} C$ | Nutrients flow in |
| $C \xrightarrow{F} \emptyset$ | Nutrients flow out |
| $\sum_{n=0}^{B} \left( b^0 + V^j \xrightarrow{\alpha p_V P_n} (B-n)V^j + \sum_{k=1}^{n} V^{m+k} \right)$ | Interaction, phage wins, |
| | $P_n$ probability of $n$ mutuant phages |
| $b^0 + V^j \to \alpha(1 - p_V)(1 - \eta)b^0$ | Interaction, bacterium survives |
| $b^0 + V^j \to \alpha(1 - p_V)\eta b^i$ | Interaction, bacterium survives and acquires a spacer |
| $\sum_{n=0}^{B} \left( b^i + V^j \to \alpha p_V(i,j)P_n(B-n)V^j + \sum_{k=1}^{n} V^{m+k} \right)$ | Interaction, phage wins, |
| | $P_n$ probability of $n$ mutant phages |
| $b^i + V^j \xrightarrow{\alpha(1 - p_V(i,j))} b^i$ | Interaction, bacterium survives |
| $b^i \xrightarrow{r} b^0$ | Bacterium loses spacer |

$$
\begin{aligned}
\frac{dP(n_B^0, \{n_B^i\}, \{n_V^j\}, C, t)}{dt} &= g(C+1)(n_B^0 - 1)P(n_B^0 - 1, \{n_B^i\}, \{n_V^j\}, C+1, t) \\
&+ \sum_{k=1}^{m} g(C+1)(n_B^k - 1)P(n_B^0, \{n_B^{i \neq k}\}, n_B^k - 1, \{n_V^j\}, C+1, t) \\
&+ F(n_B^0 + 1)P(n_B^0 + 1, \{n_B^i\}, \{n_V^j\}, C, t) \\
&+ \sum_{k=1}^{m} F(n_B^k + 1)P(n_B^0, \{n_B^{i \neq k}\}, n_B^k + 1, \{n_V^j\}, C, t) \\
&+ \sum_{\ell=1}^{m} F(n_V^\ell + 1)P(n_B^0, \{n_B^i\}, \{n_V^{j \neq \ell}\}, n_V^\ell + 1, C, t) \\
&+ F(C+1)P(n_B^0, \{n_B^i\}, n_V, C+1, t) \\
&+ FC_0 P(n_B^0, \{n_B^i\}, n_V, C-1, t) \\
&+ \sum_{\ell=1}^{m} \alpha(1 - p_V)(1 - \tfrac{\eta}{m})n_B^0(n_V^\ell + 1)P(n_B^0, \{n_B^i\}, \{n_V^{j \neq \ell}\}, n_V^\ell + 1, C, t) \\
&+ \sum_{k=1}^{m}\sum_{\ell=1}^{m} \tfrac{\alpha(1 - p_V)\eta}{m}(n_B^0 + 1)(n_V^\ell + 1)P(n_B^0 + 1, \{n_B^{i \neq k}\}, n_B^k - 1, \{n_V^{j \neq \ell}\}, n_V^\ell + 1, C, t) \\
&+ \sum_{k=1}^{m}\sum_{\ell=1}^{m} \alpha(1 - p_V(k,\ell))n_B^k(n_V^\ell + 1)P(n_B^0, \{n_B^{i \neq k}\}, n_B^k, \{n_V^{j \neq \ell}\}, n_V^\ell + 1, C, t) \\
&+ \sum_{k=1}^{m}\sum_{\ell=1}^{m}\sum_{n=0}^{B} \alpha p_V(k,\ell)P_n(n_V^\ell - (B-n) + 1)(n_B^k + 1) \\
&\quad P(n_B^0, \{n_B^{i \neq k}\}, n_B^k + 1, \{n_V^{j \neq \ell}\}, n_V^\ell - (B-n) + 1, C, t) \\
&+ \sum_{k=1}^{m}\sum_{n=0}^{B} \alpha p_V P_n(n_V^\ell - (B-n) + 1)(n_B^0 + 1) \\
&\quad P(n_B^0 + 1, \{n_B^i\}, \{n_V^{j \neq \ell}\}, n_V^\ell - (B-n) + 1, C, t) \\
&+ \sum_{k=1}^{m} r(n_B^k + 1)P(n_B^0 - 1, \{n_B^{i \neq k}\}, n_B^k + 1, \{n_V^j\}, C, t) \\
&- \left( F(n_B^0 + \sum_{k=1}^{m} n_B^k + \sum_{\ell=1}^{m} n_V^\ell + C + C_0) + gC(n_B^0 + \sum_{k=1}^{m} n_B^k) \right. \\
&\quad \left. + \alpha \sum_{\ell=1}^{m} n_V^\ell(n_B^0 + \sum_{k=1}^{m} n_B^k) + r \sum_{k=1}^{m} n_B^k \right) P(n_B^0, \{n_B^i\}, \{n_V^j\}, C, t)
\end{aligned}
\tag{10}
$$

Each term is included only if all respective population quantities are $\geq 0$; for instance, the first term is only included if $n_B^0 > 0$, since if $n_B^0 = 0$ then $n_B^0 - 1 < 0$ and the entire term is negative and non-physical. This is important to note especially in terms containing $n_V - (B - n) + 1$, which are only included if $n_V > B - n - 1$.

## Clone size master equation

We can also write master equations for the total number of clones of size $k$. These equations describe the population-level distribution of clone sizes instead of the size of a single typical clone as in **Equations 123 and 147**.

The master equation for $b_k$, the number of bacteria clones of size $k$, is:

$$\partial_t b_k = \underbrace{gC[(k-1)b_{k-1} - kb_k]}_{\text{Bacteria growth}} + \underbrace{(F + r)[(k+1)b_{k+1} - kb_k]}_{\text{Removal of spacer-containing bacteria by chemostat outflow or spacer loss}}$$

$$+ \underbrace{\alpha \sum_\ell \ell v_\ell \left[(k+1)b_{k+1}p_V(k+1,\ell) - kb_k p_V(k,\ell)\right]}_{\text{Phage predation - reduces clone size by 1 for each successful infection}} \qquad (11)$$

$$+ \underbrace{\alpha \eta n_B^0 n_v^{k*}(1 - p_V)\left[b_{k-1} - b_k\right]}_{\text{New bacteria clones from spacer acquisition}}$$

The last term representing spacer acquisition is complicated: $n_V^{k*}$ is the total number of phages which contain protospacers that match bacteria clones of size $k - 1$, multiplied by the probability of getting that protospacer if infected by that phage. In other words, these are the phages that bacteria clones of size $k - 1$ could acquire spacers from to increase to size $k$, multiplied by the probability of acquiring a particular spacer from those phages. In a model where phages are all identical with a large number of protospacers $m$, $n_V^{k*} = n_V/m$, where $n_V$ is the total number of phages and $1/m$ is the probability of acquiring a particular spacer. If instead phages can only have a single protospacer, $n_V^{k*}$ is not known in general, but approximations may be found; for instance, if there are $m$ phage types present and all phage clones are the same size, $n_V^{k*}$ is again equal to $n_V/m$.

The corresponding master equation for $v_\ell$, the number of phage clones of size $\ell$, is:

$$\partial_t v_\ell = \underbrace{(F + \alpha(n_B^0 + n_B^s))[(\ell+1)v_{\ell+1} - \ell v_\ell]}_{\text{Phage death from chemostat flow and adsorption}}$$

$$+ \underbrace{\alpha \sum_k k b_k \left[\sum_{n=0}^{B} P_n(\ell - (B-n))v_{\ell-(B-n)}p_V(k, \ell - (B-n)) - \ell v_\ell p_V(k, \ell)\right]}_{\text{Phage burst after infecting spacer-containing bacteria (subtracting mutant phages which become new clones)}}$$

$$+ \underbrace{\alpha n_B^0 p_V \left[\sum_{n=0}^{B} P_n(\ell - (B-n))v_{\ell-(B-n)} - \ell v_\ell\right]}_{\text{Phage burst after infecting bacteria without spacers}} \qquad (12)$$

$$+ \underbrace{\delta_{\ell,1}\alpha B(1 - e^{-\mu L_p})\left[\sum_{k',\ell'}p_V(k',\ell')k'b_{k'}\ell' v_{\ell'} + p_V n_B^0 \sum_{\ell'}\ell' v_{\ell'}\right]}_{\text{New phage clones created by mutation from both spacer-containing and naive bacteria}}$$

The phage master equation assumes that all new mutants are unique and that they are a type not currently present in the population.

$P_n$ is a binomial distribution giving the probability of $n$ mutant phages per burst (**Equation 9**).

$p_V$ is the probability of phage successfully infecting bacteria, and in general is an unknown function of both $k$ and $\ell$. All details of immunity are contained in $p_V$; later we discuss certain choices of $p_V$ and their implications for the model and population dynamics. $p_V$ with no functional arguments is the probability of phage success against bacteria with no spacers, a constant.

## Mean-field model results

### Total population size

These mean-field equations for total bacteria, bacteria with spacers, phages, and nutrients are the backbone of our analytic model results. Throughout this work, we assume that clone dynamics take place in a background of total populations at steady state; these equations describe the model that we solve to find these steady-state quantities underlying results in 'Bacteria and phages dynamically coexist and coevolve', 'Phages drive stable emergent sequence diversity', 'What determines the fitness and establishment of new mutants?', 'Cross-reactivity leads to dynamically unique evolutionary states', and 'Dynamics are determined by diversity'.

The total number of bacteria with spacers, $n_B^s$, is equal to $\sum k b_k$, and the total number of phages, $n_V$, is $\sum \ell v_\ell$ and $\partial_t n_V = \sum \ell \partial_t v_\ell$, and summing over $k$ and $\ell$ in *Equations 11 and 12* gives mean-field equations:

$$\dot{n_B^s} = (gC - F - r)n_B^s + \alpha(1 - p_V)\eta n_B^0 n_V - \alpha n_B^s n_V p_V^a \tag{13}$$

$$\dot{n_V} = -(F + \alpha(n_B^0 + n_B^s))n_V + \alpha B(p_V^a n_B^s n_V + p_V n_B^0 n_V) \tag{14}$$

We can also write mean-field equations for the total number of bacteria without spacers, $n_B^0$, and the total nutrients, $C$. The total number of bacteria $n_B$ is $n_B^s + n_B^0$.

$$\dot{n_B^0} = (gC - F)n_B^0 - \alpha p_V n_B^0 n_V - \alpha(1 - p_V)\eta n_B^0 n_V + r n_B^s \tag{15}$$

$$\dot{C} = F(C_0 - C) - gC n_B \tag{16}$$

$$\dot{n_B} = (gC - F)n_B - \alpha(p_V^a n_B^s n_V + p_V n_B^0 n_V) \tag{17}$$

The probability of phage success against bacteria with spacers, averaged across the whole population, is given by $p_V^a = \frac{\sum_{k,\ell} k b_k \ell v_\ell p_V(k,\ell)}{n_B^s n_V}$.

$p_V^a$ is not known in general, but it can be simplified if certain assumptions are made about the population. In particular, we begin by assuming that immunity is all-or-nothing: if a bacterium has a matching spacer to a phage, the phage success probability is reduced to $p_V(1 - e)$ (where $e$ is the 'spacer effectiveness'), but if a bacterium has a spacer that is not exactly matching, that spacer confers no immunity and the phage probability of success is $p_V$, the same as against naive bacteria. This amounts to defining $p_V$ in terms of the spacer type $i$ and protospacer type $j$ as follows:

$$p_V(i,j) = \begin{cases} p_V(1 - e) & \text{if } i = j \\ p_V & \text{if } i \neq j \end{cases} \tag{18}$$

If each bacterium and phage can have only one spacer or protospacer, then the number of bacteria or phage with a particular spacer type $i$ is $n_B^i$ and $n_V^i$, respectively, and

$$p_V^a = p_V \left(1 - \frac{e}{n_B^s n_V} \sum_i n_V^i n_B^i \right)$$

*Equations 13*–17 can be solved analytically at steady state if we further assume $p_V^a = p_V(1 - e/m)$, where $m$ is the average number of bacterial clones at steady state. This assumption is described in detail in 'Measuring diversity'. The solution is exactly the mean-field analytic solution described in *Bonsma-Fisher et al., 2018* with the simple replacement of the parameter $e$ with $e/m$.

### Phage extinction threshold

This section relates to *Figure 1E* in 'Bacteria and phages dynamically coexist and coevolve'.

We reported in our previous work (*Bonsma-Fisher et al., 2018*) that *Equations 13–15*–17 experience a change in fixed point at a critical threshold of the phage infection success probability $p_V$: below $p_V^0 = \frac{1}{B} \left( \frac{gf}{(1-f)\alpha} + 1 \right)$, phages are driven extinct (*Figure 1E*). This is the same extinction threshold reported by *Payne et al., 2018* as the cutoff for achieving herd immunity in a well-mixed bacterial

population. To a first approximation, phages must successfully infect every $1/B$ bacteria they encounter, but if bacteria are growing quickly, then phage must do better to overcome bacterial growth. The correction gives $\frac{1}{B}\left(1 + \frac{\text{bacterial birth rate}}{\text{phage birth rate}}\right) = \frac{1}{B}\left(1 + \frac{\text{phage generation time}}{\text{bacterial generation time}}\right)$ as reported in *Payne et al., 2018*. To see this, we note that the bacterial birth rate in our model is $gC \approx gC_0 f$ when phage population sizes are small, and the phage birth rate in our model is $\alpha B p_V n_B \approx \alpha n_B$ since $B p_V \approx 1$. When phage population sizes are small, the total number of bacteria is $n_B \approx C_0(1 - f)$. Combining these expressions, we find $\frac{\text{bacterial birth rate}}{\text{phage birth rate}} \approx \frac{fgC_0}{\alpha(1-f)C_0} = \frac{fg}{\alpha(1-f)}$ as in our original expression.

## Steady state

Simulation results are calculated and presented at steady state unless otherwise specified. We define steady state to be the simulation run-time divided by 5, choosing simulation run-times so that mean-field quantities have equilibrated by the steady-state time. We run large population simulations longer than small population simulations: these simulations have less frequent interactions on average because of decreased $\alpha$ and take longer to equilibrate (*Figure 8*).

We set the total bacterial generations to be $10000(\log C_0 - 3)$, rounded to the nearest thousand, with a minimum of 10,000 generations (see *Table 2*). The following table lists simulation length and assumed steady-state time $t_{ss}$ for different values of $C_0$.

*Figure 9* shows the mean number of bacterial clones ($m$) at steady state as a function of the initial number of phage clones. For most parameters, the steady-state $m$ is independent of the initial m. There is a slight dependence on initial $m$ at high $C_0$.

## Single-clone mean-field dynamics

This section presents mean-field equations for the population size of single bacteria and phage clones. The steady-state bacteria and phage clone sizes (*Equations 25 and 26)* are, like the mean-field results in 'Total population size', used extensively in further analytic results. In particular, predicted mean clone sizes are used as a cutoff to measure the time to extinction for large clones (Appendix 3) and to calculate diversity in 'Analytic approximations for diversity' (main text 'Phages drive stable emergent sequence diversity'). These expressions are also used to calculate the fitness of new phage mutants in Appendix 3.

We can write mean-field equations for individual clones $n_B^i$ and $n_V^j$ as well:

$$\dot{n_V^j} = -(F + \alpha(n_B^0 + n_B^s))n_V^j + \alpha B P_0 p_V n_B^0 n_V^j + \alpha B P_0 \sum_i p_V(i,j) n_B^i n_V^j + \alpha B \sum_j \sum_{k \neq i} p_V(j,k) \mu^{\Delta_{ik}} n_B^j n_V^k \quad (19)$$

$$\dot{n_B^i} = (gC - F - r)n_B^i - \alpha \sum_j p_V(i,j) n_B^i n_V^j + \alpha \eta n_B^0 n_V^i (1 - p_V) \quad (20)$$

If $p_V(i,j)$ is binary and all spacers are equally effective (*Equation 18*), $\dot{n_V^j}$ and $\dot{n_B^i}$ can be simplified:

$$\dot{n_V^j} = -(F + \alpha n_B)n_V^j + \alpha B P_0 p_V n_V^j (n_B - e n_B^j) + \alpha B \sum_j \sum_{k \neq i} p_V(j,k) \mu^{\Delta_{ik}} n_B^j n_V^k \quad (21)$$

$$\dot{n_B^i} = (gC - F - r)n_B^i - \alpha p_V n_B^i (n_V - e n_V^i) + \alpha \eta n_B^0 n_V^i (1 - p_V) \quad (22)$$

In *Equations 19 and 21*, mutations that arise from phage $i$ decrease $n_V^i$, which is captured in that $B$ becomes $BP_0 = Be^{-\mu L}$ for type $i$ is effectively lower, since mutations go to different phage types.

The last term in *Equations 19 and 21* is for mutations that happen in all other phages besides type $i$ that convert type $k$ to type $i$. Any particular phage $k$ has a certain mutational distance from phage $i$, $\Delta_{ik}$, which means $\Delta_{ik}$-specific mutations must happen to get from type $k$ to type $i$. This happens with probability $\mu^{\Delta_{ik}}$. Going forward we assume that this term is small, which is true if the overall mutation rate $\alpha B p_V^a \mu n_V n_B$ is sufficiently low and/or the space of protospacer types is sufficiently large so that mutations are almost always to new types not present in the population.

We solve the coupled time-dependent system given by *Equations 21 and 22* numerically, assuming all other populations ($n_B$, $n_B^0$, $n_V$, and $C$) are at their deterministic steady-state value and ignoring the last term of *Equation 21*. These solutions are plotted alongside mean clone sizes from a simulation in *Figure 10* ('Numerical deterministic prediction'). The deterministic solution matches the mean clone size well at early times but does not capture the effects of clone extinction at later times. By

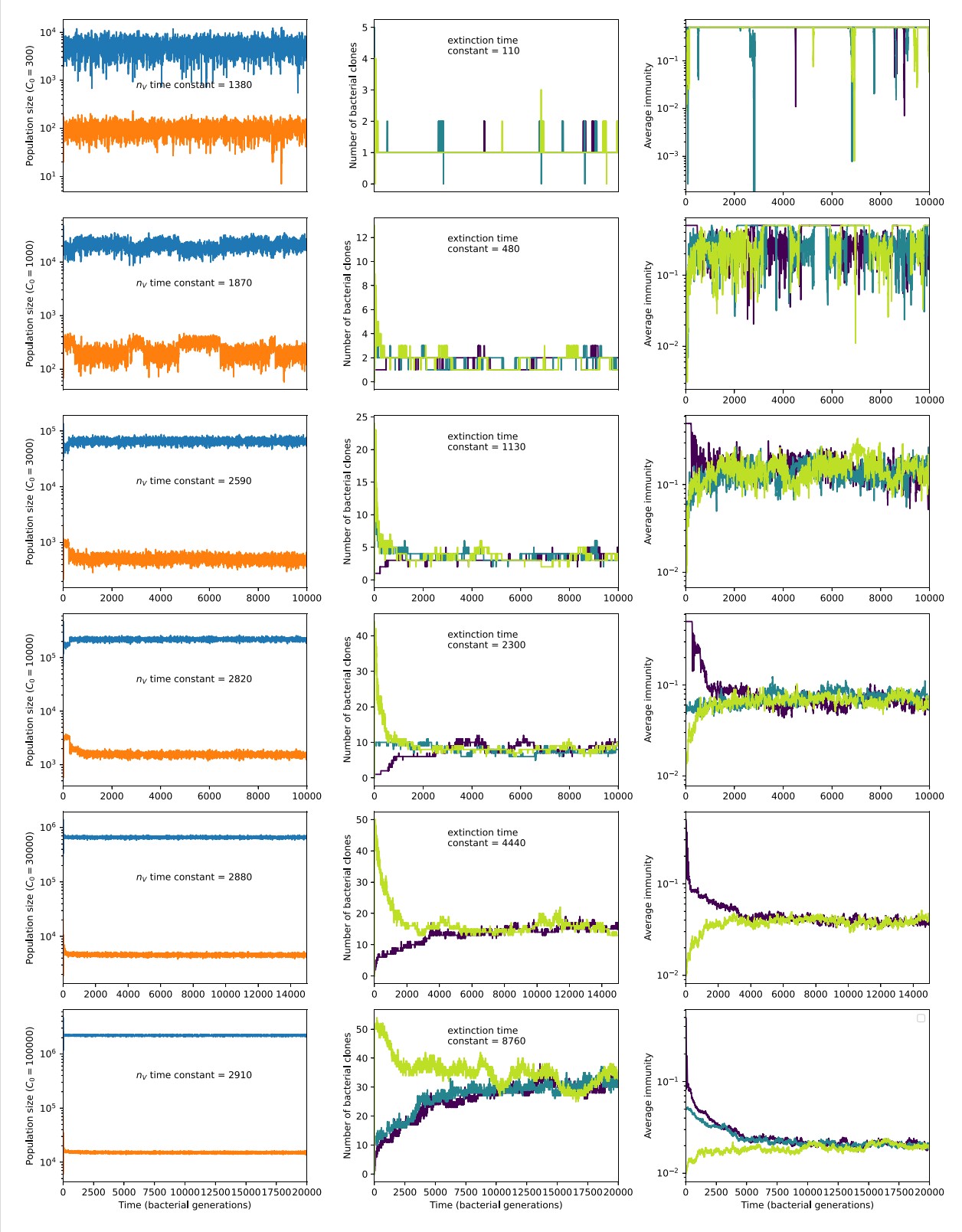

**Figure 8.** Total phage, total bacteria (left), and mean number of bacterial clones (right) vs. simulation time for five simulations with $\mu = 10^{-6}$, $e = 0.5$, $\eta = 0.001$, and $C_0$ ranging from 300 (top row) to 30,000 (bottom row). Total population sizes equilibrate very quickly, but the total number of clones can take longer at large population sizes (high $C_0$). The time constants inset are a measure of how quickly we expect each mean-field quantity to equilibrate: $n_V$ time constant is the inverse growth rate of the total phage population ($1/(-F - \alpha n_B(Bp_V - 1) - \alpha Bp_V n_B^s e/m)$) and the extinction time constant is the mean time to extinction for large phage clones (**Equation 171**), a measure of the rate of turnover of the number of clones.

**Table 2.** Simulation length.

| $C_0$ | Simulation length (bacterial generations) | Steady-state start time ($t_{ss}$) |
|---|---|---|
| 300 | 10,000 | 2000 |
| 1000 | 10,000 | 2000 |
| 3000 | 10,000 | 2000 |
| 10,000 | 10,000 | 2000 |
| 30,000 | 15,000 | 3000 |
| 100,000 | 20,000 | 4000 |
| 300,000 | 25,000 | 5000 |
| 1,000,000 | 30,000 | 6000 |

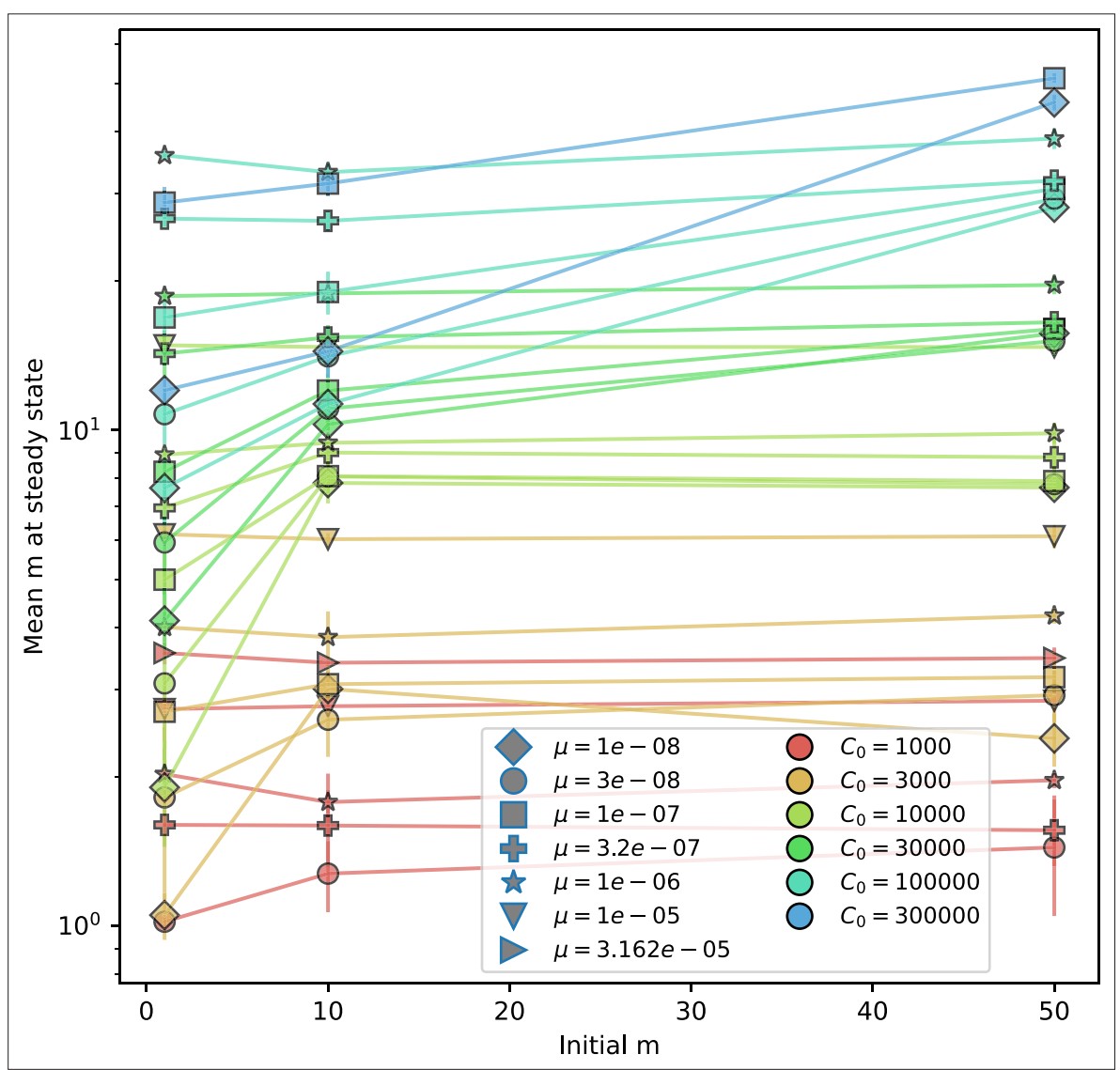

**Figure 9.** Mean number of bacterial clones after $t = t_{ss}$ bacterial generations vs. initial number of phage clones for $e = 0.8$, $\eta = 10^{-3}$. The mean is an average of 15 evenly spaced points from $t = t_{ss}$ to $t = 5t_{ss}$ bacterial generations. Error bars are the standard deviation across three or more independent simulations.

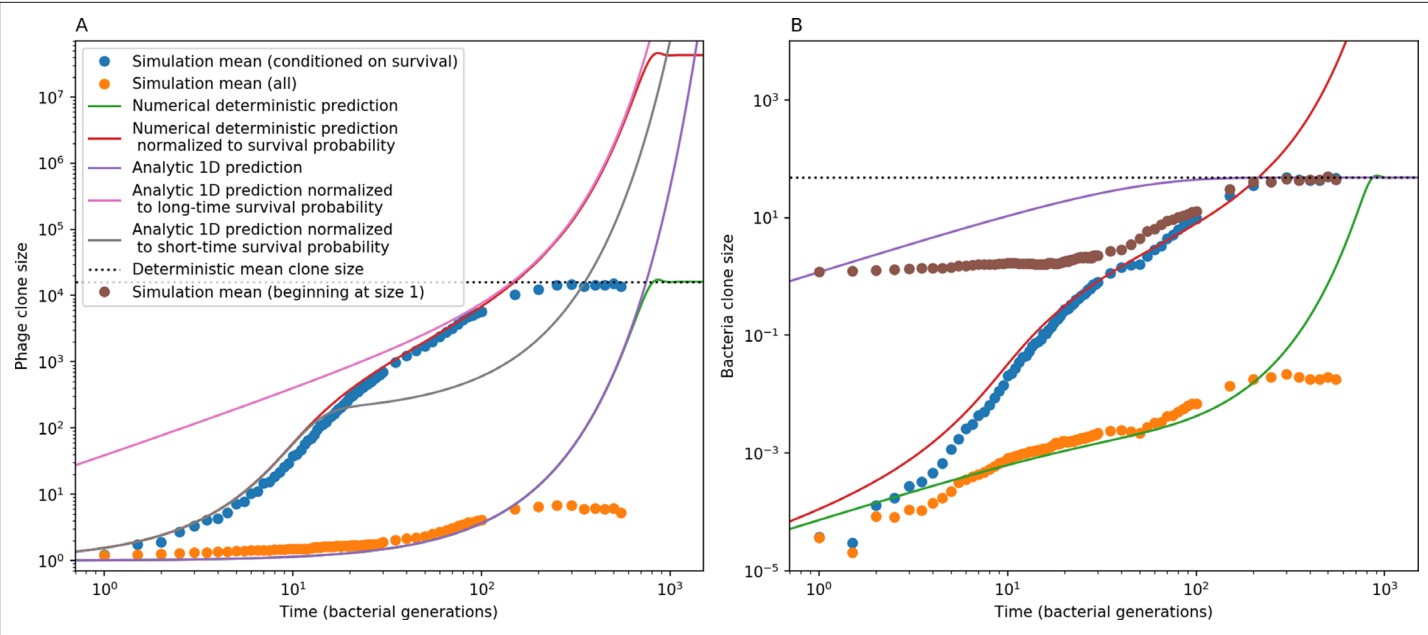

**Figure 10.** Phage (**A**) and bacteria (**B**) mean clone size in a simulation, either conditioned on survival (blue circles) or including extinct clones (orange circles). Theoretical predictions are plotted as solid lines: the time-dependent numerical solution to *Equations 21 and 22* in green, the same solution divided by the phage clone probability of survival in red, and a one-dimensional solution to *Equation 21* in (**A**) and 22 in (**B**). *Equations 25 and 26* are black dashed lines. An alternate simulation mean clone size is plotted for bacteria (brown circles) in which each clone trajectory is stacked based on the bacterial acquisition time and averaged across trajectories, conditioned on survival. Simulation parameters are $C_0 = 10000.0$, $\mu = 10^{-5}$, $\eta = 0.001$, and $e = 0.95$.

including only surviving clones in the simulation mean and normalizing the deterministic solution by the predicted fraction of surviving clones ('Numerical deterministic prediction normalized to survival probability'), the theoretical prediction matches well at early times and can be piecewise-combined with the steady-state mean clone size to give good agreement at all times (see 'Predicted clone size' dashed lines in *Figure 3—figure supplement 1*). We estimate the predicted fraction of surviving phage clones by numerically solving *Equation 153* which gives the phage clone probability of extinction in the absence of matching bacterial clones. The fraction of surviving clones is $1 - P_0(t)$ where $P_0(t)$ is the probability of extinction. Normalizing by $1 - P_0(t)$ does not give good agreement at long times because $P_0(t)$ goes to 1 as $t$ goes to infinity, causing the solution to diverge at long times. While individual trajectories do eventually go extinct, the mean clone size conditioned on survival reaches the deterministic steady state at long times.

We consider bacteria clones in the background of phage clones; that is, time 0 for a bacterial clone starts when the matching phage clone arises by mutation. This is the case in the numerical solution for the coupled bacteria-phage clone system given by *Equations 21 and 22*. For this reason, normalizing the bacterial clone size prediction by the phage clone probability of extinction also gives good agreement to the simulation mean conditioned on bacterial clone survival. (If a bacterial clone size is 0 but its corresponding phage clone has not yet gone extinct, that 0 will count in the mean clone size, but after the phage goes extinct, a 0 in the bacterial clone size will not be included.)

We can also solve *Equations 21 and 22* analytically by assuming $n_B^i$ and $n_V^i$ are constant, respectively; this amounts to removing the dependence of phage clone dynamics on bacteria clone dynamics and vice versa.

If we assume $n_B^i = 0$, we get the following solution for $n_V^i(t)$:

$$n_V^i(t) = n_V^i(0)\mathrm{e}^{s_0 t} \tag{23}$$

where $s_0 = \alpha B p_V n_B \mathrm{e}^{-\mu L} - F - \alpha n_B$ is the average growth rate of phage clones and $t$ is in minutes. This is the same growth rate as is in *Equation 156* with a small correction: $B \rightarrow B\mathrm{e}^{-\mu L}$; the burst of an individual phage clone is reduced by phages that mutate to a new type.

Likewise if we assume that $n_V^i$ is constant, we get the following solution for $n_B^i(t)$:

$$n_B^i(t) = n_B^i(0)e^{s_B t} + \frac{\delta}{s_B}\left(e^{s_B t} - 1\right) \tag{24}$$

where $s_B = gC - F - r - \alpha p_V(n_V - n_V^i)$ is the average growth rate for bacterial clone $i$ and $\delta = \alpha\eta n_B^0 n_V^i(1 - p_V)$ is the rate of spacer acquisition for that clone.

*Equations 23 and 24* are plotted in *Figure 10* ('Analytic 1D prediction'). *Equation 23* matches the phage clone size not conditioned on survival at early times, but continues to grow past the steady-state phage clone size without growing bacterial clones to reign it in. *Equation 24* is not comparable to the simulation mean with time zero as the time of phage clone birth; rather it should be compared to bacterial clone growth independent of what the phages are doing. Using the steady-state mean phage clone size in *Equation 24* (purple line), we obtain a rough correspondence with the simulation mean (brown circles). The prediction does not match at intermediate times, likely because this solution assumes that phage clones are at their mean size when in fact they are likely still smaller.

*Equations 21 and 22* can be solved at steady state in terms of the total mean-field variables (ignoring the last term of *Equation 21*). These solutions (*Equations 25 and 26*) are indicated by horizontal dashed lines in *Figure 10*.

$$n_B^{i\,*} = \frac{1}{e}\left(n_B - \frac{F + \alpha n_B}{\alpha B P_0 p_V}\right) \tag{25}$$

$$n_V^{i\,*} = \frac{n_B^{i\,*}(\alpha p_V n_V - (gC - F - r))}{\alpha\eta n_B^0(1 - p_V) + \alpha p_V e n_B^{i\,*}} \tag{26}$$

## Simulation methods

This section describes how simulations were performed, shows basic simulation results, describes parameter choices ('Parameter values'), and explores the parameter dependence of stochastic population extinction ('Stochastic population extinction') that relates to results in 'Bacteria and phages dynamically coexist and coevolve'.

Simulations were written in Python and performed on the Béluga and Niagara supercomputers at the SciNet HPC Consortium (*Ponce et al., 2019*) and on a local server in our group. Our simulation code can be found at https://github.com/mbonsma/CRISPR-dynamics-model (copy archived at *Bonsma-Fisher and Goyal, 2023*). We used the tau-leaping method, a fast approximation for Gillespie simulation *Cao et al., 2006* and compared with full Gillespie simulations for some cases. Gillespie and tau-leaping methods showed similar dynamics and good agreement for total population sizes (*Figure 11*), total spacer diversity (*Figure 12*), mean phage and bacteria clone sizes (*Figure 13*), and produced the same qualitative behaviour for individual spacer types (*Figure 14*).

We initialized five simulations for each parameter combination for a total of approximately 19,800 simulations, not including simulations with cross-reactivity. A small subset of simulation parameters were run six times instead of five. Not all simulations were successfully completed either due to errors while running or because their running time was exceedingly long. Of the initialized simulations, approximately 10,000 were completed and included in analysis. Simulations with very low $\mu$ tended to either have no phage establishments (at low $C_0$) or very long running times (at high $C_0$), and simulations with very high $\mu$ tended to have higher diversity and longer running times in general. Unless otherwise noted, plots with simulation averages are an average across three to six simulations with the same parameters. We set simulations to run for a fixed number of bacterial generations intended to be long enough to allow the system to reach steady state and remain there for a long time to generate statistics ('Steady state').

We model spacers and protospacers as a binary sequence of length $L$ as in *Han et al., 2013*; *Han and Deem, 2017*. When a phage reproduces and creates a burst of $B$ phages, we draw $BL$ numbers from a binomial distribution with probability $\mu$. Successes in this draw designate bits that will mutate (flip) in the newly created phages. It is possible but very rare for more than one mutation to happen in the same new phage: this occurs with probability $1 - e^{-\mu L} - \mu L e^{-\mu L} \approx (\mu L)^2$ whereas a single mutation occurs with probability $\mu L e^{-\mu L} \approx \mu L$. Multiple mutations occur approximately $\mu L$ times as often as single mutations; for the largest value of $\mu$ we use, they represent 0.3% of events.

Our simulation code can be found on GitHub.

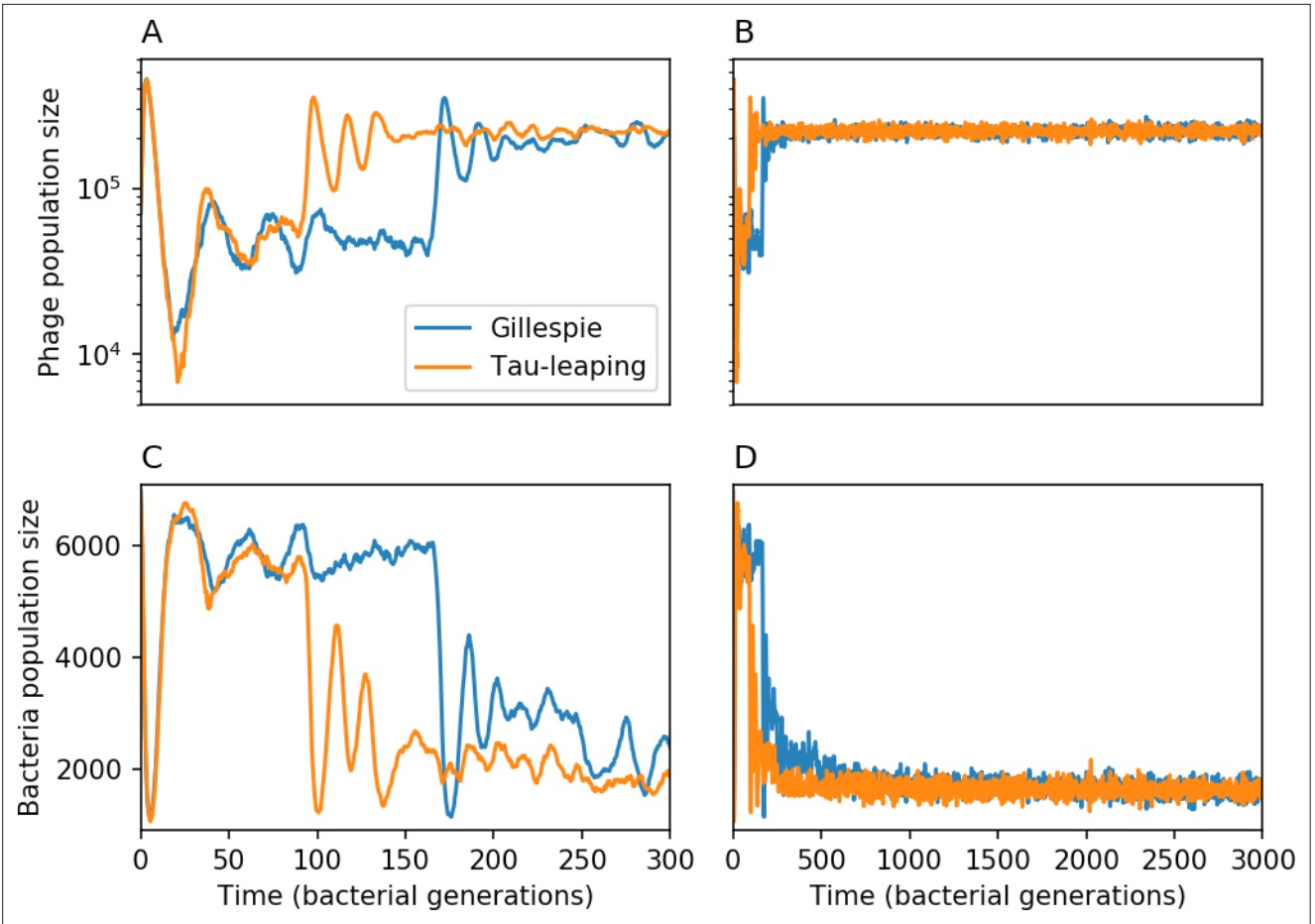

**Figure 11.** Total phage ($n_V$) and total bacteria ($n_B$) as a function of time for a Gillespie simulation and a tau-leaping simulation with the same parameters. Total phage is shown at early times (**A**) and late times (**B**), and total bacteria at early times (**C**) and late times (**D**). The simulation parameters are $C_0 = 10^4$, $\mu = 10^{-6}$, $\eta = 0.001$, and $e = 0.95$. Early time dynamics differ slightly in this example, but the long-time behaviour and steady-state values are similar.

### Parameter values

Parameter values are as above unless otherwise indicated. Representative values estimated for *S. thermophilus* bacteria in lab conditions.

Parameter descriptions and values are listed in *Table 3*. The burst size for phage that target *S. thermophilus* has been measured at between 140 and 200 (*Lucchini, 1999*) and 80 for phage 2972 (*Levin et al., 2013*). We use a burst size of 170. (*Vaningelgem et al., 2004*) measured the maximum growth rate of *S. thermophilus* in milk at 42°C to be $2.4 \times 10^{-2}$ min$^{-1}$. This corresponds to $gC_0$ in our model; we choose $g$ based on $C_0$ so that $gC_0 = 2.4 \times 10^{-2}$ min$^{-1}$.

The parameter $\alpha$ in our model is the phage adsorption rate constant divided by the culture volume. The rate of adsorption for phage is of the order of $10^{-8}$ min$^{-1}$ ml *Delbrück, 1940*; this is similar to the values used in *Levin et al., 2013* and *Weissman et al., 2018*. In order to explore regimes of different total population sizes, we decrease $\alpha$ as we increase $C_0$ in order to maintain stable coexistence between bacteria and phage; this is equivalent to decreasing the culture volume as $C_0$ decreases. For example, if $C_0 = 10^5$, we set $\alpha = 2 \times 10^{-2}/C_0 = 2 \times 10^{-7}$, implying a culture volume of approximately $50\mu$l.

*Levin et al., 2013* estimated the frequency of phage mutants that escape CRISPR targeting by *S. thermophilus* to be between $5 \times 10^{-7}$ to $5 \times 10^{-5}$. These measurements are the fractions of phages from lysate that can evade CRISPR targeting of different unique spacers. This is analogous to $\mu L$ in our model, which is the probability of a mutation occurring in a newly burst phage. We use values of

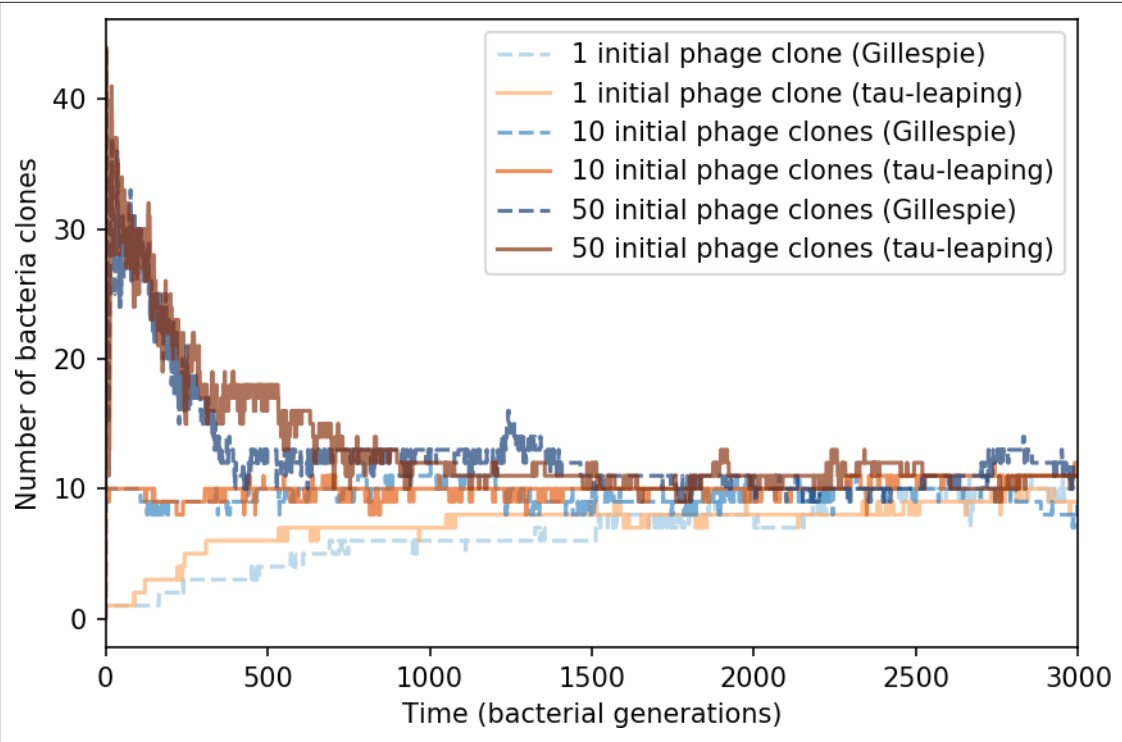

**Figure 12.** Number of bacterial clones ($m$) vs. simulation time for three sets of simulations, each beginning with 1, 10, or 50 phage clones. Gillespie simulations are dashed blue lines, and tau-leaping simulations are solid orange and red lines. The simulation parameters are $C_0 = 10^4$, $\mu = 10^{-6}$, $\eta = 0.001$, and $e = 0.95$.

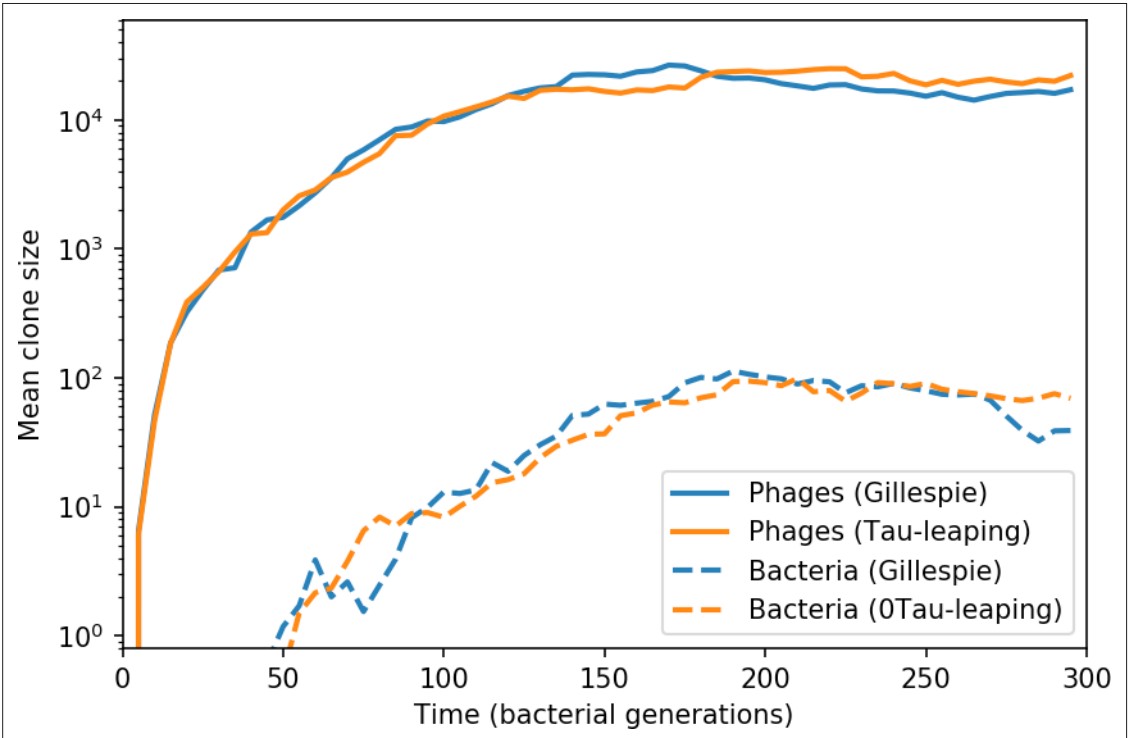

**Figure 13.** Average population size of phage clones (solid lines) and bacterial clones (dashed lines) in a Gillespie and tau-leaping simulation with the same parameters. The simulation parameters are $C_0 = 10^4$, $\mu = 10^{-6}$, $\eta = 0.001$, and $e = 0.95$.

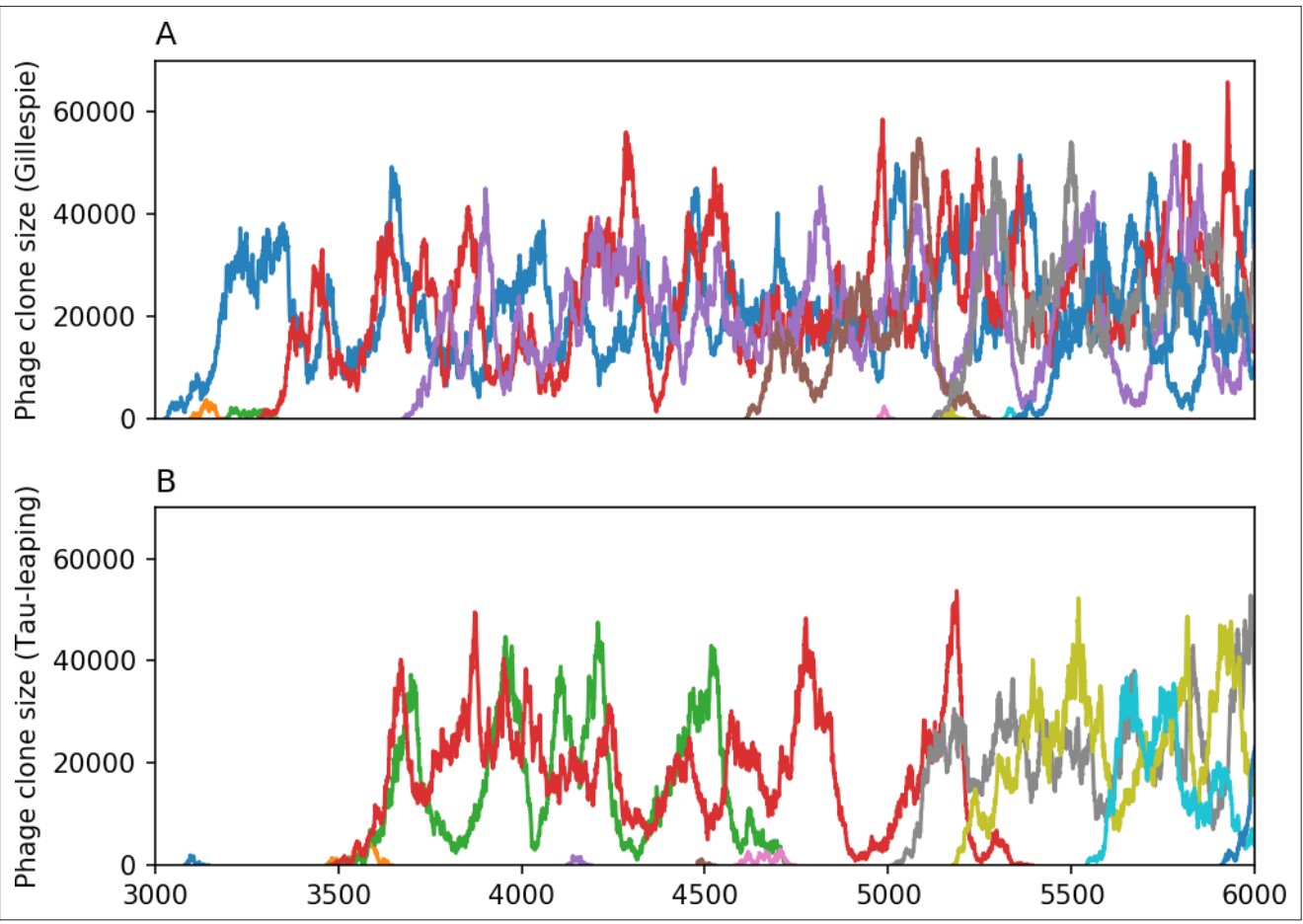

**Figure 14.** 10 spacer trajectories for a Gillespie simulation (**A**) and tau-leaping simulation (**B**). The first 10 trajectories that surpass $n_V^i = 1000$ are shown. The simulation parameters are $C_0 = 10^4$, $\mu = 10^{-6}$, $\eta = 0.001$, and $e = 0.95$.

**Table 3.** Model parameters.

| Parameter | Description | Value |
|---|---|---|
| $\frac{1}{gC_0}$ | Bacterial doubling time | 41.7 min |
| $C_0$ | Inflow nutrient concentration in | $10^2$ to $10^{-6}$ |
| | Units of bacterial cell density | |
| $\alpha$ | Phage adsorption rate | $2 \times 10^{-2}/C_0$ |
| $B$ | Phage burst size | 170 |
| $F$ | Chemostat flow rate | $0.3gC_0$ |
| $p_V$ | Probability of phage success | 0.02 |
| | for bacteria without spacers | |
| $e$ | Spacer effectiveness | 0.1 to 0.95 |
| $r$ | Rate of spacer loss | $0.04gC_0$ |
| $\eta$ | Probability of spacer acquisition | $10^{-5}$ to $10^{-2}$ |
| $\mu$ | Phage mutation rate per base per generation | $10^{-8}$ to $10^{-4}$ |
| $L$ | Phage protospacer length in nucleotides | 30 |

Parameter values are as above unless otherwise indicated. Representative values estimated for *Streptococcus thermophilus* bacteria in lab conditions.

$\mu$ between $10^{-8}$ and $10^{-4}$ corresponding to $\mu L$ between $3 \times 10^{-7}$ and $3 \times 10^{-3}$ to encompass the physiological range of phage mutation rates as well as unusually high mutation rates.

We generally fix the probability of successful phage infection against naive bacteria at $p_V = 0.02$ (though $p_V$ is varied in *Figure 1E* and in *Bonsma-Fisher et al., 2018*). This is consistent with the value of $10^{-2}$ used in *Doekes et al., 2021* and *Berngruber et al., 2013*. Several other models of CRISPR immunity have assumed phage success probability to be much higher, typically close to 1 (*Childs et al., 2014*; *Childs et al., 2012*; *Weissman et al., 2018*). Increasing $p_V$ does not qualitatively alter our results beyond changing the relative population sizes of phage and bacteria; at high values of $p_V$ bacteria population sizes are small and they become more likely to experience stochastic extinction (see *Figure 1E*). A low value of $p_V$ can be taken to reflect the presence or effectiveness of other anti-phage defence systems.

The rate of spacer loss in our model, $r$, can be thought of as a phenomenological parameter since the true rate of spacer loss is not well understood (*Weissman et al., 2018*). For comparison, Jiang et al. estimate the rate of loss of function of the entire CRISPR system in *S. epidermis* at $10^{-4}$ to $10^{-3}$ per individual per generation (*Jiang et al., 2013*). We rescale the parameter $r$ as a function of bacterial growth rate, which means that the rescaled parameter $R = r/(gC_0) = 0.04$ is constant per bacterial generation. Our loss rate is an order of magnitude higher than the rate in *Jiang et al., 2013*.

We vary the parameters $\eta$ and $e$ to explore different 'strengths' of CRISPR immunity. The parameter $e$ is spacer effectiveness: when $e = 0$, spacers provide no immunity and bacteria with spacers are functionally no different from bacteria without spacers. When $e = 1$, a spacer-containing bacterium that encounters a phage with a matching protospacer is guaranteed to survive. We vary $e$ between 0.1 and 0.5 to explore different regimes of CRISPR effectiveness. The parameter $\eta$ is the probability that a naive bacterium will acquire a spacer if it is infected but not killed by a phage. Rates of naive spacer acquisition vary widely, and we vary $\eta$ between $10^{-5}$ and $10^{-2}$, with the value of $\eta$ constant within a simulation. For comparison with measured acquisition rates, Pyenson et al. measured naive acquisition in *Staphylococcus aureus* to be approximately $10^{-6}$ to $10^{-7}$ (*Pyenson and Marraffini, 2020*). Acquisition rates may hundreds of times higher in primed acquisition (*Staals et al., 2016*), and Heler et al. measured four orders of magnitude difference in spacer abundances shortly after infection, likely a result of differences in acquisition rate (*Heler et al., 2019*).

The flow rate $F$ was picked in order to get a stable fixed point where phage and bacteria coexist. Stability conditions approximately correspond to those derived in our previous work (*Bonsma-Fisher et al., 2018*).

## Stochastic population extinction

This section relates to results in 'Bacteria and phages dynamically coexist and coevolve'.

Even when bacteria and phage are within the deterministic coexistence regime, populations may experience stochastic extinction (main text *Figure 1F*). Simulations are run for a fixed number of generations (see 'Steady state') or until bacteria or phage go extinct. Population extinction on the timescale of our simulation run-time is restricted to low values of $C_0$ (no phage population extinctions happen for $C_0 > 3000$, and no bacteria extinctions happen for $C_0 > 300$). Phages are prone to population extinction at higher population sizes than bacteria because of their large burst size which makes their dynamics more noisy (see 'Long-time approximation for $P_0(t)$').

Extinction probability is strongly related to total population size, with smaller populations being more likely to go extinct across all parameters (*Figure 1—figure supplement 1* and *Figure 1—figure supplement 2*). However, there are differences that depend on parameters and initial conditions. First, high spacer effectiveness increases the likelihood of phage extinction and decreases the likelihood of bacterial extinction (*Figure 1—figure supplement 1*). High spacer effectiveness correspondingly increases the time to extinction for bacteria (*Figure 1—figure supplement 4*) and decreases the time to extinction for phages (*Figure 1—figure supplement 3*). Bacteria also appear to be less likely to go extinct at high values of $\eta$ (*Figure 1—figure supplement 4*). Phages have longer times to extinction and lower extinction probability at high values of $\mu$ (*Figure 1—figure supplement 3*). And finally, phages have a longer time to extinction if the initial number of phage clones ($m_{init}$) is high (*Figure 1—figure supplement 6*), but this effect disappears at low $\eta$ (*Figure 1—figure supplement 5*), perhaps because bacteria do not acquire spacers quickly enough for the initial phage diversity to make a difference before the number of clones equilibrates, which happens rapidly at low total population size (see upper rows of *Figure 8*).

## Simulation results

### Measuring diversity

This section describes in detail our method for measuring diversity in simulations, relating to results in 'Phages drive stable emergent sequence diversity'. It provides justification for matching the number of large phage clones to the number of bacteria clones and discusses the assumptions made and where they may break down.

A measure for the overall sequence diversity in the population is the total number of unique clones. In general, the bacterial diversity and phage diversity need not be the same, but it turns out that the number of bacterial clones (which we call $m$) closely tracks the number of 'large' phage clones across a wide range of parameters.

To measure the number of large phage clones in a simulation, we scale the observed phage clone size distribution by the probability of extinction for each clone size below a size cutoff given by *Equation 115*. We multiply each observed number of clones of size $n$ by the probability of extinction (*Equation 159*) to the power of the clone size $n$ (*Equation 27*): this gives a scaling factor for each clone size that predicts how many clones of that size will survive at long times (*Equation 28*). $P_n^{\text{large}}$ is plotted alongside the full normalized histogram of clone sizes for one simulation in *Figure 15*.

$$1 - P_0(n) = 1 - \left(1 - \frac{2s_0}{B(s_0 + \delta_0)}\right)^n \tag{27}$$

$$P_n^{\text{large}} = \begin{cases} P_n(1 - P_0(n)) & \text{if } n < N_{\text{est}} \\ P_n & \text{if } n \geq N_{\text{est}} \end{cases} \tag{28}$$

We then calculate the number of large phage clones at steady state as $N_{\text{total}} \times \sum_n P_n^{\text{large}}$, the total number of phage clones multiplied by the fraction of clones that survive for long times. Note that $P_n^{\text{large}}$ is not normalized; it is a quantity $\leq 1$ representing the proportion of the clone size distribution that corresponds to large clones. The predicted number of large phage clones is plotted against

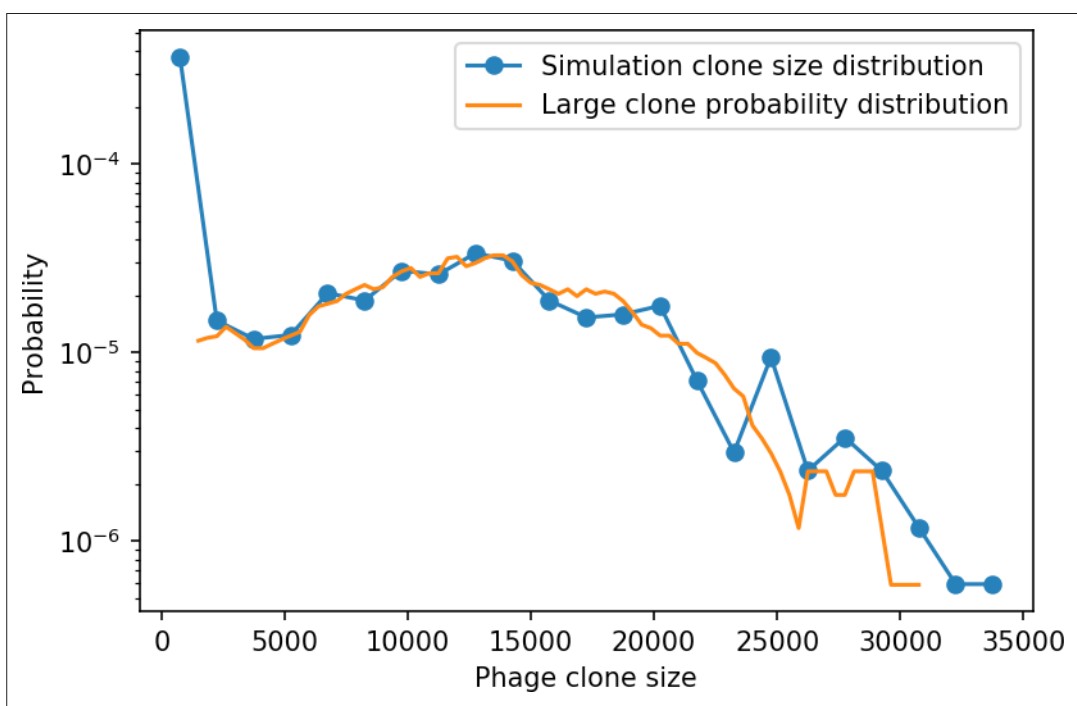

**Figure 15.** Phage clone size distribution from 15 combined time points for a simulation with the parameters $C_0 = 10^4$, $e = 0.95$, $\eta = 0.01$, and $\mu = 3 \times 10^{-6}$. The blue points are the values of the full normalized phage clone size histogram with a bin width of 1500. The orange line is given by $P_n^{\text{large}}$ in *Equation 28* smoothed with a running average of window size 3000. Both distributions are scaled by the total number of phage clones.

the observed mean number of bacterial clones in Figure 22. There is extremely good agreement at all parameters except the lowest value of $\eta$, $\eta = 10^{-5}$, where the estimated number of large phage clones tends to be larger than $m$. In this regime, phage clones go extinct because of clonal interference before bacteria are able to acquire spacers (Figure 16 and Figure 17). At low $\eta$ and high $\mu$, phage clone fitness declines more rapidly with less influence from bacterial spacer acquisition than at high $\eta$ and moderate $\mu$: Figure 18 shows the ratio of phage clone initial fitness to phage clone fitness at the mean bacteria spacer acquisition time, measured from simulations, vs. $\eta/\mu$. At most parameters, however, phage clone fitness has not changed much by the time bacteria are acquiring spacers.

There is an $\eta$-dependent 'floor' on the minimum size a phage clone must be before bacteria begin to acquire spacers (Figure 16). We can estimate the size of a phage clone at the time of first spacer acquisition by calculating the mean time of acquisition from an exponentially growing population. Let $n(t)$ be the size of a phage clone relative to the time of mutation. We assume phage clones grow exponentially with rate $s_0$ at early times, where $s_0 = \alpha n_B(Bp_V - 1) - F$ is the average initial growth rate of phage clones: $n(t) = e^{s_0 t}$.

The probability that no acquisitions have happened by time $t$ is given by $P_0$ in a Poisson process. Let $a = \alpha \eta (1 - p_V) n_B^0$ be the rate of spacer acquisition (see Equation 15), then the probability of no acquisitions by time $t$ is $P_0(t) = e^{-a \int_0^t n(t')dt'} = e^{-\frac{a}{s_0}(e^{s_0 t} - 1)}$. Now, the probability that an acquisition happens between time $t$ and $t + dt$ is then

$$P(t) = P_0(t)an(t) = ae^{s_0 t}e^{-\frac{a}{s_0}(e^{s_0 t} - 1)} \tag{29}$$

Equation 29 is shown as a function of $t$ for four simulations with different values of $\eta$ in Figure 19. The probability has a sharp peak: there is a particular time related to phage clone growth at which the first acquisition is most likely — intuitively, this is what leads to the appearance of a sharp floor in phage clone size at first acquisition. Interestingly, the mean time of first acquisition is non-monotonic in $\eta$: the mean time is smallest for the lowest and highest values of $\eta$.

We calculate the mean time from Equation 29:

$$\langle t \rangle = \int_0^\infty ae^{s_0 t}e^{-\frac{a}{s_0}(e^{s_0 t} - 1)}t\,dt \tag{30}$$

$$\langle t \rangle = \frac{1}{s_0}e^{\frac{a}{s_0}}\Gamma(0, \frac{a}{s_0}) \tag{31}$$

where $\Gamma(0, \frac{a}{s_0}) = \int_{\frac{a}{s_0}}^\infty \frac{e^{-t}}{t}dt$ is the incomplete gamma function. We can plug this time into $n(t) = e^{s_0 t}$ to estimate the mean phage clone size at first acquisition; the measured clone size is plotted as a function of this prediction in Figures 20 and 21. The clone size depends on the phage growth rate and bacterial acquisition rate, which are themselves primarily dependent on the total bacteria population size and the acquisition probability parameter $\eta$. If $a << s_0$ (true for our parameters), then $e^{\frac{a}{s_0}} \approx 1$ and $\Gamma(0, \frac{a}{s_0}) > 1$, meaning that the time of first acquisition is larger than $1/s_0$. ($\Gamma(0, \frac{a}{s_0}) > 1$ for $\frac{a}{s_0} \lesssim 0.25$.) Since $1/s_0$ is the approximate time at which phages are safe from stochastic extinction due to drift (see 'Clone fitness'), this means that phage clones are out of the stochastic extinction regime by the time bacteria begin to acquire spacers.

The definition of large phage clones we just described depends on the measured simulation distribution of phage clone sizes. We can instead approximate the number of large phage clones as the product of the rate at which phage clones become established and the rate at which large phage clones go extinct. This statement is summarized in Equation 32. Numerically solving this equation for $m$ gives a prediction for the total number of bacterial clones and total number of large phage clones.

$$m = \underbrace{\left[\frac{2s_0}{B(s_0 + \delta_0)}\right]}_{\text{phage establishment fraction}} \underbrace{\alpha B(1 - e^{-\mu L})p_V n_V n_B(1 - \frac{e\nu}{m})\frac{1}{gC_0}}_{\text{phage mutation rate}} \underbrace{\frac{2n_V^i(1 - \ln\frac{n_V^i}{n_V})gC_0}{(B-1)^2\beta + \delta}}_{\text{large phage clone time to extinction}} \tag{32}$$

$s_0 = \beta_0(B - 1) - \delta_0 = \alpha n_B(Bp_V - 1) - F$

$n_V^i = $ deterministic mean phage clone size at steady state

$\beta_0 = n_B\alpha p_V$

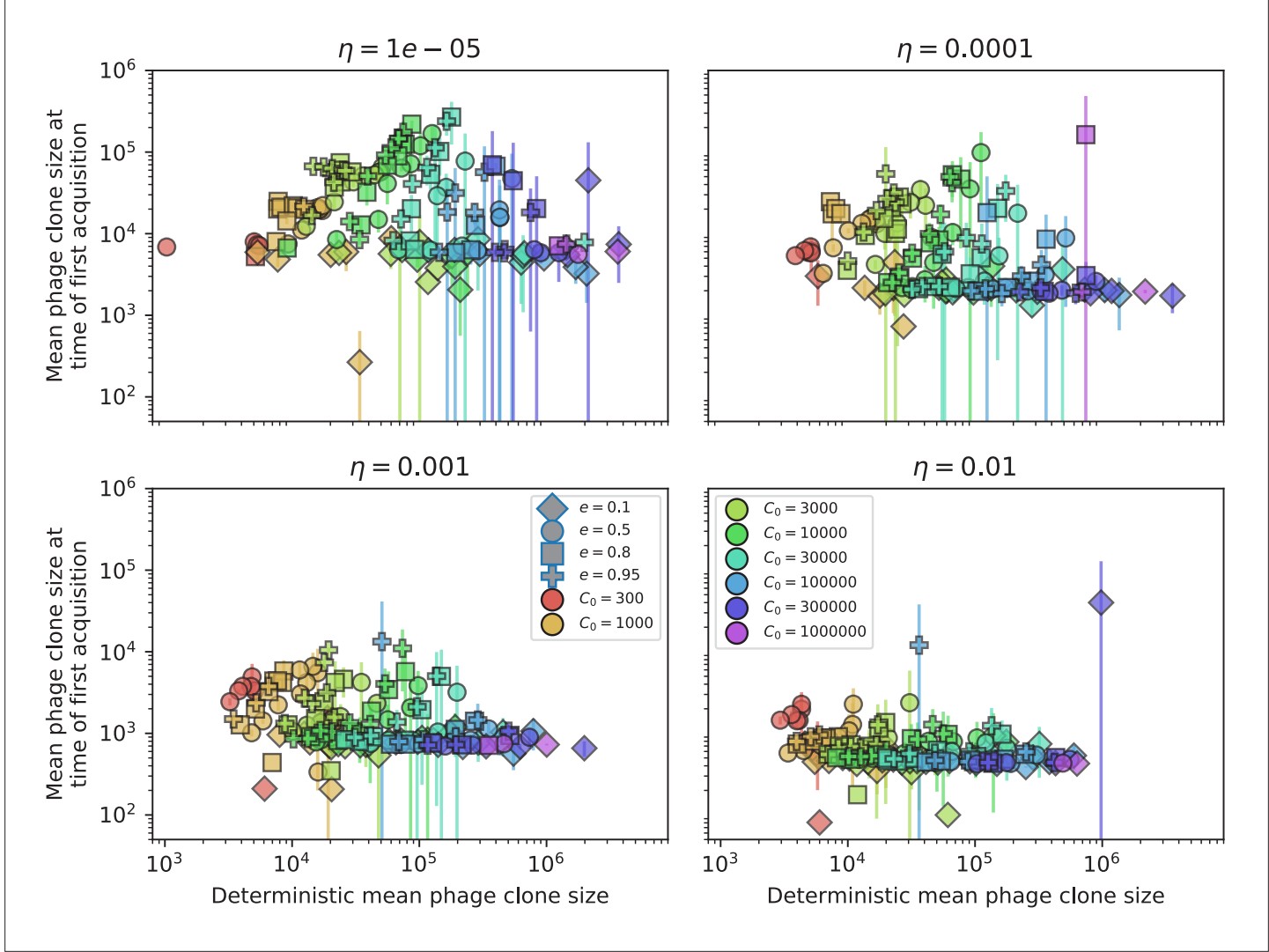

**Figure 16.** Mean phage clone size at the time of first spacer acquisition vs. the deterministic mean phage clone size. For each phage clone trajectory, the clone size at the time of first spacer acquisition is recorded and these are averaged across each simulation. Error bars are the standard deviation across three or more independent simulations.

$$\delta_0 = F + \alpha n_B(1 - p_V)$$

$$\beta = n_B \alpha p_V - \alpha p_V e n_B^i$$

$$\delta = F + \alpha n_B(1 - p_V) + \alpha p_V e n_B^i$$

We now describe each of the three terms in **Equation 32**. The phage establishment rate is the phage mutation rate multiplied by the fraction of new phage mutants which become established. Derivation of the phage establishment fraction and phage time to extinction can be found in 'Long-time approximation for $P_0(t)$' (**Equation 158**). Derivation of the mean time to extinction for large phage clones can be found in 'Neutral time to extinction from backward master equation' (**Equation 171**).

The phage mutation rate is the mean-field phage reproduction rate $\alpha B n_V(p_V^a n_B^s + p_V n_B^0)$ multiplied by the probability of one or more mutations per burst $(1 - e^{-\mu L})$.

We assume that $p_V^a = p_V(1 - e/m)$, which is true if all clones are equal in size (i.e., $n_V^i = n_V/m$ and $n_B^i = n_B^s/m$) or if the deviations from equal size are uncorrelated between matching bacteria and phage clones. This assumption means that the average immunity in the population is approximately $e/m$ which is accurate across a wide range of parameters: **Figure 22** and **Figure 23** compare the full

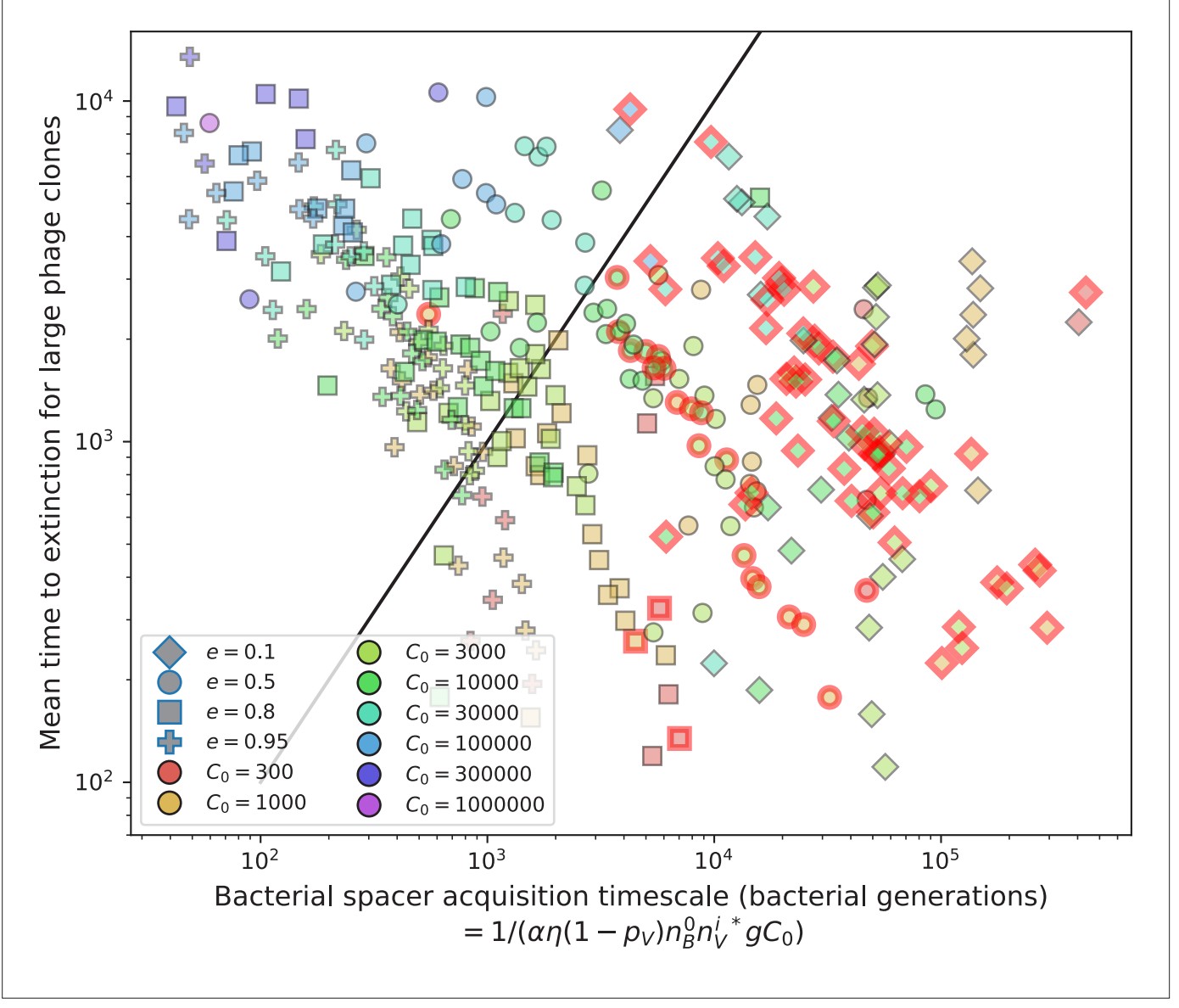

**Figure 17.** Mean time to extinction for phage clones vs. the timescale of bacteria spacer acquisition given by $1/D$ where $D = \alpha\eta(1 - p_V)n_V^{i\,*}n_B^0 gC_0$. Points outlined in red are simulations where the ratio of large phage clones to bacterial clones exceeds 1.2. Phage clones experience clonal interference at low $\eta$: they go extinct faster than bacteria acquire spacers.

average immunity (*Equation 33*) with $e/m$, where $m$ is the number of bacterial clones present at a given time point in a simulation. The assumption breaks down at low $\eta$ and high $\mu$ (points below the line in the upper right). Intuitively this happens when the sizes of matching clones become anti-correlated because bacteria acquire few spacers while phages acquire many mutations. The resulting matching pairs have more mismatched clone sizes than the mean number of bacterial clones would suggest. (In this particular case, phage clones are smaller than $n_V/m$.) Conversely, at large population sizes and large spacer acquisition rates, matching clone sizes become correlated, leading to average immunity $> e/m$. *Figure 24* shows clone size distributions for four simulations with increasing $\eta$, showing that as $\eta$ increases the phage clone distribution becomes more narrow and the matching clone pairs become more correlated.

$$\text{Average immunity} = 1 - \frac{\sum_{i,j} n_B^i n_V^j p_V(i,j)}{p_V \sum_{i,j} n_B^i n_V^j} = \frac{e}{n_B^s n_V} \sum_i n_V^i n_B^i \qquad (33)$$

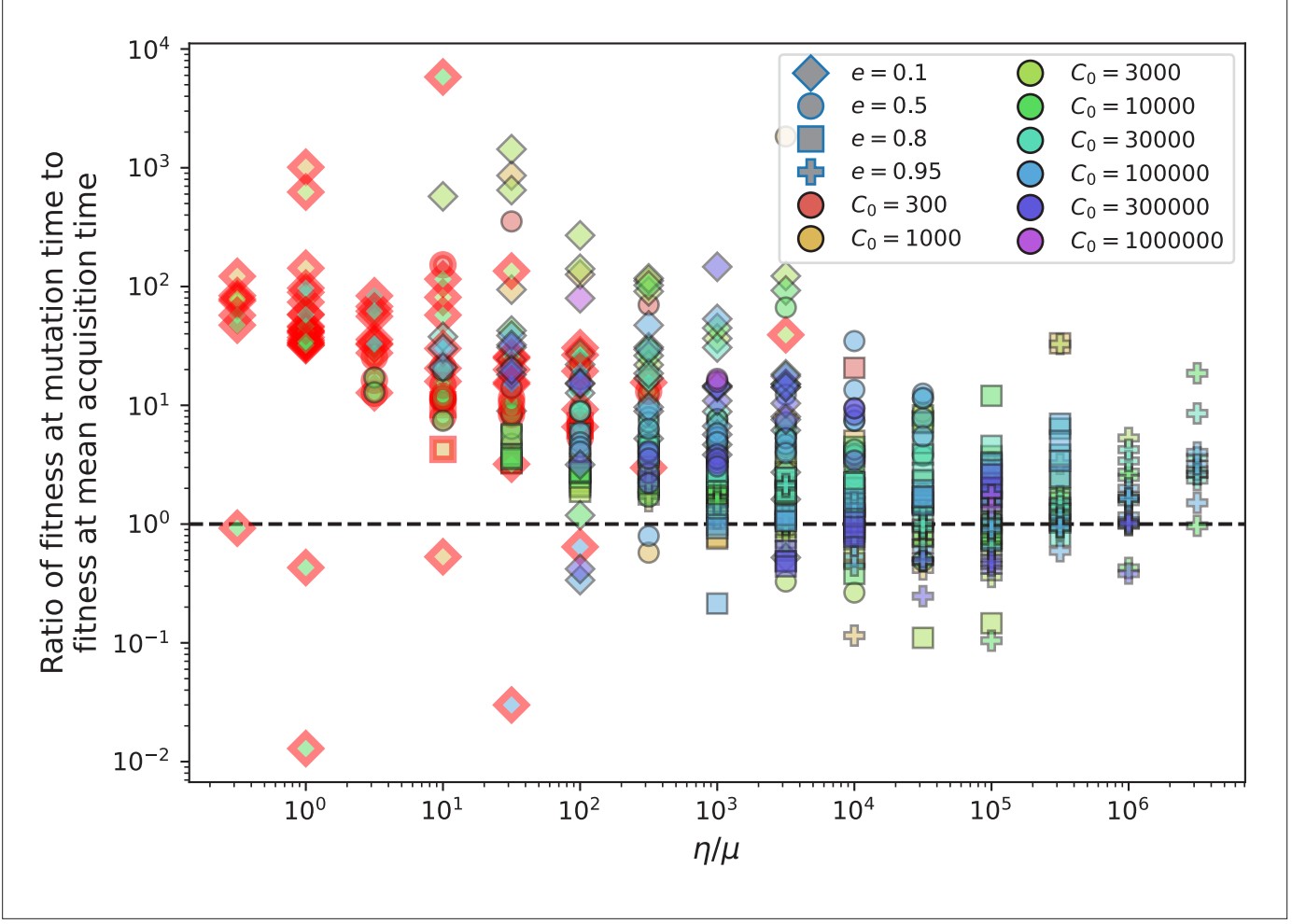

**Figure 18.** Ratio of average phage clone initial fitness to the average phage clone fitness at the mean bacterial spacer acquisition time vs $\eta/\mu$. Points outlined in red are simulations where the ratio of large phage clones to bacterial clones exceeds 1.2. The phage fitness is the average per capita growth rate of phage clones conditioned on survival. Phage clones experience clonal interference at low $\eta$ and high $\mu$.

For several parameter combinations, there is no $m$ that satisfies *Equation 32*. This generally occurs for parameters resulting in small total population sizes. In these cases, no solution is found because the predicted mutation rate and/or the predicted establishment fraction and/or the predicted mean time to extinction for large clones are too low (*Figure 25*).

### Analytic approximations for diversity

This section describes how we approximate our measurement of diversity to arrive at *Equation 87*, an analytic approximation for the number of bacterial clones at steady state. The first sections describe approximations for each of the components of diversity; the combination is summarized in 'Approximation for $m$'. This is the mathematical background for results in 'Phages drive stable emergent sequence diversity'.

We want to understand more intuitively how diversity depends on system parameters. Here we describe analytic approximations for diversity $m$ and its components. *Equation 32* contains $m$ both explicitly and implicitly, so we approximate each of the three components to arrive at an analytic approximation for $m$.

We can make some initial simplifications by cancelling terms (note that $s_0 + \delta_0 = \beta_0(B-1)$ and $\beta_0 = n_B \alpha p_V$):

$$m = \left(\frac{2s_0}{B-1}\right)(1 - e^{-\mu L})n_V(1 - \frac{e\nu}{m})\frac{2n_V^i(1 - \ln\frac{n_V^i}{n_V})}{(B-1)^2\beta + \delta} \tag{34}$$

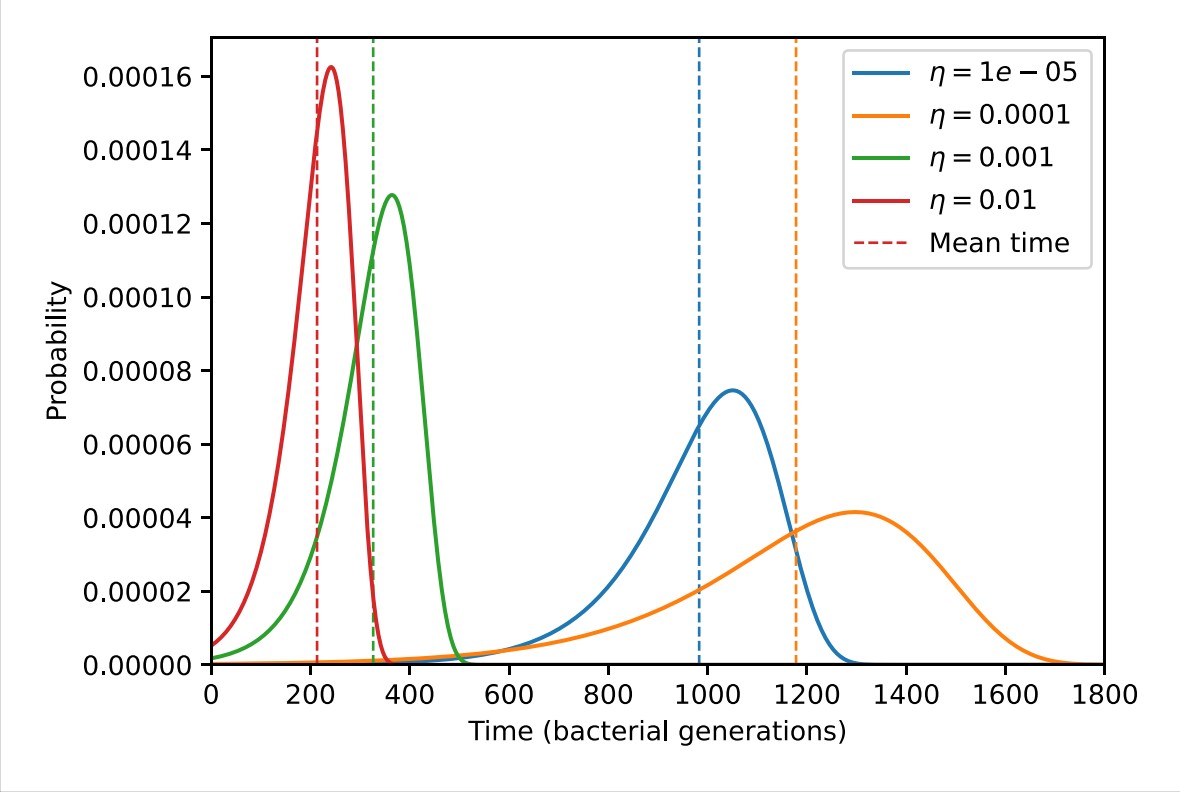

**Figure 19.** Probability of the first spacer acquisition happening at time $t$ for four simulations with different values of $\eta$ and $C_0 = 10^4$, $\mu = 10^{-5}$, and $e = 0.95$. The mean of each distribution is shown as a vertical dashed line.

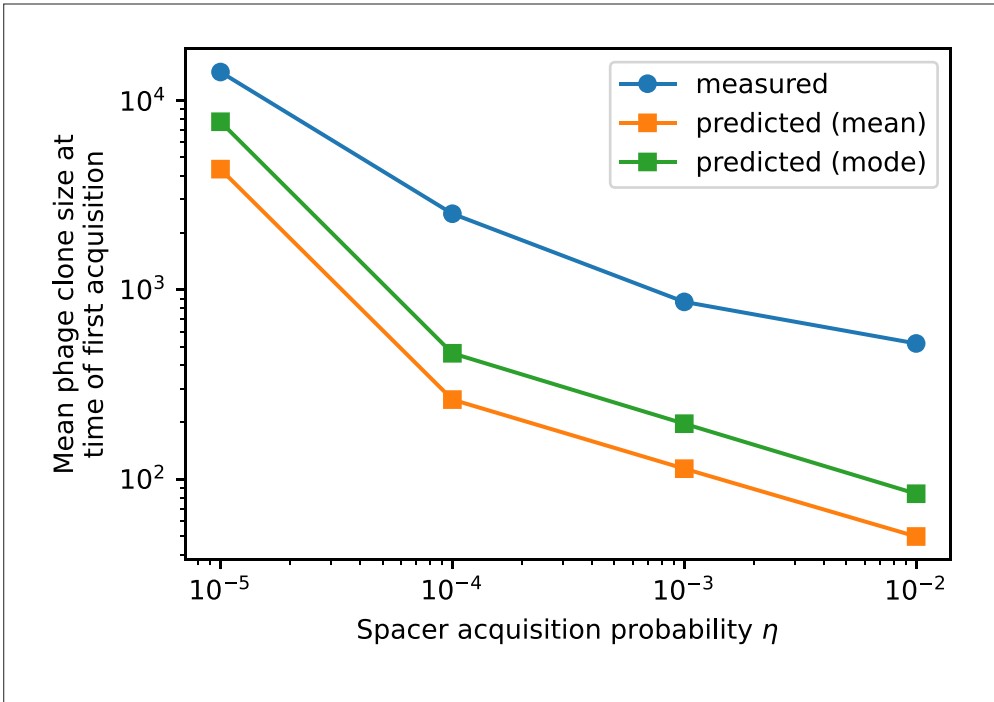

**Figure 20.** Mean phage clone size at time of first spacer acquisition for simulation data, the predicted with *Equation 31*, and the prediction with the mode of the distribution given by *Equation 29* for four simulations with different values of $\eta$ and $C_0 = 10^4$, $\mu = 10^{-5}$, and $e = 0.95$.

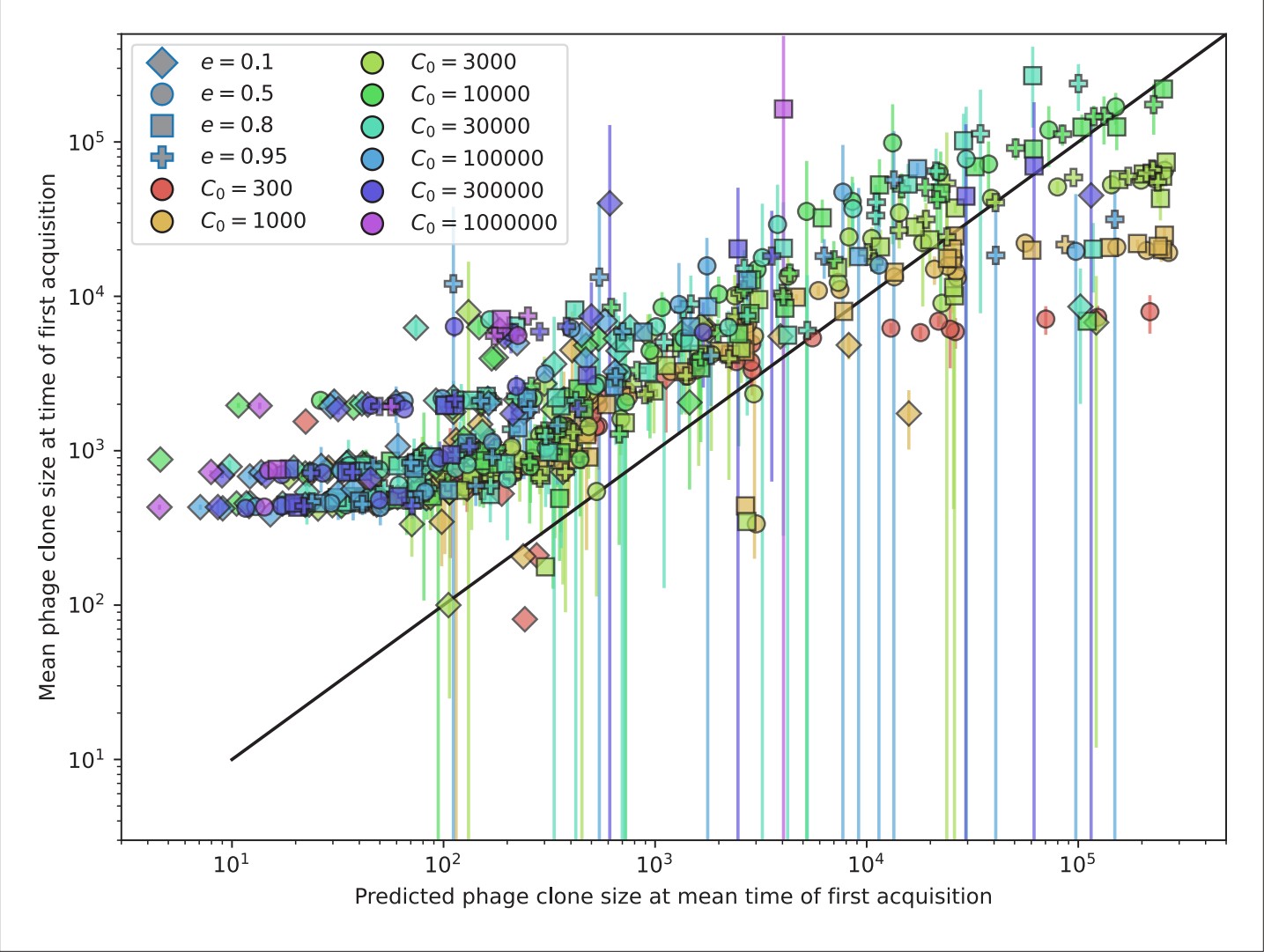

**Figure 21.** Measured mean phage clone size at the time of first spacer acquisition vs. the prediction given by $e^{s_0\langle t\rangle}$ of **Equation 31**. Error bars are the standard deviation across three or more independent simulations.

Now expanding $s_0$ and the denominator and cancelling and collecting terms:

$$m = \frac{4(\alpha n_B(Bp_V - 1) - F)(1 - e^{-\mu L})n_V n_V^i(1 - \ln\frac{n_V^i}{n_V})}{(B-1)(B(B-2)\beta + \alpha n_B + F)}(1 - \frac{e\nu}{m}) \tag{35}$$

In theory, $n_V^i \approx n_V/m$ and $n_B^i \approx n_B^s/m$. To see this, we start with the solutions for the deterministic mean bacteria and phage clone sizes (**Equations 25 and 26**, reprinted here):

$$n_B^{i\,*} = \frac{1}{e}\left(n_B - \frac{F + \alpha n_B}{\alpha B P_0 p_V}\right) \tag{36}$$

$$n_V^{i\,*} = \frac{n_B^{i\,*}(\alpha p_V n_V - (gC - F - r))}{\alpha\eta n_B^0(1 - p_V) + \alpha p_V e n_B^{i\,*}} \tag{37}$$

From the deterministic mean-field equation for $n_V$ (**Equation 14**), we have that $F + \alpha n_B = \alpha B p_V n_B(1 - \frac{e\nu}{m})$. Substituting this in **Equation 36**, we get

$$n_B^{i\,*} = \frac{1}{e}\left(n_B - \frac{\alpha B p_V n_B(1 - \frac{e\nu}{m})}{\alpha B P_0 p_V}\right) = \frac{n_B}{e}\left(1 - \frac{(1 - \frac{e\nu}{m})}{P_0}\right) \tag{38}$$

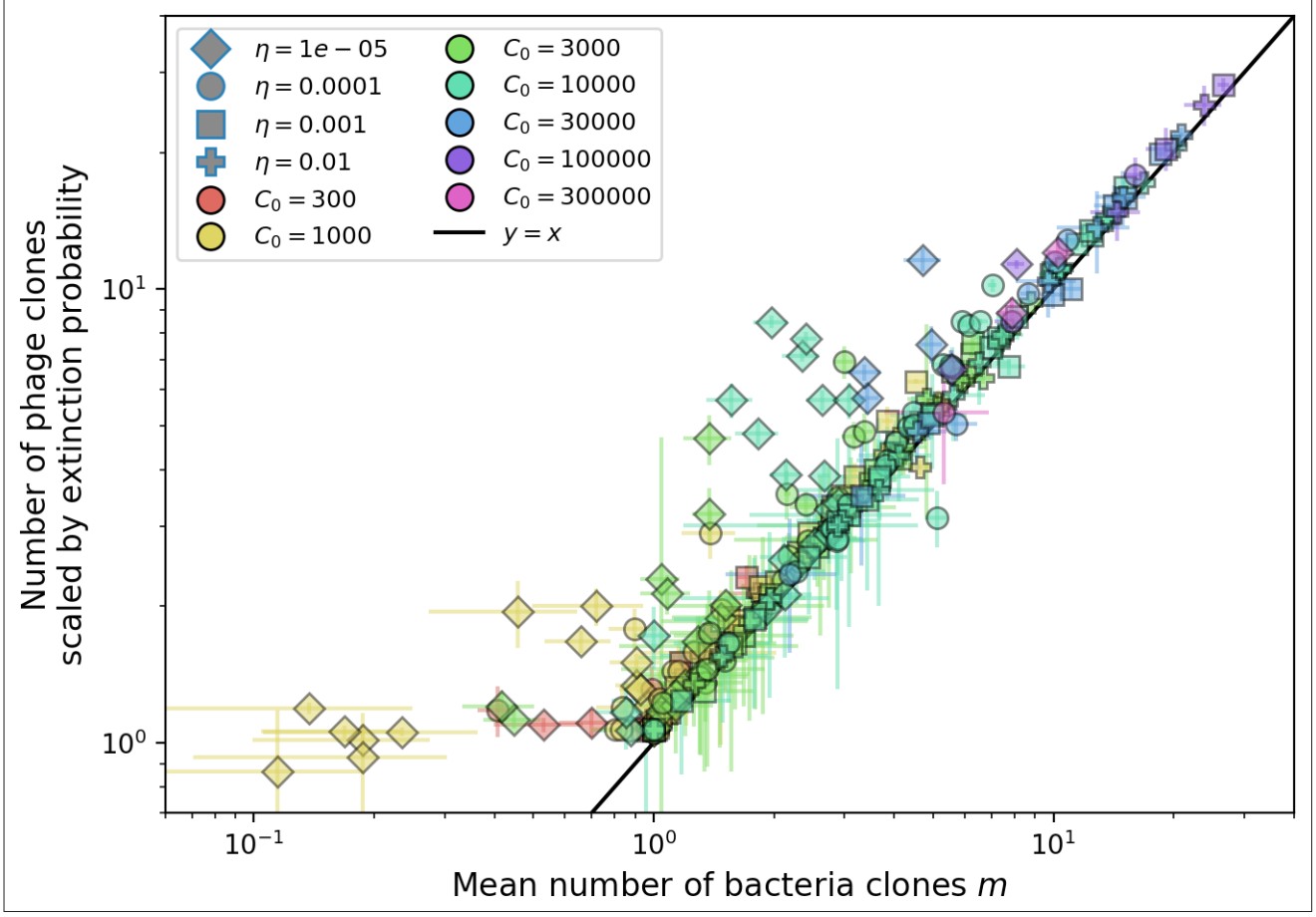

**Figure 22.** Mean number of large phage clones vs. mean number of bacterial clones in simulations. For each simulation, we take a subset of 15 evenly spaced timepoints at steady state and calculate the size and number of phage clones present. We scale the observed clone sized distribution with *Equation 28* and calculate the mean number of large phage clones by multiplying the total number of clones with the fraction of large phage clones given by $\sum_n P_n^{\text{large}}$. We use the simulation mean total population sizes to calculate $s_0$ and $\delta_0$ in *Equation 27*. We obtain the mean number of bacterial clones by averaging the number of clones present at 15 evenly spaced timepoints at steady state. Error bars are the standard deviation across three or more independent simulations.

We can neglect $P_0$, both because it is always close to 1 and because, at steady state, the effect of mutants leaving type $i$ is partially balanced by mutants entering type $i$.

$$n_B^{i\,*} \approx \frac{n_B \nu}{m} = \frac{n_B^s}{m} \tag{39}$$

Similarly, from the deterministic mean-field equation for $n_B^s$ (*Equation 13*), we have $gC - F - r = \alpha n_V p_V \left(1 - \frac{e}{m}\right) - \alpha n_V (1 - p_V) \eta \frac{n_B^0}{n_B^s}$. Substituting this in *Equation 37*, we get

$$n_V^{i\,*} = \frac{n_B^{i\,*}(\alpha p_V n_V - (\alpha n_V p_V \left(1 - \frac{e}{m}\right) - \alpha n_V (1 - p_V) \eta \frac{n_B^0}{n_B^s}))}{\alpha \eta n_B^0 (1 - p_V) + \alpha p_V e n_B^{i\,*}} \tag{40}$$

Substituting $n_B^0 = n_B(1 - \nu)$ and $n_B^s = n_B \nu$, we find $n_V^{i\,*} = n_V/m$.
Replacing $n_V^i$ and $n_B^i$ with $n_V/m$ and $n_B^s/m$, respectively:

$$m = \frac{4(\alpha n_B(B p_V - 1) - F)(1 - e^{-\mu L})\frac{n_V^2}{m}(1 - \ln\frac{1}{m})}{(B - 1)(B(B - 2)\alpha p_V n_B(1 - \frac{e\nu}{m}) + \alpha n_B + F)}(1 - \frac{e\nu}{m}) \tag{41}$$

Now we want to solve *Equation 41* for $m$. This is difficult because the total population sizes $n_B$, $n_V$, $C$, and $\nu$ also depend on $m$. To find the $m$ dependence of $n_V$, $n_B$, $C$, and $\nu$, we note that we can

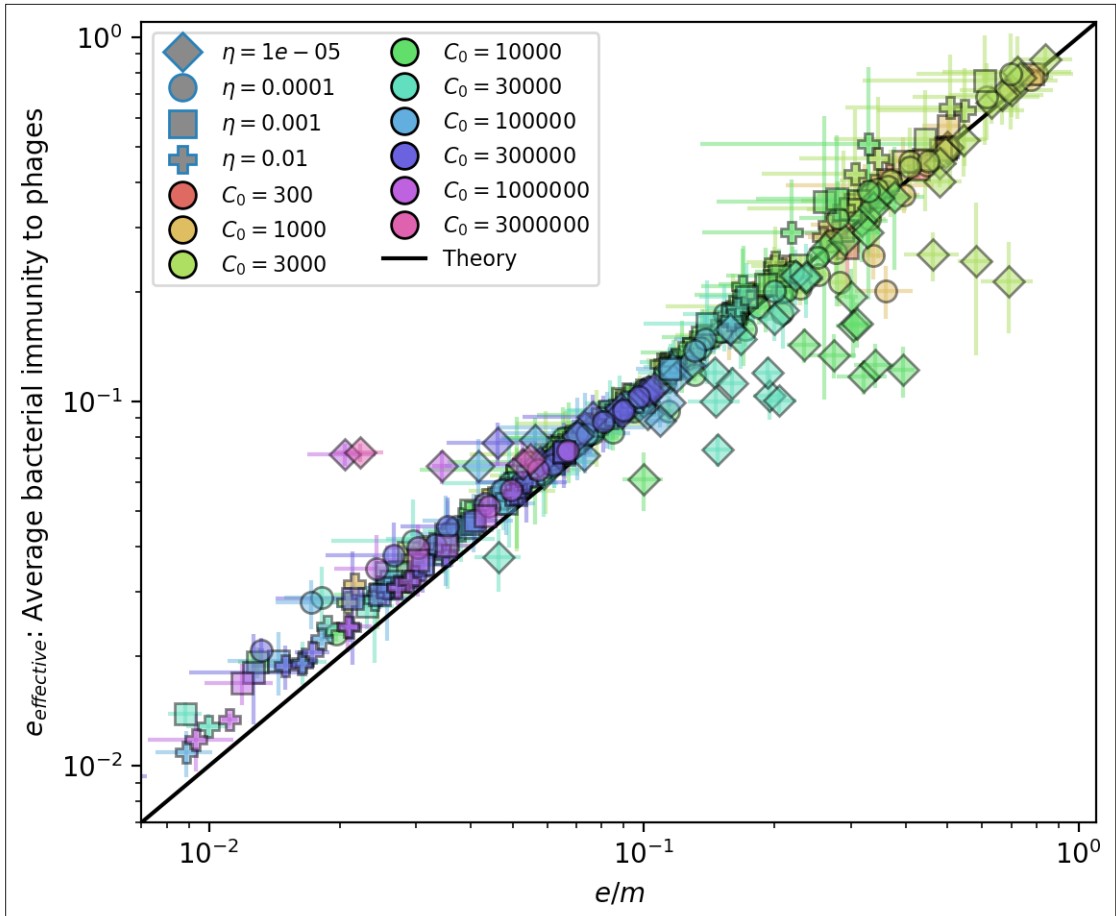

**Figure 23.** Effective $e = \frac{e}{n_B^s n_V} \sum_i n_V^i n_B^i$ vs $e/m$ across all simulations where $m \geq 1$ on average. Error bars are the standard deviation across three or more independent simulations. The solid black line is $y = x$.

approximate the steady-state solutions to *Equations 13–17* as the corresponding solutions described in *Bonsma-Fisher et al., 2018* with the simple replacement of the parameter $e$ with $e/m$. This is because $e/m$ is a good approximation for the full average immunity at most parameters (*Figure 23*), and average immunity enters these equations in place of $e$ in the model without phage mutations. This is analogous to average adaptive immunity replacing the rate of innate immunity in the Lotka-Volterra system described by *Iranzo et al., 2013*.

We write the steady-state solutions for *Equations 14 and 17* below, where we have defined $p_V(i,j)$ as in *Equation 18* and approximated average immunity as $e/m$. For simplicity we rescale $n_B$ and $n_V$ by $C_0$: $n_V = C_0 y^*$ and $n_B = C_0 x^*$. Here $p = p_V \alpha/g$. These steady-state solutions are shown in *Figure 26*.

At low average immunity (high diversity), the population behaves as if there were no CRISPR system at all. *Figure 26* shows total phage, total bacteria, and the fraction of bacteria with spacers as a function of average immunity (effective $e$). At low average immunity, total population sizes are the same as in a population without CRISPR entirely. Interestingly, the effective no-CRISPR case does not necessarily correspond to no CRISPR spacers: $\nu > 0$ even when effective $e \approx 0$ for these parameters. This is because spacer acquisition and growth of that clone can still happen even if the spacer confers no fitness benefit.

$$x^* = \frac{f p_V}{p} \frac{1}{B p_V (1 - \frac{e\nu^*}{m}) - 1} \tag{42}$$

$$y^* = \frac{(f-1)p(B p_V(\frac{e\nu^*}{m} - 1) + 1) - f p_V}{p(\frac{e\nu^*}{m} - 1)(p(B p_V(\frac{e\nu^*}{m} - 1) + 1) - p_V)} \tag{43}$$

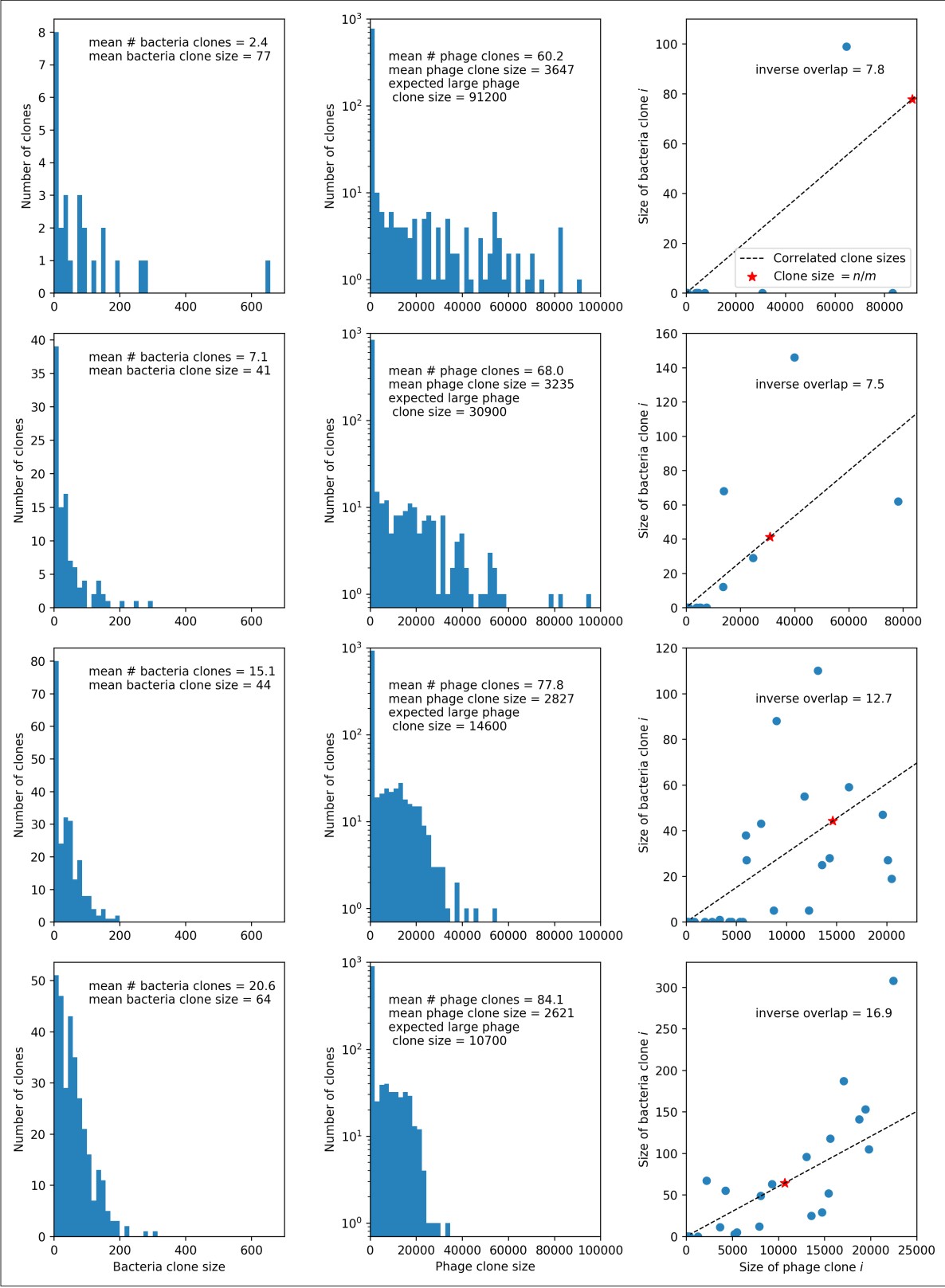

**Figure 24.** Four simulations with $C_0 = 10000$, $\mu = 10^{-5}$, and $e = 0.95$. From top to bottom, $\eta$ increases by a factor of 10 in each row, from $\eta = 10^{-5}$ in the top row to $\eta = 10^{-2}$ in the bottom row. The first two columns show clone size distributions combined from 15 time points between 2000 and 10,000 bacterial generations. Bacteria are in the left column and phages in the middle column. The third column shows the pairwise clone sizes of matching clones at the last sampled time point (9467 generations). The expected large phage clone size is the total phage population divided by the

*Figure 24 continued on next page*

*Figure 24 continued*

mean number of bacterial clones. The inverse overlap is $\frac{e}{e_{\text{eff}}} = \frac{n_B^s n_V}{\sum_i n_V^i n_B^i}$, which we assume is $\approx m$ as shown in **Figure 23**. The dashed line in the third column indicates the line that clone size pairs would fall on if they were perfectly correlated, and the red star indicates the mean large clone size for phages and the mean clone size for bacteria.

We have defined $x^*$ and $y^*$ in terms of $\nu^*$, which is given in the following implicit cubic equation, where $R = r/(gC_0)$:

$$0 = (1 - \nu)\left[-p_V\frac{e\nu}{m} - \eta(1 - p_V)\right]\left[(1 - f)p(p_VB(1 - \frac{e\nu}{m}) - 1) - fp_V\right]$$
$$+R\nu p_V(1 - \frac{e\nu}{m})(Bpp_V(1 - \frac{e\nu}{m}) - p + p_V)$$

(44)

This cubic equation is analytically solvable (ignoring the dependence of $m$ on $\nu$), but the full solutions in terms of all parameters are cumbersome.

Only one of the three solutions of **Equation 44** is physical in the parameter range we use (real-valued and properly bounded):

$$\nu^* = -\frac{\left(1 + i\sqrt{3}\right)\sqrt[3]{\sqrt{\left(-27a^2d + 9abc - 2b^3\right)^2 + 4\left(3ac - b^2\right)^3} - 27a^2d + 9abc - 2b^3}}{6\sqrt[3]{2}a}$$
$$+ \frac{\left(1 - i\sqrt{3}\right)\left(3ac - b^2\right)}{3\,2^{2/3}a\sqrt[3]{\sqrt{\left(-27a^2d + 9abc - 2b^3\right)^2 + 4\left(3ac - b^2\right)^3} - 27a^2d + 9abc - 2b^3}} - \frac{b}{3a}$$

(45)

where the coefficients are

$$a = B(\frac{e}{m})^2 fpp_V^2(f + R - 1)$$

(46)

$$b = -\frac{e}{m}fp_V(p(f(B(p_V(\frac{e}{m} + \eta + 1) - \eta) - 1) + B(\eta - p_V(\frac{e}{m} + \eta - 2R + 1)) - R + 1) + p_V(f + R))$$

(47)

$$c = fp\left[Bp_V^2(\frac{e}{m}(f - 1)(\eta + 1) + (f - 1)\eta + R)\right.$$
$$\left. - (\frac{e}{m} - 1)(f - 1)p_V(B\eta + 1) - (2B + 2)(f - 1)\eta p_V + (f - 1)\eta - p_V(f + R - 1)\right]$$
$$+ fp_V(\frac{e}{m}fp_V - f\eta + p_V(f\eta + R))$$

(48)

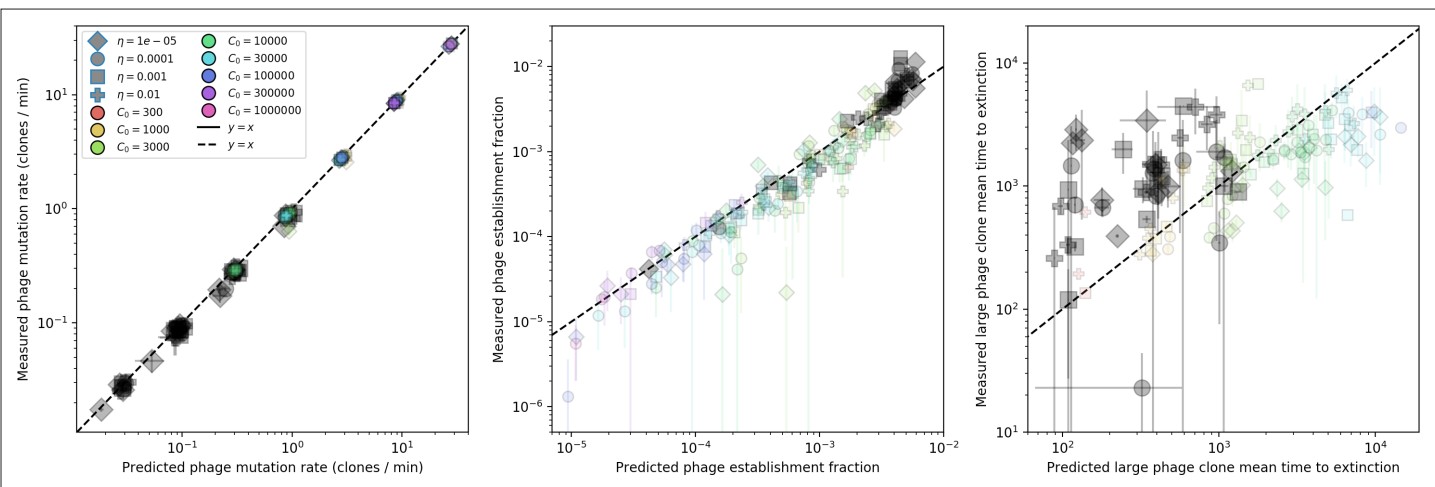

**Figure 25.** Measured simulation phage clone mutation rate, establishment fraction, and mean time to extinction as a function of the theoretical prediction for each. Highlighted in grey are parameter combinations for which no theoretically predicted $m$ could be determined; the predicted quantity is instead calculated with the simulation mean $m$. Error bar are the stnadard deviation across three or more independent simulations.

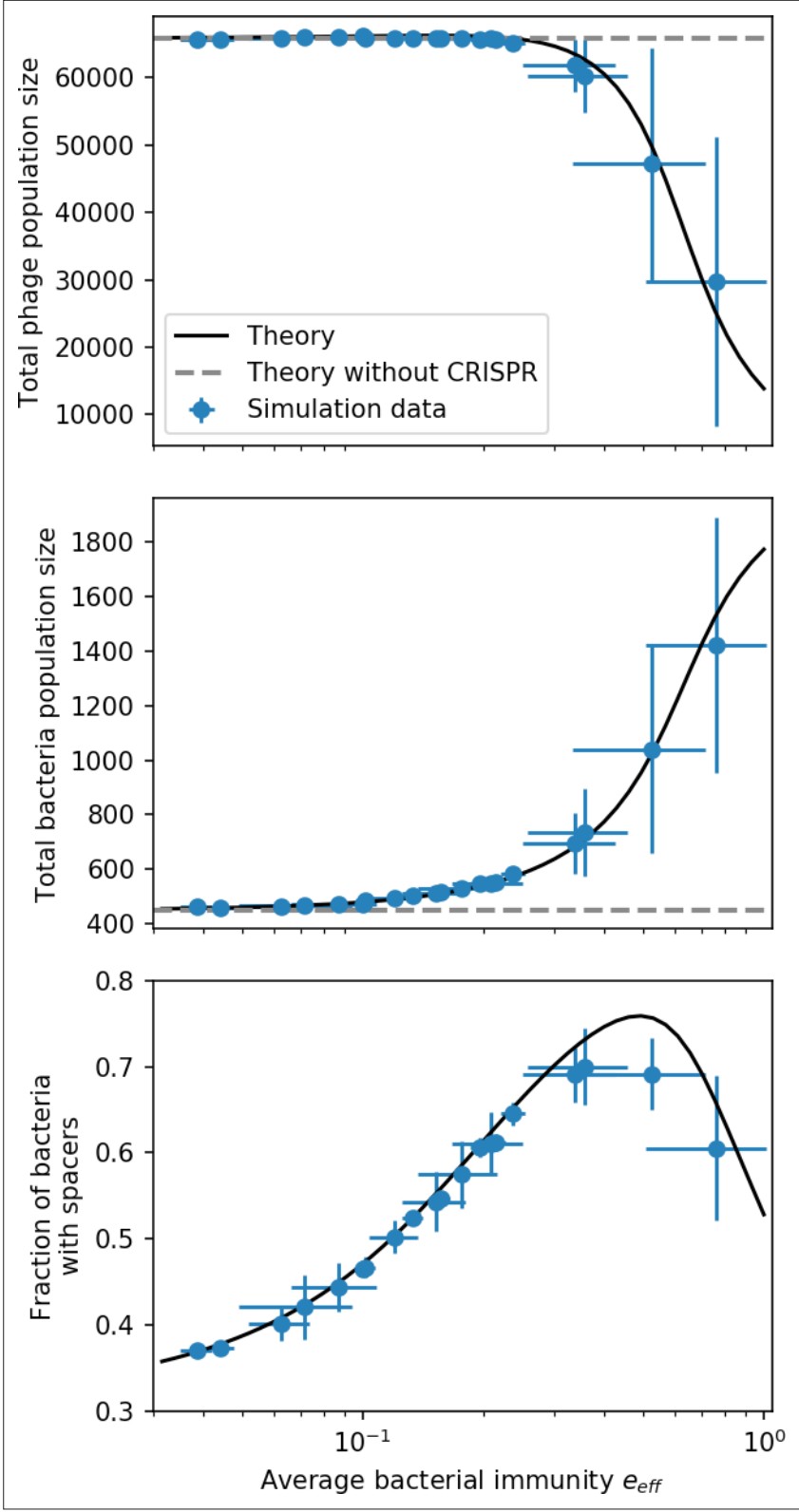

**Figure 26.** Total phage, total bacteria, and fraction of bacteria with spacers as a function of average immunity (effective $e$). Blue points are simulation results. Error bars are standard deviation across three or more independent simulations. The solid black line is the solution given by *Equations 42*–44 (from *Bonsma-Fisher et al., 2018*) with the parameter $e$ replaced by effective $e$. The horizontal grey dashed line corresponds to the no-CRISPR ($e = 0$) mean-field solution (derived in *Bonsma-Fisher et al., 2018*).

$$d = -f\eta(p_V - 1)((f-1)p(Bp_V - 1) + fp_V) \tag{49}$$

## Dominant balance approximations for $\nu$

The cubic equation for $\nu$ given by **Equation 44** can be simplified in certain parameter regimes using dominant balance. We write **Equation 44** as $a\nu^4 + b\nu^2 + c\nu + d = 0$ with the coefficients as written above. Comparing the numerical values of the coefficients in different parameter regimes, we arrive at **Table 4** which outlines three different approximations for $\nu$ obtained by dropping coefficients of the original cubic equation. Broadly speaking, at low $\eta$ and low $e$, both the cubic and quadratic coefficients ($a$ and $b$) are small, so $\nu^* \approx -d/c$, while at high $\eta$ and low $e$ the cubic coefficient ($a$) can be dropped, and at high $e$ and low $\eta$ the constant $d$ can be dropped. **Figures 27–29** show total population sizes as a function of the three approximations in **Table 4**.

For small effective $e$ and small $\eta$, $\nu \approx -d/c$:

$$\nu \approx \frac{\eta(1 - p_V)(\alpha(1-f)(Bp_V - 1) - fg)}{\eta(1 - p_V)(\alpha(1-f)(B(\frac{e}{m}+1)p_V - 1) - fg) + \alpha p_V(Bp_V - 1)(R - \frac{e}{m}(1-f)) + gp_V(\frac{e}{m}f + R)} \tag{50}$$

We define two combined parameters: $A = \frac{(Bp_V - 1)(1-f)\alpha}{fg}$, and $\hat{\eta} = \eta(1 - p_V)$. Each of these has an intuitive meaning: $A > 1$ is the stability condition for phage existence, and $\hat{\eta}$ is the probability of spacer acquisition following escape from naive phage predation, so it can be thought of as the overall spacer acquisition probability at the start of an infection. With these variable substitutions, we get

$$\nu \approx \frac{\hat{\eta}(A - 1)}{p_V R\left(\frac{1-f+Af}{f(1-f)}\right) + \hat{\eta}\left(A - 1 + \frac{ABp_V\frac{e}{m}}{(A-1)(Bp_V - 1)}\right) - \frac{e}{m}p_V(A - 1)} \tag{51}$$

Now we notice that the deterministic total phage population size when $e = 0$ is given by

$$n_{V \text{no CRISPR}} = \tilde{n}_V = \frac{gC_0 f(1-f)(A - 1)}{\alpha p_V(1 - f + Af)} \tag{52}$$

We also include $n_B$ for $e = 0$ for completeness:

$$n_{B \text{no CRISPR}} = \tilde{n}_B = \frac{C_0(1-f)}{A} \tag{53}$$

We can replace some terms in $\nu$ with $\tilde{n}_V$ by comparison with **Equation 52**:

$$\nu \approx \frac{1}{1 + \frac{r}{\hat{\eta}\alpha\tilde{n}_V} - \frac{e}{m}\left(\frac{p_V}{\hat{\eta}} - \frac{ABp_V}{(A-1)(Bp_V - 1)}\right)} \tag{54}$$

where $r = RgC_0$ is the spacer loss rate per minute. The second term in the denominator is the balance of spacer acquisition and spacer loss per naive bacterium: $\hat{\eta}\alpha\tilde{n}_V$ is the rate of spacer acquisition per naive bacterium in the low average immunity limit.

---

**Table 4.** $\nu$ approximations.

| | $e_{\text{effective}}$ | | | | |
|---|---|---|---|---|---|
| | ≤0.01 | 0.01 to 0.05 | 0.05 to 0.1 | 0.1 to 0.5 | 0.5 to 1 |
| $\eta$ | | | | | |
| $\leq 10^{-5}$ | $-\frac{d}{c}$ | $-\frac{d}{c}$ | $-\frac{d}{c}$ | $\frac{-b+\sqrt{b^2-4ac}}{2a}$ | $\frac{-b+\sqrt{b^2-4ac}}{2a}$ |
| $10^{-5}$ to $10^{-4}$ | $-\frac{d}{c}$ | $-\frac{d}{c}$ | $\frac{-c+\sqrt{c^2-4bd}}{2b}$ | $\frac{-b+\sqrt{b^2-4ac}}{2a}$ | |
| $10^{-4}$ to $10^{-2}$ | $-\frac{d}{c}$ | $-\frac{d}{c}$ | $\frac{-c+\sqrt{c^2-4bd}}{2b}$ | | |
| $\geq 10^{-2}$ | $-\frac{d}{c}$ | $\frac{-c+\sqrt{c^2-4bd}}{2b}$ | $\frac{-c+\sqrt{c^2-4bd}}{2b}$ | | |

Note: the drop-$a$ solution also works for below $e = 0.1$ for all values of $\eta$ (since $\frac{2bd^2}{c} \approx 0$), but the $-d/c$ solution is simpler and so preferred for the very low $e$ range.

---

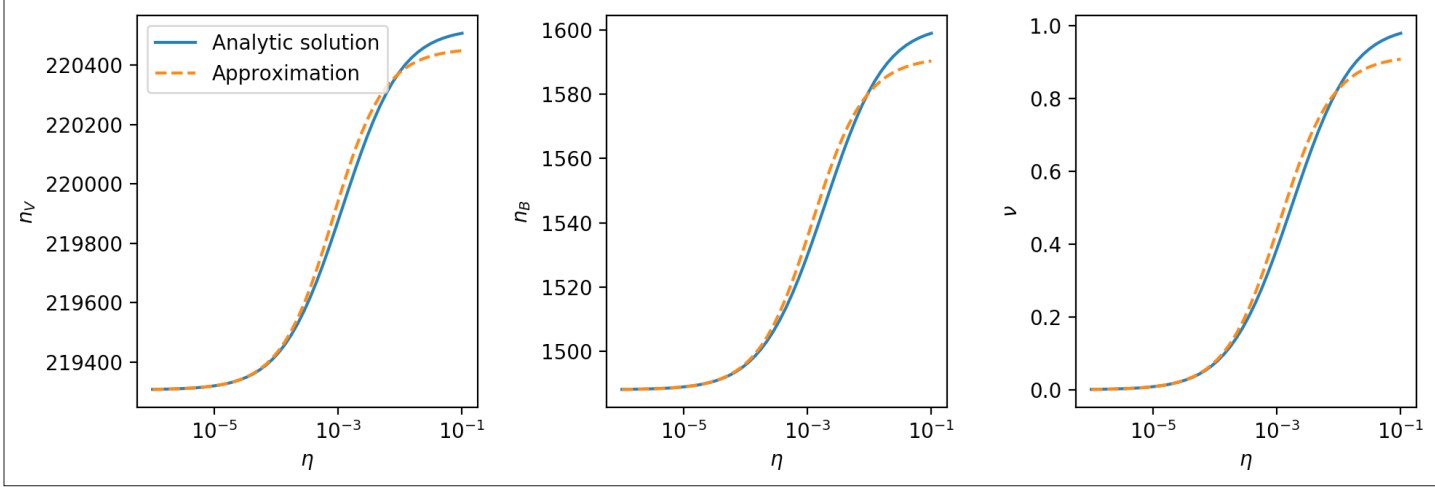

**Figure 27.** $n_V$, $n_B$, and $\nu$ vs. $\eta$ for $e = 0.05$, approximating $\nu \approx -d/c$.

The third term in the denominator is negative for the parameters we use, but this is not necessarily true in general. It is negative if $\hat{\eta}AB < (A - 1)(Bp_V - 1)$, which for the values of $A$ and $Bp_V$ we use (fixed across all simulations) is true for $\eta < 0.0113$.

From this expression for $\nu$ we learn the following:

- $\nu$ decreases if $r$ goes up. More spacer loss means a smaller fraction of bacteria with spacers.
- $\nu$ increases if spacer acquisition goes up (provided $e/m$ is small).
- $\nu$ increases if $\frac{e}{m}$ increases: higher CRISPR effectiveness means higher $\nu$.

For $e/m \to 0$, we define $\tilde{\nu}$:

$$\tilde{\nu} = \frac{1}{1 + \frac{r}{\hat{\eta}\alpha\tilde{n}_V}} \tag{55}$$

**Equation 54** expression breaks down at high $\eta$ where the true value of $\nu$ is largely independent of $e$. In this case, when both $\frac{e}{m}$ is small and $\hat{\eta}$ is large, we can drop the third term in the denominator in **Equation 54**.

There is a discontinuity in **Equation 54** at the value of $e/m$ where the denominator equals zero. This critical value of $e/m$, $\frac{e}{m}^*$, is given by **Equation 56**.

$$\frac{e}{m}^* = \frac{(A - 1)(Bp_V - 1)(\alpha\hat{\eta}\tilde{n}_V + r)}{\alpha\tilde{n}_V p_V((A - 1)(Bp_V - 1) - AB\hat{\eta})} \tag{56}$$

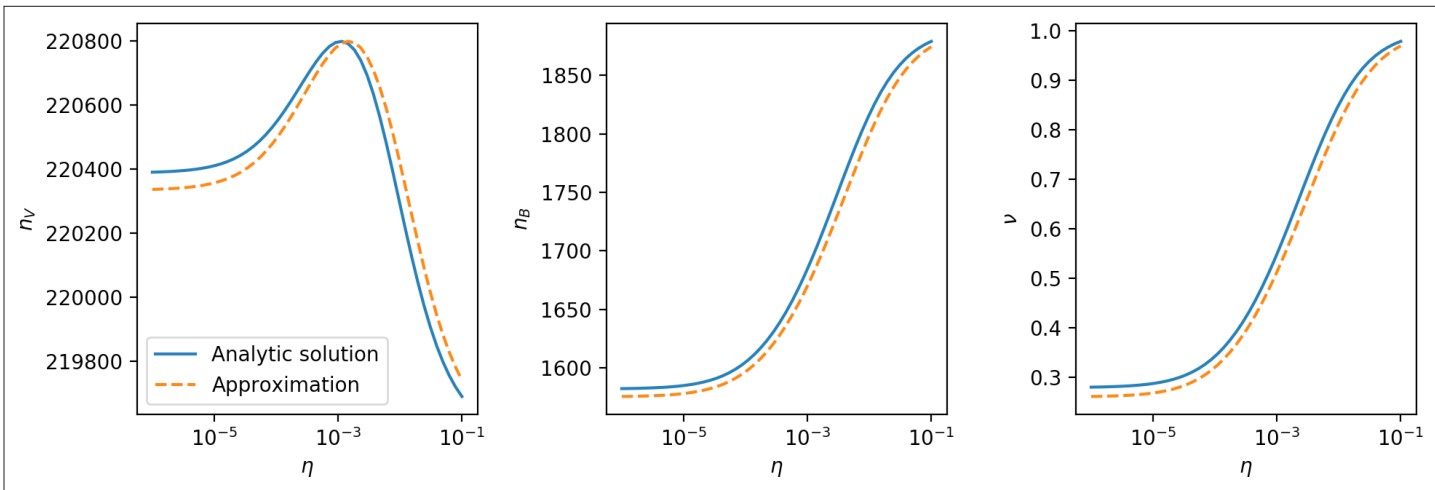

**Figure 28.** $n_V$, $n_B$, and $\nu$ vs $\eta$ for $e = 0.15$, approximating $\nu \approx \frac{-c + \sqrt{c^2 - 4bd}}{2b}$.

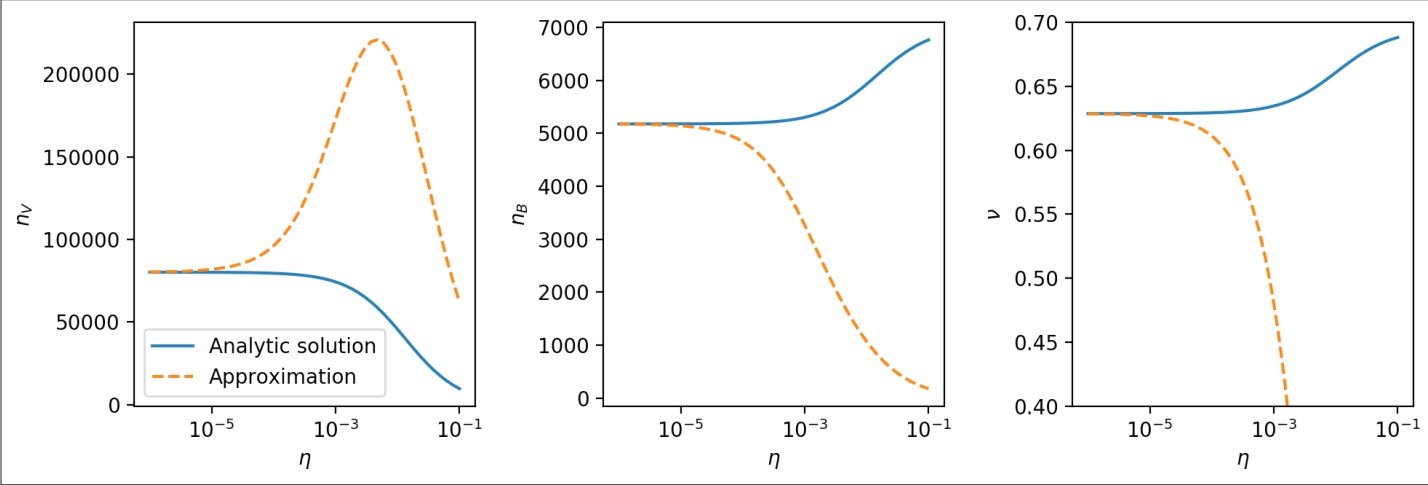

**Figure 29.** $n_V$, $n_B$, and $\nu$ vs. $\eta$ for $e = 0.8$, approximating $\nu \approx \frac{-b + \sqrt{b^2 - 4ac}}{2a}$.

Taking a series expansion in $\hat{\eta}$ and keeping the first two terms:

$$\frac{e*}{m} \approx \frac{r}{\alpha \tilde{n}_V p_V} + \frac{\hat{\eta}}{p_V} + \frac{ABr\hat{\eta}}{\alpha \tilde{n}_V p_V (A - 1)(B p_V - 1)} \tag{57}$$

At high $\eta$, the middle term dominates, and at small $\eta$, the first two terms dominate (since the third term is $\approx 9.8\hat{\eta}$ and $1/p_V = 50$). This transition governs the change between the low $e$ and high $e$ regimes; *Equation 58* is plotted in *Figure 3—figure supplement 2* and *Figure 3—figure supplement 3*.

$$\frac{e*}{m} \approx \frac{r}{\alpha \tilde{n}_V p_V} + \frac{\hat{\eta}}{p_V} \tag{58}$$

If we substitute this critical value of $e/m$ into the mean-field equations and take $\eta$ to 0, we recover the no-CRISPR mean-field solutions for $n_V$, $n_B$, and $C$. Since this also corresponds to $\nu = 0$, this seems to explain why several simulation values of the phage establishment probability go to 0 around effective $e = 0.1$ for small $\eta$ and small $C_0$ (*Figure 30*). Despite the presence of spacers with non-zero effectiveness, the population appears to behave as if there is no CRISPR near this point, which removes the

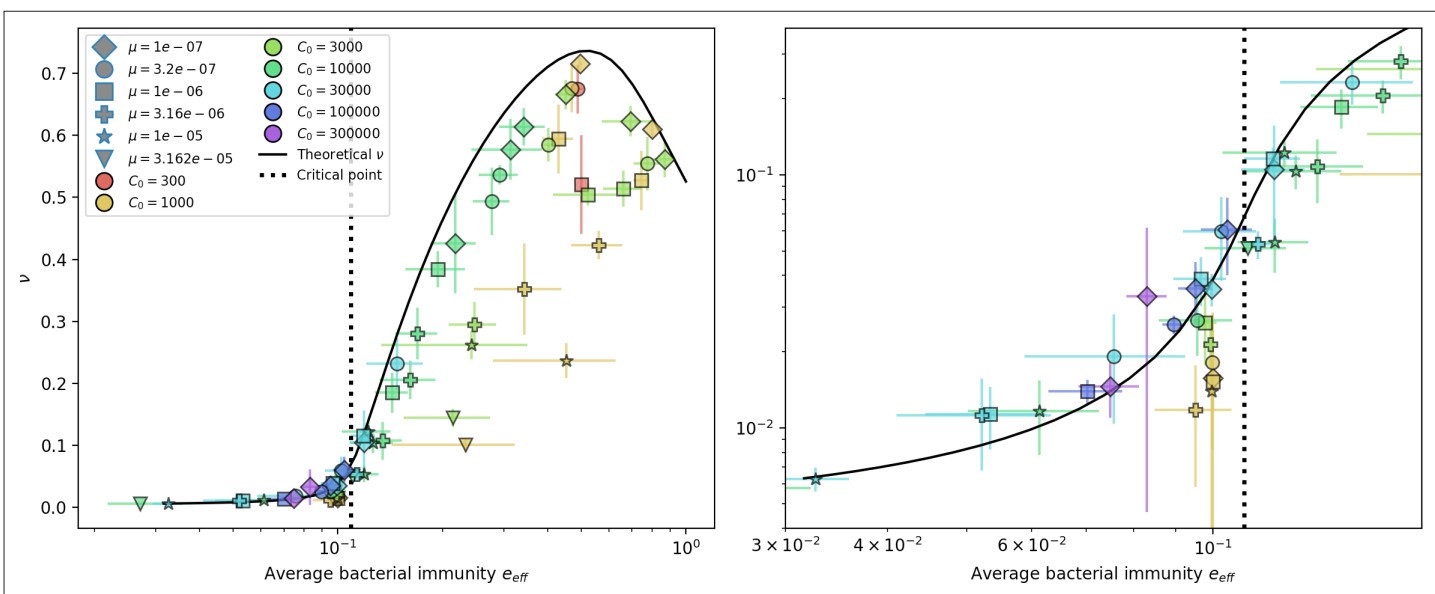

**Figure 30.** Fraction of bacteria with spacers $\nu$ vs. effective $e$. The solid line is the full numerical theoretical solution. The dashed black line is given by *Equation 58*. Error bars are the standard deviation across three or more independent simulations.

fitness advantage of new phage mutants and lowers their establishment probability. We also see large fluctuations in total bacteria population size near this critical point (**Figure 31**).

Another way to understand the critical point is to look at the difference in relative fitness between bacteria with and without spacers, $n_B^s$ and $n_B^0$:

$$\frac{\dot{n_B^s}}{n_B^s} - \frac{\dot{n_B^0}}{n_B^0} = -r\left(1 + \frac{n_B^s}{n_B^0}\right) + \alpha n_V p_V e_{eff} + \alpha(1 - p_V)\eta n_V\left(\frac{n_B^0}{n_B^s} - 1\right) \tag{59}$$

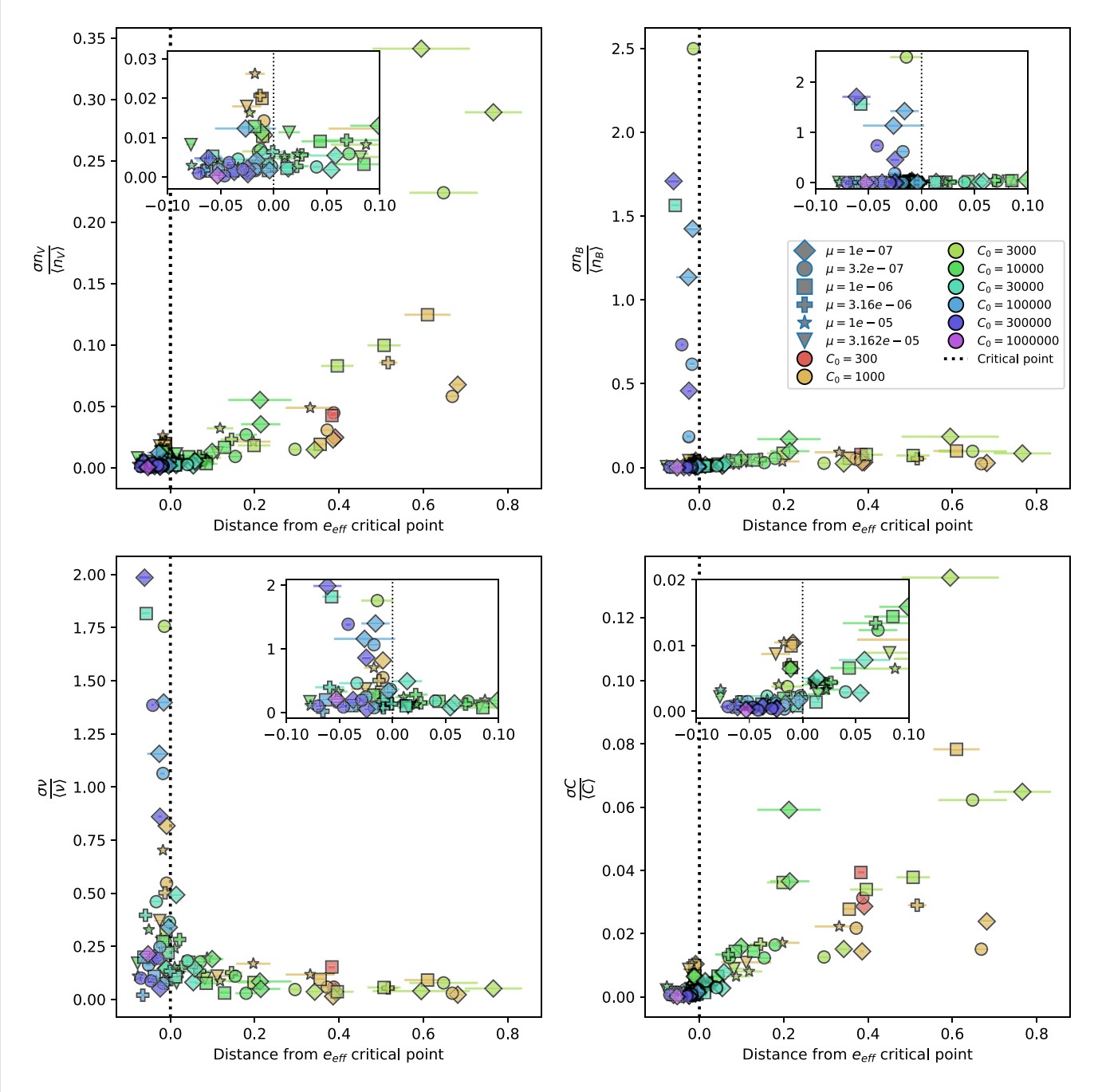

**Figure 31.** Population standard deviation divided by population mean as a function of the distance from the average immunity critical point given by **Equation 58**. Insets show a smaller x-axis range for the same quantities. Total phage (top left), total bacteria (top right), fraction of bacteria with spacers (bottom left) and total nutrients (bottom right) are plotted. X error bars are the standard deviation across three or more independent simulations.

Now if effective $e$ is given by **Equation 58**:

$$\frac{\dot{n_B^s}}{n_B^s} - \frac{\dot{n_B^0}}{n_B^0} = -r\frac{n_B^s}{n_B^0} + \alpha(1-p_V)\eta n_V \frac{n_B^0}{n_B^s}$$

(60)

**Equation 60** indicates that at the critical value of effective $e$, the fitness difference is exactly given by the balance of spacer acquisition and spacer loss. Below this critical point, there are additional constant negative terms, and for the same values of $n_B^s$ and $n_B^0$, the relative fitness of spacer-containing bacteria is lower than naive bacteria; above this point, it is higher. At the critical point, the immunity boost from spacers exactly cancels out spacer loss. It is interesting that for the same reasons that many parameters collapse onto the same curves as a function of average immunity, the value of this critical point is largely independent of the parameters of CRISPR immunity and occurs at a value of effective $e \approx 10^{-1}$ regardless of $\mu$, $e$, or $C_0$.

## Large $e$ approximation for $\nu$

In the limit of $e \to 1$ and small $\eta$, we find the following approximate solution for $\nu$ by setting $e = 1$ and taking a series expansion in $\eta$.

$$\nu \approx \frac{\alpha(Bp_V - 1)(f + R - 1) + g(f + R)}{\alpha Bp_V(f + R - 1)} + \frac{gR\hat{\eta}}{p_V(f + R - 1)(\alpha(Bp_V - 1)(f + R - 1) + g(f + R))}$$

(61)

**Equation 61** is plotted in **Figure 3—figure supplement 2** in red. The second term proportional to $\hat{\eta}$ is tiny compared to the first for all values of $\eta$ we use.

## Approximation for $T_{\text{ext}}$

If we evaluate the mean phage time to extinction ('Neutral time to extinction from backward master equation') at the deterministic mean phage clone size, we can approximate $n_V^i \approx \frac{n_V}{m}$ and $n_B^i \approx \frac{n_B^s}{m}$ as before:

$$T_{\text{ext}} \approx \frac{2\frac{n_V}{m}(1 + \ln m)gC_0}{B(B-2)\alpha p_V n_B(1 - \nu\frac{e}{m}) + F + \alpha n_B}$$

(62)

We expand $n_V$ and $n_B$ in powers of $e/m$:

$$n_V = \frac{gC_0 f(1-f)(A-1)}{\alpha p_V(1 - f + Af)} + \frac{e\nu}{m}C_0\left[-\frac{fg}{\alpha p_V} - \frac{1-f}{p_V(1 - f + Af)} + \frac{ABf(1-f)}{(1 - f + Af)^2}\right] + \mathcal{O}(e/m)^2$$

(63)

$$n_B = \frac{C_0(1-f)}{A} + \frac{e\nu}{m}\left[\frac{BFp_V}{\alpha(Bp_V - 1)^2}\right] + \mathcal{O}(e/m)^2$$

(64)

Note that $\nu$ never appears without $e/m$ beside it in these expressions, so we will use the zeroth-order expression for $\nu$ (**Equation 55**) which still allows us to keep $e/m$ to first order elsewhere.

$$\tilde{\nu} = \frac{1}{1 + \frac{r}{\hat{\eta}\alpha\bar{n}_V}} + \mathcal{O}(e/m)$$

(65)

Substituting **Equations 63–65** into $T_{\text{ext}}$:

$$T_{\text{ext}} \approx \frac{2(1+\ln m)}{f(B-1)}\frac{\tilde{n}_V}{m}\left(1 - \frac{1}{Bp_V}\right)$$
$$+ \frac{e}{m^2}(1 + \ln m)\left[\frac{2C_0^2 fg\hat{\eta}(1-A)((Bpv-2)(g - \frac{A^2 fg}{1-f}) + \frac{2gA}{1-f}(1 + f(Bp_V - 2)))}{\alpha p_V^2 B(B-1)(1 + \frac{Af}{1-f})^2(\hat{\eta}gfC_0(1-A) - pvr(1 + \frac{Af}{1-f}))}\right]$$

(66)

The second term is order $\frac{e}{m^2}$, which we drop, giving the following approximation for the mean phage time to extinction (plotted in **Figure 32**):

$$T_{\text{ext}} \approx \frac{2(1 + \ln m)}{f(B - 1)}\frac{\tilde{n}_V}{m}\left(1 - \frac{1}{Bp_V}\right)$$

(67)

Here again we are using the phage population size independent of CRISPR ($\tilde{n}_V$). Notably this is independent of $\eta$ and $e$ (except implicitly in $m$) but is still extremely close to the full theoretical quantity. From **Equation 67** we learn:

- $T_{\text{ext}}$ is insensitive to the changes in total phage population size caused by changing CRISPR immunity.
- $T_{\text{ext}}$ increases if the phage clone size $\frac{\tilde{n}_V}{m}$ increases.

- $Bp_V$ is the effective burst size: for every phage adsorption, $Bp_V$ new phages emerge on average. This can be thought of as the phage growth rate. $1 - \frac{1}{Bp_V}$ lies between 0 and 1; if $Bp_V$ is larger, $T_{\text{ext}}$ is also larger.
- The burst size $B$ also appears alone in the denominator ($B - 1$). A larger burst size means larger fluctuations: for larger $B$, more deaths tend to happen between birth events at steady-state, so extinction happens sooner on average (shorter $T_{\text{ext}}$ from $B - 1$ in the denominator).
- The normalized chemostat flow rate $f$ lies between 0 and 1 and is a death rate for phages; if $f$ is larger $T_{\text{ext}}$ is smaller.

The $1 + \ln m$ term is more difficult to parse. A hand-wavy intuition is that while larger $m$ sometimes means smaller clone size and shorter extinction times ($\frac{\bar{n}_V}{m}$), it is also related to increased phage success in the population, and so the $1 + \ln m$ in the numerator softens the straight inverse dependence on $m$ through clone size.

### Approximation for $P_{\text{est}}$

Now we approximate the probability of phage clone establishment.

$$P_{\text{est}} = \frac{2s_0}{B(s_0 + \delta_0)} \tag{68}$$

Recall that $s_0 = \beta_0(B - 1) - \delta_0 = \alpha n_B(Bp_V - 1) - F$, $\beta_0 = n_B \alpha p_V$, and $\delta_0 = F + \alpha n_B(1 - p_V)$. These all depend on $n_B$, so plugging in **Equation 42** for $n_B$, we get

$$P_{\text{est}} = \frac{2e\nu}{m(B - 1)} \tag{69}$$

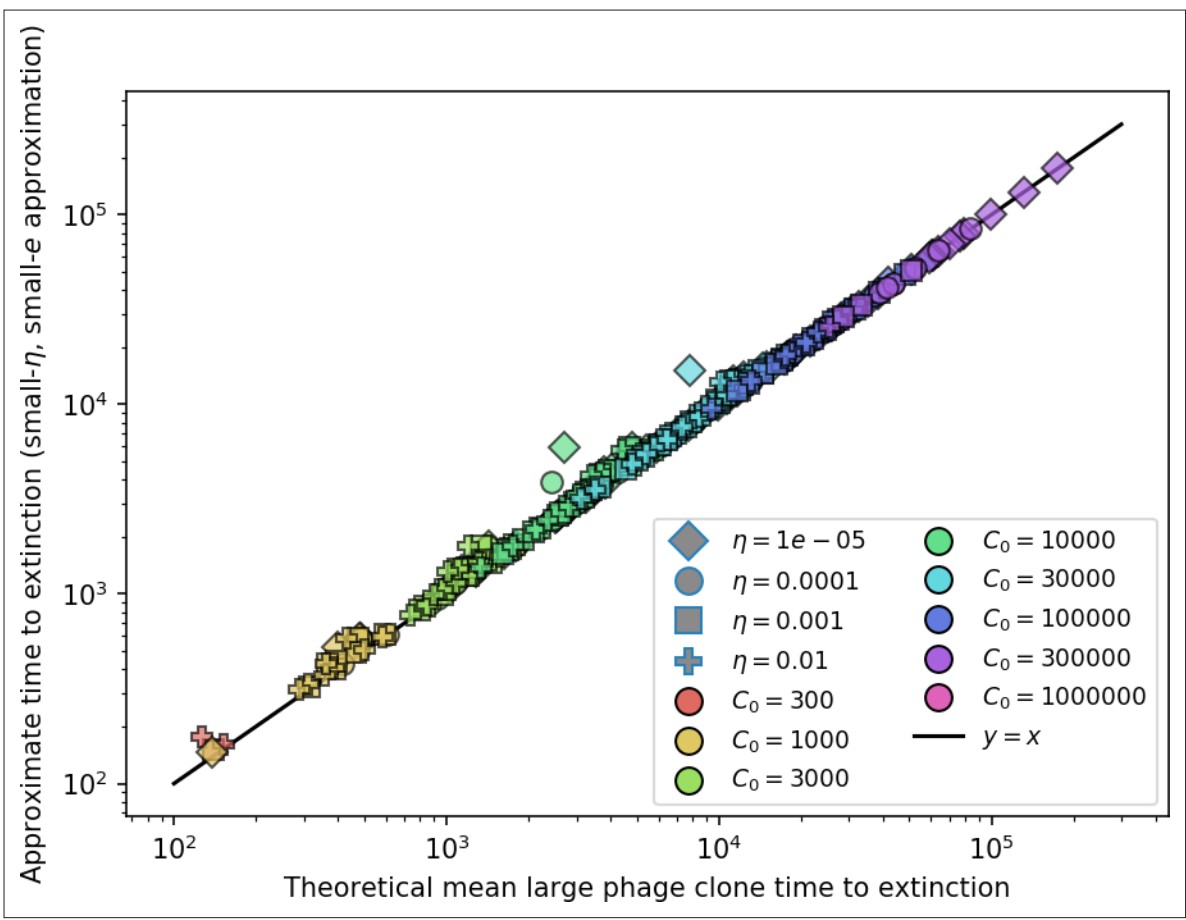

**Figure 32.** Approximate phage time to extinction vs. numerically calculated theoretical time to extinction, where the approximate time to extinction is given by (**Koskella, 2014**). The full theoretical predicted value of **m** is used in the approximate expression.

*Equation 69* with $\nu$ given by *Equation 54* is plotted in blue in *Figure 3—figure supplement 2*. The same expression with $e = 0$ is plotted in green. There is a discontinuity in $\tilde{\nu}$ when the third term in the denominator becomes large, which is why the blue lines do not extend to large effective $e$. These two approximations bound the full solution at low effective $e$.

### Approximation for phage mutation rate

The effective phage mutation rate (new phages per minute, $\bar{\mu}$) is given by *Equation 70*. In general, $\bar{\mu}$ depends both explicitly on $e$ and $m$ and implicitly through $n_V$, $n_B$, and $\nu$. We do the same small $e/m$ approximation for $\bar{\mu}$ as above, where we expand $n_V$, $n_B$, and $\nu$ in $e/m$ ($n_V \approx \tilde{n}_V + \frac{e}{m} n_V^1$, etc.). The zeroth order approximation ($e/m = 0$) does not change $\bar{\mu}$ much at all across the whole range of parameters we investigated (*Equation 71* and *Figure 33*).

$$\bar{\mu} = \alpha B(1 - e^{-\mu L}) p_V n_V n_B (1 - \frac{e\nu}{m}) \frac{1}{g C_0} \tag{70}$$

$$\bar{\mu} = \frac{\alpha B(1 - e^{-\mu L}) p_V}{g C_0} \left[ \tilde{n}_V \tilde{n}_B + \frac{e}{m}(n_V^1 \tilde{n}_B + \tilde{n}_V n_B^1 - \tilde{\nu} \tilde{n}_V \tilde{n}_B) \right] + \mathcal{O}(e/m)^2 \tag{71}$$

Approximation for $m$

Finally, we do the same approximation for the complete expression for $m$, expanding all variables in powers of $e/m$. The result turns out to be the same as if we combined the individual approximations shown above.

$$m = \underbrace{\frac{2e\nu}{m(B-1)}}_{\text{phage establishment fraction}} \underbrace{\alpha B(1 - e^{-\mu L}) p_V n_V n_B (1 - \nu \frac{e}{m})}_{\text{phage mutation rate}} \underbrace{\frac{2\frac{n_V}{m}(1 + \ln m)}{B(B-2)\alpha p_V n_B(1 - \nu\frac{e}{m}) + F + \alpha n_B}}_{\text{large phage clone time to extinction}} \tag{72}$$

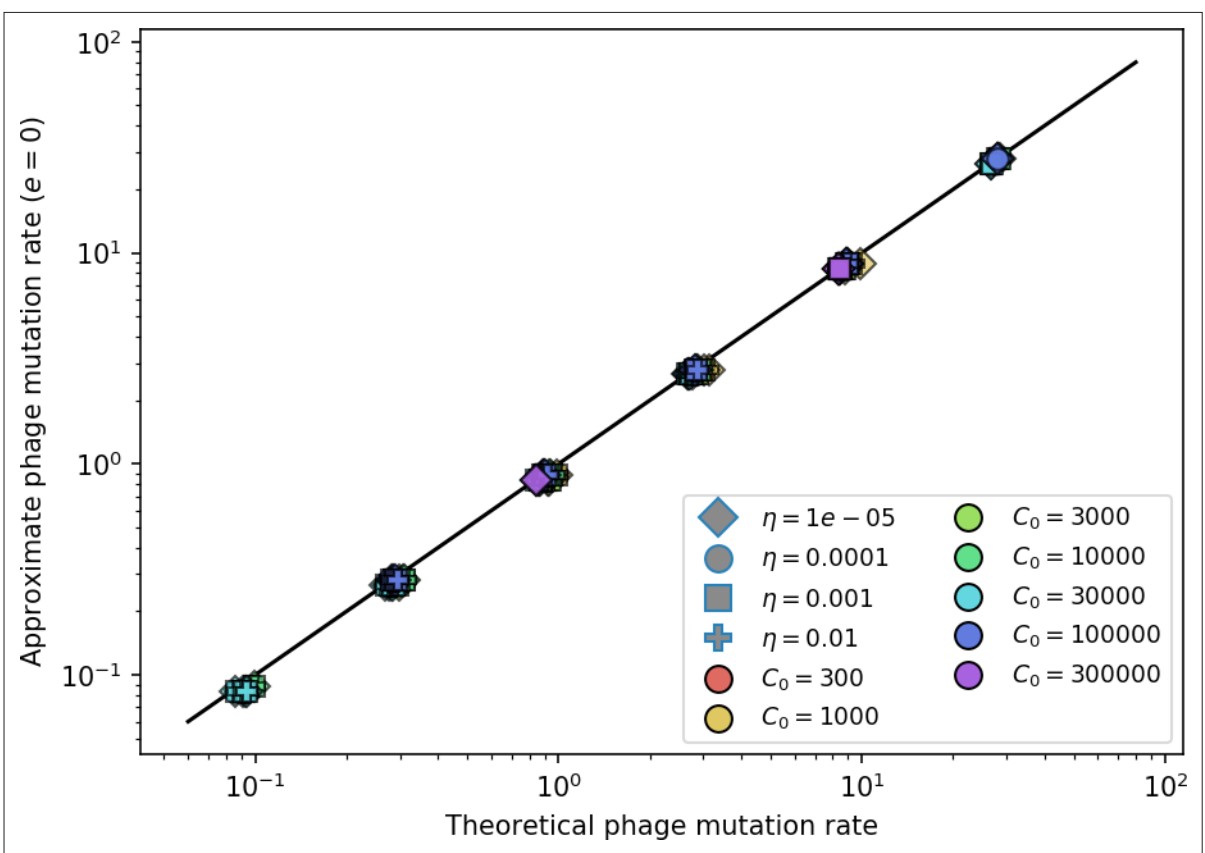

**Figure 33.** Approximate phage mutation rate for $e = 0$ vs theoretical phage mutation rate.

$$m = \frac{4(1-e^{-\mu L})\tilde{\nu}\tilde{n}_V^2}{(B-1)^2}\frac{e(1+\ln m)}{m^2}$$
$$+\frac{4\tilde{n}_V(1-e^{-\mu L})\left(BC_0 fgp_V\left(2(B-1)\tilde{\nu}n_V^1+(B-1)\nu^1\tilde{n}_V+\tilde{\nu}^2(-\tilde{n}_V)\right)+\alpha n_B^1\tilde{\nu}\tilde{n}_V(Bp_V-1)^2\right)}{(B-1)^3 BC_0 fgp_V}\frac{e^2(1+\ln(m))}{m^3}$$
$$+\mathcal{O}(e/m)^3 \tag{73}$$

Let's look at the first term by itself, neglecting $(e/m)^2$ and higher. Rearranging to collect $m$ terms:

$$\frac{m^3}{(1+\ln m)} = \frac{4e(1-e^{-\mu L})\tilde{\nu}\tilde{n}_V^2}{(B-1)^2} \tag{74}$$

Let us let the right-hand side of **Equation 74** equal $a$, a new parameter that is independent of $m$. Now we are approximating:

$$m^3 = a(1+\ln m) \tag{75}$$

Changing variables to

$$z = 1 + \frac{\ln a}{3} + \frac{\ln z}{3} \tag{76}$$

We solve this perturbatively. Let $z_0 = 1 + \frac{\ln a}{3}$, then let $z = z_0(1+\delta)$ where we assume $\delta$ is small. Now we are solving for the perturbation

$$z_0(1+\delta) = z_0 + \frac{1}{3}\ln(z_0(1+\delta)): \tag{77}$$

We approximate $\ln(1+\delta) \approx \delta$ for small $\delta$. This gives

$$\delta \approx \frac{\ln z_0}{3z_0 - 1} \tag{78}$$

$$m \approx \left(az_0\left(1 + \frac{\ln z_0}{3z_0 - 1}\right)\right)^{\frac{1}{3}} \tag{79}$$

$$m \approx \left[\frac{a(\ln(a)+3)\left(\ln(a)+\ln\left(\frac{1}{3}(\ln(a)+3)\right)+2\right)}{3(\ln(a)+2)}\right]^{\frac{1}{3}} \tag{80}$$

If $a$ is large, the leading order contribution to $m$ is $a^{1/3}$.

$$a = \frac{4e\tilde{\nu}\tilde{n}_V^2(1-e^{-\mu L})}{(B-1)^2} \tag{81}$$

Measured $m$ vs. $a$ is shown in **Figure 34**. To get more insight into the dependence of $a$ and $m$ on parameters, we make some approximations. First, since $\mu L << 1$, we approximate $(1 - e^{-\mu L}) \approx \mu L$. Then substituting $\tilde{\nu}$ in $a$:

$$a \approx \frac{4e\mu L\frac{\hat{\eta}\alpha\tilde{n}_V}{\hat{\eta}\alpha\tilde{n}_V+r}\tilde{n}_V^2}{(B-1)^2} \tag{82}$$

Substituting in $\tilde{n}_V$:

$$\tilde{n}_V = \frac{C_0}{\alpha p_V}\frac{(Bp_V-1)(1-f)\alpha - fg}{1+(Bp_V-1)\alpha/g} \tag{83}$$

$$a \approx \frac{4e\mu L\frac{\hat{\eta}\alpha\left(\frac{C_0}{\alpha p_V}\frac{(Bp_V-1)(1-f)\alpha-fg}{1+(Bp_V-1)\alpha/g}\right)}{\hat{\eta}\alpha\left(\frac{C_0}{\alpha p_V}\frac{(Bp_V-1)(1-f)\alpha-fg}{1+(Bp_V-1)\alpha/g}\right)+r}\left(\frac{C_0}{\alpha p_V}\frac{(Bp_V-1)(1-f)\alpha-fg}{1+(Bp_V-1)\alpha/g}\right)^2}{(B-1)^2} \tag{84}$$

Now we do an expansion assuming large burst size, $B >> 1$. Expanding in $1/B$:

$$a \approx \frac{4g^3 C_0^3(1-f)^3 e\eta\mu L(1-p_V)}{\alpha^2 B^2 p_V^2(gC_0(1-f)\eta(1-p_V)+p_V r)} + \mathcal{O}\left(\frac{1}{B}\right)^3 \tag{85}$$

Now if $\eta$ is small, specifically if $p_V r > \eta(1-p_V)gC_0(1-f)$, we can expand in $\eta$. This assumption roughly amounts to assuming bacterial survival followed by spacer acquisition is rare compared to phage predation and spacer loss.

$$a \approx \frac{4g^3 C_0^3(1-f)^3 e\mu L}{\alpha^2 B^2 p_V^3 r}\left[\eta(1-p_V) - \frac{gC_0(1-f)}{p_V r}\eta^2(1-p_V)^2 + ...\right] \tag{86}$$

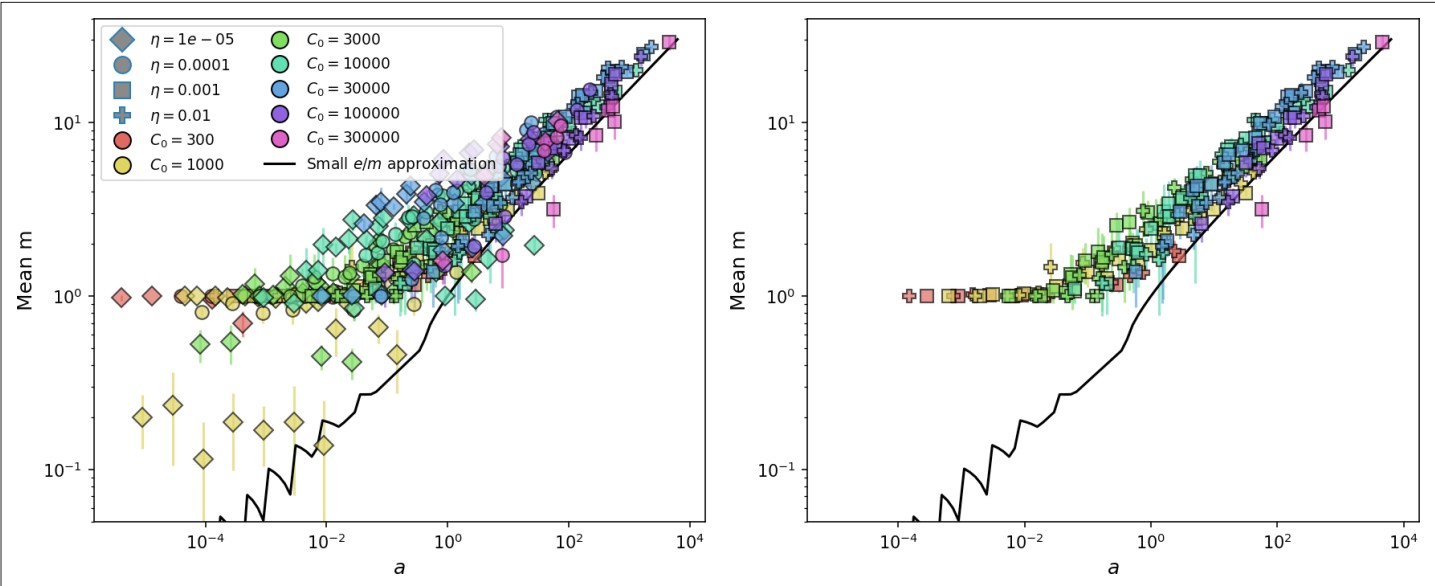

**Figure 34.** Measured mean $m$ in simulations vs. $a$ as given by **Equation 81**. The solid line is $m$ vs. $a$ solved numerically using **Equation 113**. The right panel shows the same but with the two lowest values of $\eta$ removed. Error bars are the standard deviation across three or more independent simulations.

The 0th-order term captures the trend across a wide range of parameters and gives some insight into parameter dependence (**Figure 2C**). To summarize: For $\mu L \ll 1$, $B \gg 1$ and $p_V r > \hat{\eta} g C_0 (1 - f)$:

$$a \approx \frac{4e\mu L\hat{\eta}(gC_0(1 - f))^3}{B^2\alpha^2 p_V^3 r} \tag{87}$$

**Equation 87** implies that $m$ goes like $(e\mu\eta)^{1/3}$, a non-intuitive dependence.

- $m$ increases as the overall phage mutation rate increases.
- $m$ increases as $\nu$ increases.
- $m$ increases for the same reasons that phage time to extinction increases, except that it depends on the total phage population size (instead of the phage clone size).

## Speed of evolution

This section provides more detail on our calculation of the speed of phage evolution presented in 'Dynamics are determined by diversity'.

The speed of evolution of a population is often taken as the rate at which its genetic structure changes over time. For example, Betts et al. measure the rate of evolution in a bacteria-phage coevolution experiment by calculating the Euclidean genetic distance between populations at different time points (**Betts et al., 2018**), and Rouzine and Rozhnova calculate the rate of substitution (**Rouzine and Rozhnova, 2018**). Speed of evolution is sometimes defined more functionally as the rate of change of a population's fitness over time (see **Desai and Fisher, 2007**; **Desai et al., 2007**; **Rouzine and Rozhnova, 2018**).

In our context, the average population fitness does not change over time at steady state (more like a Red Queen scenario than an arms-race scenario) and so we use a measure of genetic distance to calculate the speed of evolution. Instead of Euclidean distance (L2 norm), we use Hamming distance or edit distance (L1 norm). This distance metric is more representative of the process involved in changing sequences since the sequence [1,1] is mutationally twice as far from [0,0] as [0,1]. Our sequence space has as many dimensions as the number of genetic sites, and a population's 'centre of mass' can be calculated by weighting each sequence ('position') in this space by the frequency of each subpopulation with that sequence. Our effective space is dimension $L = 30$ since each protospacer has 30 sites that can mutate. Each site can have the value 0 or 1 meaning there are $2^{30}$ possible sequences, but a sequence can differ by at most 30 changes from another sequence. The maximum distance possible in our model using the L1 norm is 30.

For example: if $L = 2$ and there are 20 phages with the sequence [0,1] and 10 phages with the sequence [1,1], then the centre of mass in the two-dimensional space is $[\frac{1}{3}, 1]$:

$$R_x = \frac{1}{30}(20 \times 0 + 10 \times 1) = 1/3x$$

$$R_y = \frac{1}{30}(20 \times 1 + 10 \times 1) = 1y$$

If the ancestor phage is [0,0], then the centre of mass distance is $1 + 1/3 = \frac{4}{3}$. If the ancestor sequence is at one of the corners (i.e. only values of 0 or 1), calculating the centre of mass in this manner is the same as calculating the population-weighted average distance from the ancestor: 20 phages are 1 mutation away, 10 are 2 mutations away, therefore the average distance is $(20 + 20)/30 = \frac{4}{3}$. However, if the ancestor sequence is not at a corner (i.e. if comparing two centres of mass), the sum of distances gives larger values than the difference of centres of mass. The sum of distances is a measure of the spread, while the change in centre of mass is a measure of the speed of evolution.

We can look at the distance of the phage and bacteria population in simulations from the ancestor sequence over time to get an idea of how the population moves through sequence space. *Figures 35 and 36* show the mutational distance for phage and bacteria over time from either the ancestor phage sequence or a centre of mass starting point later in the simulation, for low (*Figure 35*) and high (*Figure 36*) values of $\eta$.

Since the maximum distance in our model is 30, there comes a point in distance where mutations back in the direction of the ancestor become likely. The fraction of available mutations $f_m$ that move away from the ancestor decreases as the distance increases: $f_m = \frac{L - \text{distance}}{L}$. For $L = 30$, this means that at a distance of 15 the population is equally likely to move towards or away from the ancestor. This is apparent in *Figure 35* – the distance from the ancestor phage reaches 15 and then begins to decrease. In other simulations, the distance continues to increase and does not come close to 15 (*Figure 36*).

## Measuring speed of evolution

The speed of evolution in this framework is the distance the population travels in spacer space in a certain amount of time. In principle, this is straightforward to calculate, but we find that the distance between populations does not increase linearly with the time interval used to calculate the distance (*Figure 37* left panel). This means that measured speed will depend on the time interval used to calculate it (*Figure 37* right panel), and there is no clear way to choose one particular time interval over another. We go to the

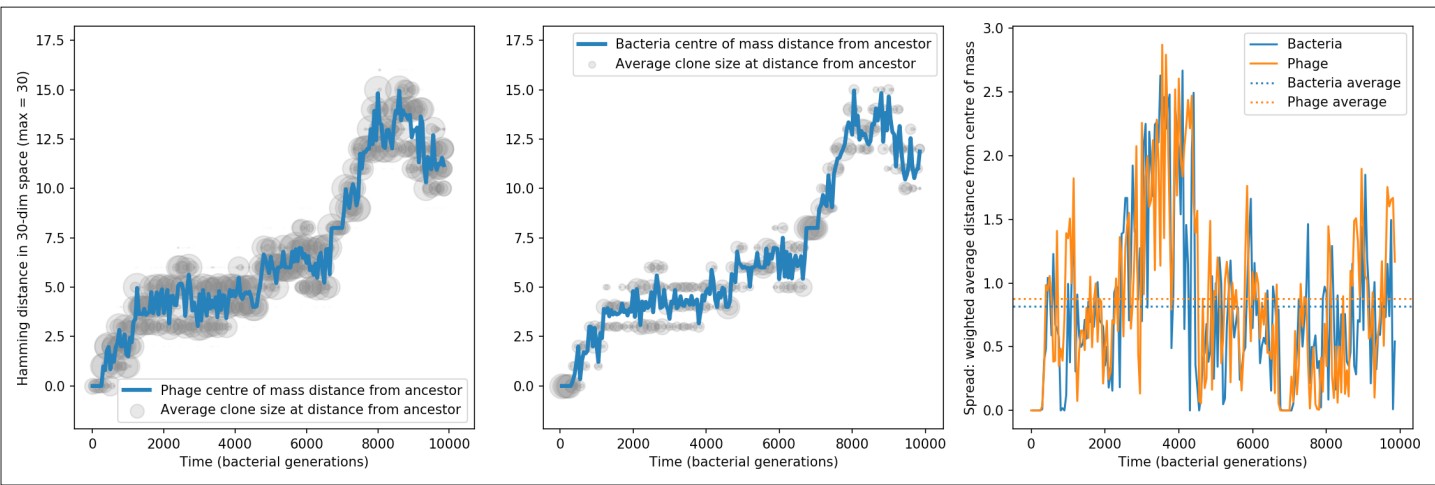

**Figure 35.** Phage and bacteria centre of mass distance from the original phage and bacteria sequences. The centre of mass distance is plotted in blue for phage (left) and bacteria (centre). Grey circles represent the size of clonal subpopulations at each distance from the ancestor sequence (arbitrary scale). The third panel shows the weighted average distance of the population from the centre of mass at that timepoint, a measure of the spread in sequences present at any time. In this simulation $C_0 = 10^4$, $\eta = 10^{-5}$, $\mu = 10^{-6}$, $e = 0.95$, and mean $m = 2.8$.

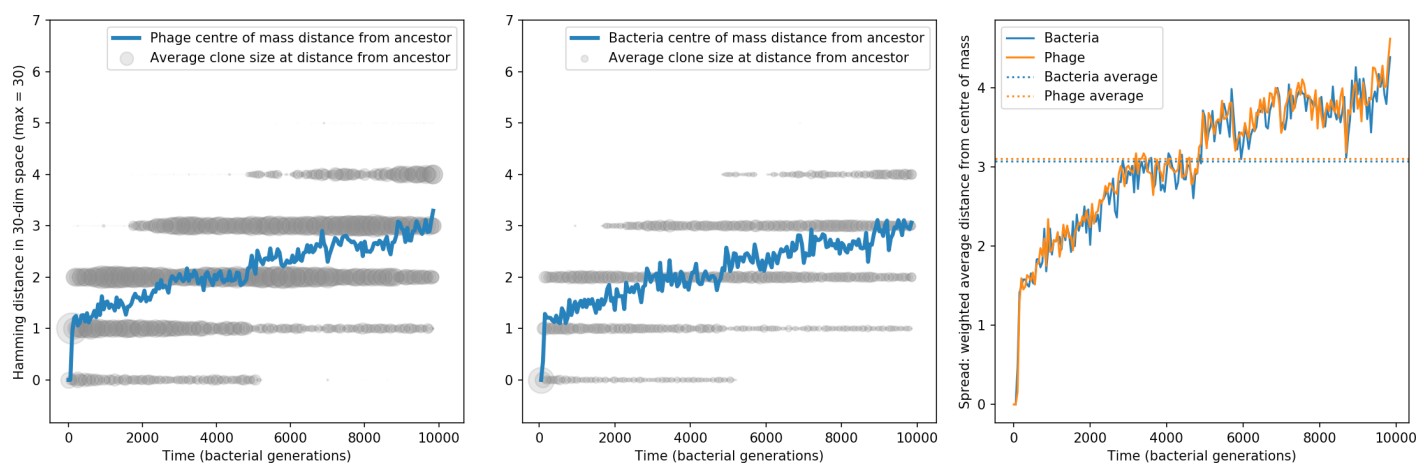

**Figure 36.** Phage and bacteria centre of mass distance from the original phage and bacteria sequences. The centre of mass distance is plotted in blue for phage (left) and bacteria (centre). Grey circles represent the size of clonal subpopulations at each distance from the ancestor sequence (arbitrary scale). The third panel shows the weighted average distance of the population from the centre of mass at that timepoint, a measure of the spread in sequences present at any time. In this simulation $C_0 = 10^4$, $\eta = 10^{-2}$, $\mu = 10^{-6}$, $e = 0.95$, and mean $m = 14.5$.

long $\Delta t$ limit and measure the maximum distance from the ancestor reached by a population in a set number of bacterial generations. The maximum distance reached by bacteria and phage populations is highly correlated with each other within a simulation and is also repeatable across independent simulations (**Figure 38**). Speed in this definintion increases with phage mutant initial fitness (**Figure 39**).

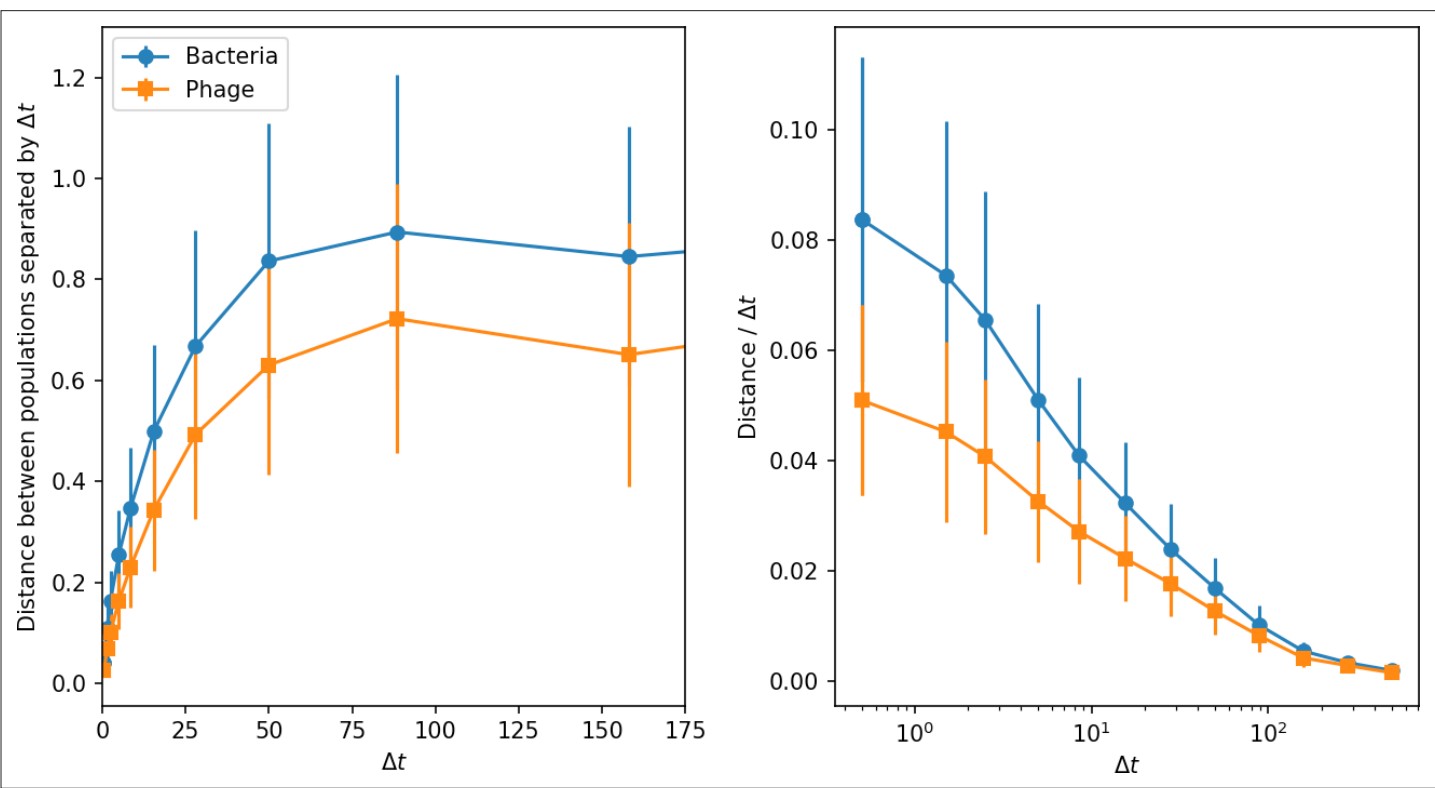

**Figure 37.** Phage and bacteria centre of mass distance from the centre of mass at time $t - \Delta t$ (left) and the distance divided by the time interval $\Delta t$ (right). Distances are averaged over the entire simulation at steady state; error bars are standard deviation. Simulation parameters are $C_0 = 10^4$, $\eta = 10^{-2}$, $\mu = 10^{-6}$, $e = 0.95$, and mean $m = 14.5$.

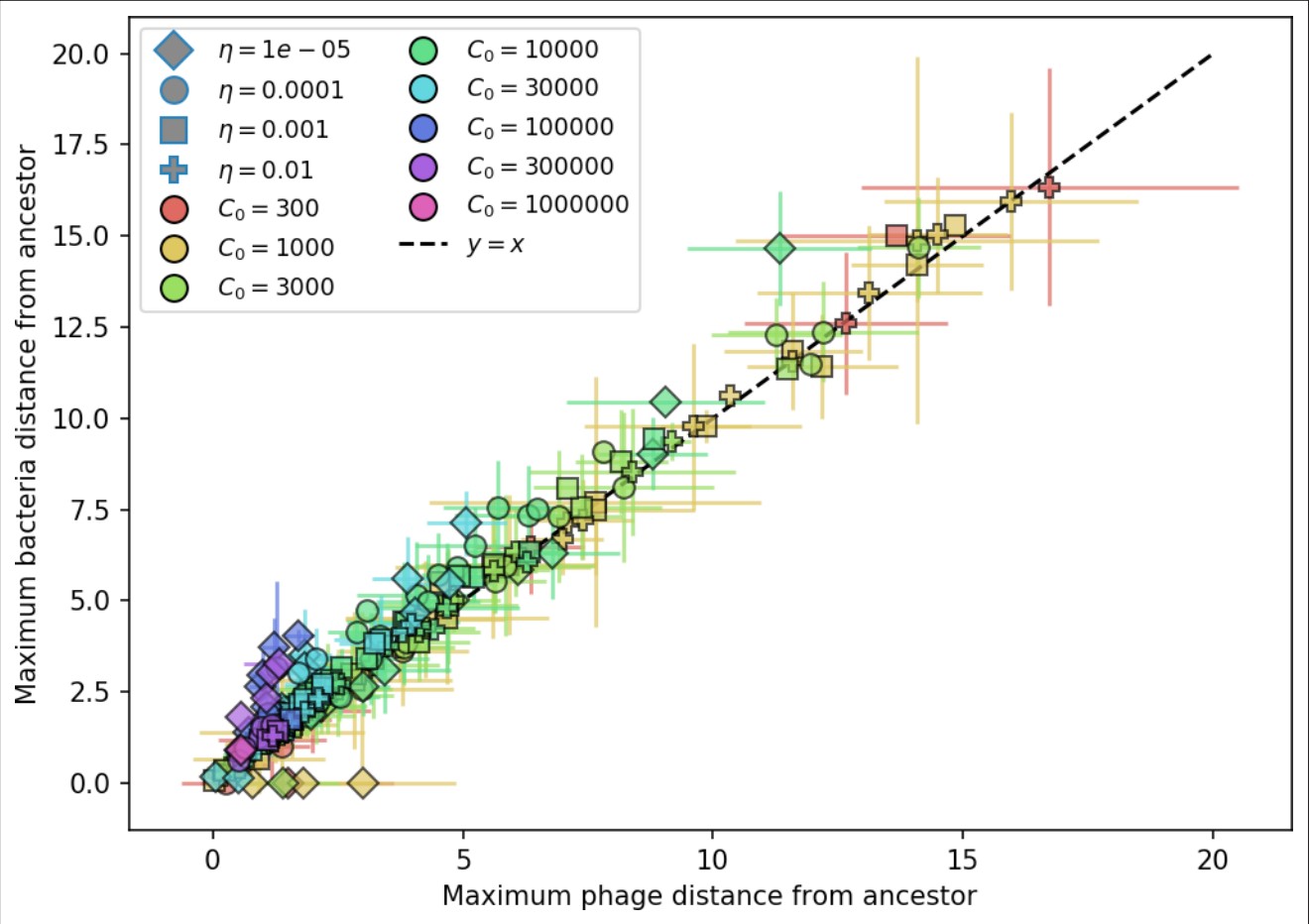

**Figure 38.** Maximum distance from ancestor population for bacteria vs. phage. The maximum distance is highly correlated, indicating that the bacteria population tracks the phage population closely. Error bars are the standard deviation across three or more independent simulations.

The total number of phage establishments in a simulation determines the maximum distance that the population can move away from the ancestor, but if diversity is high, many of those establishments will not actually move the population further away. A simple approximate scaling for this process is that total distance is proportional to $\frac{1}{m}$: if there are $m$ established clones and one of them is the furthest away from the ancestor, there is a $\frac{1}{m}$ probability that the next establishment will come from the furthest-away clone. This is similar to *Childs et al., 2012* where if they have $v$ protospacers per phage (and one dominant bacterial species with a spacer), there is only a $1/v$ chance that a random mutation will actually be an escape mutation.

Since $m$ is a product of mutation rate, establishment probability, and extinction time, plotting distance per establishment vs. $m$ is equivalent to plotting speed vs. time to extinction (*Equation 88*). We have $m = P_{\text{est}}\bar{\mu}T$, where $P_{\text{est}}$ is the probability of establishment for a new phage clone, $\bar{\mu}$ is the effective phage mutation rate, and $T$ is the mean time to extinction for large phage clones. Let the mutational distance from the ancestor be $\Delta = v\tau$, where $v$ is the 'speed' and $\tau$ is a fixed time interval used to measure distance.

$$
\begin{aligned}
\frac{\Delta}{P_{\text{est}}\bar{\mu}\tau} &\propto \frac{1}{m} \\
\frac{v\tau}{P_{\text{est}}\bar{\mu}\tau} &\propto \frac{1}{P_{\text{est}}\bar{\mu}T} \\
v &\propto \frac{1}{T}
\end{aligned}
\tag{88}
$$

*Figure 6—figure supplement 2* shows distance per establishment vs. $m$ and speed vs. $T$. Relating distance and speed to the time to extinction gives further insight into the underlying parameter dependencies: we know from other approximations ('Analytic approximations for diversity') that $T \propto \tilde{n}_V \frac{1+\ln m}{m}$,

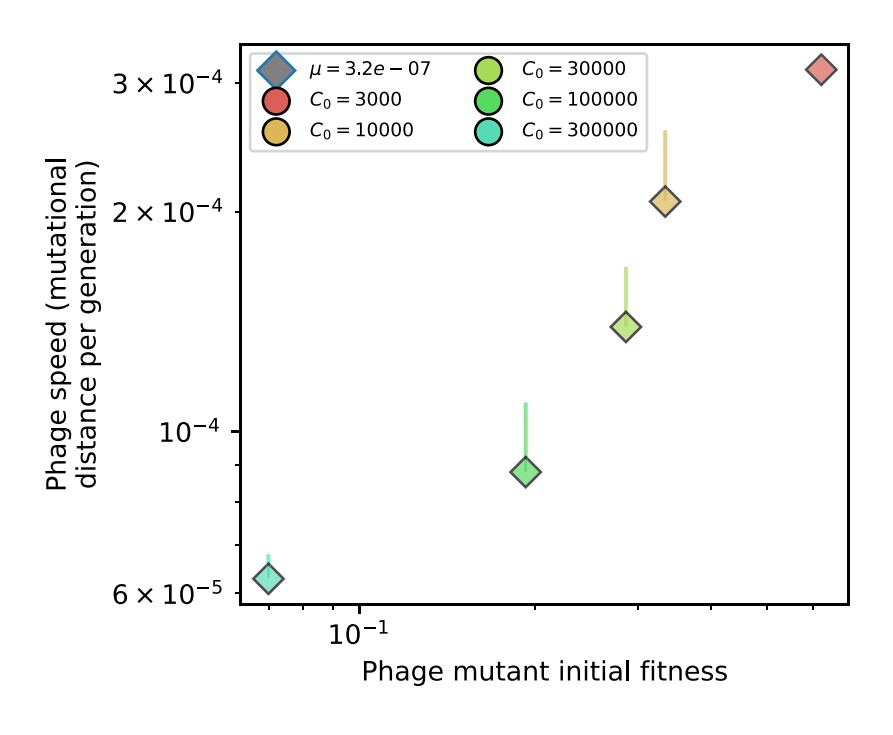

**Figure 39.** Phage mutational distance per generation vs. initial phage mutant fitness for simulations with $e = 0.95$, $\eta = 10^{-3}$, and $\mu = 3 \times 10^{-7}$. Error bars are the standard deviation across multiple independent simulations and are shown in the positive direction only.

and for small $m$ we can substitute our approximation for $m$, giving $T \propto \frac{1}{e\mu\eta}^{\frac{1}{3}}$ and $v \propto e\mu\eta^{\frac{1}{3}}$. Applying the same approximation here as for $m$ above (**Equation 87**), with $T_{ext} \approx \frac{2\tilde{n}_V}{f(B-1)m}\left(1 - \frac{1}{Bp_V}\right)$ (**Figure 6**):

$$v \approx \alpha B f \left( \frac{e\mu\eta L(1 - p_V)}{2\alpha^2 B^2 r} \right)^{\frac{1}{3}} \tag{89}$$

## Spread in sequence space

We can ask how spread out the population is in sequence space – how far is the average clone from the centre of mass? When there are more clones (larger $m$), the population spread is larger (**Figures 40 and 41**). The spread also depends on initial conditions, at least on the timescale of our simulations: simulations that start with one phage clone (**Figure 40**) have lower spread than simulations that start with 50 phage clones (**Figure 41**). In principle, waiting for a very long time in simulations that begin with 1 clone should produce results in line with simulations that start with 50 clones as different clans diverge from each other over time. This suggests that there are multiple features with which we could define steady state, and they do not all reach true steady state at the same time. Total population size equilibrates fastest, followed by diversity (**Figure 8**), and then spread in sequence space.

## Number and size of clone clans

Watching a simulation visualization of spacer and protospacer types, it is apparent that over time several different lineages may appear and become separated from each other in genome space (**Figure 42**). What determines how many separate 'clans' there are at steady state? What is the characteristic size of a clan? How does this depend on parameters? To address this question, we perform agglomerative clustering on protospacer and spacer sequences to identify separated groups of sequences. We use the L1 norm to define distance, which as described above is the most natural distance metric for sequence changes by mutation. Clusters are grouped using the 'single' linkage criterion in scikit-learn AgglomerativeClustering: the cluster distance is the minimum of the distances

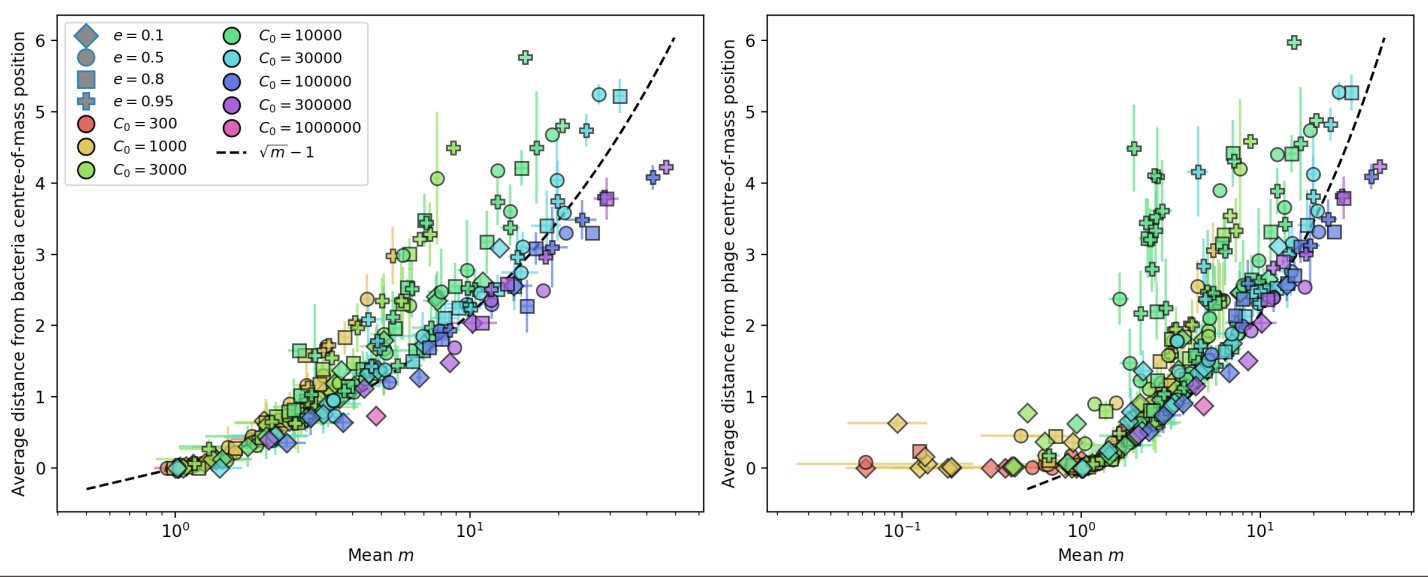

**Figure 40.** Average population distance from the centre of mass at steady state for bacteria (left) and phages (right) vs. mean $m$ for simulations with one original phage clone ancestor. The dashed line is $\sqrt{m}-1$, a purely phenomenological choice. Error bars are the standard deviation across three or more independent simulations.

between all observations of the two sets. In other words, a collection of sequences that differ from any other sequence in the cluster by one mutation are all grouped together, even if some of the sequences differ by more than one from each other. This is a reasonable criterion for identifying groups of sequences that are related by descent and are actively feeding each other with mutations.

*Figures 43 and 44* show the resulting dendrogram from this clustering process for phages and bacteria at one time point for one initial phage clone and 10 initial phage clones. To determine the number of clusters, we use a distance threshold of 2. This separates groups that have no members closer than two mutations away. In reality, these groups may still be linked by descent more recently than other groups (apparent in the dendrograms), but a distance of 2 or more means these groups are likely to remain separate going forward in time since they lack a connecting clone that can be accessed by a single mutation.

Simulations with an initial number of clones greater than 1 converge to a steady-state number of clusters at a different rate than beginning with one clone since clones from a single ancestor remain more related on average than clones from multiple different ancestors (*Figures 45 and 46*).

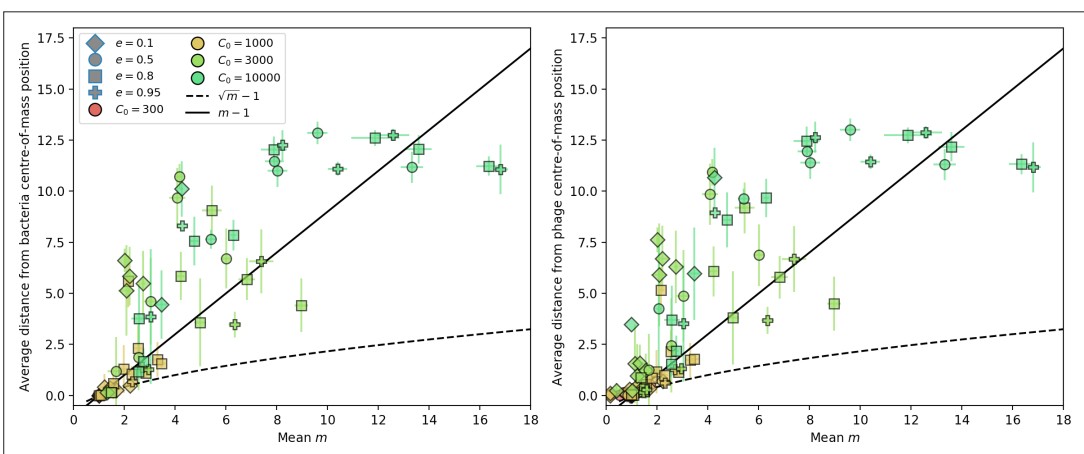

**Figure 41.** Average population distance from the centre of mass for bacteria (left) and phages (right) vs. mean $m$ for simulations with 50 original phage clones. The dashed line is $\sqrt{m}-1$ and the solid line is $m-1$. Error bars are the standard deviation across three or more independent simulations.

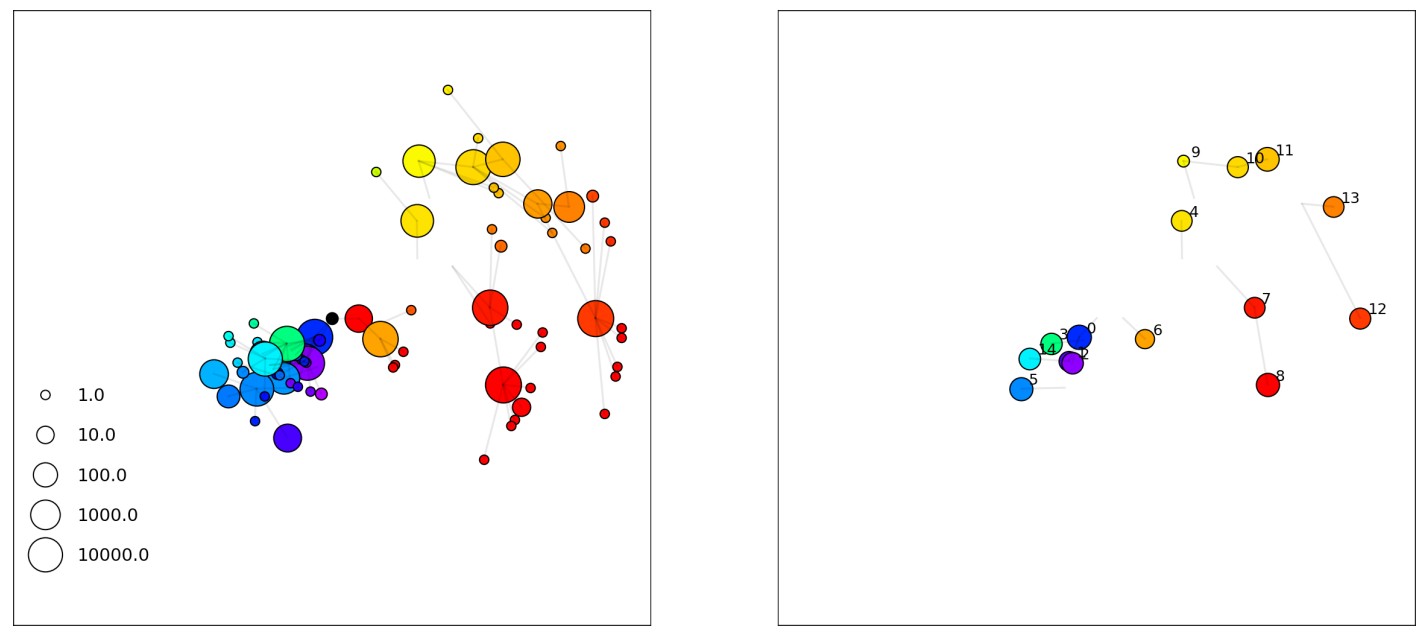

**Figure 42.** A frame from a simulation movie at 5000 generations with $C_0 = 10^4$, $\eta = 10^{-3}$, $\mu = 10^{-5}$, $e = 0.95$, initial $m = 1$. Phages are on the left, bacteria with spacers on the right.

We calculate the average number of clans at steady state and the average clan size for bacteria and phages across all simulations. The number of clans is proportional to $m$; at high mutation rates the number of clans is approximately $m/2$. Low mutation rates mean that it takes a long time for the number of clans to equilibrate, which can be seen by contrasting the mutation rate dependence of 10 initial clones (*Figure 47*) with 1 initial clone (*Figure 48*).

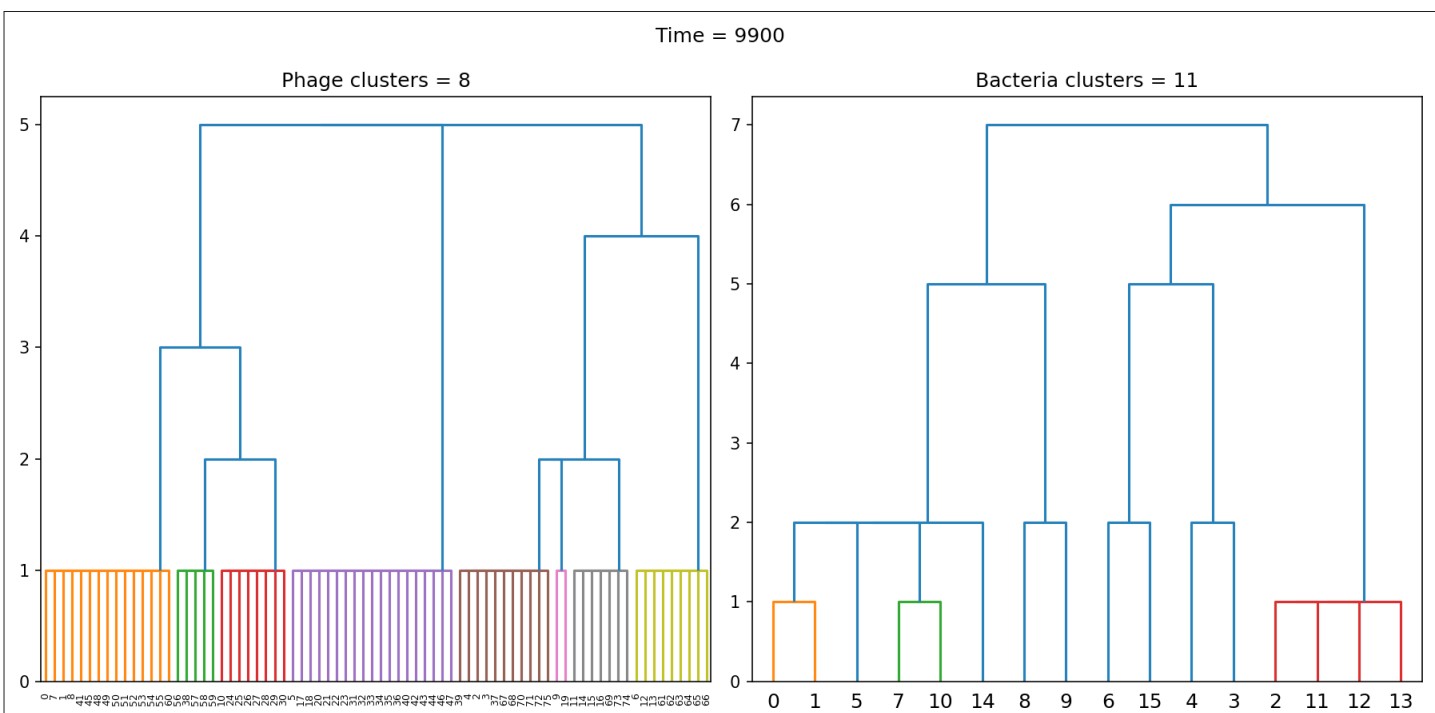

**Figure 43.** Dendrogram resulting from agglomerative clustering with the L1 norm and linking clusters using the minimum distance between members. The number of clusters is determined with a cutoff at a distance of 2. $C_0 = 10^4$, $\eta = 10^{-3}$, $\mu = 10^{-5}$, $e = 0.95$, initial $m = 1$.

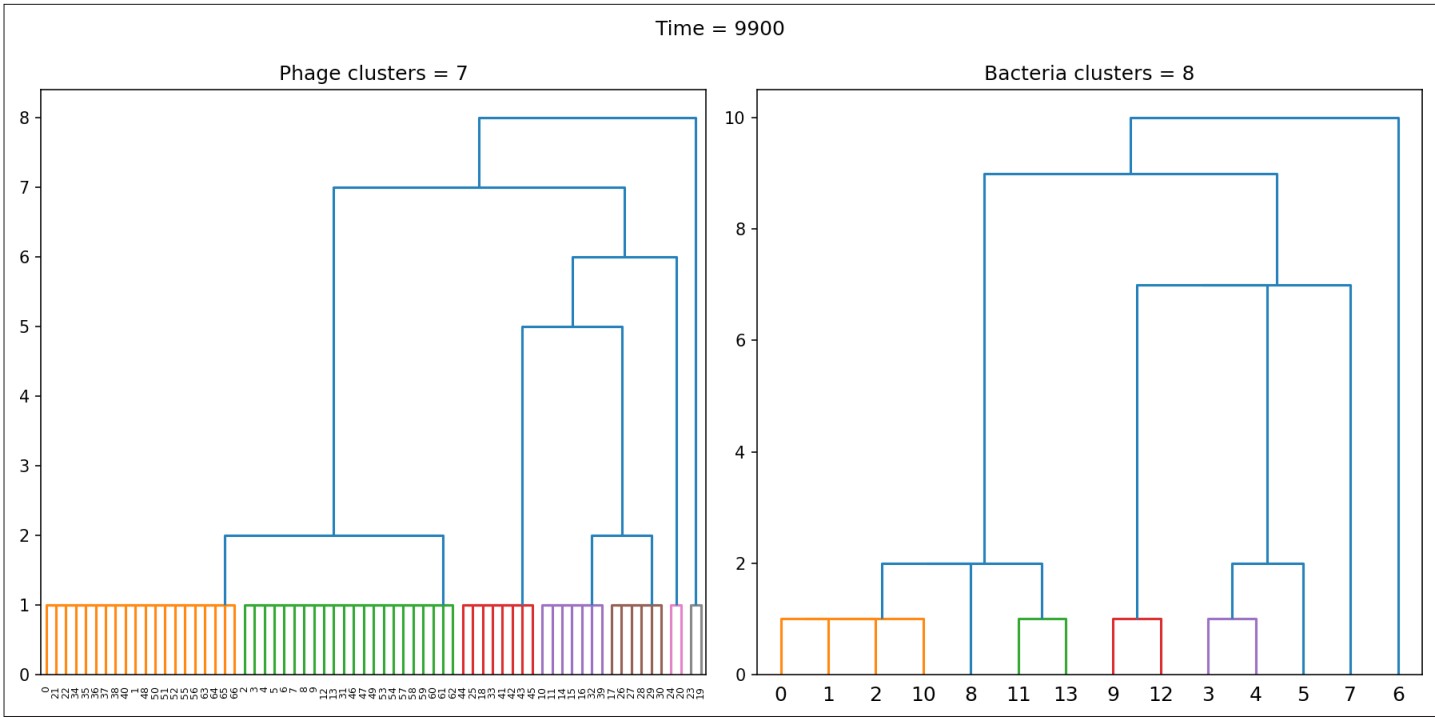

**Figure 44.** Dendrogram resulting from agglomerative clustering with the L1 norm and linking clusters using the minimum distance between members. The number of clusters is determined with a cutoff at a distance of 2. $C_0 = 10^4$, $\eta = 10^{-3}$, $\mu = 10^{-5}$, $e = 0.95$, initial $m = 10$.

The regimes of low and high clan number in our model are similar to regimes of virus-host coevolution identified in *Marchi et al., 2019*: at low virus mutation rates and low mutational jump distances, trajectories move in a straight line in antigenic space, while at higher jump distances, trajectories become more diffusive, the way ours look. At even higher mutation rates and jump sizes, trajectories split into multiple stable coexisting lineages, which is what we see at high diversity.

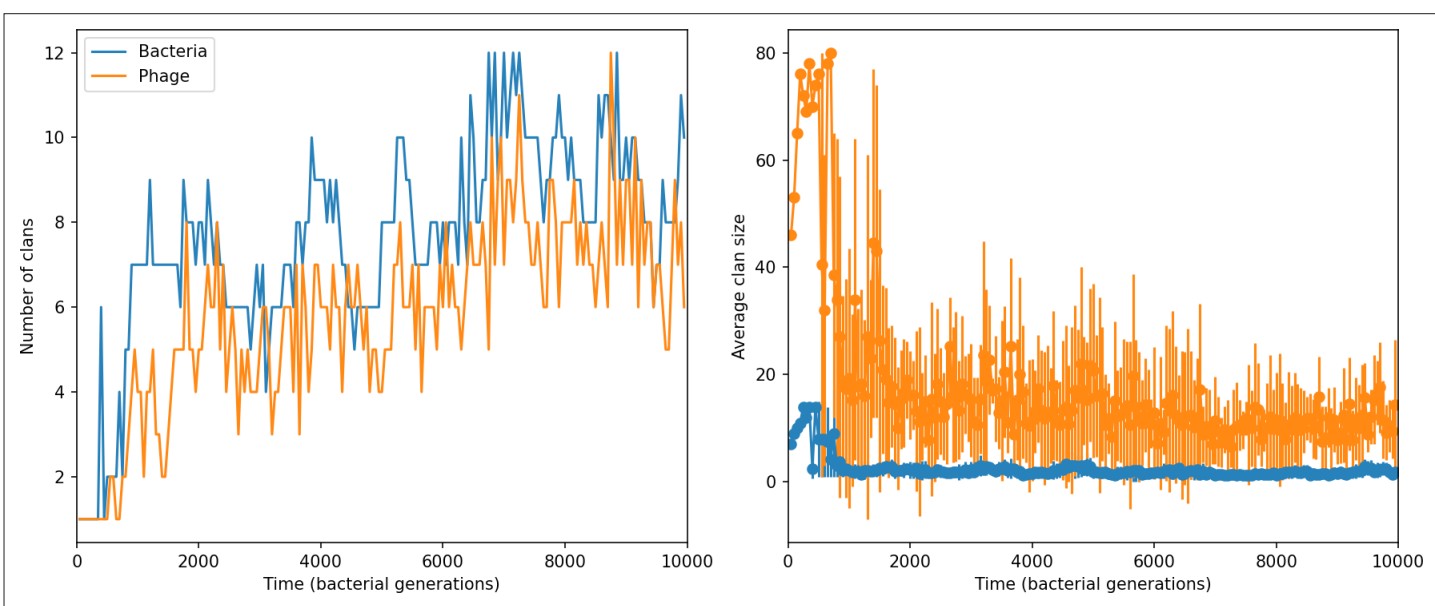

**Figure 45.** Clan number and size over time in a simulation with $C_0 = 10^4$, $\eta = 10^{-3}$, $\mu = 10^{-5}$, $e = 0.95$, initial $m = 1$.

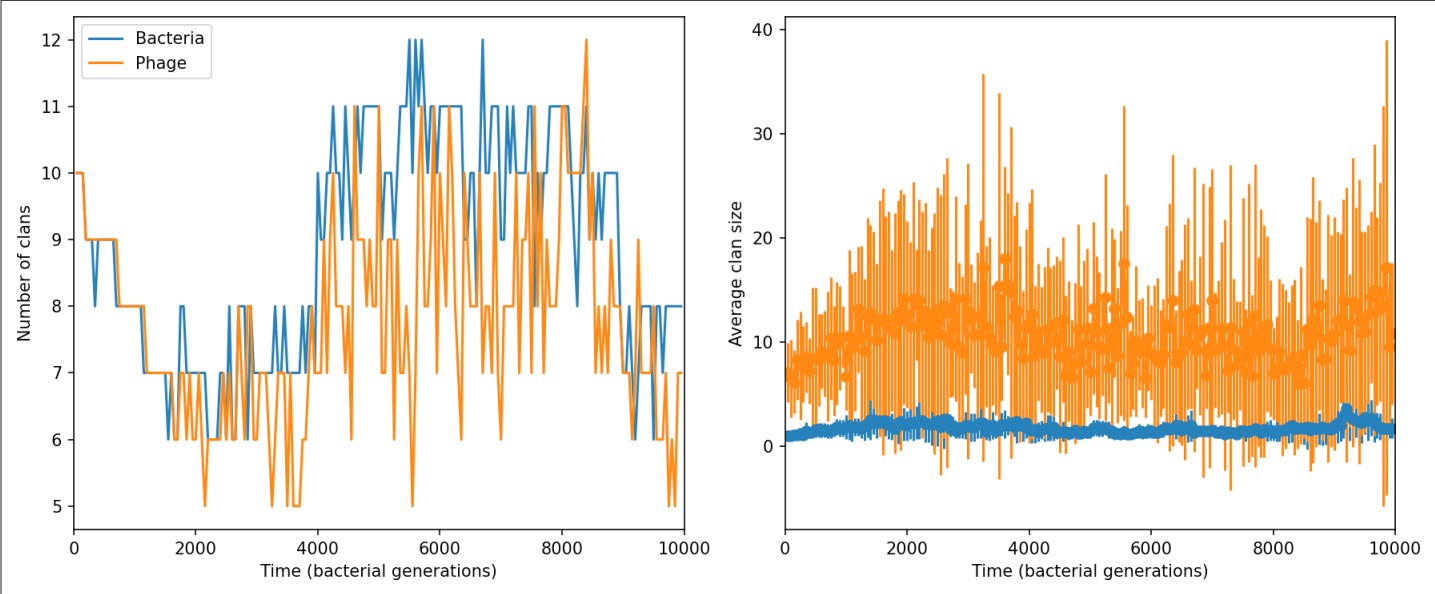

**Figure 46.** Clan number and size over time in a simulation with $C_0 = 10^4$, $\eta = 10^{-3}$, $\mu = 10^{-5}$, $e = 0.95$, initial $m = 10$.

Based on their parameters and regimes, we expect that increasing cross-reactivity relative to mutation rate in our model should bring our results closer to the ballistic regime.

## Cross-reactivity

In this section, we present additional methods and results for simulations with two types of cross-reactivity relating to results in 'Cross-reactivity leads to dynamically unique evolutionary states'.

In our standard simulations and theory, we assume that $p_V(i,j)$ is binary (*Equation 6*): any mismatch between protospacers and spacers means bacteria no longer have any immune advantage against phage. Here, we discuss results of simulations where $p_V$ includes some cross-reactivity so that the more mutations a protospacer contains, the less immunity bacteria have against it.

We explore two types of cross-reactivity: one in which we define $p_V(i,j)$ as an exponential function of the mutational distance between a spacer and protospacer (*Equation 7*), and one in which $p_V(i,j)$

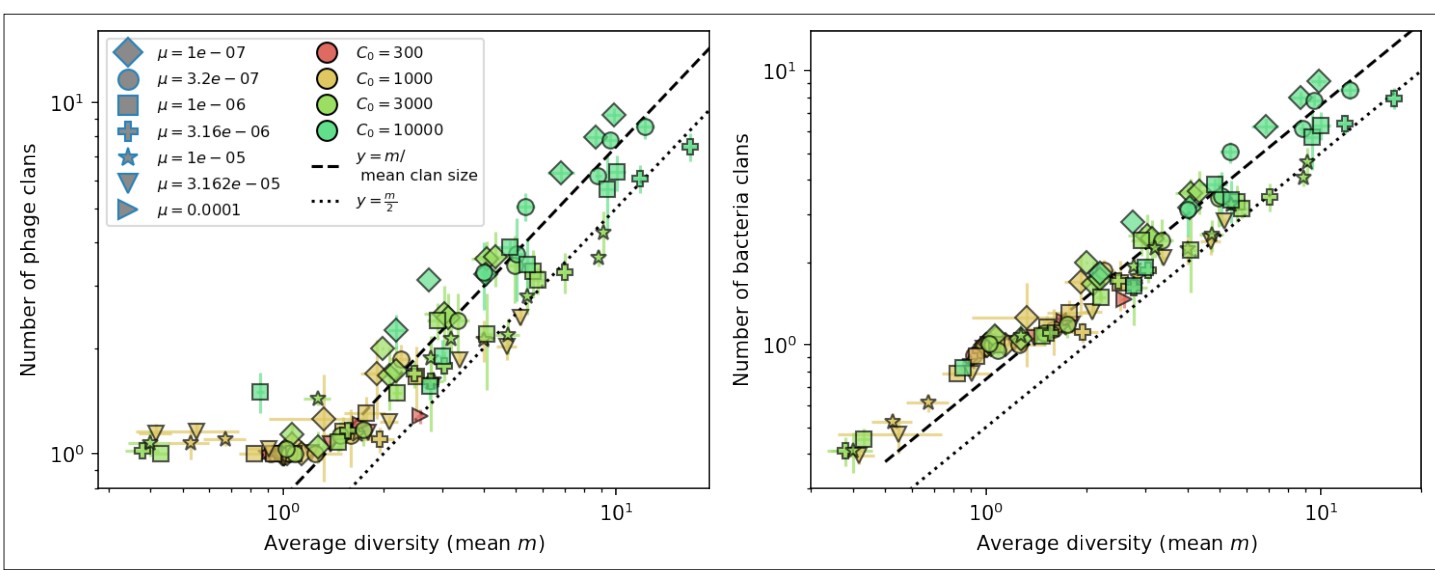

**Figure 47.** Average clan number vs. $m$ for all simulations that begin with 10 clones. The dashed lines are $m$ divided by the mean bacterial clan size ($\approx 1.3$) and $m/2$. Error bars are the standard deviation across three or more independent simulations.

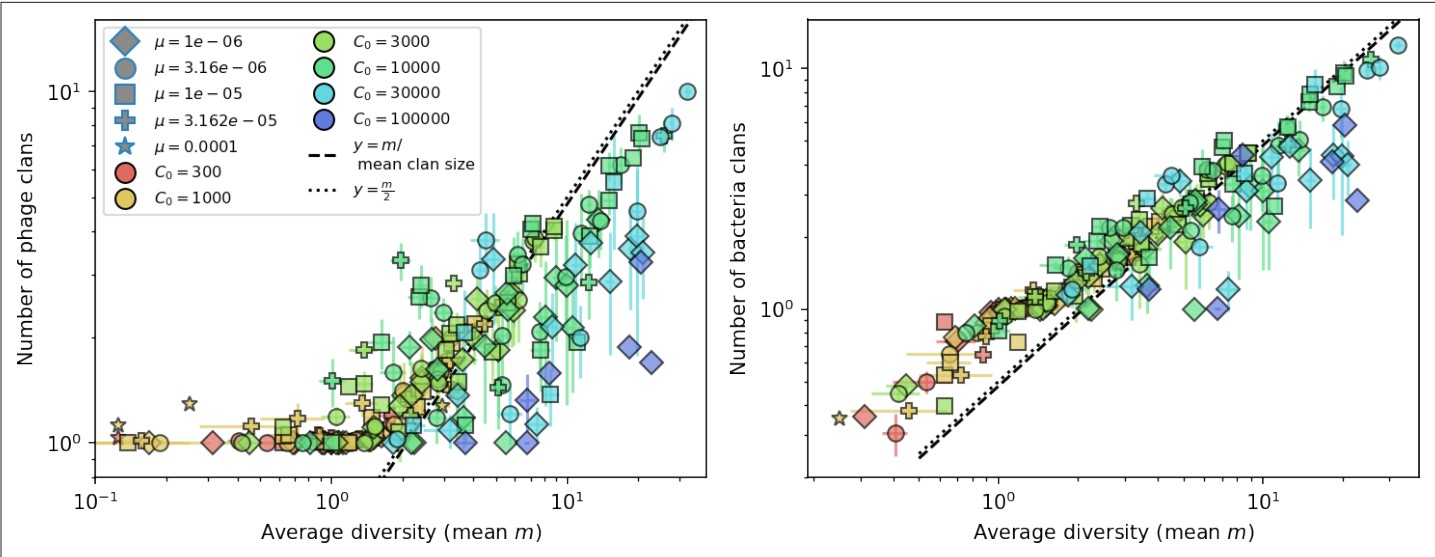

**Figure 48.** Average clan number vs. $m$ for all simulations that begin with one clone with $\mu \geq 10^{-6}$. The dashed lines are $m$ divided by the mean bacterial clan size and $m/2$. Error bars are the standard deviation across three or more independent simulations.

is a $\theta$-function of mutational distance (*Equation 8*). The exponential definition of cross-reactivity is the same as that used in *Marchi et al., 2019*; *Yan et al., 2019* to model viruses evolving in abstract antigen space (*Marchi et al., 2019*) and in genomic distance space (*Yan et al., 2019*). Note that our usual binary definition of $p_V$ is a special case of *Equations 7 and 8* with $\theta = 0$. *Figure 49* shows these different definitions of $p_V(i,j)$ as a function of $d_{ij}$.

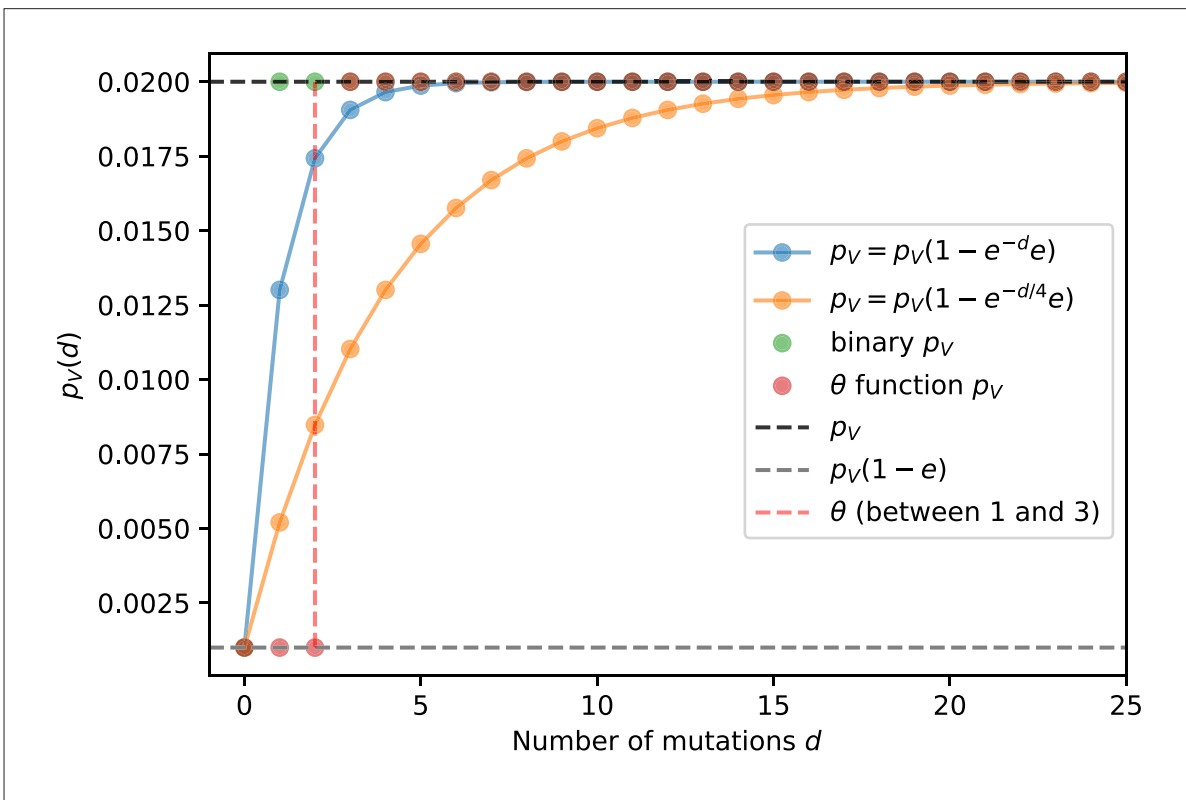

**Figure 49.** Phage infection success probability $p_V$ as a function of mutational distance between spacers and protospacers for different definitions and degrees of cross-reactivity. Definitions are plotted for $p_V = 0.02$, $e = 0.95$.

Cross-reactivity leads to a higher average bacterial immunity against phages for the same parameters (*Figures 50 and 51*). This means that our assumption that effective $e \approx e/m$, valid for nearly all parameters without cross-reactivity, breaks down when we introduce cross-reactivity. Importantly, this is a quasi-steady-state effect: if we start simulations with more initial clones, clone clans are genetically separated and both mean $m$ and average immunity converge to the binary $p_V$ case (*Figure 50* right panel).

Phages are less likely to establish when there is cross-reactivity because phage escape mutations are incomplete (*Figure 5A*). However, the clones that do survive grow quicker: simulations with high cross-reactivity appear to have a faster mean growth rate for clones, regardless of conditioning on survival (*Figure 51*).

Adding cross-reactivity causes some unexpected dynamical behaviours (*Figure 52*, *Figure 53*). We observed a travelling wave regime in some simulations with large spikes in phage clone size (*Figures 54 and 55*). This happens because cross-reactivity creates a fitness gradient for new phage mutants such that some mutants are much more fit and establish much more quickly.

To see this happening, let us look closely at the simulation in *Figure 54* at time 3565 (*Figure 56*). There are three large phage clones and two large bacteria clones (soon to be three). Phage clone 0 is the largest and phage clone 2 is the smallest. Phage clone 0 and phage clone 1 are both either 0 or 1 mutation away from bacteria clones 0 and 1 (*Figure 57*). Because $\theta = 1$, they are all equally fit; effectively they could be considered one large clone. Phage 2, however, is 1 mutation from bacteria clone 1 but 2 mutations from bacteria clone 0, so it now has a much higher fitness than the other two phage clones and grows quickly. But now bacteria clone 1 is also fit against this new mutant, so it too continues growing. This push-pull keeps going with new phage mutants experiencing runaway fitness for a while. This also leads to an asymmetry in clone identity between the populations: because a bacterial clone may be immune to a phage clone with a different but related sequence, we see

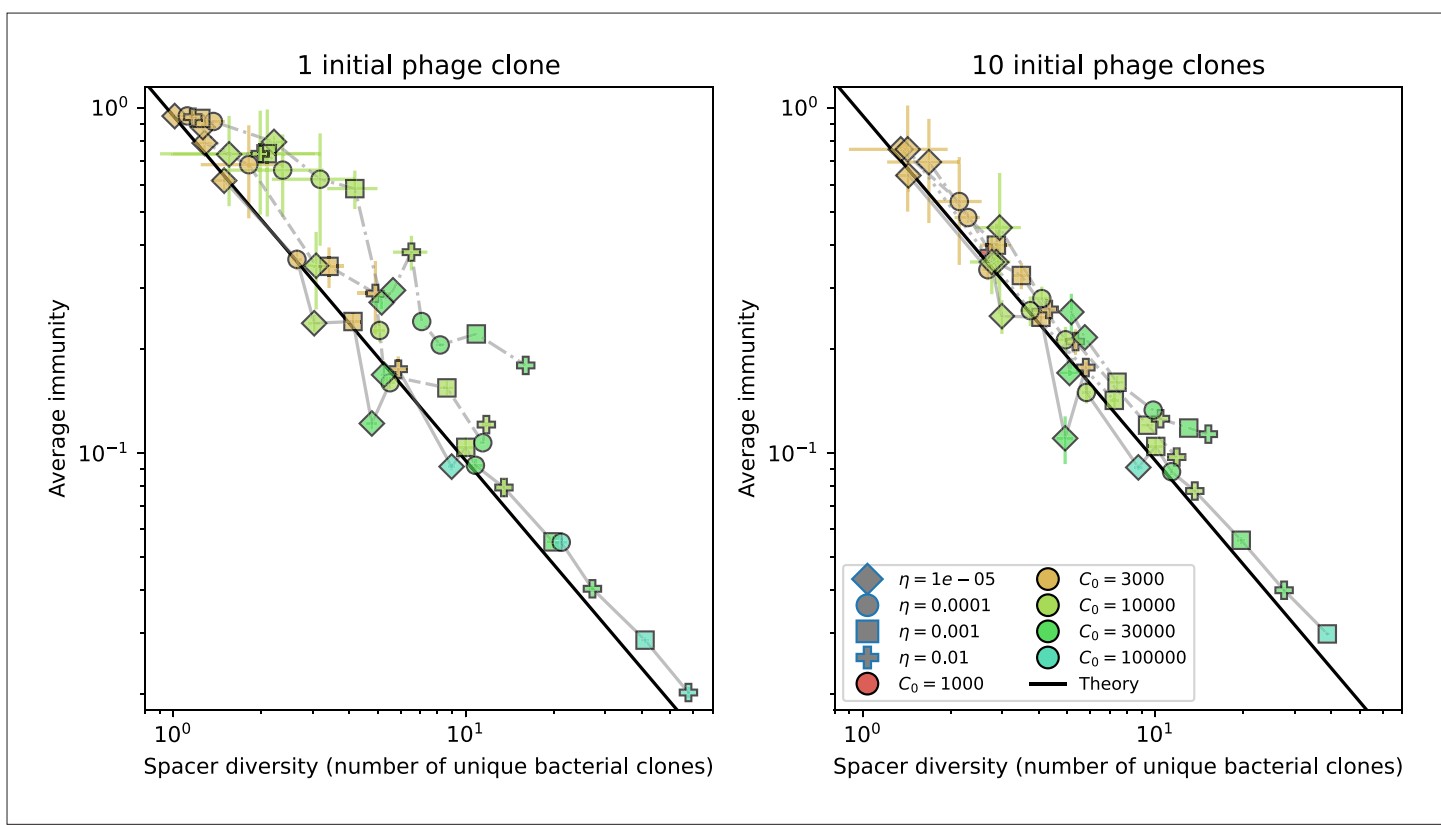

**Figure 50.** Average immunity vs. diversity with different degrees of cross-reactivity for simulations with $e = 0.95$, $\mu = 10^{-6}$. Dashed lines are simulations with cross-reactivity, solid line is simulations without cross-reactivity. Error bars are the standard deviation across three or more independent simulations.

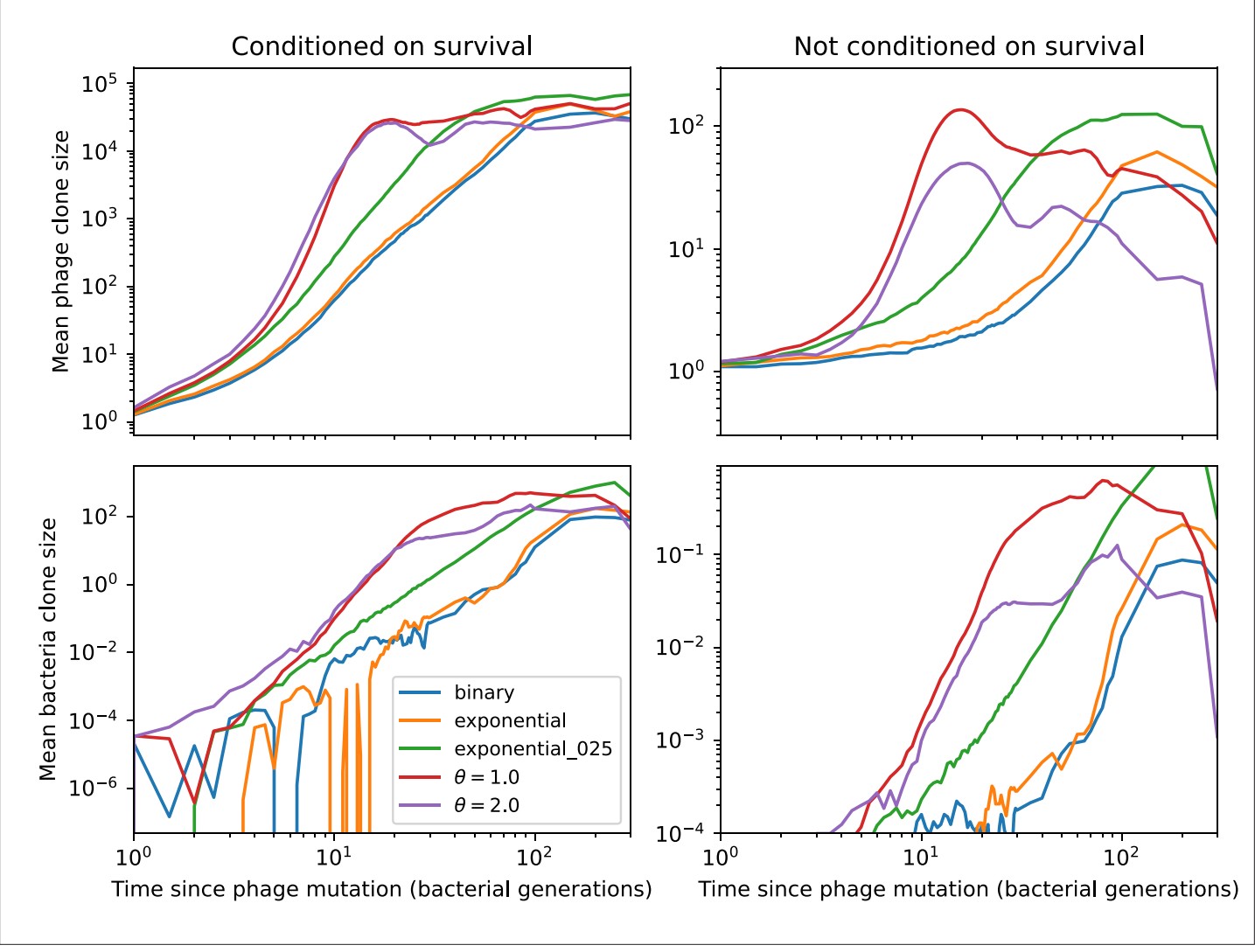

**Figure 51.** Mean phage clone size (top) and mean bacteria clone size (bottom) relative to time of phage mutation for different definitions of $p_V$. These simulations begin with one initial phage clone and parameters $C_0 = 10^4$, $\mu = 10^{-6}$, $\eta = 0.0001$, $e = 0.95$.

bacteria clones becoming large *after* their matching phage clone has gone extinct because they are still resistant to an existing phage clone that is genetically related (**Figures 58 and 59**).

This fitness and growth pattern can be seen as a function of the marginal immunity of bacterial clones to the phage population and of the bacterial population to individual phage clones (**Figure 59**). Average immunity for individual clones, which we call marginal immunity in analogy with **Nourmohammad et al., 2016**, captures the fitness differences between individual clones that determine their dynamical behaviour. Without cross-reactivity, bacteria clones are only immune to at most one phage clone at a time, though their overall marginal immunity still changes smoothly as the matching phage clone changes size (**Figure 60B**). When cross-reactivity is added, bacteria may be immune or partially immune to multiple phage clones, and calculating marginal immunity summarizes all of these combined effects on the fitness of a particular clone. Bacteria clones grow larger once their marginal immunity is high, and phage clones grow large once bacterial marginal immunity against them is low. These opposing forces generate Lotka-Volterra oscillations even in the case with no cross-reactivity, though these oscillations are more rapid and persistent when cross-reactivity is added (**Figure 60C and D**). Phage clones grow quickly when bacteria have low marginal immunity to their sequence, while bacteria clones grow quickly when they have high marginal immunity.

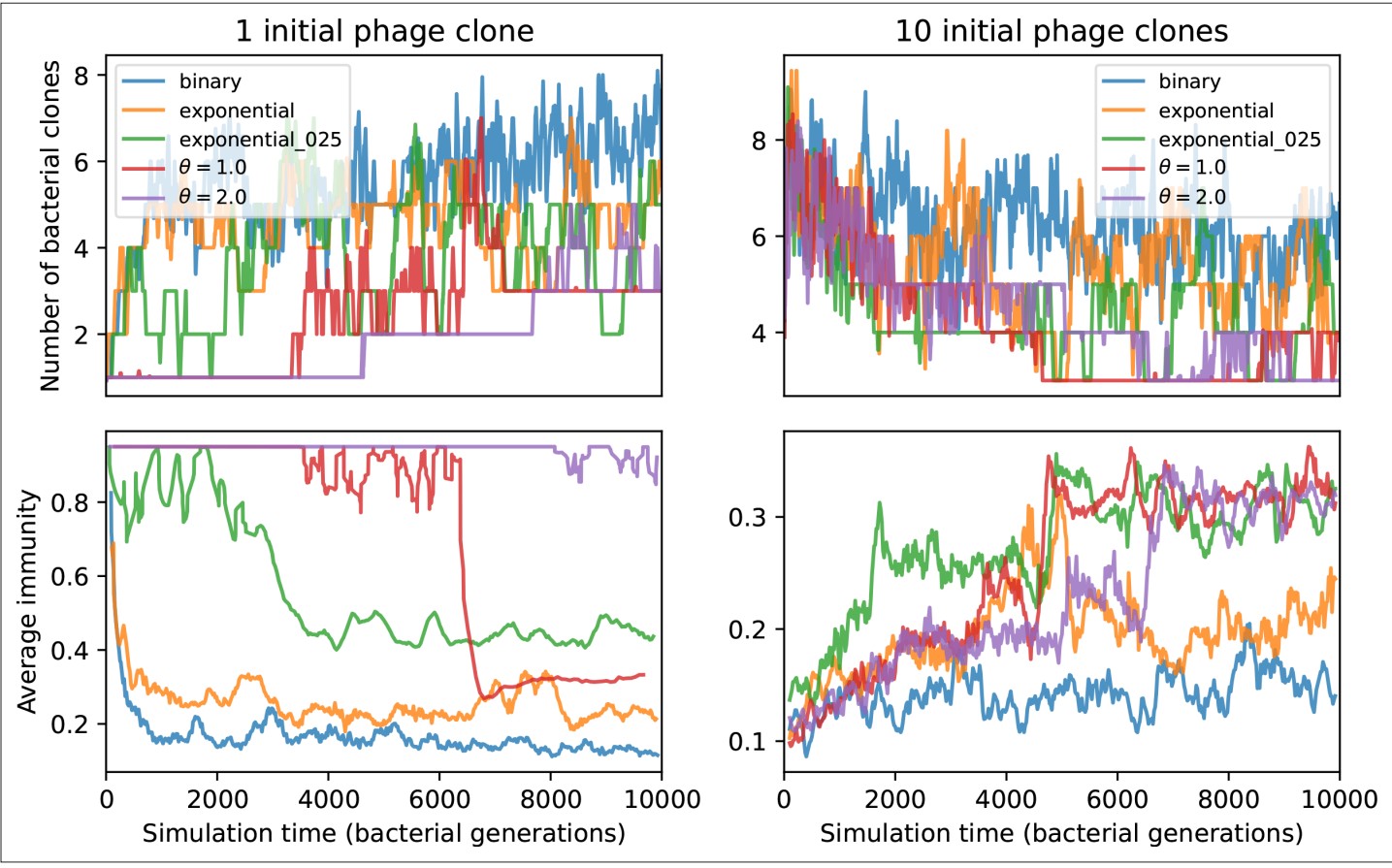

**Figure 52.** Number of bacterial clones (top) and average immunity (bottom) for simulations beginning with either 1 phage clone (left) or 10 phage clones (right). These simulations have parameters $C_0 = 10^4$, $\mu = 10^{-6}$, $\eta = 0.0001$, $e = 0.95$.

Clones grow more quickly on average in the travelling wave regime than in other regimes in the same simulation (*Figure 58*).

In *Figure 4*, we show phylogenies of four simulations with different amounts and types of cross-reactivity. We used DendroPy *Sukumaran and Holder, 2010* and Toytree *Eaton and Matschiner, 2020* to construct and plot phylogenies for different simulations. *Figure 61* shows simulations for the same parameters as in the main text but a tenfold higher phage mutation rate: most simulations have multiple coexisting lineages, and the travelling wave regime is less long-lived before periods of low turnover. In the low-turnover regime, a relatively small number of large clones that are all outside of each other's cross-reactivity radius can coexist for a very long time, oscillating out of phase from each other (*Figures 62 and 63*).

The durability and turnover of immune memory is qualitatively different in the different regimes of the simulation with cross-reactivity shown in *Figure 54*. We calculated time-shifted average immunity for each of the three regimes (initial regime with no turnover, travelling wave regime, and persistent oscillation regime) and found that turnover was extremely low in all but the travelling wave regime, which had a rapid decay of immune memory to zero (*Figure 64*). In contrast, the immune memory for the same parameters without cross-reactivity had a much more gradual long-term decay towards zero immunity (*Figure 65*). In the travelling wave regime, different amounts of cross-reactivity result in different time-shifted immunity patterns. Turnover is slow but smooth with exponential cross-reactivity, while turnover is much more rapid with step-function cross-reactivity (*Figure 66*). The differences in rate of turnover can also be seen by comparing timescale of sequence change in *Figure 4*.

Adding cross-reactivity appears to slightly decrease average clan sizes (*Figure 67*). This is what we intuitively expect since cross-reactivity means phages are under pressure to get as far away from other clones as possible.

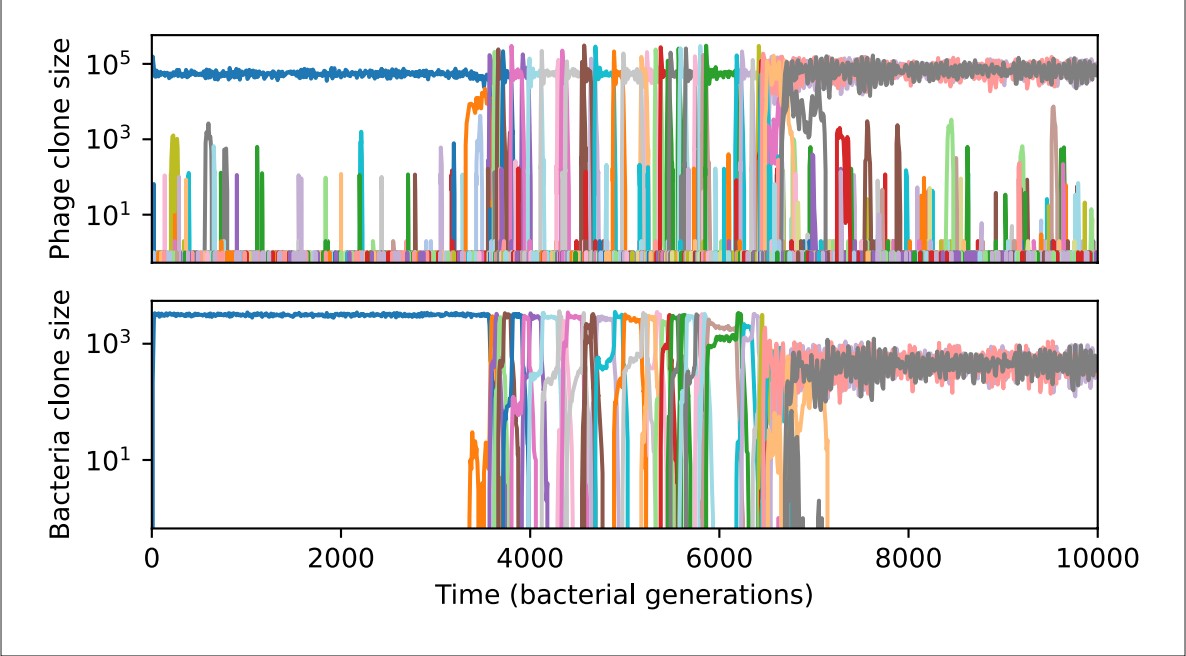

**Figure 53.** Phage clone size (top) and bacteria clone size (bottom) in a simulation with cross-reactivity (step function CRISPR effectiveness with $\theta = 1$). This simulation has one initial phage clone and parameters $C_0 = 10^4$, $\mu = 10^{-6}$, $\eta = 10^{-4}$, and $e = 0.95$. Later times in this simulation are shown in **Figure 55**, and earlier times in this simulation are shown in **Figure 54**.

## Data analysis

This section contains details of our methods for analysing sequencing data from three published papers – groundwater metagenome data used in calculating spacer turnover (Metagenomic CRISPR

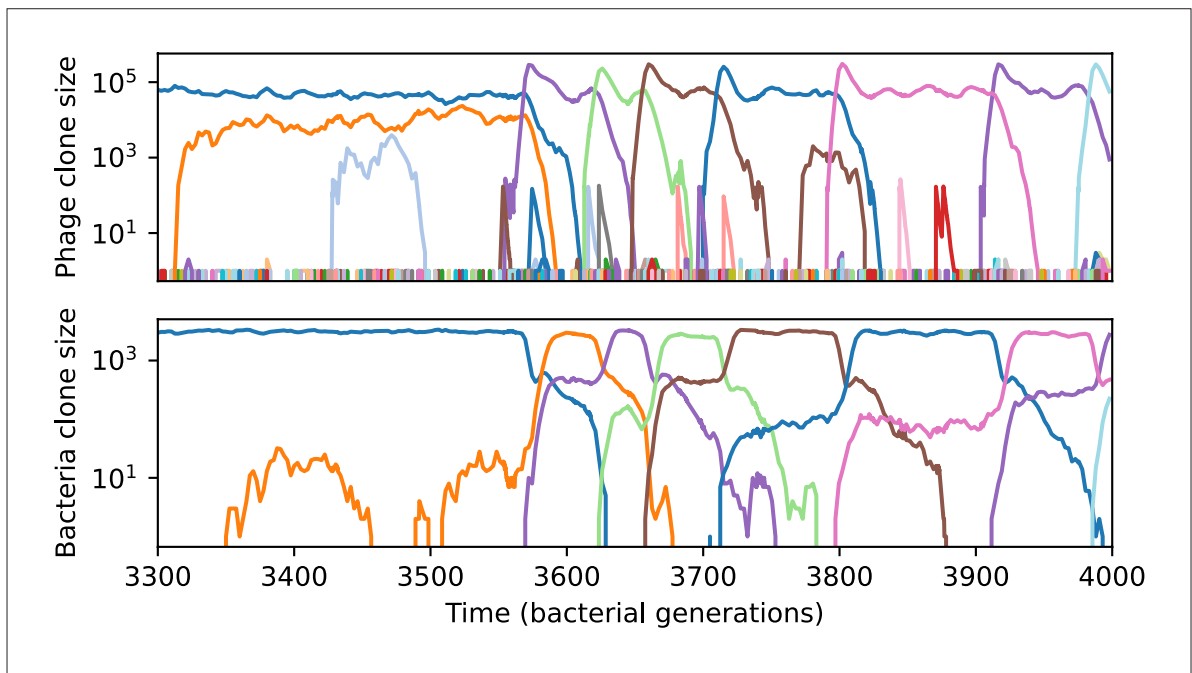

**Figure 54.** Phage clone size (top) and bacteria clone size (bottom) in a simulation with cross-reactivity (step function CRISPR effectiveness with $\theta = 1$). This simulation has one initial phage clone and parameters $C_0 = 10^4$, $\mu = 10^{-6}$, $\eta = 10^{-4}$, and $e = 0.95$. Later times in this simulation are shown in **Figure 55**.

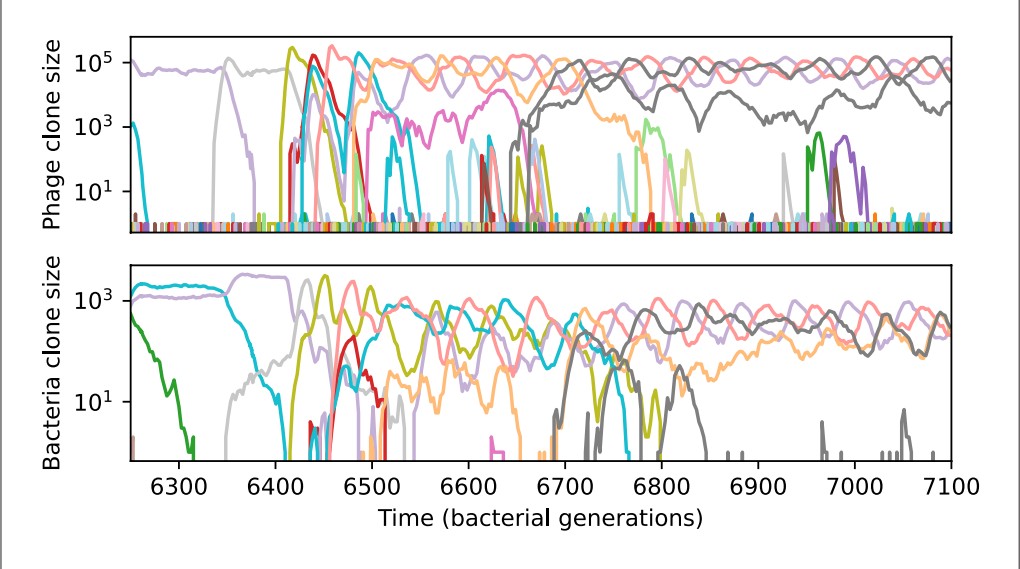

**Figure 55.** Phage clone size (top) and bacteria clone size (bottom) in a simulation with cross-reactivity (step function CRISPR effectiveness with $\theta = 1$) showing the switch between a travelling wave and low turnover regime. This simulation has one initial phage clone and parameters $C_0 = 10^4$, $\mu = 10^{-6}$, $\eta = 10^{-4}$, and $e = 0.95$. Earlier times in this simulation are shown in **Figure 54**.

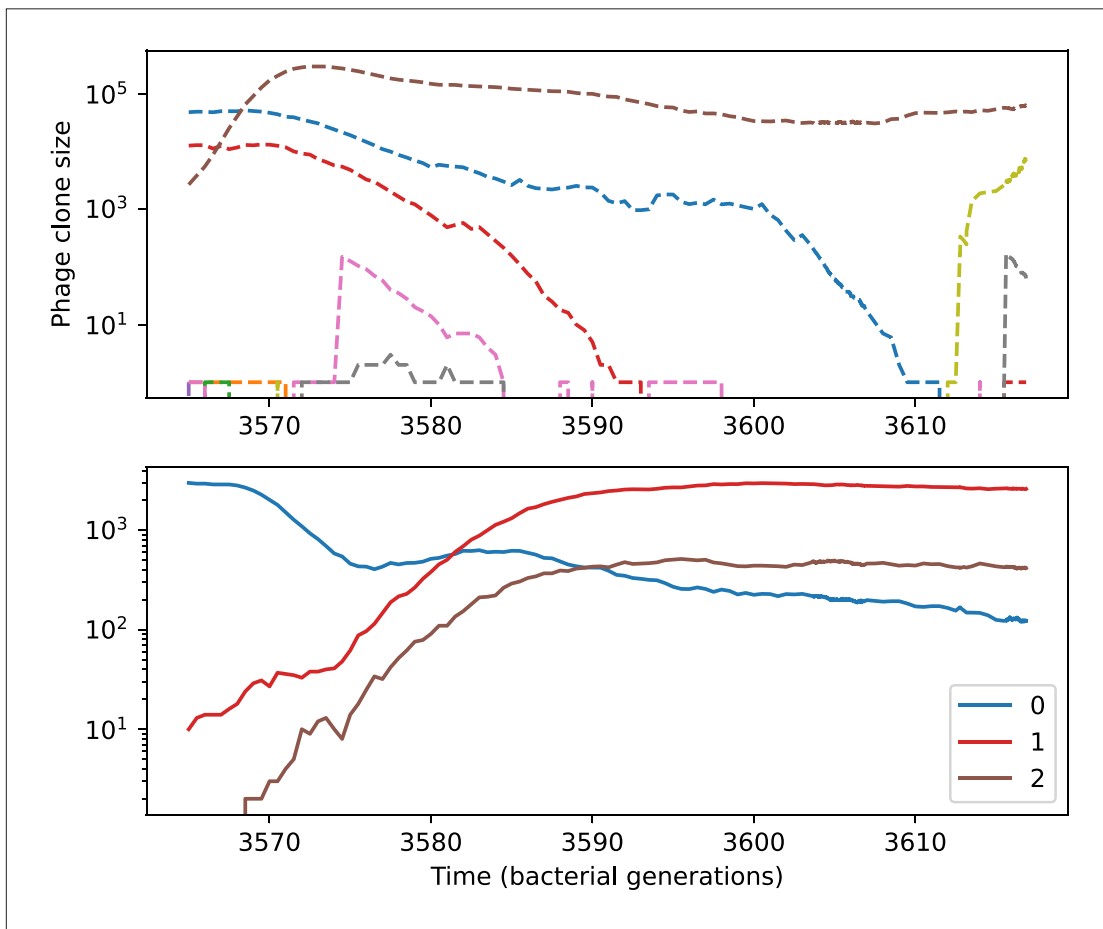

**Figure 56.** Phage clone size (top) and bacteria clone size (bottom) for a short time window of the simulation shown in **Figure 54**. Large phage and bacteria clones are numbered in the legend; these numbers correspond to the numbers in **Figure 57**.

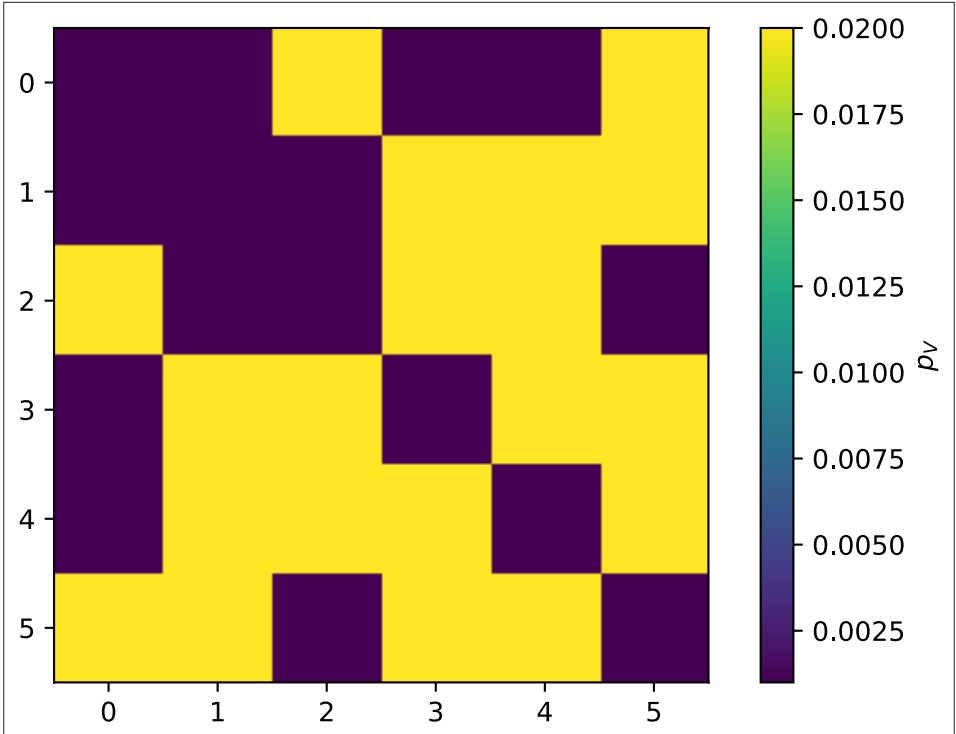

**Figure 57.** Phage infection success probability matrix for each clone shown in *Figure 56*. Dark blue is low infection success, light blue is high infection success.

spacer analysis from *Burstein et al., 2016*), a laboratory long-term coevolution experiment used for spacer turnover and time-shifted average immunity ('Laboratory coevolution experimental data'), and wastewater treatment plant metagenome data used for spacer turnover and time-shifted average immunity ('Experimental data from wastewater treatment plant'). Methods and toy models relating to calculating average immunity in data are in 'Theoretical considerations when calculating average immunity from data'.

## Metagenomic CRISPR spacer analysis from *Burstein et al., 2016*

This dataset is the source for *Figure 6F* in 'Dynamics are determined by diversity'.

We detected spacer sequences in metagenomic data from *Burstein et al., 2016*. We analysed the 0.1 $\mu m$ filter dataset which contains metagenomic reads at six time points, found under accession PRJNA268031. We searched for matches to the CRISPR repeats identified in the study using blast; we accessed a list of repeats from the study's Supplementary data 2. There were 144 unique repeat sequences. Reads are 150 bp long, and we kept only blast results where a repeat had two or more matches to a read in order to accurately detect spacers and to reduce spurious matches to repeats. To further ensure match quality, we kept only alignments that were at least 85% similar to repeat queries, unless another match to the same repeat was present on the read. This 85% threshold was chosen for consistency with previous spacer identification studies (*Paez-Espino et al., 2013*; *Paez-Espino et al., 2015*; *Skennerton et al., 2013*).

It is possible that multiple unique repeats match the same read. For each matched read, we kept the repeat match that had the highest alignment identity (fractional alignment length times percent identity). We detected spacers between repeats, then used the detected spacer length to extract spacers on either side of matched repeats. Because we required at least two repeat matches, at most three spacers could be detected from a read, and at minimum two spacers were detected. We detected a total of 37,963 spacers of which there were 17,491 unique sequences. The most abundant sequence was present 572 times. After grouping within each unique repeat by spacer sequence similarity with an 85% average similarity threshold, there were 12022 unique types with the largest type having an abundance of 616.

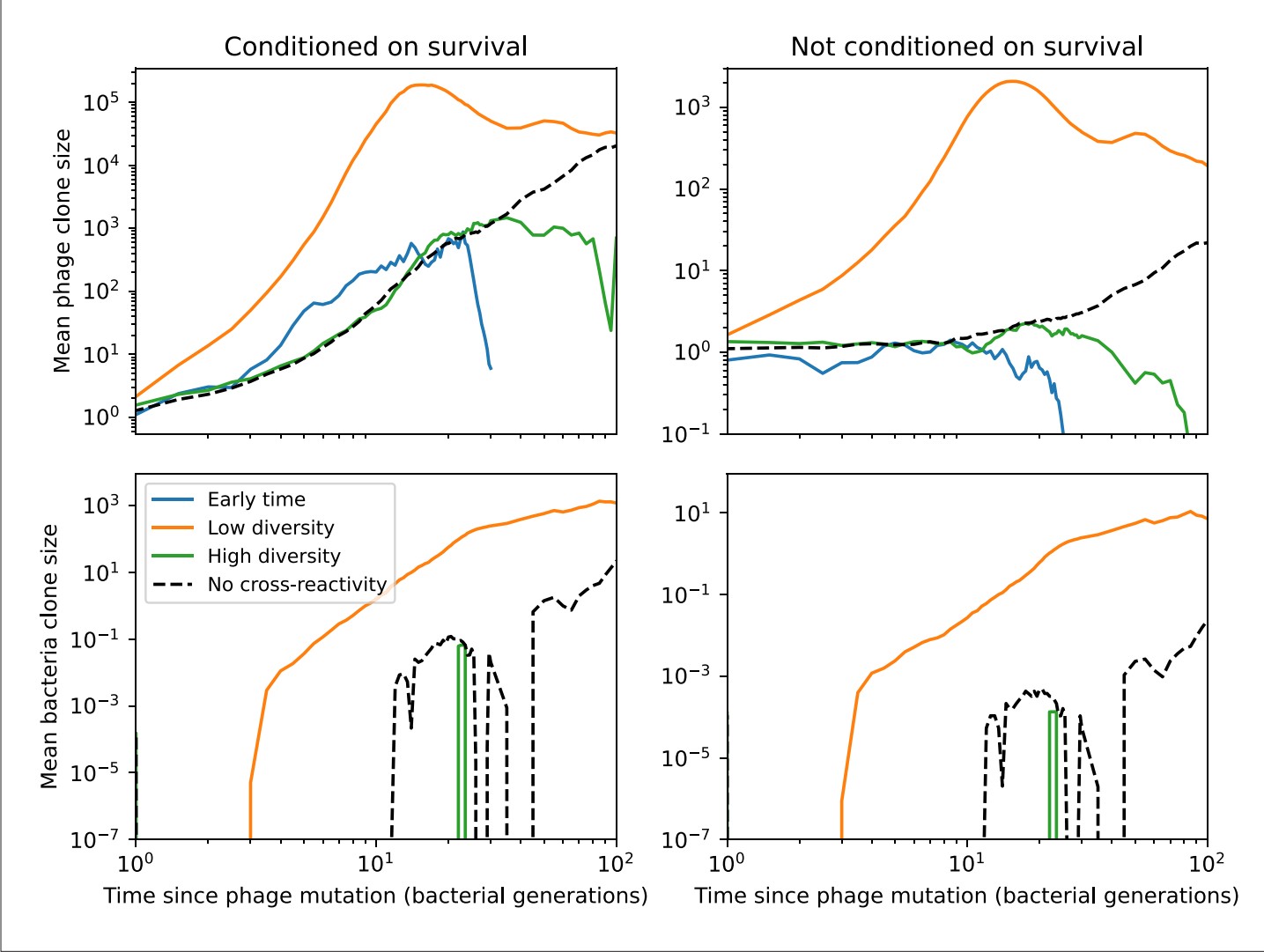

**Figure 58.** Mean phage clone size (top) and mean bacteria clone size (bottom) relative to the time of phage clone mutation, either normalized to surviving clones (left) or averaged over all trajectories (right) for the simulation shown in **Figure 54** ($C_0 = 10^4$, $\mu = 10^{-6}$, $\eta = 10^{-4}$). Clones in the travelling wave regime (4000–6200 generations, orange) grow much more quickly than clones in the initial low-turnover regime (1000–3200 generations, blue) or the final low turnover regime (7000–10,000 generations, green). The black dashed line is the mean clone size for a simulation with the same parameters but without cross-reactivity.

## Theoretical considerations when calculating average immunity from data

The methods described here are used to scale the y-axis of *Figure 7C, D* in 'Time-shifted average immunity calculated from data reveals distinctive patterns of turnover'.

In our simulations and model, we assume that each phage has a single protospacer and each bacterium has a single spacer. In reality, phages can have hundreds to thousands of possible protospacers, and bacteria can also acquire tens to hundreds of spacers (*Pavlova et al., 2021*). If we assume that average immunity plays out at the organism level; that is, if a bacterium that contains one or more spacers matching one or more phage protospacers is immune to that phage, then calculating average immunity in practice requires knowing something about the typical numbers of protospacers and spacers in organisms.

First, let us consider the effect of changing the number of protospacers while keeping spacer array length constant at 1. This is a reasonable model for experimental data at short timescales when most bacteria acquire only one new spacer. For a set $i$ of observed spacers and a set $j$ of observed protospacers with abundances $n_i$ and $n_j$, if each spacer and protospacer are assumed to belong to one

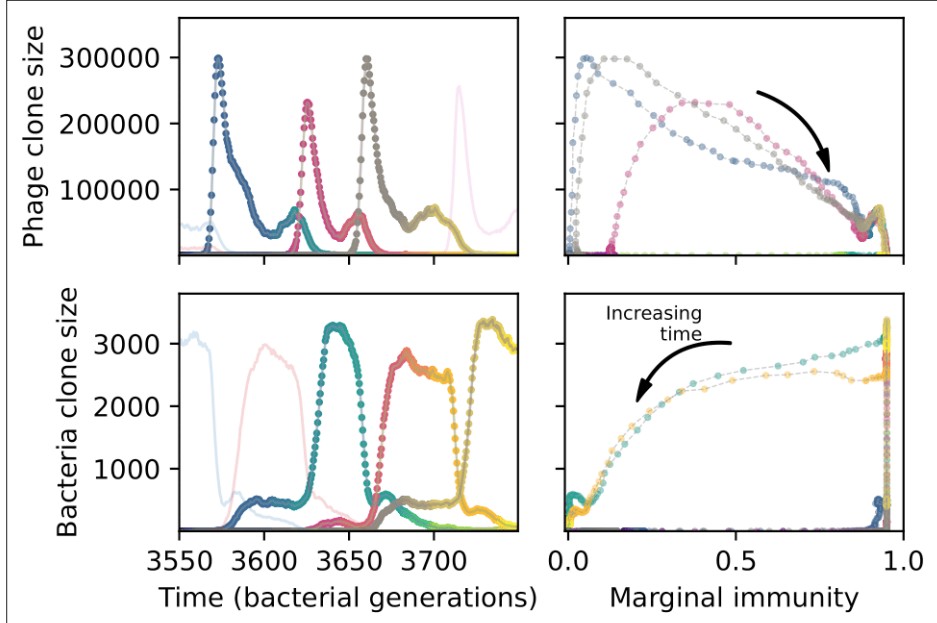

**Figure 59.** A slow-switching cross-reactivity regime for the simulation shown in *Figure 54*. (Left) A subset of clone trajectories for phages (top) and bacteria (bottom) in a simulation with cross-reactivity (step function CRISPR effectiveness with $\theta = 1$). Three trajectories are highlighted and coloured to show increasing time. The population is in a regime where matching clones are offset: because clones that are one mutation apart have the same complete immune overlap, the highlighted clones have large bacterial clone size well after the matching phage clone goes extinct. (Right) The clone size of the highlighted trajectories shown as a function of the bacterial marginal immunity against a particular phage clone (top) or the marginal immunity of a particular bacteria clone (bottom).

organism only, then the average immunity is given by *Equation 90*. We assume for simplicity that any matching spacer provides perfect immunity, that is, $p_V(i,j) = p_V(1 - \delta_{ij})$.

$$1 - \frac{\sum_{i,j} n_i n_j p_V(i,j)}{p_V \sum_{i,j} n_i n_j} = 1 - \frac{p_V \sum_{i,j} n_i n_j (1 - \delta_{ij})}{p_V \sum_{i,j} n_i n_j} = \frac{\sum_{i,j} n_i n_j \delta_{ij}}{N_i N_j} \tag{90}$$

where $N_i$ denotes $\sum_i n_i$.

If instead the same set of protospacers is divided among fewer phages so that each phage has $a$ protospacers on average, then the total number of phages is $N_j/a$. The numerator of average immunity remains the same since each bacterium still contains only a single spacer. Average immunity is exactly the same as before but multiplied by the average number of protospacers per phage (*Equation 91*).

$$\frac{\sum_{i,j} n_i n_j \delta_{ij}}{N_i \frac{N_j}{a}} = \frac{a \sum_{i,j} n_i n_j \delta_{ij}}{N_i N_j} \tag{91}$$

This agrees with the intuition that more spacer targets per phage increases the immune potential of the CRISPR system.

In data from *Paez-Espino et al., 2015*, we find quantitative agreement between the range of average immunity values in their data and the types of values in our simulations once we apply this simple transformation with $a = 696$, since there are 231 possible CRISPR1 protospacers and 465 possible CRISPR3 protospacers in the phage genome (determined by searching for the canonical PAM for each CRISPR locus).

If bacteria contain multiple spacers and phages contain multiple protospacers, the combinatorics of average immunity gets more interesting. Now, if a bacterium contains multiple spacers that target the same phage, we assume this does not increase its immunity (although in reality there may be a benefit to multiple matching spacers).

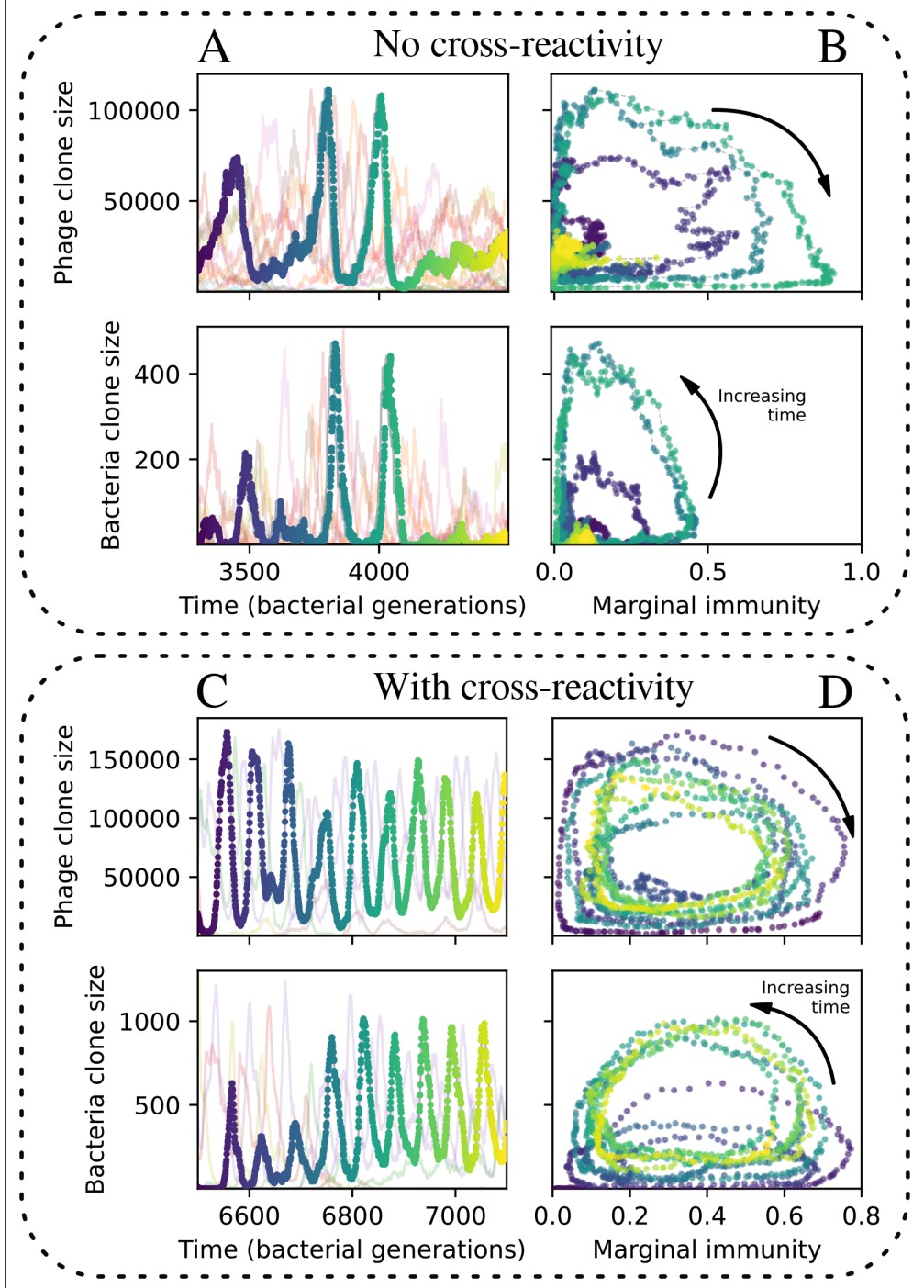

**Figure 60.** Cross-reactivity leads to persistent oscillations. (**A**) A subset of clone trajectories for phages (top) and bacteria (bottom) in a simulation with no cross-reactivity. Transient oscillations occur. One trajectory is highlighted and coloured to show increasing time. (**B**) The highlighted trajectory in (**A**) is shown as a function of the marginal immunity for phages (top) and bacteria (bottom). Clones experience an oscillating fitness that depends on their overlap from the other population. Arrows indicate the direction of increasing time in the oscillation. (**C**) A subset of clone trajectories for phages (top) and bacteria (bottom) in a simulation with cross-reactivity (step function CRISPR effectiveness with $\theta = 1$). The population is in a regime where several clones experience persistent and rapid oscillations. One trajectory is highlighted and coloured to show increasing time. (**D**) The highlighted trajectory in (**C**) is shown as a function of the marginal immunity for phages (top) and bacteria (bottom). Clones experience an oscillating fitness that depends on their overlap from the other population. Arrows indicate the direction of increasing time in the oscillation. For all simulations $C_0 = 10^4$, $e = 0.95$, $\eta = 10^{-4}$, and $\mu = 10^{-5}$.

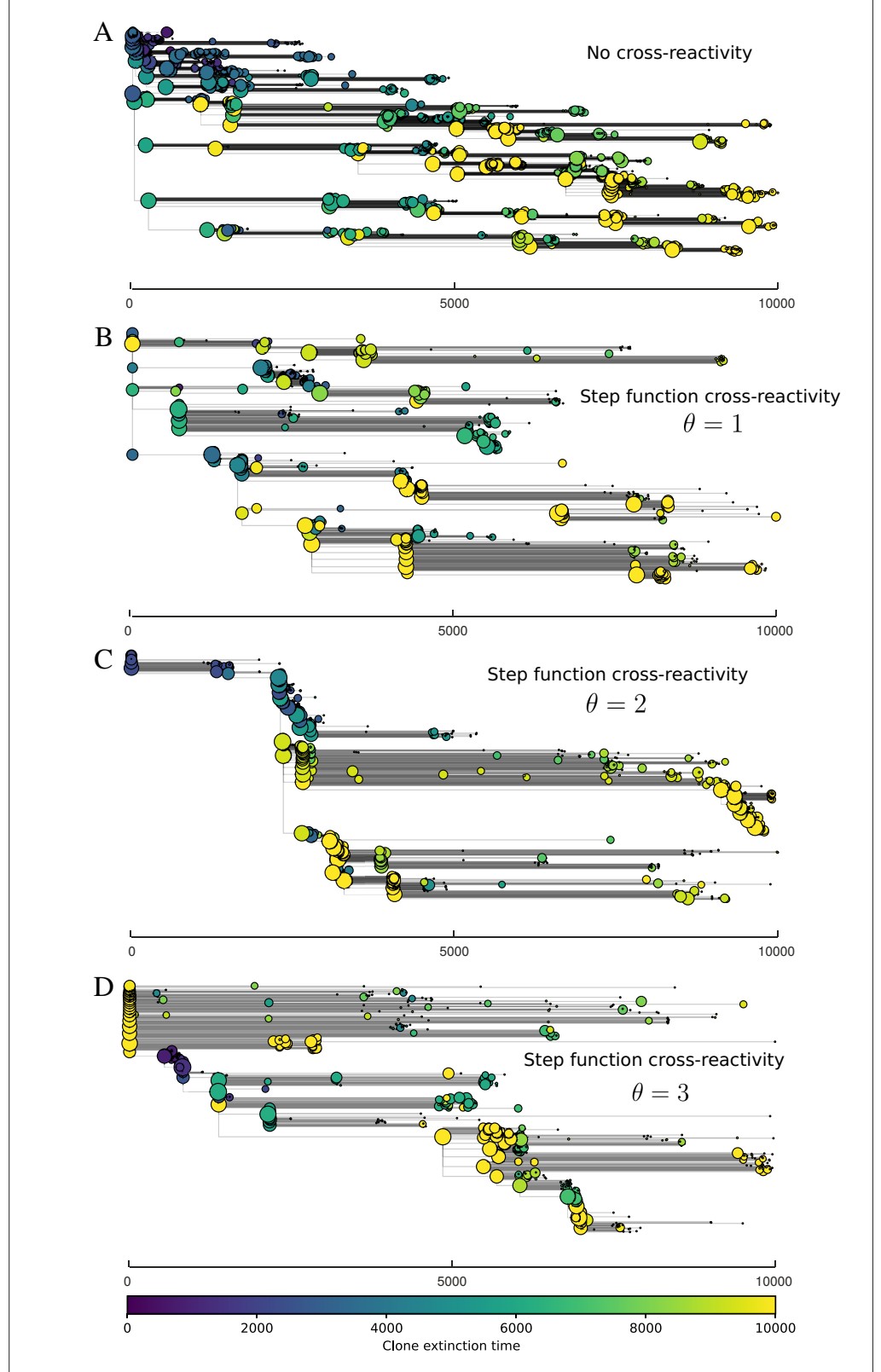

**Figure 61.** Phage clone phylogenies for four simulations with different cross-reactivities and a higher mutation rate than shown in main text *Figure 4*: no cross-reactivity (**A**) and step-function cross-reactivity with $\theta = 1$ (B, $\theta = 2$) (**C**), and $\theta = 3$ (**D**). All simulations share all other parameters: $C_0 = 10^4$, $\eta = 10^{-4}$, $\mu = 10^{-5}$, $e = 0.95$. Phage clones are plotted at the first time they pass a population size of 2 (to remove clutter from many new mutations

*Figure 61 continued on next page*

*Figure 61 continued*

destined for extinction), and the size of each circle is logarithmically proportional to the maximum size reached by that clone. Colours indicate the time of extinction of each clone. For each simulation with cross-reactivity, the left inset shows phage (top) and bacteria (bottom) clone sizes over time; colours indicate unique clone identities.

We simulated sets of spacers and protospacers, randomly divided them into arrays of different sizes, and calculated the organism-level average immunity in each case. *Figure 68* shows the average immunity values for several array lengths with different assumptions for how the set of spacer sequences and array lengths are distributed. We chose a set of spacer 'sequences' where each sequence is a letter of the alphabet, then constructed phage protospacer arrays by randomly drawing 5 unique letters 50 times to create 50 phages. We then sampled letters from the alphabet either uniformly or from an exponential distribution (where 'A' is the number 1, etc.). The exponential sampling captures the fact that in practice spacer abundance distributions are highly non-uniform. To create bacteria spacer arrays, we sampled from the set of spacers without replacement, either creating arrays of constant size or arrays with the same mean size but with their length either normally or exponentially distributed.

We developed a simple theoretical model for the change in average immunity as array length increases. The baseline average immunity when bacteria are assumed to have single spacers we denote $a_1$, and define a constant $C = 1 - a_1$. Then as array length increases, the total number of bacteria decreases and the overlap increases, but by a smaller factor than the population size decrease: the remaining average immunity that could be gained is reduced with the same fraction as the original average immunity so that the average immunity for an array of length $n + 1$ is $a_{n+1} = 1 - (1 - a_n)C$. Plotting $1 - a_n$ vs. $n$ on a log plot gives a straight line. Plugging in $a_1 = 1 - C$ gives $a_n = 1 - C^n$, plotted as a black dashed line in *Figure 68*. The theory agrees well with the simulated results except for when array length is exponentially distributed; in this case, we believe the presence of several very long and very short arrays means that short-array bacteria do not get the coverage of multiple spacers while the long-array bacteria have redundant spacers. This functional form for average immunity as a function of array length is very similar to a quantity derived by *Iranzo et al., 2013* — they calculated that the

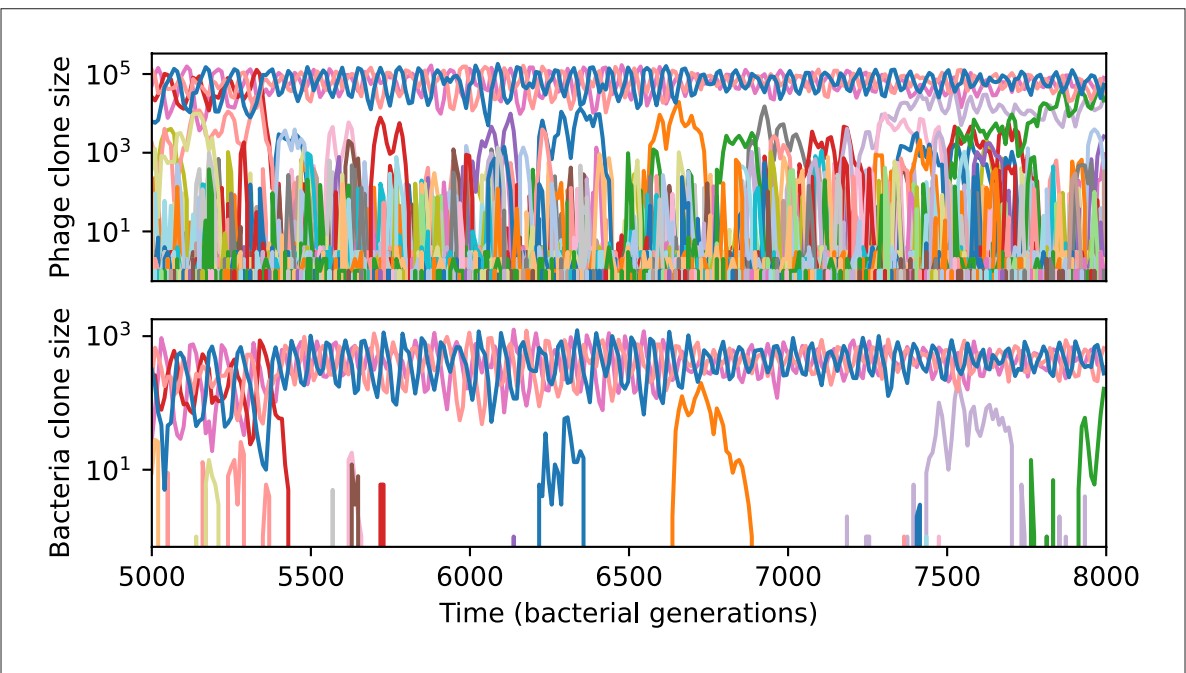

**Figure 62.** Phage clone size (top) and bacteria clone size (bottom) for a short time window of the $\theta = 2$ simulation shown in *Figure 61C*. Each of the large trajectories oscillating out of phase between 5000 and 7000 generations is at least three mutations away from all of the others; they are all outside each other's cross-reactivity radius.

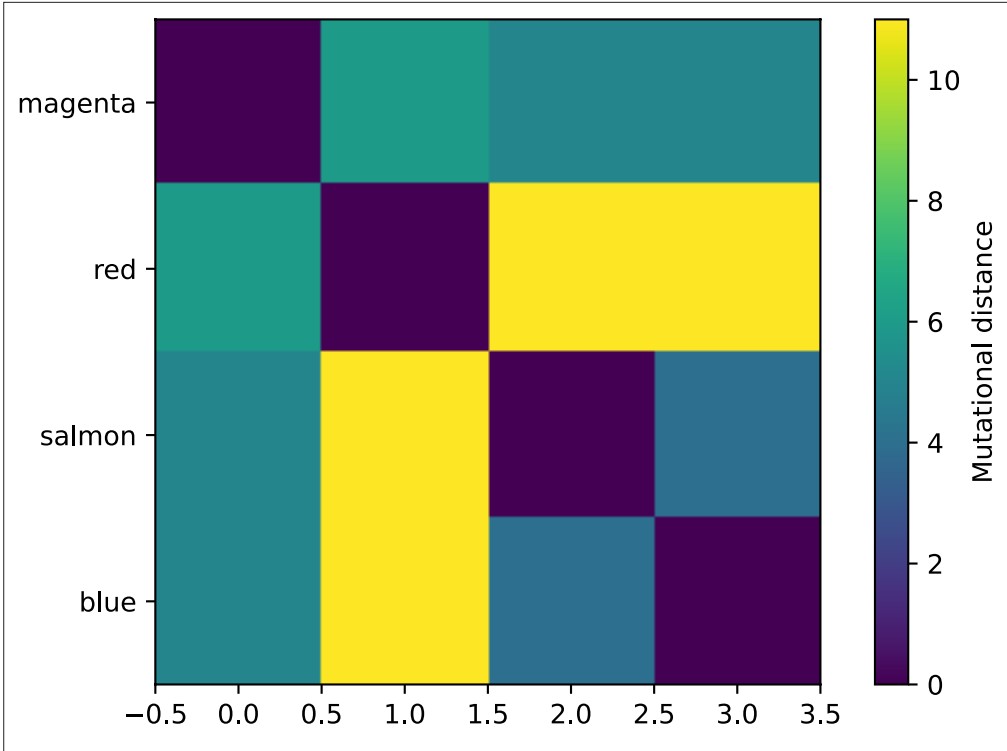

**Figure 63.** Matrix of mutational distance between each of the four largest phage clones shown in *Figure 62*; colours of those trajectories are labelled on the y axis. Each clone is at least three mutations away from all other large clones.

probability that a random bacterium is immune to a random phage, each with a random set of spacers and protospacers is $p_c = 1 - (1 - \alpha \frac{n}{N_t})^{N_s}$, where $n$ is the average bacterial array length, $N_t$ is the total number of unique protospacers, and $N_s$ is the number of protospacers per phage. The constant $\alpha$ is a scaling factor that measures the degree of correlation between matching spacer and protospacer abundances. Their scaling is as a function of the *phage* array length instead of bacterial, but the same intuition applies.

An interesting result of this analysis is that the relative immune benefit from gaining a spacer decreases as immunity increases: a CRISPR array twice as long does not provide double the immunity. A similar diminishing-returns result has also been observed in theoretical models of vertebrate immunity where the relative decrease in immune susceptibility and increase in immune memory both decrease as the number of infections increases over an organism's lifetime (*Mayer et al., 2019*). Long CRISPR arrays are also subject to a dilution effect: spacers compete to form complexes with limited *Cas* protein machinery, and if the number of spacers is high, there is a higher chance that the needed spacer is not available in high enough numbers during an infection (*Martynov et al., 2017*; *Bradde et al., 2020*).

## Average immunity negatively correlates with diversity regardless of array size

This section describes the toy model we present in results 'Pathogen and host diversity must be considered together.

In our simulations, we observed an inverse proportionality between average immunity and bacterial diversity (*Figure 50*). We wondered if this trend would hold for array sizes and numbers of protospacers larger than 1. Our intuition is that this trend should hold for any arrangement of spacers and protospacers into arrays, provided the following two statements are true: (a) bacteria and phage diversity is coupled; that is, phage protospacer diversity matches bacterial spacer diversity, and (b) diversity is larger than CRISPR array length.

We tested this with a toy model, described in 'Pathogen and host diversity must be considered together'. Conceptually, there are two regimes of CRISPR immunity and phage evolution to consider.

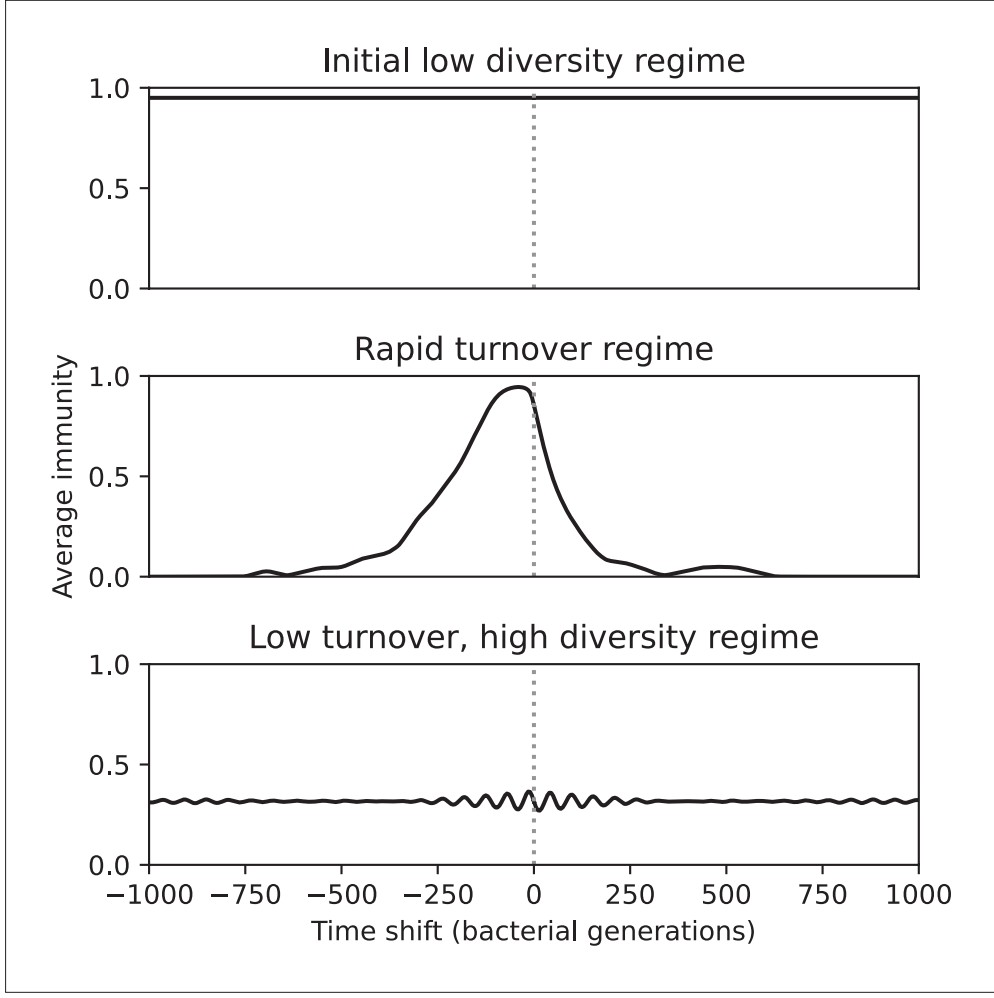

**Figure 64.** Time-shifted average immunity for three regimes of the simulation shown in **Figure 54** ($C_0 = 10^4$, $\mu = 10^{-6}$, $\eta = 10^{-4}$, and $e = 0.95$, step-function cross-reactivity with $\theta = 1$). The initial low-diversity regime (1000–3200 generations) and the low turnover, high diversity regime (7000–10,000 generations) had extremely low turnover, while the travelling wave regime (4000–6200 generations) had high average immunity near 0 delay that rapidly decayed to zero both in the past and future.

In an 'evolutionarily late' regime, bacteria are exposed to many diverse phages for which any one spacer may target only a few phage genotypes. In an 'evolutionarily early' regime, most phages are clonally similar and protospacer diversity is generated through escape mutations. In this scenario, we can imagine a set of phages that share most of their protospacers but have a small number of recently mutated protospacers. This regime can further be divided into a slow-mutation 'successive sweeps' regime in which each new mutation takes place in the background of the previous mutant and a fast-mutation 'clonal interference' regime in which many unique mutations coexist in the same background.

We explored these three scenarios by generating synthetic sets of protospacers and spacers and varying the total diversity by changing the size of the pool of available sequences, distributing sequences exponentially in abundance to qualitatively match data **Bonsma-Fisher et al., 2018**. We randomly assigned sequences to individual phages and bacteria, then calculated average immunity in three scenarios of protospacer origin. First, we divided protospacers randomly among 100 phages so that each phage had $x$ unique protospacers to mimic an evolutionarily diverse population. In the second scenario, we created a set of phages that mimic a population undergoing successive sweeps by changing the first protospacer, then the first and second, and so on. In the third scenario, the set of phages share the first $x - 1$ protospacers and only the last protospacer is varied; this mimics an initially

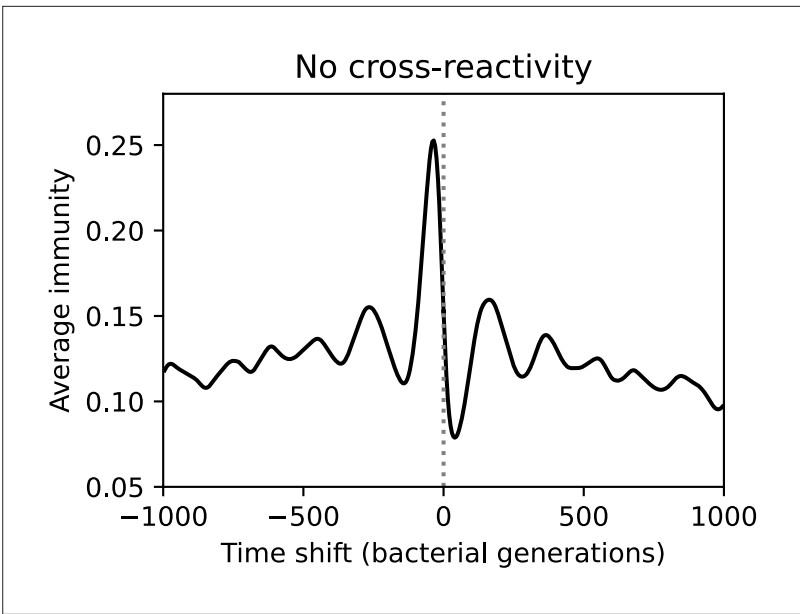

**Figure 65.** Time-shifted average immunity for the corresponding simulation to **Figure 64** without cross-reactivity ($C_0 = 10^4$, $\mu = 10^{-6}$, $\eta = 10^{-4}$, and $e = 0.95$). Peak average immunity is low because of high diversity, and immunity decays very gradually to zero in both the past and future.

clonal population undergoing rapid mutations. In all scenarios, we sampled uniformly from the pool of protospacers to generate 1000 spacer sequences that were divided into bacterial arrays with either a constant array size (**Figure 69B–D**), or a Gaussian-distributed array size (**Figure 70**). We repeated the bacterial array assortment process 20 times for each total spacer diversity and combination of array sizes.

We calculated average immunity in these synthetic scenarios, assuming for simplicity that any matching spacer provides perfect immunity, that is, $e = 1$, $p_V(i,j) = p_V(1 - \delta_{ij})$, or $p_V(I,J) = p_V[I \cap J = \varnothing]$. For the set of bacteria $n_B$ and phages $n_V$, the average immunity for the population is

$$1 - \frac{\sum_{I,J} n_B^I n_V^J p_V(I,J)}{p_V \sum_{I,J} n_B^I n_V^J} = 1 - \frac{p_V \sum_{I,J} n_B^I n_V^J [I \cap J = \varnothing]}{p_V \sum_{I,J} n_B^I n_V^J} = \frac{\sum_{I,J} n_B^I n_V^J [I \cap J \neq \varnothing]}{N_B^I N_V^J} \tag{92}$$

where $N_B^I = \sum_I n_B^I$ and $N_V^I = \sum_I n_V^I$. In our base model, bacteria and phages are each limited to a single spacer or protospacer, whereas now $I$ and $J$ label the full set of spacers or protospacers in an organism. $n_B^I$ is the number of bacteria with the same unique set of spacers $I$ and the Iverson bracket $[I \cap J \neq \varnothing]$ gives 1 if any of the spacers in $I$ matches any of the protospacers in $J$ and 0 if there is no overlap between sets $I$ and $J$.

The 1-spacer 1-protospacer curve in **Figure 69B** represents the same situation as our simulations: one spacer and one protospacer (but with a random distribution of sequences), and the trend matches our simulations exactly with average immunity inversely proportional to diversity. This strict inverse dependence changes shape as the number of spacers and protospacers per organism changes.

We compared these toy model results to the experimental setup of **Common et al., 2020**. In their experiments, they combined equal proportions of bacterial strains with a different CRISPR spacer with one phage strain that is either targeted by all spacers or escaped from one of the spacers. There is then one susceptible bacterial strain per mixture when the escape phage is used. All experiments also contained one surface mutant bacterial strain that is always immune to the phage.

We can directly calculate the expected initial average immunity based on this setup. For $m$ bacterial CRISPR strains, one surface mutant strain, and one escape phage, total bacteria $N_b$, total phage $N_v$, each bacterial strain has abundance $N_b/m$ and $v_j = N_v$.

We calculate average immunity assuming $e = 1$ : $\frac{\sum_{i,j} b_i v_i \delta_{ij}}{N_i N_j}$

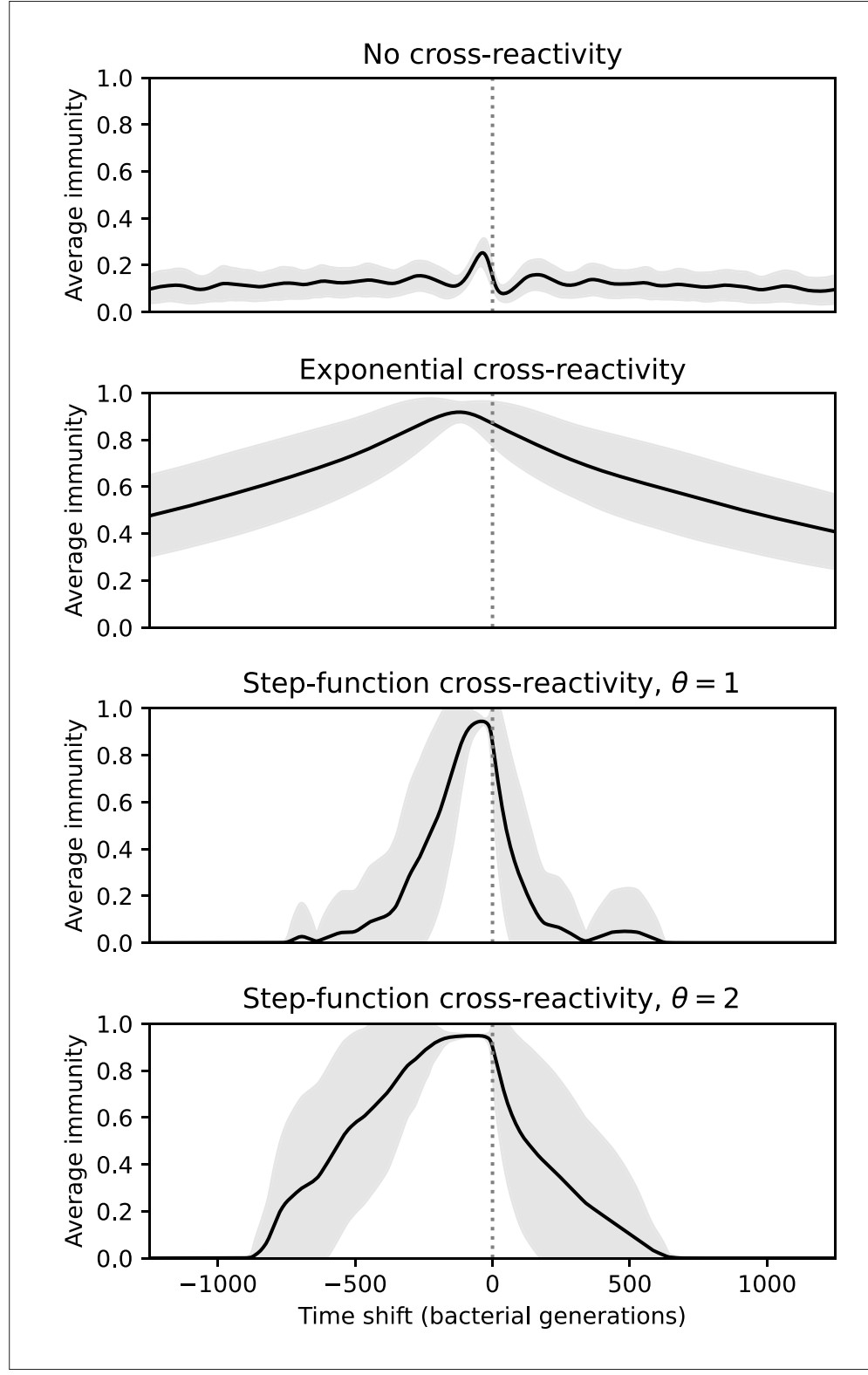

**Figure 66.** Time-shifted average immunity for four simulations with the same parameters but different types of cross-reactivity: no cross-reactivity (top), exponential cross-reactivity (middle top), step-function cross-reactivity with $\theta = 1$ (middle bottom), and $\theta = 2$ (bottom). Shared parameters are $C_0 = 10^4$, $\mu = 10^{-6}$, $\eta = 10^{-4}$, and $e = 0.95$. Only the travelling-wave regime of each simulation with cross-reactivity was used to compare turnover in this regime. The grey shaded region is the standard deviation across averaged data.

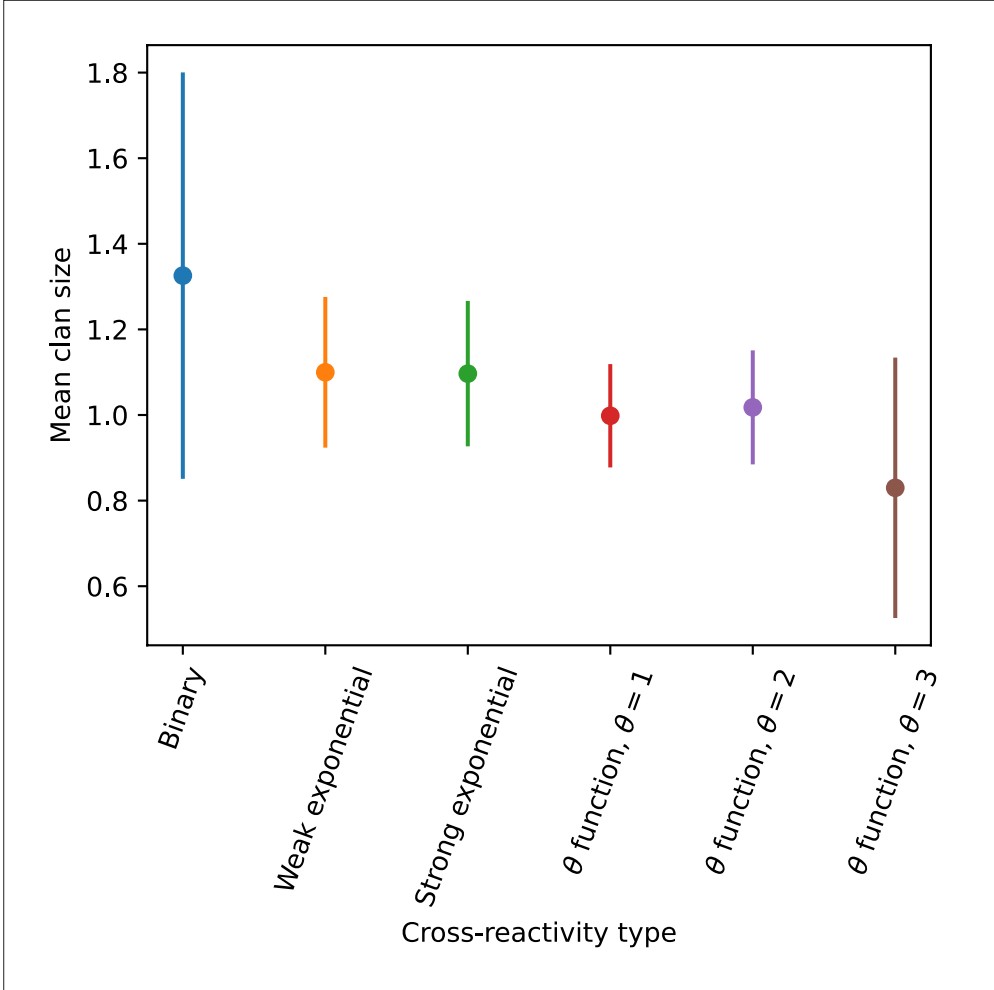

**Figure 67.** Average clan size across simulations with different parameters for different degrees of cross-reactivity with $m_{\text{init}} = 10$. Error bars are the standard deviation across three or more independent simulations.

We can include the surface mutant or not; it slightly shifts the total diversity but the trend is unaffected. For instance, for $m = 3$ CRISPR clones, the phage is able to infect one of them, so average immunity is $\frac{N_v N_b/m + N_v N_b/m}{N_b N_v} = \frac{2}{m} = \frac{2}{3}$. If we included the surface mutant in equal proportions, we would get 3/4 instead.

### Laboratory coevolution experimental data

This dataset is the source for *Figure 6E* in 'Dynamics are determined by diversity' and *Figure 7C* in 'Time-shifted average immunity calculated from data reveals distinctive patterns of turnover'.

We analysed experimental data from *Paez-Espino et al., 2015*. In this experiment, *S. thermophilus* bacteria were mixed with phage 2972 and allowed to coevolve until phage extinction, up to 232 days in one replicate. Bacteria and phage whole-genome shotgun sequencing was performed at irregular intervals (13 time points for series MOI-2B). This data is publicly available in the NCBI Sequence Read Archive under the accession PRJNA275232.

We analysed data from the MOI-2B series and detected spacers in the CRISPR1 and CRISPR3 loci by searching raw reads for matches to the *S. thermophilus* repeats (GTTTTTGTACTCTCAAGATT TAAGTAACTGTACAAC for CRISPR1 and GTTTTAGAGCTGTGTTGTTTCGAATGGTTCCAAAAC for CRISPR3) using BLAST.

The expected structure of the CRISPR locus is 36 nt repeats interspaced with 30 nt spacers. Reads are 100 nt long (Illumina), and so at most two complete spacers can be detected per read (one full

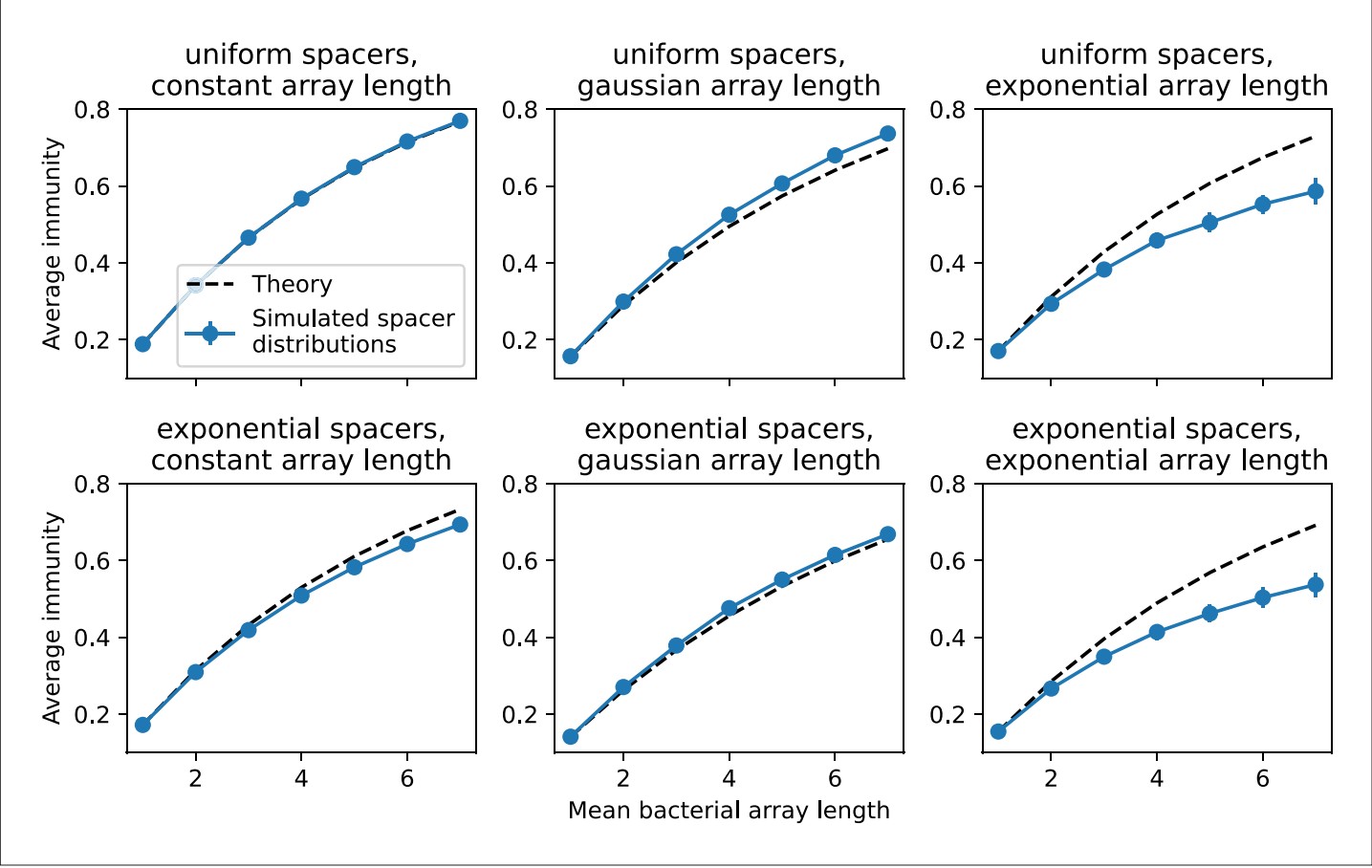

**Figure 68.** Average immunity vs. bacterial spacer array length for simulated distributions of protospacers and spacers. We simulate 50 phages, each with 5 protospacers represented by letters from the alphabet, uniformly sampled. We simulate 420 bacterial spacers, drawn from the alphabet either uniformly (top row) or following an exponential distribution with mean 6 (bottom row). We construct arrays by sampling from the set of 420 spacers without replacement, either creating arrays of constant length (left column), or of variable length with a gaussian distribution (middle column) or exponential distribution (right column). Average immunity is calculated as in *Equation 90* except the indices run over all arrays and not over individual sequences, and the presence of any matching pair gives perfect immunity. Blue points are average results over 50 simulations; error bars are standard deviation. The black dashed curve is given by $a_n = 1 - C^n = 1 - (1 - a_1)^n$.

repeat match near the centre of the read). To maximize the number of genuine spacers detected while removing low-quality matches, we kept only full-length alignments to the repeat (length 36), unless a shorter alignment was also present on the same read as a full-length alignment (for instance, if the repeat is partially present at the start or end of a read). If a single repeat match was present on a read, we extracted 30 nt on either side and labelled these spacers. We set a minimum spacer length to be 26 nt ($\approx 0.85 \times 30$) and did not keep spacer sequences at the start or end of a read that were shorter than this. If two repeat matches were present, we extracted the spacer sequence between the matches.

To detect wild-type spacers, we searched for matches to the CRISPR repeats on the *S. thermophilus* DGCC7710 reference genome (accession NZ_CP025216) and extracted the sequences between repeat matches. We also searched raw reads from the day 1 data of the control replicate (no phages) for matches to the repeats and performed the same spacer extraction procedure.

We grouped all extracted spacers using the AgglomerativeClustering method from SciPy with an 85% similarity threshold and the 'average' linkage criterion. We performed this grouping separately for CRISPR1 and CRISPR3 spacers. Each spacer sequence was then labelled with a group type identifier. *Figure 71* shows our detected spacer counts after grouping by 85% average similarity and eliminating single counts and matches to wild-type spacers, compared with the reported results from *Paez-Espino et al., 2015*; there is good agreement between our counts and the original authors' counts.

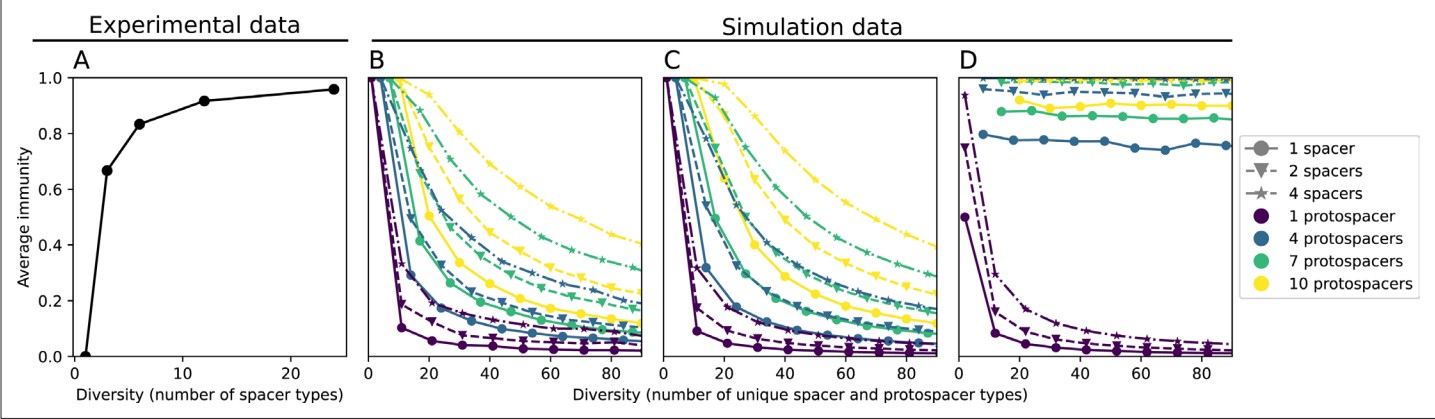

**Figure 69.** Average immunity decreases with coupled CRISPR diversity. (**A**) Inferred initial average immunity as a function of diversity in the experiment of *Common et al., 2020*. Bacterial diversity was manipulated in the experiment by combining different numbers of bacterial clones defined by unique CRISPR spacers with a single phage strain that was able to infect only one of the clones. We calculated the expected initial average immunity based on the number of clones. (**B–D**) Results of a toy model of sorting a set of protospacers and spacers into arrays of variable length, either assigning protospacers fully randomly (**B**), as sequential mutations (**C**), or changing the last protospacer in the array only (**D**).

To detect protospacers, we first blasted all reads against the *S. thermophilus* DGCC7710 reference genome and the phage 2972 reference genome (accession NC_007019) (**Figure 72**). We next blasted all unique detected spacer sequences from the previous step against all the reads, removing any query sequences that are a perfect subset of another sequence and any sequences that were more than 30% N nucleotide. This detects both potential protospacer sequences and the original spacers themselves. To isolate protospacers, we kept only results that were on reads that did not match the bacterial genome and that did not match the CRISPR1 repeat (CRISPR3 repeats were not checked; however, they represent a very small number of additional reads, less than 0.1%). We kept only results that were 26 nt or larger. If there were multiple hits on a read from the same spacer type (but different sequences), we kept only the match with the lowest e-value. We extracted the matched sequence from the read and 10 nucleotides after it to check for a PAM sequence (all spacers were stored oriented in the same direction relative to the repeat to facilitate comparison and PAM detection). Since reads are paired-end, some reads will overlap, and some spacers may be double-counted

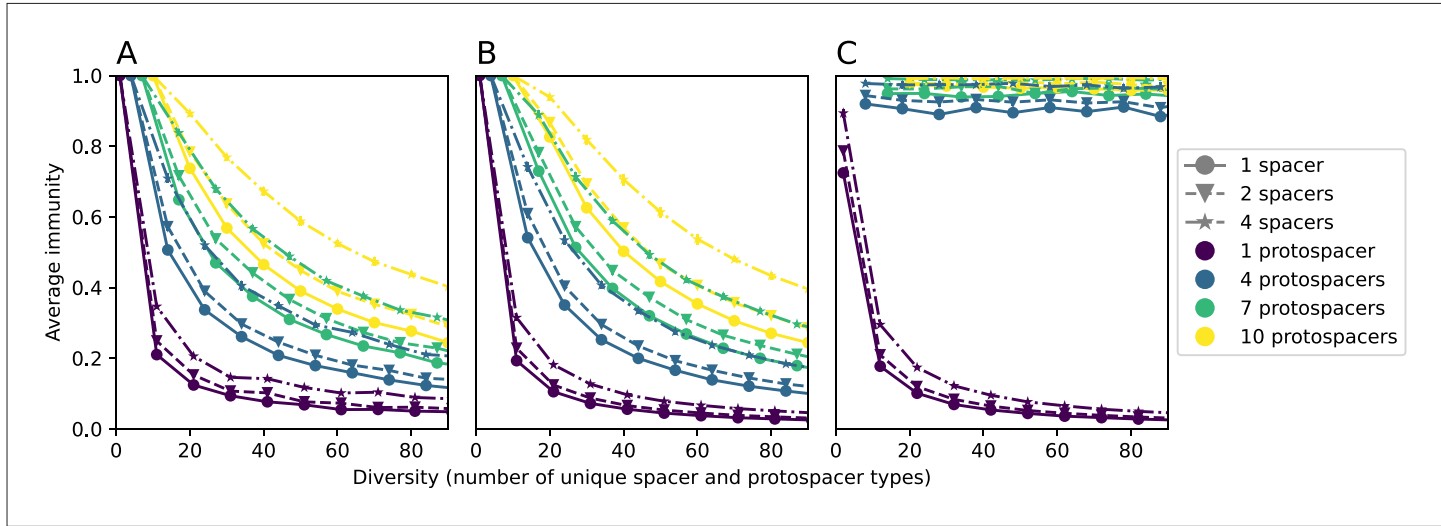

**Figure 70.** Average immunity vs. diversity (number of unique types) for simulated distributions of spacers and protospacers with different bacterial array sizes (increasing top to bottom) and different numbers of protospacers per phage (increasing left to right). Bacterial array sizes are drawn from a Gaussian distribution with mean given by the array mean for each row and a standard deviation of 2. Points are averages across 50 independent runs.

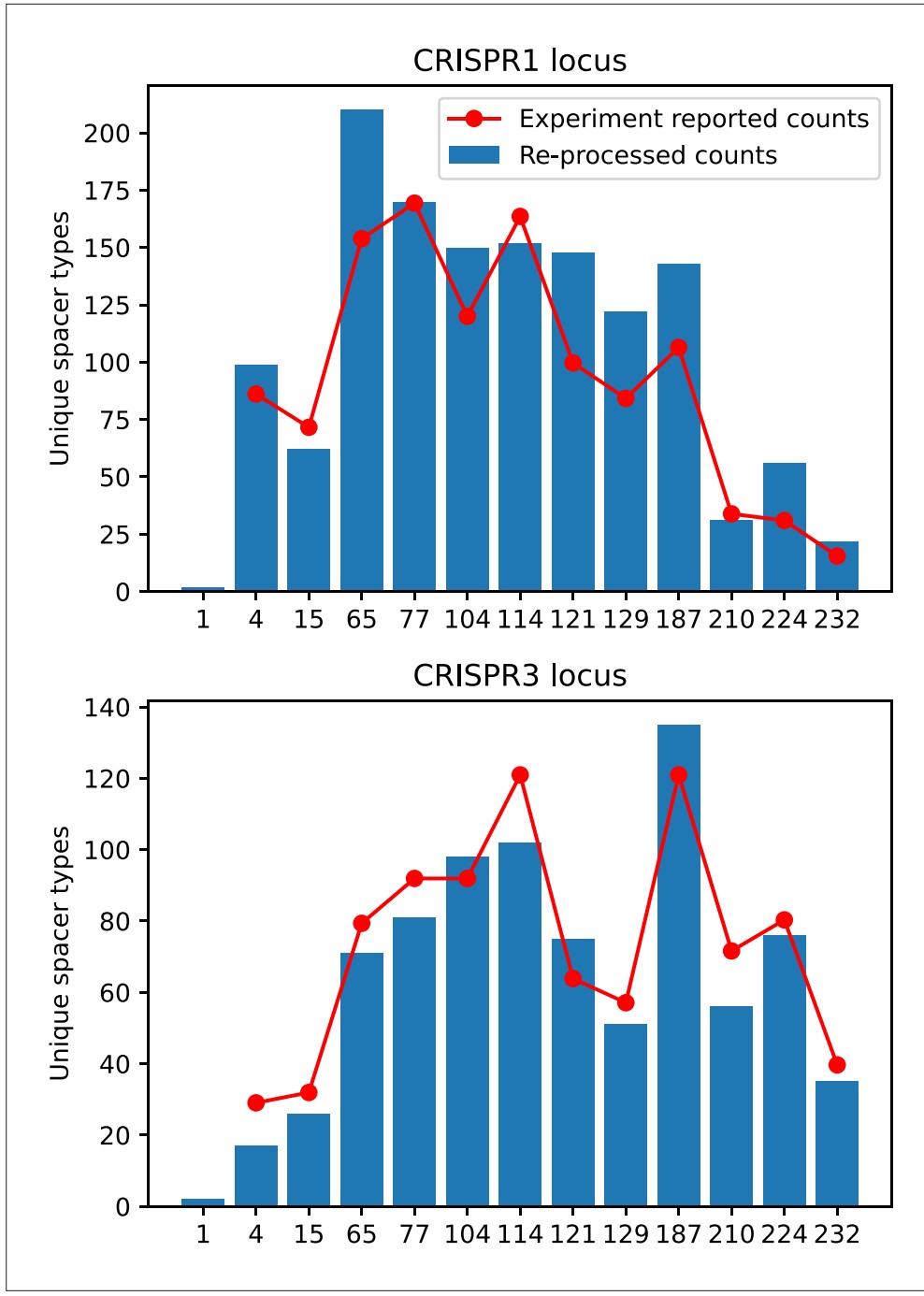

**Figure 71.** Unique spacer types detected in our analysis of data from *Paez-Espino et al., 2015* after grouping by 85% average similarity and removing single spacer counts (blue bars). Counts reported in *Paez-Espino et al., 2015* are red points.

in the overlap. We decremented the total spacer or protospacer count for a sequence by 1 if a spacer sequence was present on both ends of a paired read. We also removed sequences that contained long strings of 11 or more of the same nucleotide, assuming these to be sequencing errors. This removed around 1% of sequences.

We grouped all spacer and protospacer sequences together (separated by CRISPR locus) by several different similarity thresholds between 85 and 99% to assign type labels based on each similarity threshold.

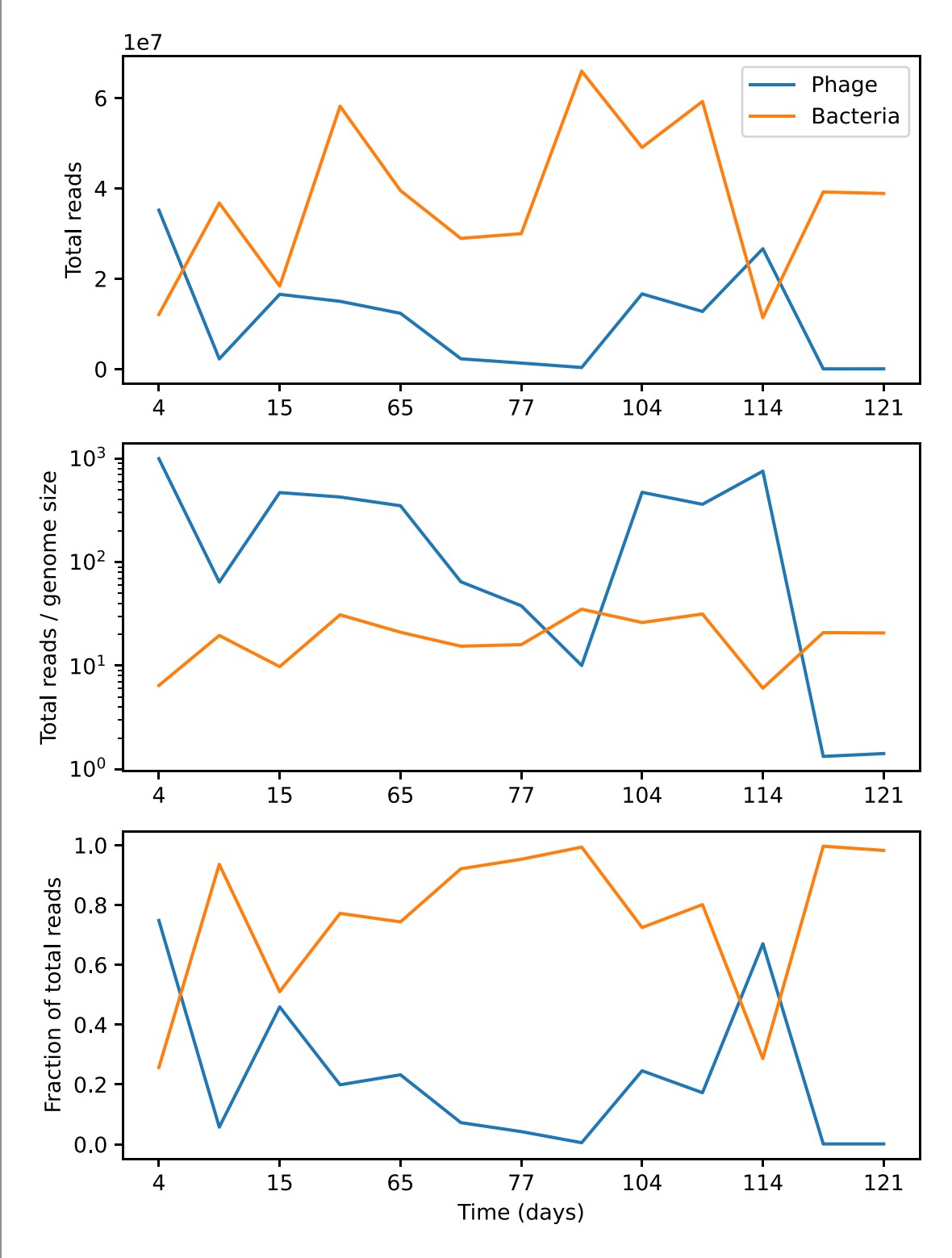

**Figure 72.** Total reads per time point that match phage (blue) or bacteria (orange). Top: total reads. Middle: total reads divided by phage and bacteria genome sizes. Bottom: fraction of total reads matching bacteria or phages.

We checked the 10 nt region downstream from each potential protospacer and removed any sequences that did not have a perfect match to the PAM: AGAAW for CRISPR1 (*Paez-Espino et al., 2013*; *Paez-Espino et al., 2015*; *Shah et al., 2013*; *Garneau et al., 2010* and GGNG for CRISPR3

*Paez-Espino et al., 2015*; *Shah et al., 2013*). Since targeting is highly sensitive to PAM mutations (*Shah et al., 2013*), we assumed that any deviation from the perfect PAM meant that the protospacer would not be successfully targeted. There are cases where the PAM sequence is incomplete because of hitting the start or end of a read; these were also assumed to be imperfect PAMs (though some would be genuine). Changing this assumption to include partial PAMs that were subsets of perfect PAMs caused total average immunity numbers to increase and the slope of each curve to decrease. Results including incomplete PAMs were qualitatively similar to the results including all protospacer matches regardless of PAM, highlighting the importance of a perfect PAM match for functional immunity (*Figure 7—figure supplement 2* and *Figure 7—figure supplement 3*; *Figure 73*).

## Calculating average immunity

To calculate average immunity between bacteria and phage, we interpolated spacer and protospacer counts between sequenced time points using the shortest interval between experimental time points

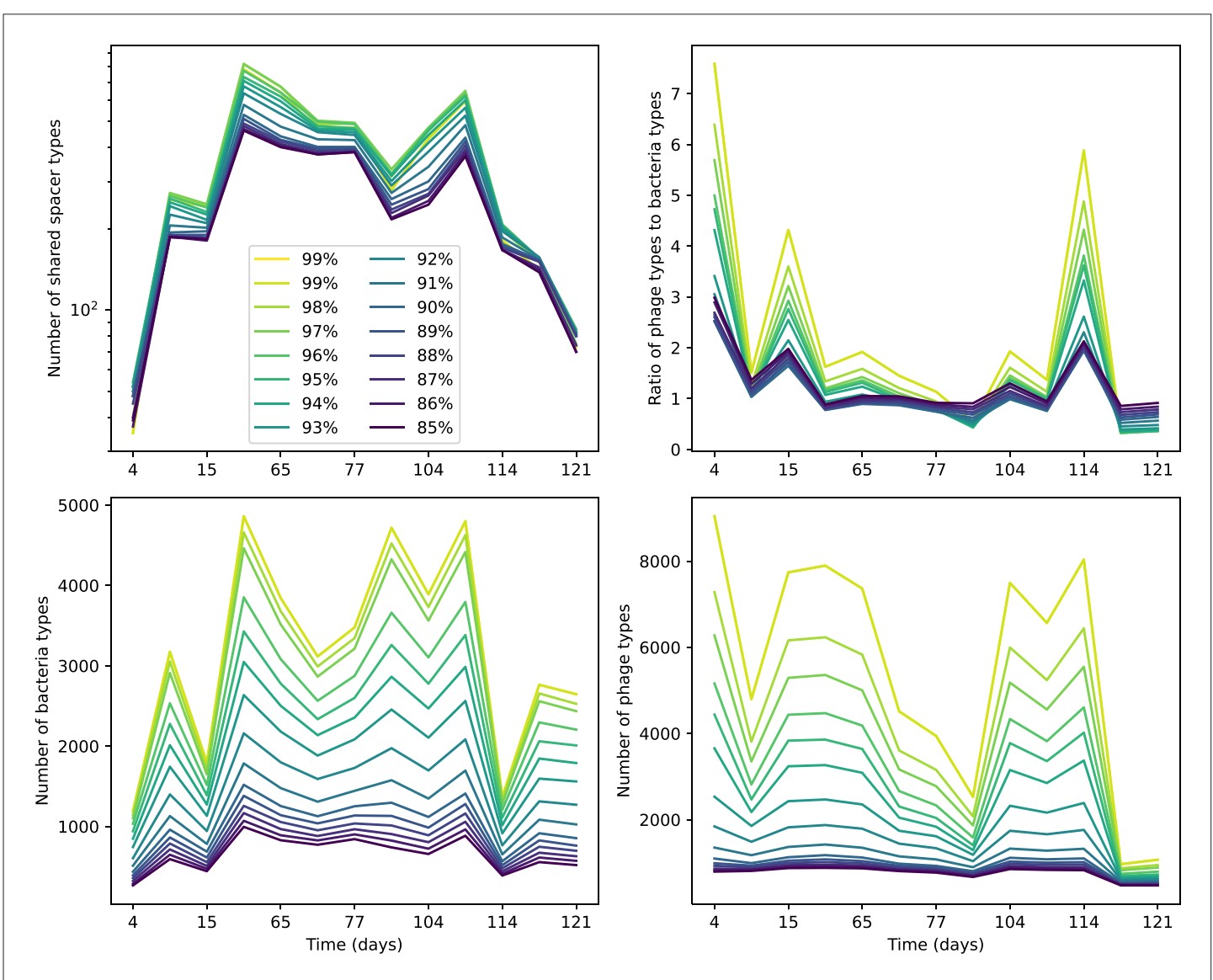

**Figure 73.** Number of shared spacer types between bacteria and phage (top left), ratio of the number of phage types to the number of bacteria types (top right), number of bacteria types (bottom left), and number of phage types (bottom right) as a function of the sampling date for data we analysed from *Paez-Espino et al., 2015*. Colours indicate different similarity grouping thresholds. Spacer counts include wild-type spacers, and protospacers are included only if they possess a perfect PAM. Data is summed over CRISPR1 and CRISPR3. All sequences are included regardless of abundance.

as the sampling frequency from the interpolated data. We removed the first time point and last two time points from the data because the first time point has low bacterial spacer counts and a very large phage population size, and at the last two time points, phage counts are very low because phages are about to go extinct (*Figure 74* and *Figure 73*). Results including all time points are given in *Figure 7— figure supplement 4*; they are qualitatively similar except for large changes at the first and last time points where only the first and last phage population are included in the average, respectively.

We calculated the time-shifted average overlap between bacteria and phage spacer types by comparing bacteria and phage types separated by a time delay and averaging over all points with the same time delay (*Figure 7—figure supplement 4*). As described in 'Theoretical considerations when calculating average immunity from data', we multiplied raw average immunity values by the total number of protospacers to account for multiple protospacers on each phage genome. The number of possible protospacers in phage 2972 for the CRISPR1 locus in *S. thermophilus* (AGAAW PAM) is 231 and the number for CRISPR3 is 465 (696 total). Note that because PAM mutations are common, the true number of protospacers may be less than this hypothetical amount, which would cause us to slightly overestimate average immunity. However, we are also assuming that each bacterium has one effective spacer, and this is likely an underestimate based on estimates of locus length from *Paez-Espino et al., 2015*.

As the similarity threshold increases, the overall overlap goes down slightly (*Figure 7—figure supplement 4* and following) because there are more total types (Figure 72). Interestingly, the number of shared types across the whole dataset is very stable regardless of the clustering threshold: there is a tradeoff between the total number of types (which increases as the similarity threshold increases) and the likelihood that types are shared (which increases as the similarity threshold decreases). At high similarity thresholds, we expect that many spurious types will be created from sequencing errors in spacers, while at low similarity thresholds, genuine escape mutants will be grouped with spacers meaning that the immunity information contained in the overlap is not accurate. There are about 700 unique protospacer sequences in the wild-type phage genome, so we expect the number of unique types to be on the order of 700; they are for bacteria, but the number of phage types in our analysis can be quite a bit higher. For all similarity thresholds, we see that bacterial immunity is higher to phages from the past than phages from the future.

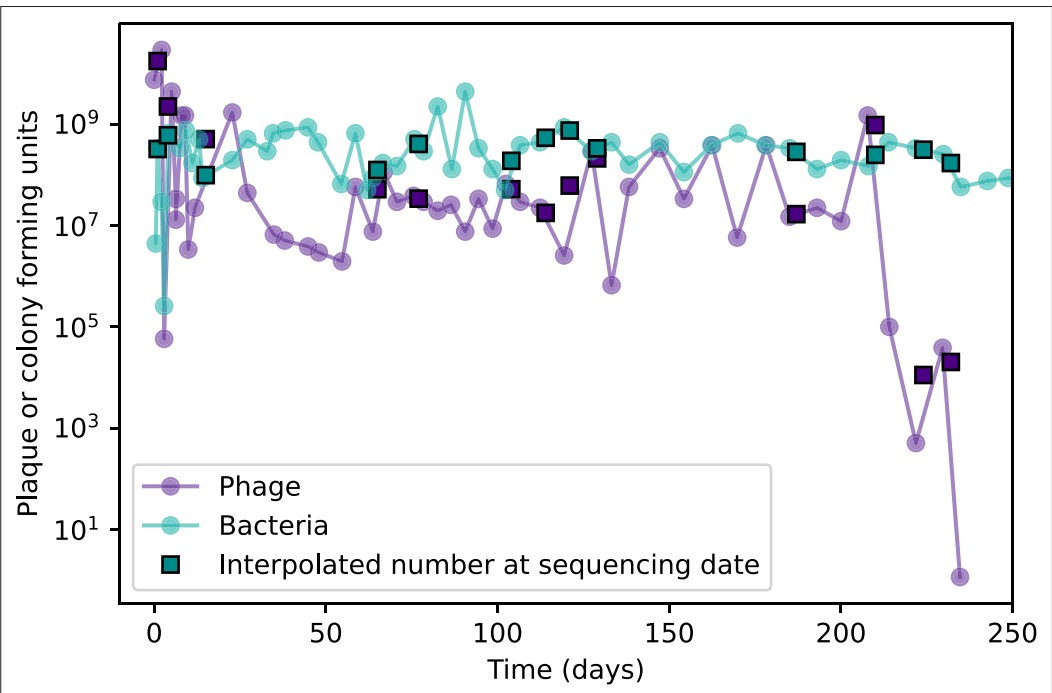

**Figure 74.** Total phage and bacteria population size for the MOI2B experiment in Paez-Espino et al.*Paez-Espino et al., 2015*. Circles are points digitized from Figure 1A in *Paez-Espino et al., 2015*; squares are the population size interpolated to match the sequencing dates in the experiment.

We experimented with different trimming thresholds for the data, removing time points from the beginning and end of the data. The early and late parts of the experiment experienced large fluctuations in population sizes, so if we are interested in steady-state behaviour, it makes sense to remove some time from the beginning and end. But, how much to remove? We can remove the points that look more different from the others either in terms of population size, total reads, or number of types: this could lead us to remove the first three and last three time points. Looking just at population size alone, it makes sense to remove the first time point and last two time points. *Figure 7—figure supplement 4* shows results with different sets of points removed to compare the resulting overlap.

There were no protospacer matches to the reference genome wild-type spacers for CR1 and only a single protospacer match to CR3, even with a lenient 85% similarity threshold. This means the phage has effectively escaped all the wild-type spacers and only the new spacers matter. However, our results differ when wild-type spacers are removed since we also included spacer sequences from the control experiment as wild-type spacers. There are many sequence variants in the control experiment wild-type spacers that do cluster with other spacers, and there are protospacer sequences that cluster with supposed wild-type spacers at lenient grouping thresholds (*Figure 75*). This means that average immunity results with and without wild-type spacers are more similar at high similarity thresholds than at low thresholds.

Our model predicts that phage population sizes should decrease as average immunity increases and that bacterial population sizes should increase as average immunity increases. We calculated average immunity at each experimental time point and compared it to the reported phage and bacterial population size at each time point. Phage and bacteria population sizes were digitized from Figure 1A in *Paez-Espino et al., 2015*. We found that for all similarity groupings, the log-transformed phage population size was significantly negatively correlated with average immunity, while there was no significant correlation with bacterial population size (*Figure 76*). This suggests that the mechanism of phage extinction is increasing bacterial immunity over time, and indeed the average immunity just before phage extinction is very high (*Figure 77*).

## Experimental data from wastewater treatment plant

This dataset is the source for *Figure 6G* in 'Dynamics are determined by diversity' and *Figure 7D* in 'Time-shifted average immunity calculated from data' reveals distinctive patterns of turnover.

We analysed metagenomic sequencing data from *Guerrero et al., 2021a*. This is a longitudinal study of a natural community of bacteria and phages in a wastewater treatment plant. Sixty samples

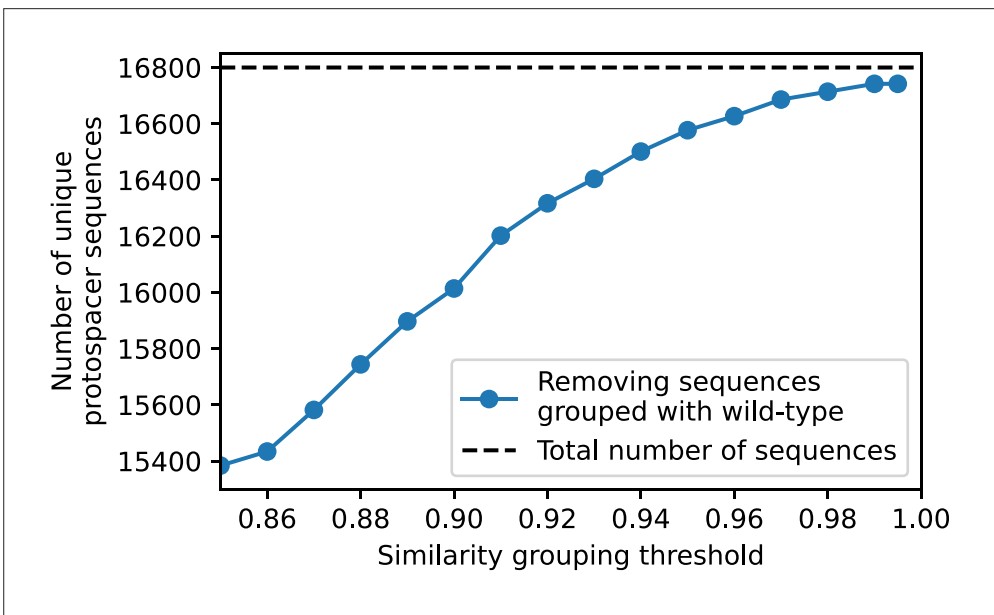

**Figure 75.** Total number of unique protospacer sequences after removing all sequences that cluster with wild-type sequences at different similarity thresholds. Only protospacers with a perfect PAM are included.

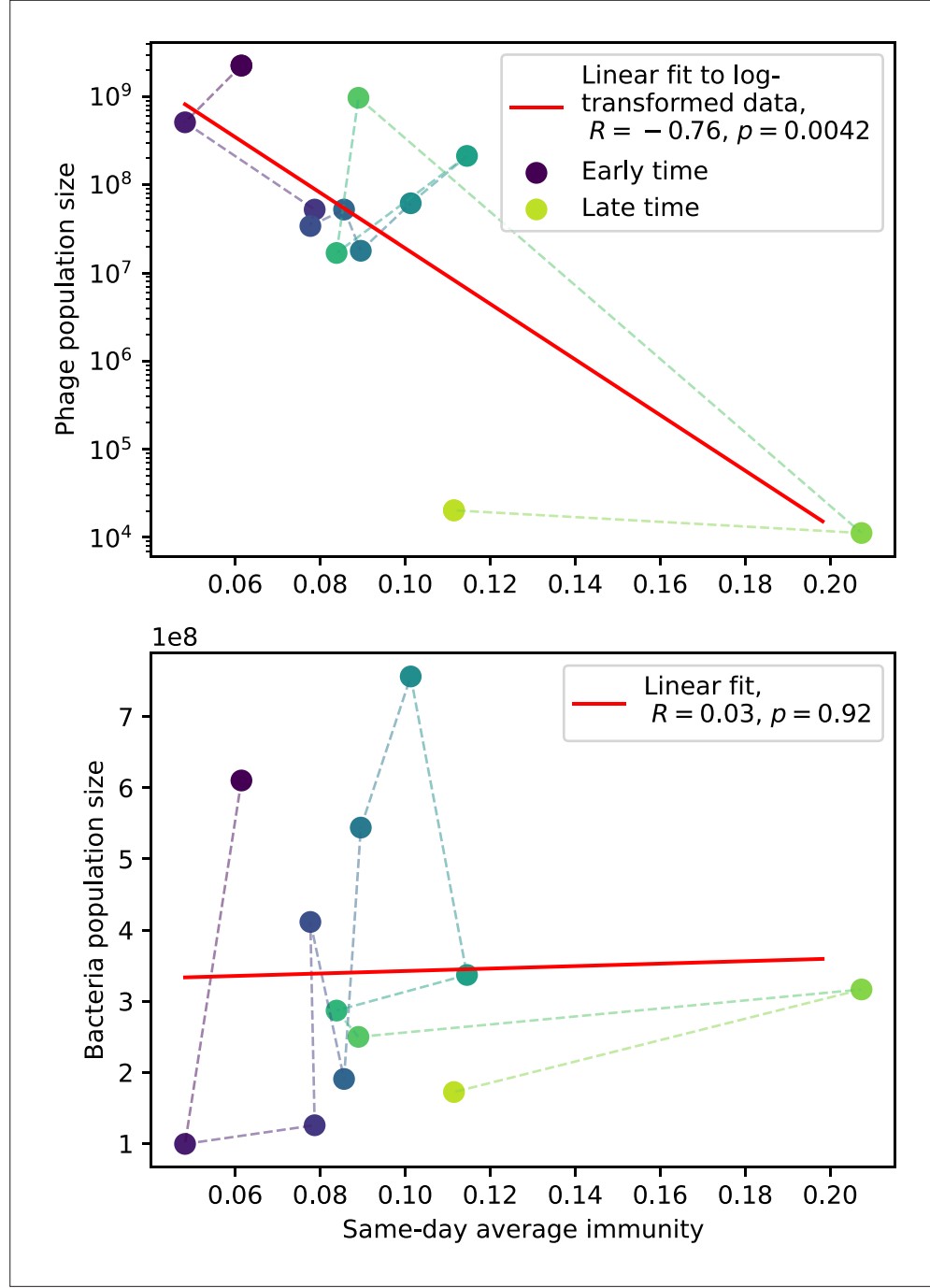

**Figure 76.** Phage (top) and bacteria (bottom) population size vs. average immunity calculated with data from *Paez-Espino et al., 2015*. Protospacers are included if they have a perfect PAM, and all wild-type spacers are included. Spacers and protospacers are grouped with an 85% similarity threshold. Colours from blue to yellow indicate increasing time.

were collected approximately biweekly over a 3-year period from a municipal sewage treatment plant, and whole-genome sequencing was performed on extracted DNA (Illumina platform, 250 bp paired-end reads). They focused on bacteria from the genus *Gordonia*, which is one of the most abundant taxa and was consistently present in samples. They detected CRISPR loci in *Gordonia* genome assemblies, then searched for matches to the two identified CRISPR repeats in all reads to detect other spacers. They constructed graphs of all the linked spacers based on spacer adjacency

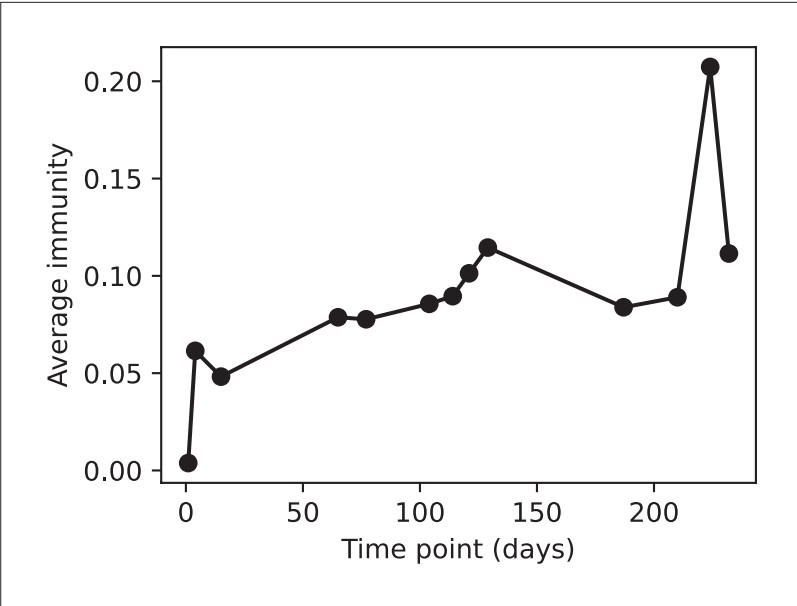

**Figure 77.** Average immunity vs time in the MOI-2B experiment. Protospacers are included if they have a perfect PAM, and all wild-type spacers are included. Spacers and protospacers are grouped with an 85% similarity threshold.

within reads. They searched for matches to spacers in the metagenomic data, then used matches to identify contigs that might be phages, resulting in two draft phage genomes (DC-56 and DS-92) that they followed over time. They searched for protospacers within these phage genomes using blast. They did not find matches between the *Gordonia* spacers and any other contigs from the dataset, suggesting that those two phages are the only two from which *Gordonia* is acquiring spacers. They used Crass and metaCRT to detect other possible CRISPR arrays within the unassembled data and other contigs; they did not find and/or do not discuss any other detected CRISPR loci. They found a broad fan-like network of CRISPR arrays within *Gordonia* that reflect new spacer acquisition, many of which matched the phage genomes. Older, conserved spacers largely did not match the phage genomes. They found that peaks in phage genome coverage (a proxy for abundance) over time were highly correlated with bacterial coverage, suggesting that their population dynamics are tightly coupled. They identified a PAM sequence, GTT (5') for the CRISPR1 locus, but did not find that mutations occurred more frequently in the protospacer regions or PAM regions of the phage genomes than elsewhere in the genome.

Following the same analysis procedure we used for data from *Paez-Espino et al., 2015* ('Laboratory coevolution experimental data'), we used the two reported *Gordonia* CRISPR repeats (GTGC TCCCCGCGCGAGCGGGGATGATCCC for CRISPR1 (type I) and ATCAAGAGCCATGTCTCGCT GAACAGCGAATTGAAAC for CRISPR2 (type III)) to search for spacers in the raw read data using BLAST.

To detect wild-type spacers, we searched for matches to the CRISPR repeats on the *Gordonia* metagenome-assembled genome (MAG) provided on the study's GitHub repository and extracted the sequences between repeat matches. We determined that spacers from the CRISPR1 locus had an average length of 32 nt and spacers from the CRISPR2 locus had an average length of 35 nt.

We detected spacers as sequences between or adjacent to repeat sequences. To maximize the number of genuine spacers detected while removing low-quality matches, we kept only full-length alignments to the repeat, unless a shorter alignment was also present on the same read as a full-length alignment (for instance, if the repeat is partially present at the start or end of a read). If a single repeat match was present on a read, we extracted 32 (CRISPR1) or 35 (CRISPR2) nt on either side and labelled these spacers. We set a minimum spacer length of 85% of the expected spacer length and did not keep spacer sequences at the start or end of a read that were shorter than this. If multiple repeat matches were present, we extracted intervening sequences as spacers.

We grouped all extracted spacers using the AgglomerativeClustering method from SciPy with an 85% similarity threshold and the 'average' linkage criterion. We performed this grouping separately for CRISPR1 and CRISPR2 spacers. Each spacer sequence was then labelled with a group type identifier.

To detect protospacers, we first blasted all reads against the *Gordonia* MAG and the two reported phage genomes (DC-56 and DS-92). We next blasted all unique detected spacer sequences from the previous step against all the reads, removing any query sequences that are a perfect subset of another sequence and any sequences that were more than 30% N nucleotide. This detects both potential protospacer sequences and the original spacers themselves. To isolate protospacers from spacers, we kept only results that were on reads that did not match the CRISPR1 or CRISPR2 repeat. Some reads matched both the bacterial genomes and the phage genomes; we removed matches for which the log-difference in e-value between phage genome match and bacteria genome match was less than 25, that is, the phage genome e-value must be lower than the bacteria e-value by a factor of $10^{25}$ to be considered not a bacteria match. *Figure 78* shows the e-value differences for reads that matched both phage and bacteria for one time point.

We kept only potential protospacers that were 85% of the expected spacer length or larger. If there were multiple hits on a read from the same spacer type (but different sequences), we kept only the match with the lowest e-value. We extracted the matched sequence from the read and 10 nucleotides on either side to check for a PAM sequence (all spacers were stored oriented in the same direction relative to the repeat to facilitate comparison and PAM detection). Consistent with the PAM reported

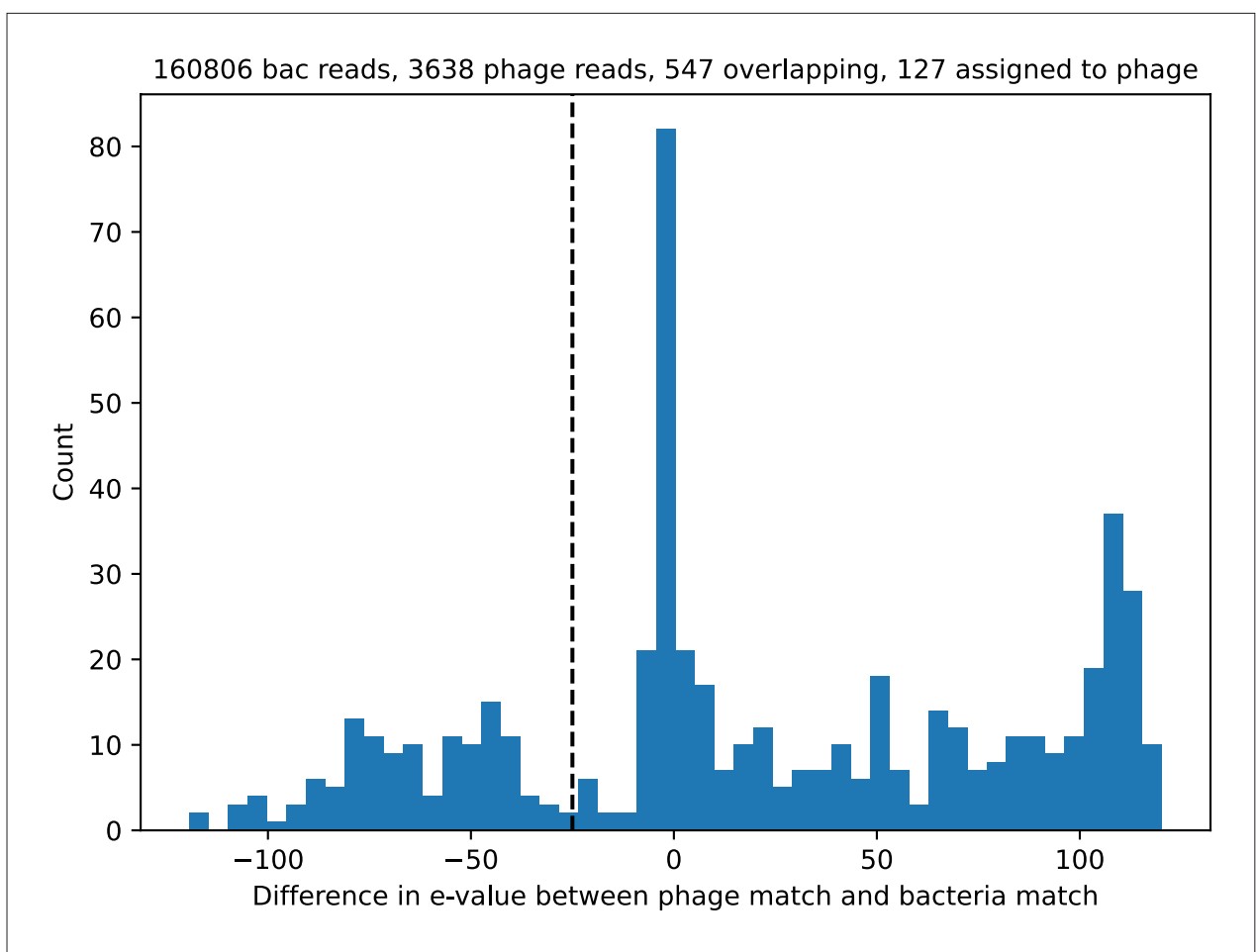

**Figure 78.** Histogram of reads that matched both the *Gordonia* MAG and either phage reference genome vs the base-10 log difference in e-value between the matches for accession SRR9260993. A positive value means the bacteria match has a lower e-value than the phage match. The vertical dashed line indicates the -25 cutoff; matches to the left were considered phage matches, matches to the right were considered bacteria matches.

by Guerrero et al., we identified a GTT PAM at the 5' end of protospacer sequences (*Figure 79*, reverse-complement).

Since reads are paired-end, some reads will overlap, and some spacers may be double-counted in the overlap. We decremented the total spacer or protospacer count for a sequence by 1 if a spacer sequence was present on both ends of a paired read.

We grouped all spacer and protospacer sequences together (separated by CRISPR locus) by several different similarity thresholds between 85 and 99% to assign type labels based on each similarity threshold.

We checked the 10 nt region upstream from each potential protospacer and removed any sequences that did not have a perfect match to the GTT PAM. Since targeting is highly sensitive to PAM mutations *Shah et al., 2013*, we assumed that any deviation from the perfect PAM meant that the protospacer would not be successfully targeted. There are cases where the PAM sequence is incomplete because of hitting the start or end of a read; these were also assumed to be imperfect PAMs (though some would be genuine). We detected 627,555 potential protospacers for the CRISPR1 locus and only 61 potential protospacers for the CRISPR2 locus. Since the CRISPR2 locus is a type III CRISPR system, we expect that there is no PAM for this system and so included all CRISPR2 protospacers; because there are so few, this choice does not affect results. After removing all CRISPR1 protospacers that did not have an adjacent match to the perfect PAM, we were left with 10,2761 protospacers in total representing 19,292 unique sequences and 537 unique types after grouping with an 85% similarity threshold. The total number of unique spacers detected from the CRISPR1 locus over the experiment was 5289 (2078 unique types at 85% similarity), while the total number from the CRISPR2 locus was an order of magnitude lower at 972 (357 unique types at 85% similarity). In total, 28,386 CRISPR1 spacers and 3353 CRISPR2 spacers were detected.

Our detection of reads that matched the *Gordonia* MAG and phage genomes is closely correlated with the reported coverage from the study (*Figure 80*).

The number of unique protospacer and spacer types we detected fluctuated considerably over the course of the study (*Figure 81*, *Figure 82*). The number of shared types as well as the ratio of phage types to bacteria types also fluctuated considerably.

About 40% of spacer and protospacer types remained throughout the time series without going extinct at an 85% similarity threshold (*Figure 83*). In data from *Paez-Espino et al., 2015*, about 20% of original types remained at the end of the experiment (*Figure 6E*); however, the fraction of types remaining decreases steadily in data from *Paez-Espino et al., 2015* unlike in this dataset.

## Calculating average immunity

To calculate average immunity between bacteria and phage, we interpolated counts between sequenced time points using a 14-day spacing (the most common sampling interval in the data). We calculated the time-shifted average overlap between bacteria and phage spacer types by comparing bacteria and phage types separated by a time delay and averaging over all points with the same time delay (*Figure 7—figure supplement 5*). As described in 'Theoretical considerations when calculating

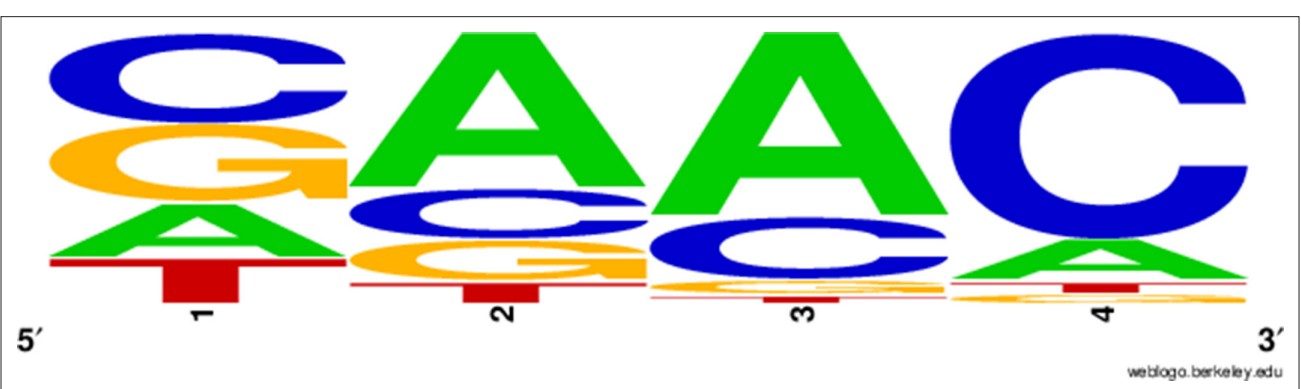

**Figure 79.** Probability logo for the four nucleotides at the 5' end of potential protospacer sequences, generated with WebLogo; the spacer sequence starts at the right edge of the logo meaning that the reverse-complement PAM is GTT.

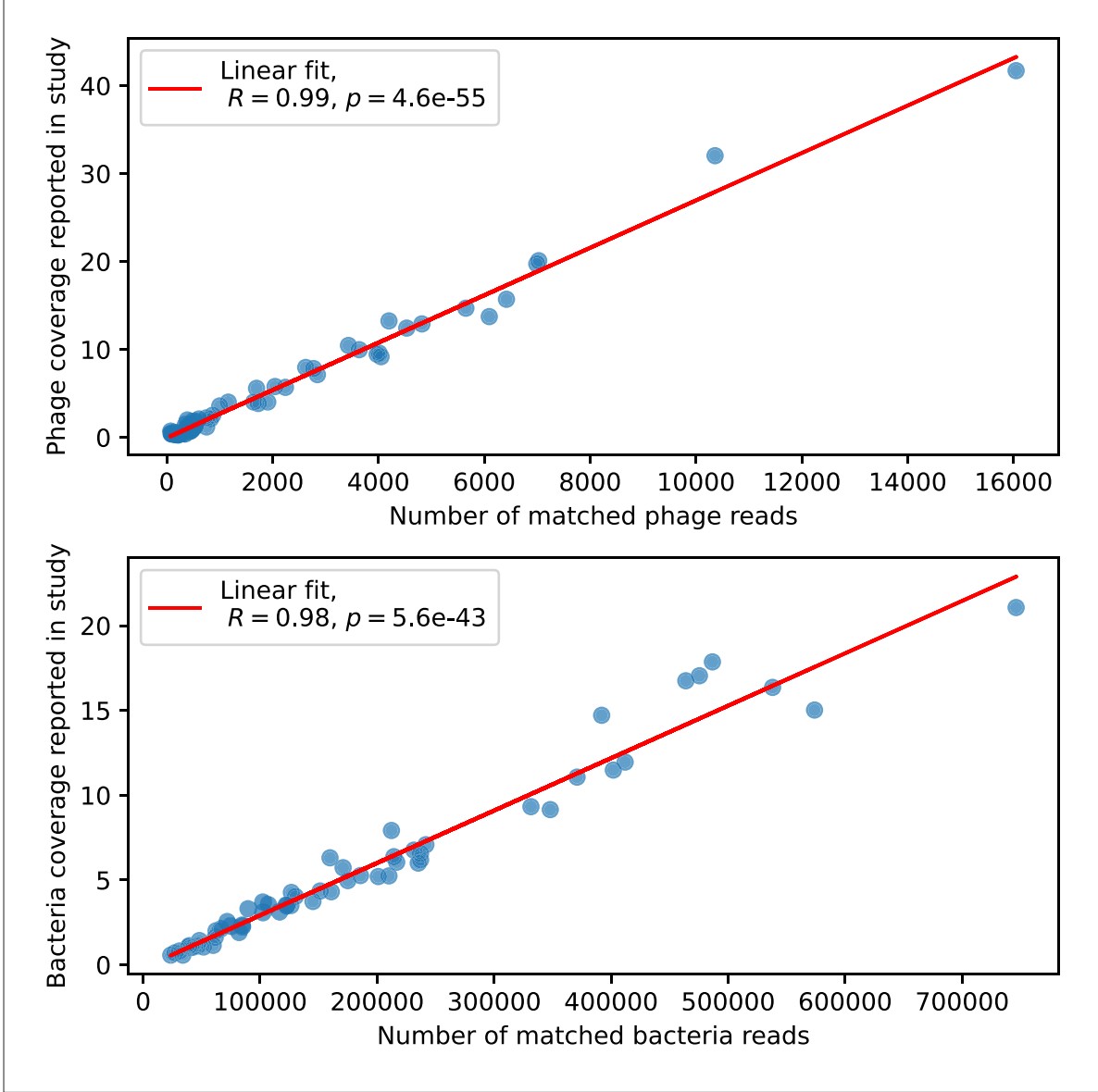

**Figure 80.** Phage genome (top) and bacteria genome (bottom) coverage, digitized from Figure 2D of *Guerrero et al., 2021a* vs. number of reads assigned to phage or bacteria from our analysis. Each marker is a separate time point.

average immunity from data', we multiplied raw average immunity values by the average number of protospacers with the GTT PAM from the phage DC-56 and DS-92 genomes (1956 protospacers) to account for multiple protospacers on each phage genome. Note that because PAM mutations are common, the true number of protospacers may be less than this hypothetical amount, which would cause us to slightly overestimate average immunity. However, we are also assuming that each bacterium has one effective spacer, and this is likely an underestimate. As the similarity threshold increases, the overall overlap goes down slightly because there are more total types. As in the data from *Paez-Espino et al., 2015*, the number of shared types across the whole dataset is insensitive to the clustering threshold: there is a tradeoff between the total number of types (which increases as the similarity threshold increases) and the likelihood that types are shared (which increases as the similarity threshold decreases) (*Figure 82*).

We experimented with different trimming thresholds for the data, removing time points from the beginning and end of the data series before calculating time-shifted average immunity

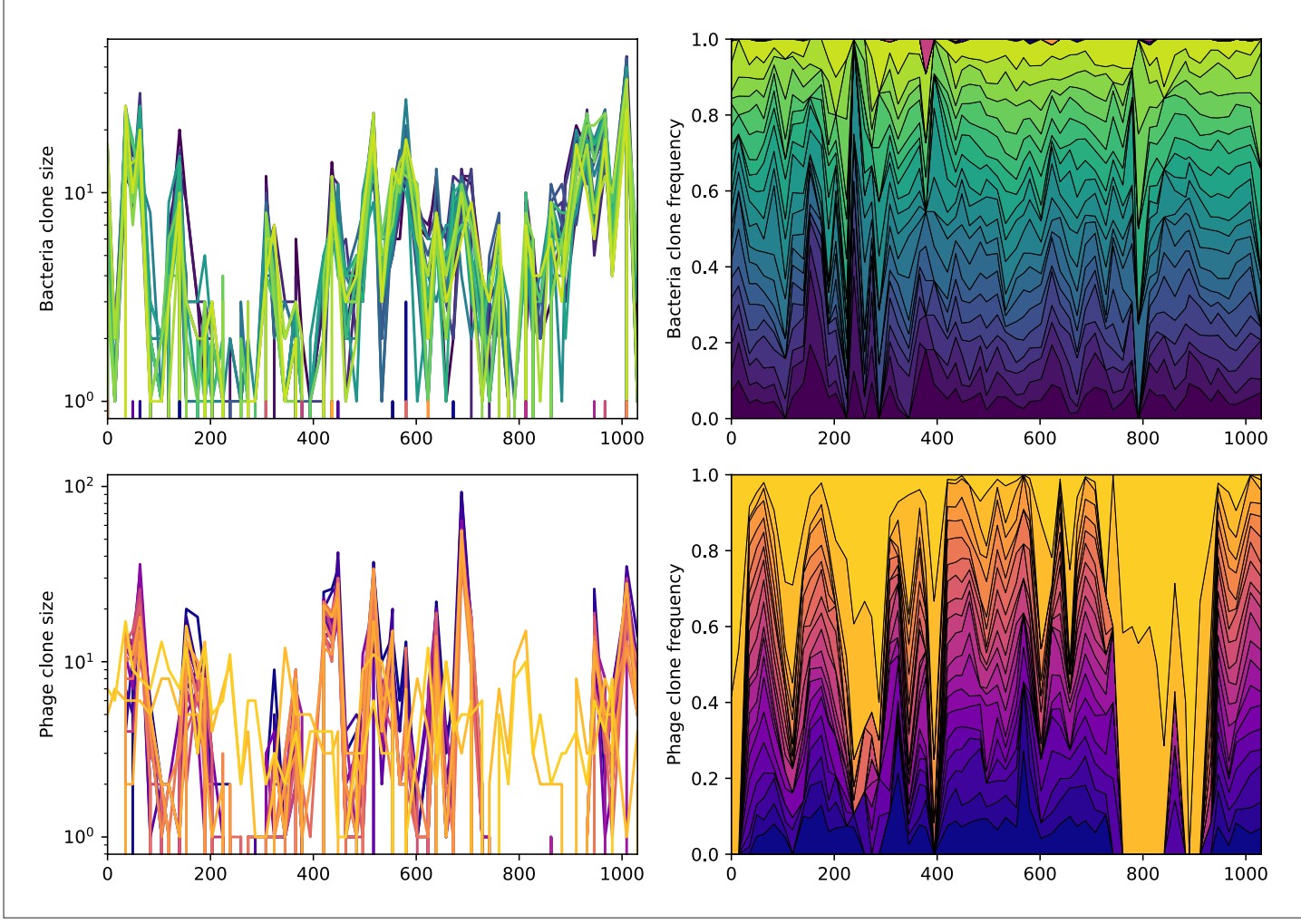

**Figure 81.** Abundance over time for the 20 largest bacteria clones (top) and phage clones (bottom) over time for data from *Guerrero et al., 2021a*. The left panels show absolute counts, and the right panels show fractional abundance for the included types. None of the top types are shared between phage and bacteria.

(*Figure 7—figure supplement 8*, *Figure 7—figure supplement 9* and *Figure 7—figure supplement 10*). For all similarity thresholds and most trimming choices, there is a peak in average immunity at zero time shift that decays in both the past and future directions (*Figure 7—figure supplement 5*). However, if enough data is trimmed from the beginning to remove the large spikes in average immunity between days 200 and 400 (*Figure 84*), the central peak is quite diminished or removed altogether (*Figure 7—figure supplement 9* and *Figure 7—figure supplement 10*). The standard deviation of average immunity is large near the zero time shift (*Figure 7—figure supplement 5*), which is because average immunity is highly variable across the time series: there is a large spike in average immunity near the middle of the time series (*Figure 84*).

Because the variability in average immunity between time points was very high, we performed a Wilcoxon signed-rank test between the average immunity at zero time delay and the average immunity at a time delay ±200 and ±500 (arbitrarily chosen). We paired each shifted time point with its corresponding average immunity at zero delay for each bacterial abundance (*Figure 85*). Using a paired test answers the following question for each time point: is the overlap between bacteria and past phages significantly lower than the overlap in the present, and is the overlap between bacteria and future phages significantly lower than the overlap in the present? The test returns a statistic that is the sum of the ranks of differences greater than zero; a larger number indicates more asymmetry between the two compared datasets. We found that past immunity after 200 days is significantly lower than present (Wilcoxon statistic $Z = 1241$, p-value $p = 0.008$), but that immunity after 500 days is

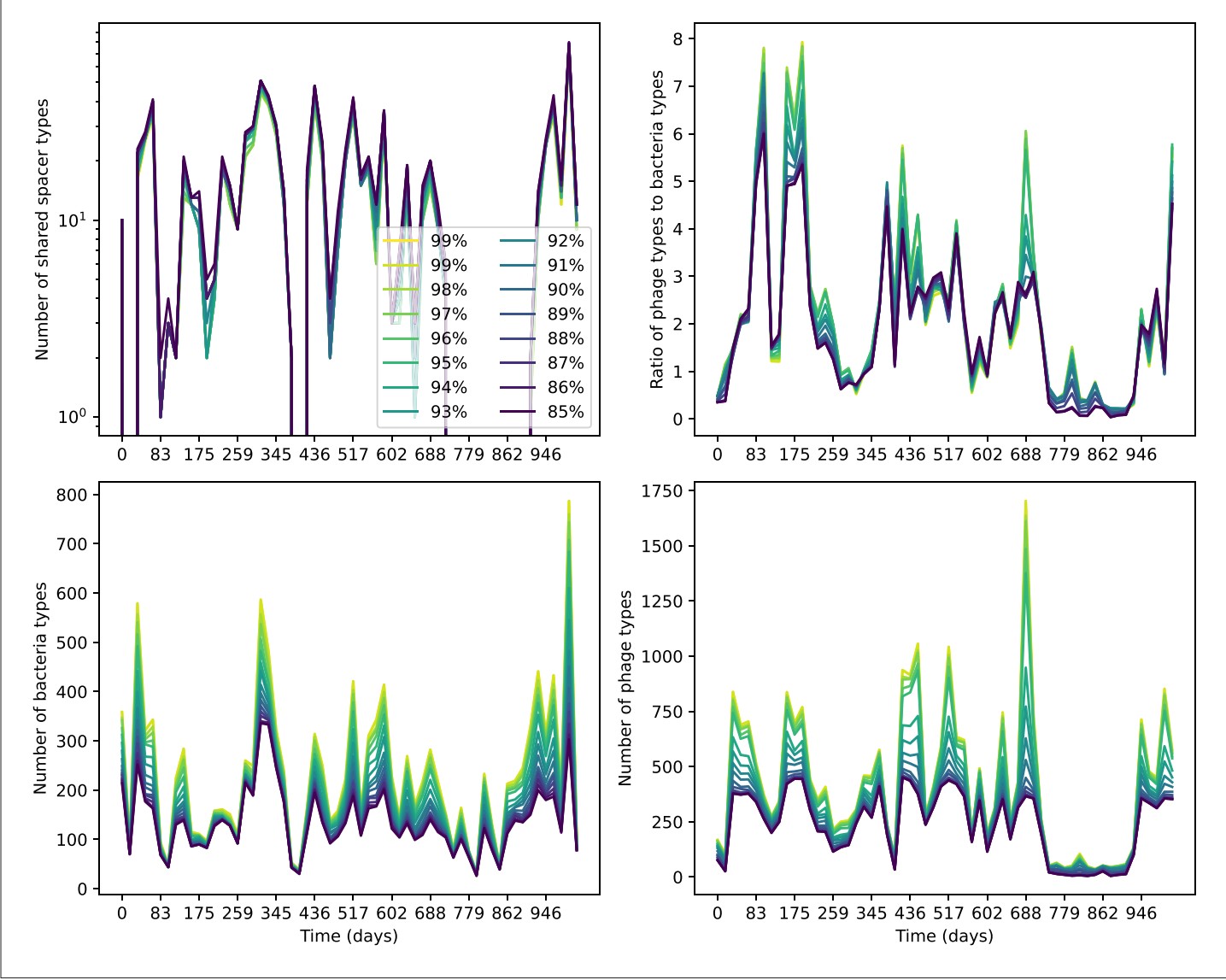

**Figure 82.** Number of shared spacer types between bacteria and phage (top left), ratio of the number of phage types to the number of bacteria types (top right), number of bacteria types (bottom left), and number of phage types (bottom right) as a function of the sampling date for data we analysed from *Guerrero et al., 2021a*. Colours indicate different similarity grouping thresholds. Spacer counts include wild-type spacers, and protospacers are included only if they possess a perfect PAM. Data is summed over CRISPR1 and CRISPR2.

not significantly lower than present ($Z = 392, p = 0.27$). When comparing all time points with present, only a small range of past immunity values were significantly lower than present (*Figure 86*). For both time delays, we found that future immunity is significantly lower than present ($Z = 580, p = 0.0012$ for 500 days and $Z = 1293, p = 0.003$ for 200 days). Interestingly, these significance values are not symmetric if we pose the question from the perspective of phage: the overlap between phage and future bacteria at 500 days is lower than the overlap for present phages ($Z = 507, p = 0.024$), and the overlap between phage and past bacteria is lower than the overlap at present ($Z = 388, p = 0.027$). This asymmetry, that bacteria are generally more immune to all past phages while phage are not more infective against all past bacteria, is qualitatively the same as that reported by Dewald-Wang et al. in an explicit time shift study of bacteria and phage immunity and infectivity in chestnut trees *Dewald-Wang et al., 2022*.

We confirmed that the past and future shifted average immunity becomes less correlated with the zero-shift average immunity as the time shift increases. *Figure 87* shows the time-shifted average

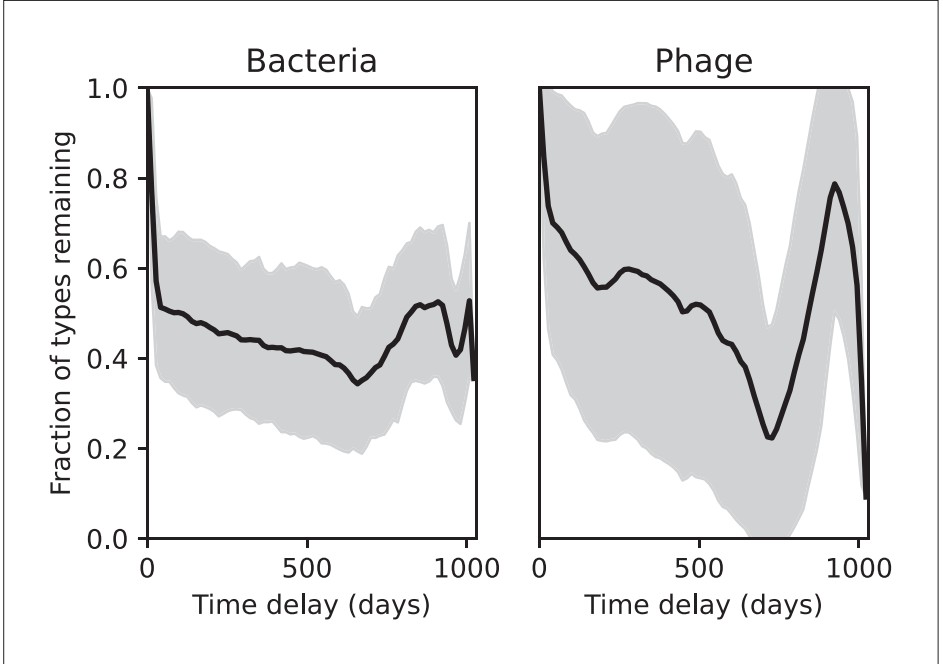

**Figure 83.** Fraction of spacer types (left) and protospacer types (right) remaining as a function of time delay, averaged over the entire time series from *Guerrero et al., 2021a*. Types are grouped with an 85% similarity threshold. The grey shaded region is the standard deviation across averaged data.

immunity as a function of the zero-shift average immunity for three time shift values, and we see that average immunity is highly correlated at small time shifts but becomes less correlated at larger shifts.

We randomly shuffled time point labels for bacteria and phages separately (*Figure 7—figure supplement 6*) and together (*Figure 7—figure supplement 7*) and calculated average immunity with

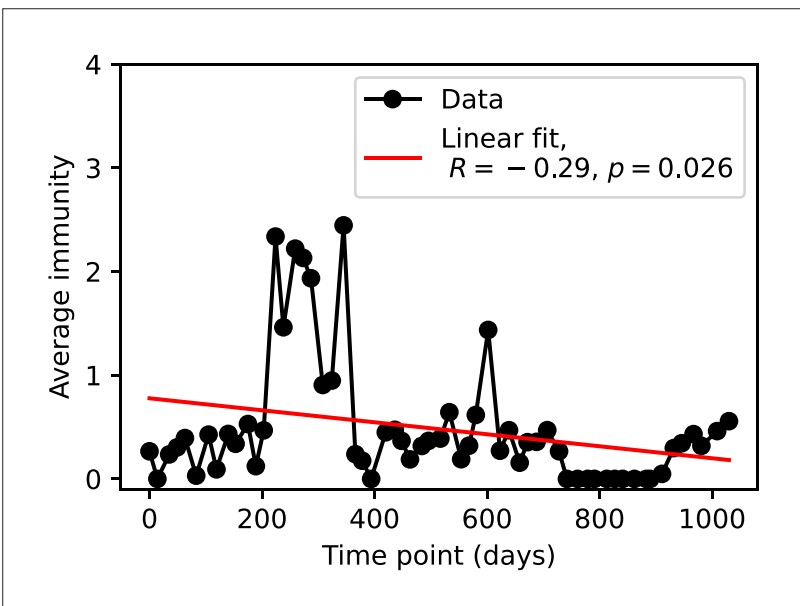

**Figure 84.** Average immunity at each time point in the time series. Raw values cannot be larger than 1, but plotted values are multiplied by 1956, the average number of protospacers with the GTT PAM from the phage DC-56 and DS-92 genomes, yielding some values larger than 1.

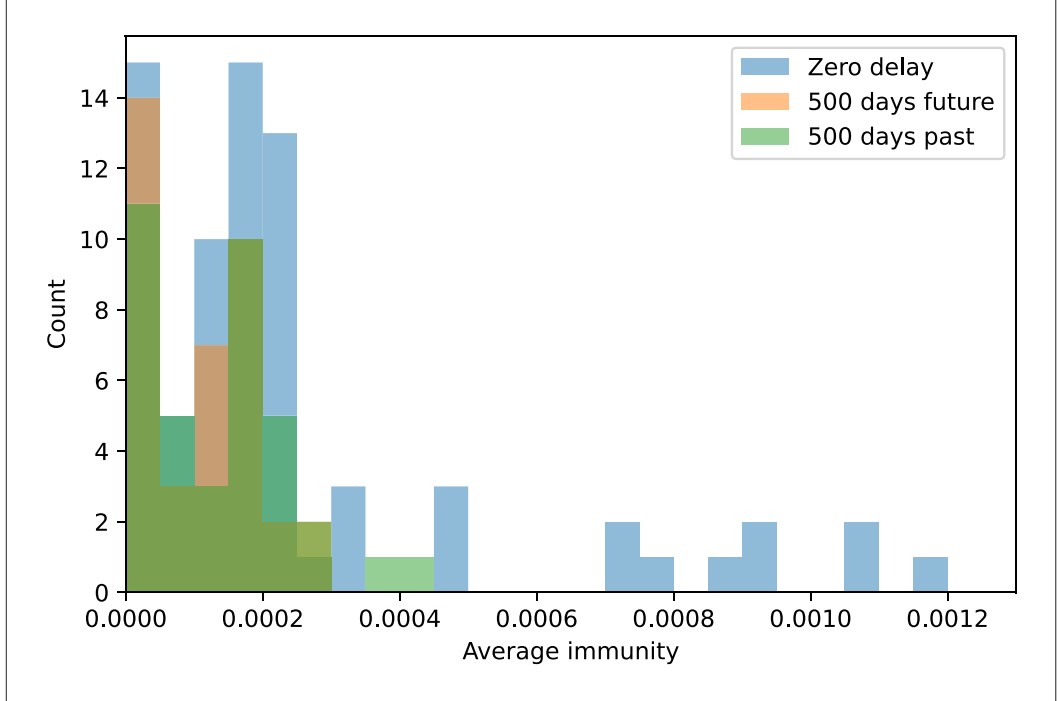

**Figure 85.** Distribution of average immunity values at zero time shift (blue) and at 500 days time shift for future phages (orange) and 500 days time shift for past phages (green).

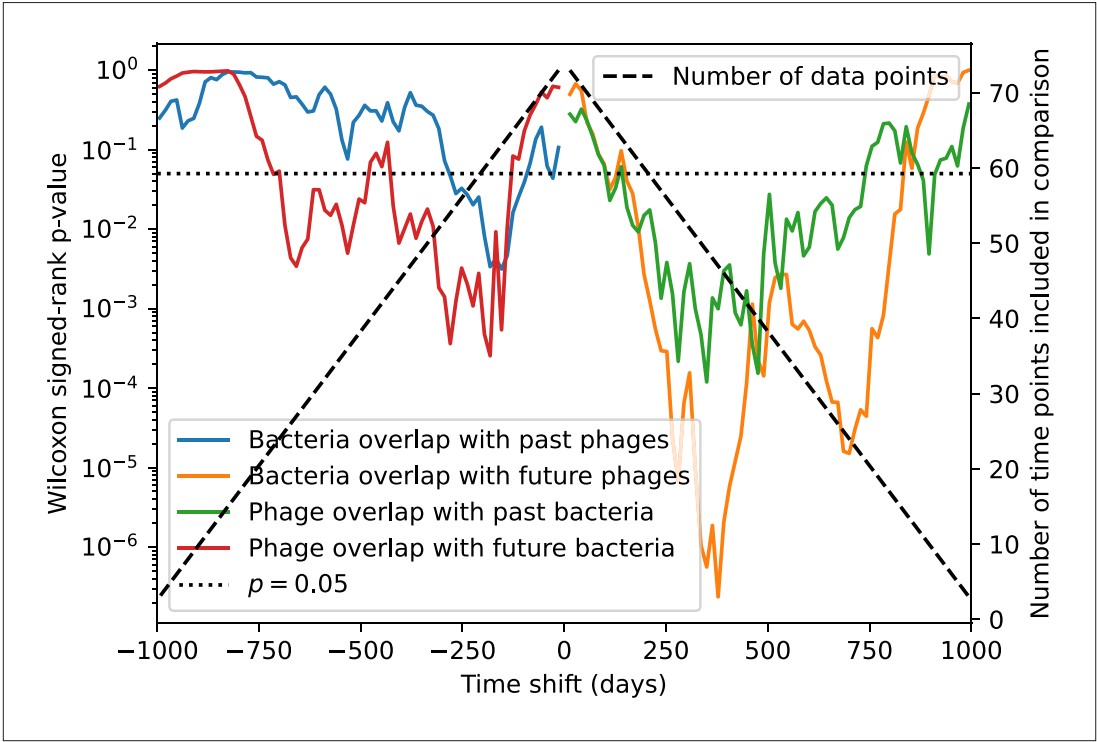

**Figure 86.** p-values for the Wilcoxon signed-rank test comparing the average immunity at zero time shift with all other time shifts. Bacteria immunity against past phages is generally not significantly lower than bacterial immunity against current phages (blue), while bacterial immunity against future phages is significantly lower than immunity against current phages for almost all time shifts (orange). The time shifts are not symmetric; bacterial overlap with past phages is not necessarily the same as phage overlap with future bacteria and vice versa. The number of points that are available to compare decreases as the time shifts get larger (black dashed line).

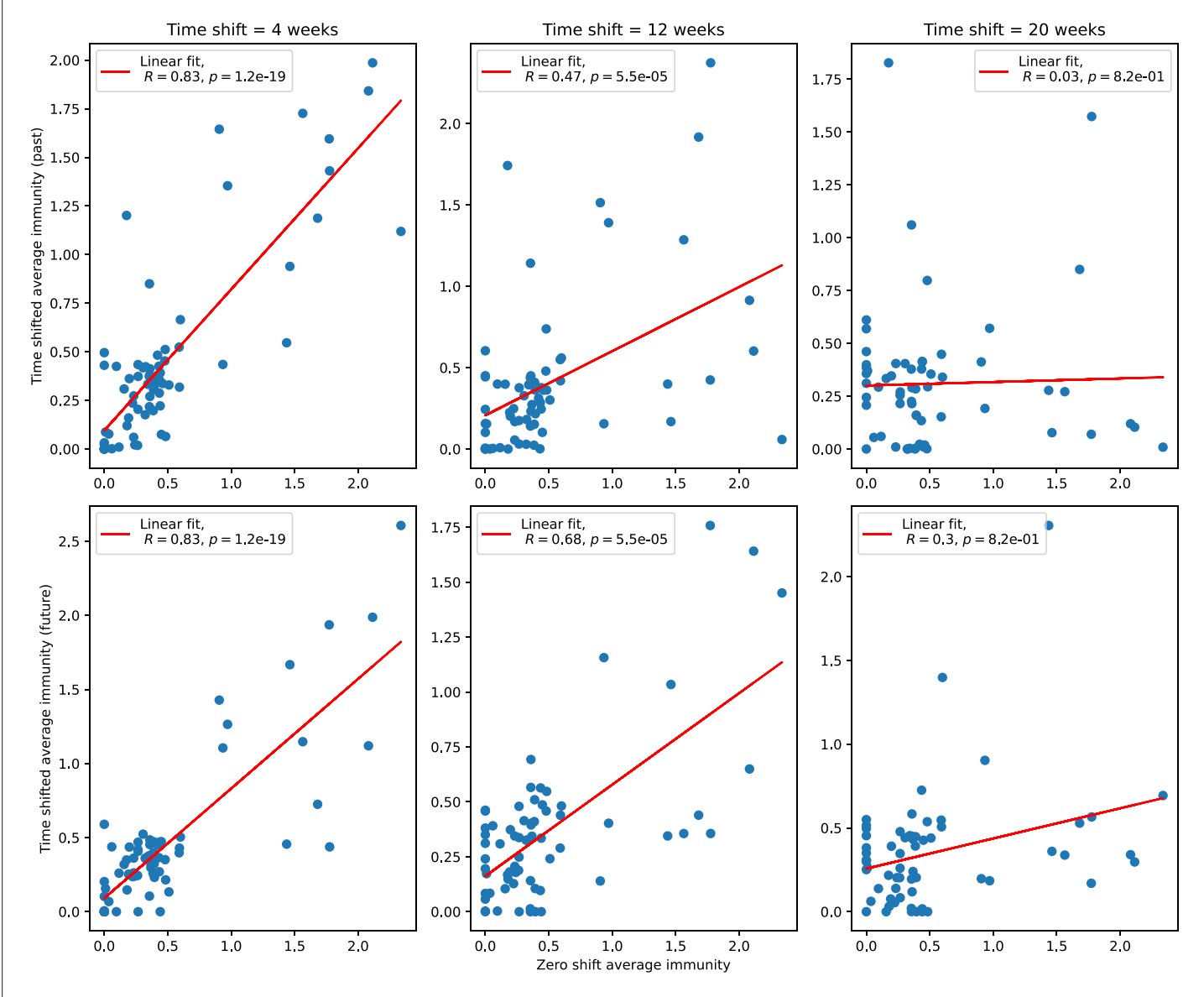

**Figure 87.** Time-shifted average immunity for each time point as a function of the zero-shift average immunity of that data point for shifts comparing past phages (top row) and future phages (bottom row).

the bootstrapped data. The average immunity trend as a function of time shift disappears in the bootstrapped data even if bacteria and phage clone abundances remain matched at each time point.

Unlike in the data from *Paez-Espino et al., 2015*, we did not find a correlation between phage or bacteria coverage and average immunity (*Figure 88*), though *Guerrero et al., 2021a* did report that bacteria and phage coverage was correlated and that periods of high phage coverage coincided with higher abundance of *Gordonia* strains lacking matching spacers, suggesting that the expected negative correlation between phage abundance and bacterial immunity may be playing out at some level but may be confounded by generally higher detected average immunity when coverage is high.

## Spacer turnover

For all sets of experimentally observed spacers (*Paez-Espino et al., 2015*, *Burstein et al., 2016*, and *Guerrero et al., 2021a*), we calculated spacer turnover over time. We interpolated spacer abundances

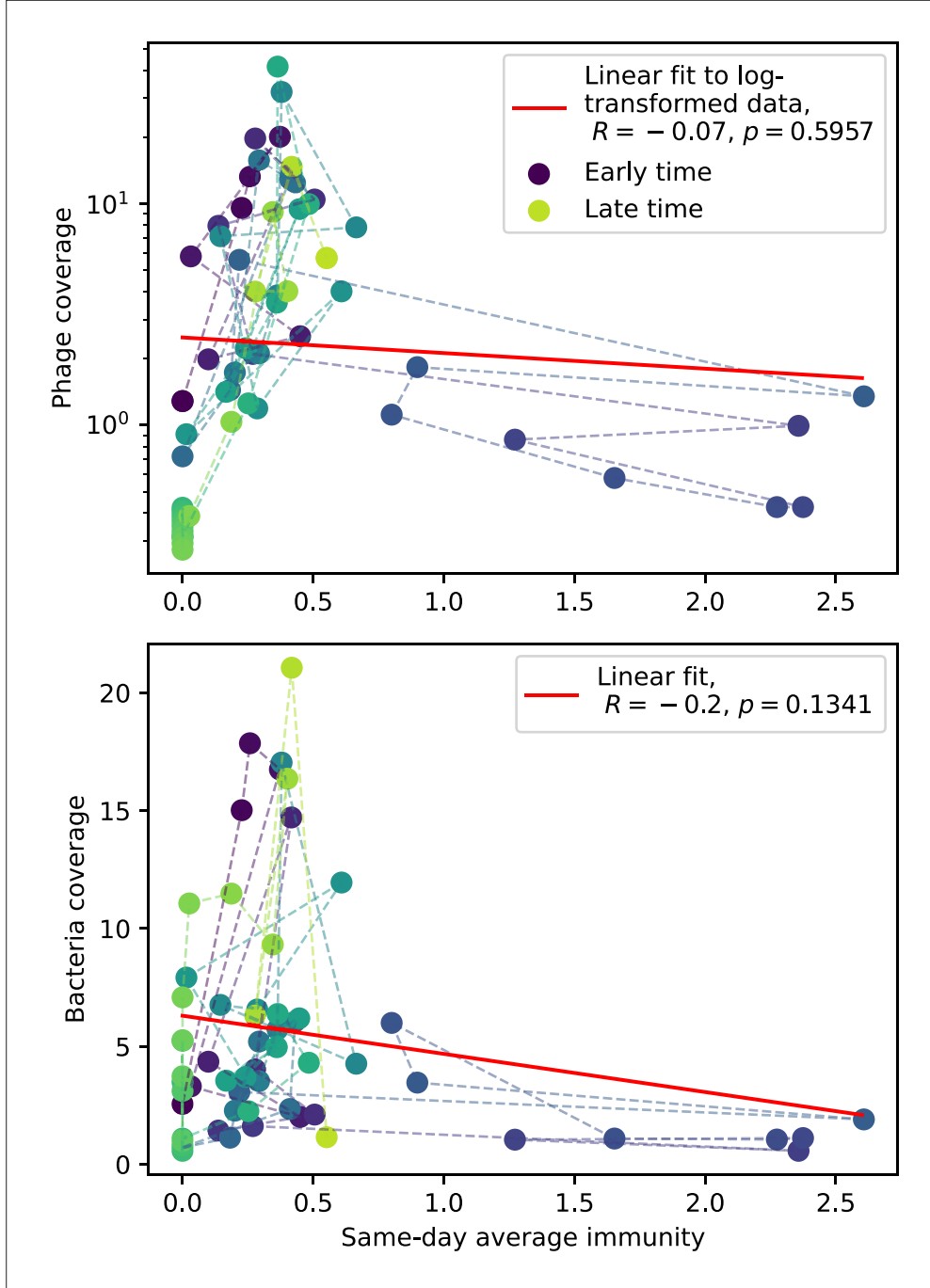

**Figure 88.** Phage (top) and bacteria (bottom) population size vs. average immunity calculated with data from *Guerrero et al., 2021a*. Protospacers are included if they have a perfect PAM, and all wild-type spacers are included. Spacers and protospacers are grouped with an 85% similarity threshold. Colours from blue to yellow indicate increasing time.

grouped by 85% similarity over time using the smallest time delay between samples, then calculated the average fraction of spacer types remaining as a function of time delay across the entire dataset.

## Acknowledgements

We acknowledge helpful discussions and comments from all the members of the Goyal and Zilman groups. We thank Vijay Kumar for help with extinction probability of phage clones.

## Additional information

### Funding

| Funder | Grant reference number | Author |
|---|---|---|
| Natural Sciences and Engineering Research Council of Canada | Vanier Canada Graduate Scholarship | Madeleine Bonsma-Fisher |
| Ministry of Colleges and Universities | Queen Elizabeth II Graduate Scholarship in Science & Technology | Madeleine Bonsma-Fisher |
| Walter C. Sumner Foundation | Walter C. Sumner Memorial Fellowship | Madeleine Bonsma-Fisher |
| Natural Sciences and Engineering Research Council of Canada | Discovery Grant RGPIN-2015 | Sidhartha Goyal |
| Natural Sciences and Engineering Research Council of Canada | Discovery Grant and RGPIN-2021 | Sidhartha Goyal |

The funders had no role in study design, data collection and interpretation, or the decision to submit the work for publication.

### Author contributions

Madeleine Bonsma-Fisher, Conceptualization, Data curation, Software, Formal analysis, Funding acquisition, Investigation, Visualization, Methodology, Writing - original draft, Writing - review and editing; Sidhartha Goyal, Conceptualization, Resources, Supervision, Funding acquisition, Methodology, Project administration, Writing - review and editing

### Author ORCIDs

Madeleine Bonsma-Fisher http://orcid.org/0000-0002-5813-4664
Sidhartha Goyal http://orcid.org/0000-0002-7452-892X

### Decision letter and Author response

Decision letter https://doi.org/10.7554/eLife.81692.sa1
Author response https://doi.org/10.7554/eLife.81692.sa2

## Additional files

### Supplementary files

• MDAR checklist

### Data availability

We analysed published genetic data from three sources: PRJNA268031 (**Burstein et al., 2016**), PRJNA275232 (**Paez-Espino et al., 2015**), and PRJNA484416 (**Guerrero et al., 2021a**). We also used phage genomes associated with **Guerrero et al., 2021a** available on GitHub at https://github.com/GuerreroCRISPR/Gordonia-CRISPR (**Guerrero et al., 2021b**). Processed spacer and protospacer sequences for all datasets, along with source code and data for all main text figures, is available on GitHub at https://github.com/mbonsma/CRISPR-dynamics-model (copy archived at **Bonsma-Fisher and Goyal, 2023**). Raw simulation data has been uploaded to Dryad: https://doi.org/10.5061/dryad.sn02v6x74.

The following dataset was generated:

| Author(s) | Year | Dataset title | Dataset URL | Database and Identifier |
|---|---|---|---|---|
| Bonsma-Fisher M, Goyal S | 2022 | Simulation data from: Dynamics of immune memory and learning in bacterial communities | https://doi.org/10.5061/dryad.sn02v6x74 | Dryad Digital Repository, 10.5061/dryad.sn02v6x74 |

The following previously published datasets were used:

| Author(s) | Year | Dataset title | Dataset URL | Database and Identifier |
|---|---|---|---|---|
| Burstein D, Sun CL, Brown CT, Sharon I, Anantharaman K, Probst AJ, Thomas BC, Banfield JF | 2016 | Rifle, CO metagenome from aquifer | https://www.ncbi.nlm.nih.gov/bioproject/PRJNA268031 | NCBI BioProject, PRJNA268031 |
| Paez-Espino D, Sharon I, Morovic W, Stahl B, Thomas BC, Barrangou R, Banfield JF | 2015 | Streptococcus thermophilus infected with lytic phage 2972 | https://www.ncbi.nlm.nih.gov/bioproject/PRJNA275232 | NCBI BioProject, PRJNA275232 |
| Guerrero LD, Pérez MV, Orellana E, Piuri M, Quiroga C, Erijman L | 2020 | Time series genome centric analysis in activated sludge microbial communities | https://www.ncbi.nlm.nih.gov/bioproject/PRJNA484416 | NCBI BioProject, PRJNA484416 |

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

## Appendix 1

### Relationship of average immunity to Morisita–Horn index

This appendix is presented to contextualize our work with the broader language of diversity in ecology.

In certain limits, average immunity can be related to the Morisita–Horn similarity index $\Psi$ (*Wolda, 1981*; *Childs et al., 2012*), where instead of comparing two populations from different timepoints or different regions, we are comparing bacteria and phage populations scaled by the immune benefit of CRISPR. If we assume that a matching spacer provides perfect immunity ($e = 1$) and there is no cross-reactivity, then $p_V(i, j) = p_V(1 - \delta_{ij})$ and average immunity is given by

$$1 - \frac{\sum_{i,j} n_B^i n_V^j p_V(i,j)}{p_V \sum_{i,j} n_B^i n_V^j} = 1 - \frac{p_V \sum_{i,j} n_B^i n_V^j (1 - \delta_{ij})}{p_V \sum_{i,j} n_B^i n_V^j} = \frac{\sum_i n_B^i n_V^i}{n_B^s n_V} \tag{93}$$

This is the Morisita overlap index between bacteria and phage without the factor of $\frac{2}{D_B + D_V}$, where $D_B$ and $D_V$ are the Simpson's diversity indices for bacteria with spacers and phages, respectively. In the limit that all phage and bacteria clones are equal sizes ($n_B^i \approx n_B^s/m$, $n_V^i \approx n_V/m$), the factor $\frac{2}{D_B + D_V} \approx m$, so we have $\Psi \approx m e_{eff}$. We found in most simulations that effective $e \approx e/m$, which implies that the Morisita overlap between bacteria and phage is constant across all parameters we study. A constant Morisita overlap implies that the population diversity evolves to attain the highest possible overlap. The full average immunity includes pairwise comparisons between all species and may be quite different from the Morisita overlap index.

## Appendix 2

## A neutral model of non-coevolving bacteria

This section provides the mathematical background for the statement that neutral diversity depends linearly on mutation rate made in 'Phages drive stable emergent sequence diversity'.

In our results for the diversity in a coevolving population of bacteria and phage, we find that diversity approximately scales like $m \approx e\mu\eta^{\frac{1}{3}}$ (Approximation for $m$). Intuitively this dependence seems quite weak, but we would like to know what to compare this to: what is the expectation for diversity in a population with mutations but without coevolution? An appropriate comparison point is a neutral model of a bacterial population that grows and dies; the birth and death rates are exactly matched to give a constant size at steady state. We also include the possibility of mutations: a bacterium may mutate to a new type with probability $\mu$ per division, and we assume that each mutation is to a new, never-before-seen type (infinite alleles model).

Let us consider a population of bacteria that divides with rate $g$, dies with rate $F$, and mutates with rate $\mu$. We can write a master equation for the number of bacterial clones of size $k$. New mutants enter at clone size 1, and we assume that the total population size is constant at $N_b = \sum_k kb_k$. Mutations effectively lower the growth rate of a particular clone.

$$\partial_t b_k = (g - \mu)[(k - 1)b_{k-1} - kb_k] + F[(k + 1)b_{k+1} - kb_k] + \delta_{k,1}\mu N_b \tag{94}$$

## Generating function solution

The generating function for the probability distribution $b_k(t)$ is $G(z, t) = \sum_k z^k b_k(t)$. Let us replace the birth and death rates by $\beta$ and $\delta$, respectively. $\beta = g - \mu$, $\delta = F$. Multiplying *Equation 94* with $\sum_k z^k$ and noting that $\partial_z G(z, t) = \sum_k kz^{k-1}b_k(t)$, we get the following differential equation:

$$\partial_t G(z, t) = \partial_z G(z, t)\left(z^2\beta - z(\beta + \delta) + \delta\right) + Dz \tag{95}$$

The term $Dz$ comes from the source term with $D = \mu N_b$.

*Equation 95* can be solved with the method of characteristics (*Van Kampen, 1981*). We parameterize the function $G(z, t)$ with a new variable $s$. Applying the chain rule:

$$\partial_s G(z(s), t(s)) = \frac{\partial G}{\partial z}\frac{\partial z}{\partial s} + \frac{\partial G}{\partial t}\frac{\partial t}{\partial s} \tag{96}$$

And by comparison with *Equation 95*, the characteristic equations are

$$\frac{\partial t}{\partial s} = 1 \tag{97}$$

$$\frac{\partial z}{\partial s} = (1 - z)(\beta z - \delta) \tag{98}$$

$$\frac{\partial G}{\partial s} = Dz \tag{99}$$

From *Equation 97* we see $t = s + C$, so we can choose $t_0 = C = 0$ and replace $s$ with $t$ going forward.

Solving the characteristic equation for $z$ by integrating both sides gives *Equation 100*.

$$\frac{1 - z}{\delta - \beta z}e^{(\beta - \delta)t} = C \tag{100}$$

At $t = 0$, $z$ will pass through some point $z_0$, so we have the initial condition $z(0) = z_0$. With $z_0$ in *Equation 100* at $t = 0$, we get *Equation 101*, where $C$ is given by *Equation 100*.

$$z_0 = \frac{C\delta - 1}{C\beta - 1} \tag{101}$$

The variation of $G$ along this $z$-$t$ curve is given by

$$\partial_z G = \frac{-Dz}{z^2\beta - z(\beta + \delta) + \delta} \tag{102}$$

which has the solution

$$G(z) = \frac{D}{\beta - \delta}\left[-\ln(1-z) + \frac{\delta}{\beta}\ln(\delta - \beta z)\right] + \Omega(C) \tag{103}$$

The constant $\Omega$ is a function of the characteristic $z$-$t$ curve (**Equation 100**). To find the particular form of $\Omega(C)$, we apply the initial condition $G(z,0) = z^{N_b}$, meaning we start with one clone of size $N_b$ at $t = 0$.

$$z^{N_b} = \frac{D}{\beta - \delta}\left[-\ln(1-z) + \frac{\delta}{\beta}\ln(\delta - \beta z)\right] + \Omega\left[\frac{1-z}{\delta - \beta z}\right] \tag{104}$$

Let us use the temporary variable $\xi = \frac{1-z}{\delta - \beta z}$. Then $z = \frac{\xi\delta - 1}{\xi\beta - 1}$, and the solution for $\Omega(\xi)$ is

$$\Omega(\xi) = \frac{\xi\delta - 1}{\xi\beta - 1}^{N_b} - \frac{D}{\beta - \delta}\left[-\ln(1 - \frac{\xi\delta - 1}{\xi\beta - 1}) + \frac{\delta}{\beta}\ln(\delta - \beta\frac{\xi\delta - 1}{\xi\beta - 1})\right] \tag{105}$$

Now we can write the full solution for $G(z,t)$, replacing $\xi$ with $\xi\epsilon$, where $\epsilon = e^{(\beta-\delta)t}$.

$$G(z,t) = \left(\frac{\epsilon\delta(1-z) - \delta + \beta z}{\epsilon\beta(1-z) - \delta + \beta z}\right)^{N_b} + \frac{D}{\beta - \delta}\left[-\ln(1-z) + \frac{\delta}{\beta}\ln(\delta - \beta z)\right]$$
$$- \frac{D}{\beta - \delta}\left[-\ln(1 - \frac{\epsilon\delta(1-z) - \delta + \beta z}{\epsilon\beta(1-z) - \delta + \beta z}) + \frac{\delta}{\beta}\ln(\delta - \beta\frac{\epsilon\delta(1-z) - \delta + \beta z}{\epsilon\beta(1-z) - \delta + \beta z})\right] \tag{106}$$

$b_k$ can be found by Taylor-expanding $G(z)$ about $z = 0$. The full distribution is cumbersome for this initial condition, but the limit as $t \to \infty$, if $\beta < \delta$, is independent of the initial condition:

$$b(k) = \frac{D}{k}\frac{\beta^{k-1}}{\delta^k} = \frac{D}{\beta k}e^{-k\ln(\delta/\beta)} \tag{107}$$

Now we have required constant $N_b = \sum_k k b_k$, which means

$$N_b = \sum_k D\frac{\beta^{k-1}}{\delta^k} = \frac{D}{\delta - \beta} = \frac{\mu N_b}{F - g + \mu} \tag{108}$$

So to maintain constant population size, $F = g$, as expected. The diversity is the total number of clones, defined as $\sum_k b_k$. We represent diversity as $m$.

This gives the following result for steady-state diversity:

$$\sum_k b_k = m = -\frac{D}{\beta}\ln\left(\frac{\delta - \beta}{\delta}\right) = -\frac{\mu N_b}{g - \mu}\ln\left(\frac{\mu}{g}\right) \tag{109}$$

In our definition of the birth and death rates, it is implied that $\mu < g$, and under that condition, the diversity is positive. If we let $\frac{\mu}{g} = u$, we can write the diversity as

$$m = -N_b\frac{u}{1-u}\ln u \tag{110}$$

For small $u$ (i.e. mutation rate much smaller than division rate, usually true), $m \approx -N_b u \ln u$. Equation 107 is equivalent to Fisher's logseries with $\alpha = \frac{D}{\beta}$ and $x = \frac{\beta}{\delta}$. This is also equivalent to Hubbell's unified neutral theory in the limit of large sample size. This result for diversity (**Equation 110**) is also approximately equivalent to the result obtained in **Vallade et al., 2003** and also in **He and Hu, 2005** when the mutation rate is within a few orders of magnitude of $g$.

## Comparison to coevolution model

In our bacteria-phage coevolution model, we find that diversity scales approximately like $m \sim a^{\frac{1}{3}}$, where

$$a \approx \frac{4e\mu L^{\frac{\hat{\eta}\alpha\tilde{n}_V}{\hat{\eta}\alpha\tilde{n}_V + r}}\tilde{n}_V^2}{(B-1)^2} \tag{111}$$

For $\mu L << 1$, $Bp_V - 1 >> 1$ and $p_V r > \hat{\eta}gC_0(1-f)$, the dependence of diversity on $a$ is given by **Equation 112**. Note that this expression correctly captures the fold change in diversity as a function of parameters, but the value itself is off by about a factor of 10.

$$a \approx \frac{4e\mu L\hat{\eta}(gC_0(1-f))^3}{(Bp_V - 1)^2\alpha^2 p_V r} \tag{112}$$

Is this dependence quite different from what we get under neutral evolution without coevolution? Since we are measuring bacterial diversity in the coevolution model, the spacer acquisition probability $\eta$ is a good analog to bacterial mutation rate in the non-coevolving model. Let us look at the change in diversity for a given change in spacer acquisition probability.

In the coevolution model, a $k-\mathrm{fold}$ change in spacer acquisition probability gives approximately a $k^{1/3}$ change in diversity, while under the non-coevolution model, a $k-\mathrm{fold}$ change in mutation rate $u$ gives a $\frac{k(1-u)}{1-ku}(1+\frac{\ln k}{\ln u})-\mathrm{fold}$ change in diversity. For small $u$, this is approximately $k(1+\frac{\ln k}{\ln u})$. The change is dependent on the mutation rate, but if $k$ is not very large and $u << 1$, then a $k-\mathrm{fold}$ change in $u$ gives approximately a $k-\mathrm{fold}$ change in diversity. A similar correspondence can be reached with a different approximation. If $a$ is large, the leading order contribution to $m$ is $a^{1/3}$, but a better approximation is given by *Equation 113*.

$$m \approx \left( a(1 + \frac{\ln a}{3}) \right)^{\frac{1}{3}} \tag{113}$$

If $a$ is large, which it typically is, then the leading term is more accurately $\frac{a\ln a}{3}^{1/3}$. This is directly comparable to the $u\ln u$ dependence of the non-coevolving model: diversity in the coevolution model goes like $u\ln u^{1/3}$, and in the non-coevolving model it goes like $u\ln u$. Either way, there's a 1/3 power in the coevolution model that isn't there in the simple model.

*Appendix 2—figure 2* compares the predicted diversity and fold change in diversity in each model under the simple approximation assumptions here. Diversity increases more rapidly with mutation rate in the simple model than in the coevolution model.

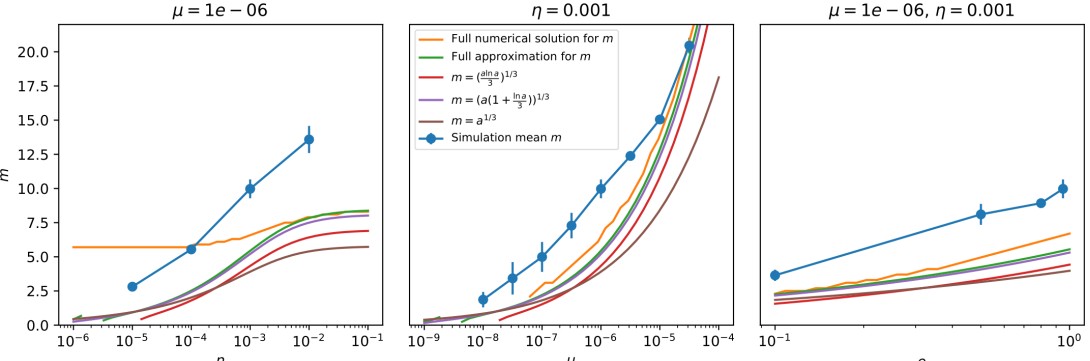

**Appendix 2—figure 1.** Diversity vs. $\eta$ (left), $\mu$ (centre), and $e$ (right) for $C_0 = 10^4$. The $\eta$ dependence of diversity is not very well predicted even by the full numerical solution, but for mutation rate and spacer effectiveness the approximate solutions do pretty well in this regime. The simulation data increase in diversity as a function of spacer acquisition probability actually goes more like $m \propto \ln\eta$.

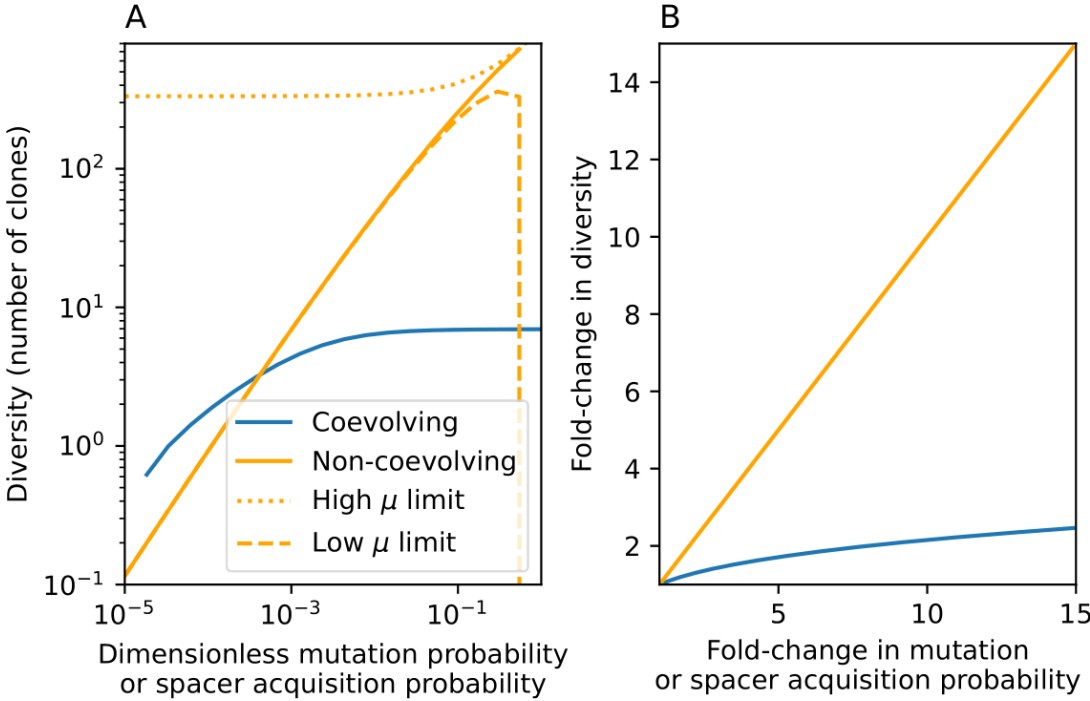

**Appendix 2—figure 2.** Approximate predictions for diversity in our coevolution model (blue) or a simple model with mutation but no coevolution (orange). (**A**) Predicted diversity as a function of spacer acquisition probability in our coevolution model as given by *Equation 111* for $C_0 = 10^4$, $e = 0.95$, and $\mu = 10^{-6}$ (blue). Predicted diversity in the non-coevolving model as given by $m = -N_b \frac{u}{1-u} \ln u$ with $N_b = 1000$ (orange). The low-$\mu$ limit is $m = -N_b u \ln u$. The high-$\mu$ limit is a series expansion in $\epsilon$ for $\mu = 1 - \epsilon$ giving $m \approx N_b(\frac{1}{3} + \frac{5\mu}{6} - \frac{\mu^2}{6})$. (**B**) Fold change in diversity as a function of fold change in mutation rate or spacer acquisition probability under the simple approximation that $m \sim \eta^{1/3}$ (blue) and $m \sim u$ (orange).

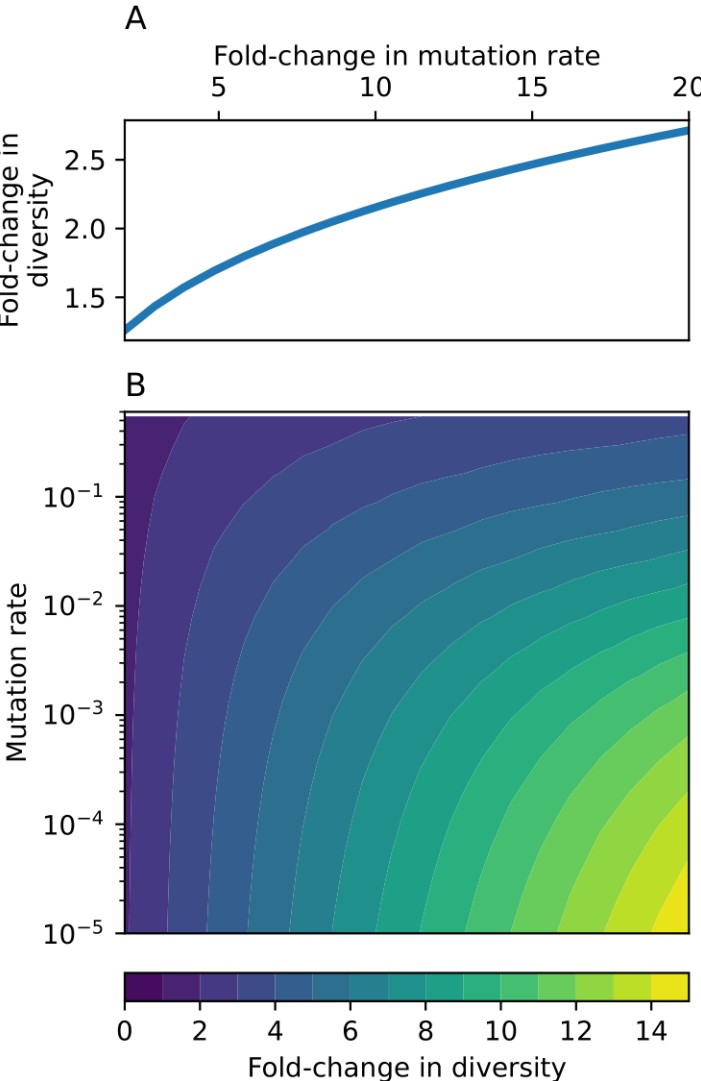

**Appendix 2—figure 3.** Fold change in diversity (number of species) as a function of mutation rate and $k$, the fold change in mutation rate. (**A**) Fold change in diversity as a function of fold change in mutation rate in the coevolution model: diversity increases by approximately a factor of $k^{\frac{1}{3}}$, independent of mutation rate (solid line). (**B**) Fold change in diversity as a function of both mutation rate and fold change in mutation rate in the model without coevolution.

## Appendix 3

## Stochastic clone dynamics

In this appendix, we calculate clone size distributions for bacteria, the probability of extinction/establishment for phage clones, and the mean time to extinction for bacteria and phage clones. These results underlie our result for diversity; in particular the phage establishment probability is a core component of our calculation of diversity in 'Phages drive stable emergent sequence diversity'. Phage establishment is also covered in 'What determines the fitness and establishment of new mutants?'.

In this section we define $p_V(i,j)$ as given by *Equation 18*.

## Clone fitness

Here we use mean-field expressions for phage and bacteria clone size from 'Single-clone mean-field dynamics to calculate the fitness of new phage mutants'. These results are used in 'Measuring diversity to justify our assumptions about the timescale of spacer acquisition relative to phage establishment'.

We investigated the fitness of new phage mutants to understand the effect of bacterial spacer acquisition on phage mutant growth. We define the fitness of phage clones to be their per-capita average growth rate: their average growth rate in bacterial generations divided by their average size. We calculate the fitness from simulation data by calculating the mean phage clone size conditioned on survival (as in *Figure 10*, 'Single-clone mean-field dynamics'), then taking the time derivative, then dividing by the mean phage clone size. (In principle, this is equivalent to first taking the time derivative of each individual clone trajectory and then averaging across all trajectories, but we found that edge effects from trajectories that go extinct skewed the result, so we first average across all trajectories before taking a derivative.) Phage clone fitness over time in a single simulation is plotted in *Figure 3—figure supplement 1* (orange markers).

We calculate the theoretical phage clone fitness by taking the time derivative of the predicted mean phage clone size and dividing by the predicted mean clone size. The predicted mean phage clone size is piecewise-defined at short times as the numerical solution of the system of *Equations 21 and 22*, and at long times as the numerical solution steady-state clone size (*Equation 26*). This prediction is plotted as a solid black line in *Figure 3—figure supplement 1*.

New phage mutants have a positive growth rate on average (initial fitness >0), and their growth rate drops to zero on average as bacteria acquire matching spacers and gain immunity to new mutants. We can define the time or size at which phage clones become 'established', after which they are safe from rapid stochastic extinction and behave like neutral clones (having a fitness that is close to 0). We define one minus the long-time limit of the probability of phage clone extinction (*Equation 159*) as the probability of establishment for new phage clones:

$$P_{\text{est}} = 1 - P_0^{N_{\text{est}}} = 1 - \left(1 - \frac{2s_0}{B(s_0 + \delta_0)}\right)^{N_{\text{est}}} \tag{114}$$

where $s_0 = \alpha B p_V - \alpha n_B - F$ is the average initial growth rate of phage clones and $\delta_0 = F + \alpha n_B(1 - p_V)$ is the average initial death rate of phage clones (more in 'Phage clone probability of extinction'). The establishment clone size can then be defined as the value of $N_{\text{est}}$ for which $P_{\text{est}} \approx 1$. Since $N_0 \frac{2s_0}{B(s_0+\delta_0)}$ is not necessarily small, we approximate $P_{\text{est}}$ as an exponential function and set $N_{\text{est}}$ as the scale of the exponent:

$$\left(1 - \frac{2s_0}{B(s_0 + \delta_0)}\right)^{N_{\text{est}}} \approx e^{-N_0 \frac{2s_0}{B(s_0+\delta_0)}} = e^{-\frac{N_0}{N_{\text{est}}}}$$

$$N_{\text{est}} \approx \frac{B(s_0 + \delta_0)}{2s_0} \tag{115}$$

*Equation 115* is the size at which phage clones become established on average (where $P_0 \approx 1/e$). This is plotted as a horizontal dashed line for one simulation in *Figure 3—figure supplement 1*. If $B = 2$ (birth-death with no bursts and positive selection), then $N_{\text{est}} = \frac{s_0+\delta_0}{s_0} \approx \delta_0/s_0$. This is the expected result for a simple birth-death process with selection as given in *Desai and Fisher, 2007*.

We can also find the time at which phage clones reach the establishment size. In the absence of matching bacterial spacers, new phage mutants grow deterministically as $n_V^i(t) = e^{s_0 t}$. We condition

on survival by dividing by the probability of establishment; for this calculation we use the long-time probability of establishment given by **Equation 156** with $s = s_0$ and $\delta = \delta_0$. This curve does not match the measured growth at short times, but in the region of phages reaching their establishment size it agrees well (solid pink line in **Figure 10**).

With these assumptions, the growth curve for phage clones is

$$n_V^i(t) \approx \frac{e^{s_0 t}}{1 - \frac{(B(\delta_0 + s_0) - 2s_0)(e^{s_0 t} - 1)}{2s_0 + B(e^{s_0 t} - 1)(\delta_0 + s_0)}} = 1 + \frac{B(e^{s_0 t} - 1)(\delta_0 + s_0)}{2s_0} \tag{116}$$

Replacing $n_V^i$ with the establishment clone size (115) and solving for $t$, we find

$$t_{\text{est}} = \frac{1}{s_0} \ln\left(\frac{2B(s_0 + \delta_0) - 2s_0}{B(s_0 + \delta_0)}\right) = \frac{1}{s_0}\left[\ln 2 + \ln\left(1 - \frac{s_0}{B(\delta_0 + s_0)}\right)\right] \tag{117}$$

Now $s_0/(B(\delta_0 + s_0)) \ll 1$, so the second logarithm can be approximated as $-s_0/(B(\delta_0 + s_0))$.

$$t_{\text{est}} \approx \frac{1}{s_0}\left[\ln 2 - \frac{s_0}{B(\delta_0 + s_0)}\right] \tag{118}$$

We can further approximate by dropping the second term entirely since $\ln 2 \gg s_0/(B(\delta_0 + s_0))$.

$$t_{\text{est}} \approx \frac{\ln 2}{s_0} \tag{119}$$

We get the same approximate value of $t_{\text{est}}$ if we take $B = 2$ directly and assume $s_0 \ll 1$. **Equation 119** is also numerically close to the mean establishment time calculated by Desai and Fisher for positive selection, $\langle \tau_{\text{est}} \rangle = \frac{\gamma}{s}$ (where $\gamma \approx 0.577216$) **Desai and Fisher, 2007**. This implies that the presence of a burst size $B > 2$ does not dramatically change the dependence of the time to establishment on phage fitness. The phage growth rate $s_0$ does still depend on $B$, however. (Note that Desai and Fisher define establishment time differently than we do, so our cases are not directly comparable.) **Equation 119** is plotted as a vertical dashed line in **Figure 3—figure supplement 1**.

The initial fitness $f_0$ of a new phage mutant can be computed analytically by using a different early-time approximation for $n_V^i(t)$, this time conditioning on survival using the short-time approximation for $P_0(t)$ given by **Equation 163**.

$$n_V^i(t) \approx \frac{e^{s_0 t}}{1 - \frac{\delta_0}{\beta_0 + \delta_0}\left(1 - e^{-(\beta_0 + \delta_0)t}\right)} = \frac{e^{s_0 t}(\beta_0 + \delta_0)}{\beta_0 + \delta_0 e^{-(\beta_0 + \delta_0)t}} \tag{120}$$

To estimate the initial fitness, we differentiate 120 with respect to $t$, divide by $n_V^i(t)$ to get the per-capita growth rate, and evaluate at $t = 0$:

$$f_0 = \frac{1}{n_V^i(t)}\frac{dn_V^i(t)}{dt}\Big|_{t=0} = s_0 + \frac{\delta_0(\beta_0 + \delta_0)e^{-(\beta_0 + \delta_0)t}}{\beta_0 + \delta_0 e^{-(\beta_0 + \delta_0)t}}\Big|_{t=0} = s_0 + \delta_0 \tag{121}$$

Interestingly, if we had not conditioned on survival, the initial per-capita growth rate would simply be $s_0$: conditioning on survival increases the apparent growth rate of phage clones by effectively ignoring their death rate.

We can evaluate $s_0 + \delta_0$ in terms of our original parameters for insight. We replace $B$ with $Be^{-\mu L}$ to capture the decrease in phage growth clone growth due to mutations away from a clone.

$$f_0 = s_0 + \delta_0 = \alpha p_V n_B (Be^{-\mu L} - 1) \tag{122}$$

The initial fitness does not directly depend on characteristics of CRISPR immunity such as $\eta$ and $e$ because new phage mutants see the bacterial population as if it did not have any CRISPR immunity; $f_0$ depends only on the total bacterial population size (**Appendix 3—figure 1**).

## Bacteria clone dynamics

In this section, we calculate clone size distributions for bacteria and the time to extinction for bacterial clones. Clone size distributions from simulations are shown in **Figure 2A**.

To solve for the dynamics of individual bacteria clones, we write a one-dimensional master equation just for $n_B^i$ (**Equation 123**).

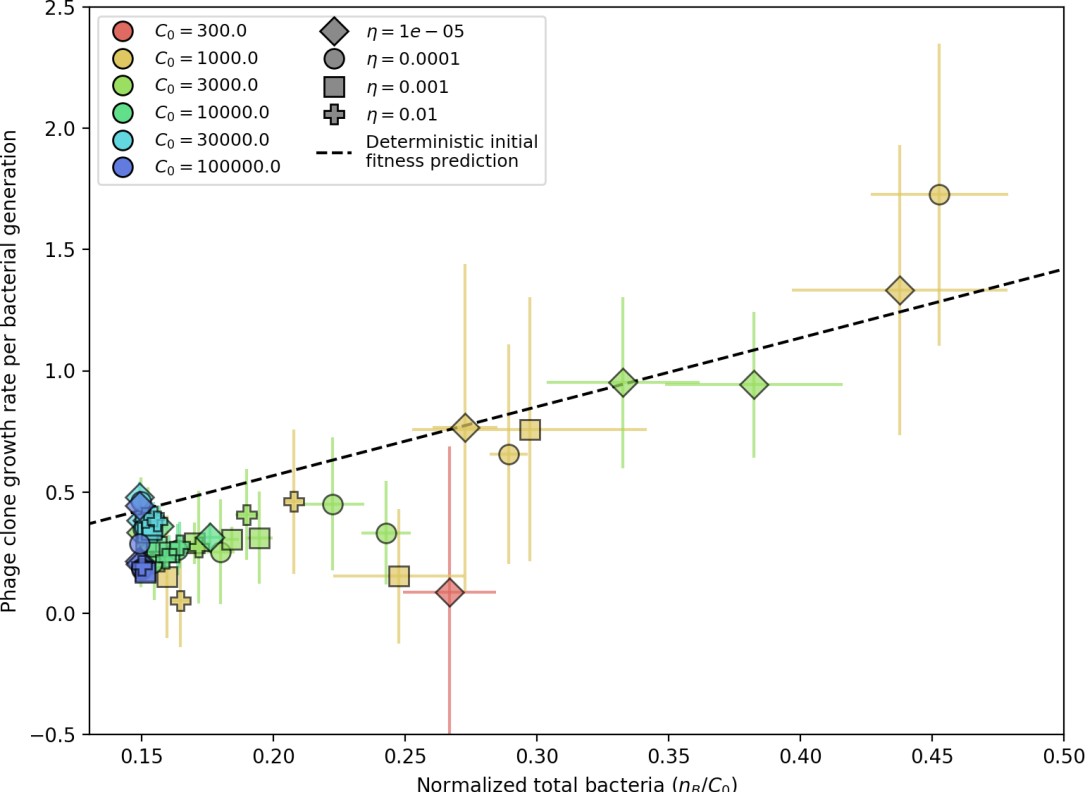

**Appendix 3—figure 1.** Phage clone initial growth rate vs. total bacteria normalized by the initial nutrient concentration $C_0$. Phage clone growth rate is as defined in **Figure 3—figure supplement 1**; for each simulation, the average phage clone growth rate is the derivative of the average phage clone size, averaged across all trajectories after steady state; plotted points and error bars are the average across three or more simulations. The phage initial fitness depends slightly on the phage mutation rate (mutants decrease the growth rate of a particular phage clone), but this dependence is slight enough that all mutation rates collapse onto the theoretical line. Here we plot data with $\mu = 10^{-6}$. The phage clone initial growth rate also does not depend on $e$ or $\eta$ because new phage mutants see the bacteria population as if it did not have any CRISPR immunity. The theoretical initial phage clone growth rate is given by **Equation 122**. The effective lower bound of $n_B/C_0$ is set by the steady-state population size without CRISPR immunity: $n_B/C_0 = \frac{fg}{\alpha(Bp_V - 1)} \approx 0.15$.

$$
\begin{aligned}
\frac{dP_n}{dt} = \; & (n+1)P_{n+1}[F + r + \alpha p_V(n_V - en_V^i)] \\
& + (n-1)P_{n-1}[gC] \\
& + P_{n-1}[\alpha \eta n_B^0 n_V^i(1 - p_V)] \\
& - nP_n[F + r + \alpha p_V(n_V - en_V^i) + gC] \\
& - P_n[\alpha \eta n_B^0 n_V^i(1 - p_V)]
\end{aligned}
\tag{123}
$$

For brevity we write $P_n = P_{n_B^i}(t|N_0)$, the probability of having $n$ bacteria of type $i$ at time $t$ given $N_0$ bacteria of type $i$ at $t = 0$.

Bacteria clone growth ($gC$), phage predation ($\alpha p_V(n_V - en_V^i)$), outflow ($F$), and spacer loss ($r$) all depend on the number of bacteria $n$, but spacer acquisition ($\alpha \eta n_B^0 n_V^i(1 - p_V)$) adds new bacteria independent of the current size of the clone.

We assume that the total population is in steady state so that the total population sizes $n_V$, $C$, and $n_B^0$ are constant. In general, $n_V^i$ is time-dependent and varies for each clone $i$, but we will assume that it is also constant at steady state.

This equation is very nearly identical in form to the bacteria clone size equation solved in our previous work **Bonsma-Fisher et al., 2018** as well as the clone size equation described in **Dessalles et al., 2022**, and we repeat our derivation of the solution here in brief.

We solve *Equation 123* using a generating function approach: $G(z,t) = \sum_n z^n P_n(t)$. Multiplying *Equation 123* by $\sum_n z^n$, we get the corresponding generating function partial differential equation:

$$\partial_t G(z,t) = \partial_z G(z,t)\left(d + bz^2 - (b+d)z\right) + DG(z-1) \tag{124}$$

Here $b$ and $d$ are the birth and death rates for bacterial clones: $b = gC$ and $d = F + r + \alpha p_V(n_V - en_V^i)$. $D$ is the rate of spacer acquisition from naive bacteria: $D = \alpha\eta n_B^0 n_V^i(1 - p_V)$.

We solve *Equation 124* using the method of characteristics *Van Kampen, 1981*. We parameterize the function $G(z,t)$ with a new variable $x$. Applying the chain rule:

$$\partial_x G(z(x), t(x)) = \frac{\partial G}{\partial z}\frac{\partial z}{\partial x} + \frac{\partial G}{\partial t}\frac{\partial t}{\partial x} \tag{125}$$

Comparing *Equation 124* with *Equation 125* gives the following characteristic equations:

$$\frac{\partial t}{\partial x} = 1 \tag{126}$$

$$\frac{\partial z}{\partial x} = (1-z)(bz-d) \tag{127}$$

$$\frac{\partial G}{\partial x} = DG(z-1) \tag{128}$$

From *Equation 126*, we see $t = x + c_1$, so we can choose $t_0 = c_1 = 0$ and replace $x$ with $t$ going forward. Solving the characteristic equation for $z$ by integrating both sides gives *Equation 129*.

$$\frac{1-z}{d-bz}e^{(b-d)t} = c_2 \tag{129}$$

At $t = 0$, $z$ will pass through some point $z_0$, so we have the initial condition $z(0) = z_0$. With $z_0$ in *Equation 129* at $t = 0$, we get *Equation 130*, where $c_2$ is given by *Equation 129*.

$$z_0 = \frac{c_2 d - 1}{c_2 b - 1} \tag{130}$$

The variation of $G$ along the $z-t$ curve is

$$\frac{\partial G}{\partial z} = -\frac{DG(z-1)}{(1-z)(bz-d)} = -\frac{DG}{(bz-d)} \tag{131}$$

Integrating both sides, we get

$$G(z) = \Omega(c_2)(bz-d)^{-\frac{D}{b}}$$

The constant $\Omega$ is a function of the characteristic $z$-$t$ curve (*Equation 129*). To find the particular form of $\Omega(c_2)$, we apply the initial condition $P_{N0}(0) = 1$ which gives $G(z,0) = z^{N_0}$, meaning that the clone starts at size $N_0$ at time $t = 0$.

$$G(z,0) = z^{N_0} = \Omega\left(\frac{1-z}{d-bz}\right)(bz-d)^{-\frac{D}{b}}$$

Let $\xi = \frac{1-z}{d-bz}$, therefore $z = \frac{\xi d - 1}{\xi b - 1}$.

$$\Omega(\xi)\left(b\left(\frac{\xi d - 1}{\xi b - 1}\right) - d\right)^{-\frac{D}{b}} = \left(\frac{\xi d - 1}{\xi b - 1}\right)^{N_0}$$

Solving for $\Omega(\xi)$:

$$\Omega(\xi) = \left(\frac{\xi d - 1}{\xi b - 1}\right)^{N_0}\left(b\left(\frac{\xi d - 1}{\xi b - 1}\right) - d\right)^{\frac{D}{b}}$$

The full solution for $G(z,t)$ can be written by replacing the constant $\Omega(c_2)$ with the expression for $\Omega(\xi)$ and replacing $\xi$ with $\xi\epsilon$, where $\epsilon = e^{(b-d)t}$ is the time-dependent part of the $z-t$ curve.

$$G(z,t) = (bz-d)^{-\frac{D}{b}}\left(\frac{\xi\epsilon d - 1}{\xi\epsilon b - 1}\right)^{N_0}\left(b\left(\frac{\xi\epsilon d - 1}{\xi\epsilon b - 1}\right) - d\right)^{\frac{D}{b}}$$

Finally, replacing $\xi$ with $\frac{1-z}{d-bz}$, we get

$$G(z,t) = (bz-d)^{-\frac{D}{b}} \left( \frac{(1-z)\epsilon d + bz - d}{(1-z)\epsilon b + bz - d} \right)^{N_0} \left( b \left( \frac{(1-z)\epsilon d + bz - d}{(1-z)\epsilon b + bz - d} \right) - d \right)^{\frac{D}{b}}$$

$G(1,t) = \sum_n P_n(t) = 1$, meaning that the total probability is conserved. Assuming $d > b$, the limit as $t \to \infty$ of $G(z,t)$ is

$$G(z) = \left( \frac{bz-d}{b-d} \right)^{-\frac{D}{b}}$$

The limit is independent of the initial clone size $N_0$ as we expect. We can construct $P_n$ at steady state by taking successive derivatives of $G(z) : P_n = \frac{1}{n!} \frac{\partial^n G}{\partial z^n} |_{z=0}$

$$P_n = \frac{1}{n! d^n} \left( \frac{d-b}{d} \right)^{D/b} \prod_{i=1}^{n} [D + (i-1)b] \tag{132}$$

$$P_0 = \left( \frac{d-b}{d} \right)^{D/b}$$

This is a negative binomial distribution with parameters $D/b$ and $b/d$. We can re-write this expression using Stirling's approximation for $n!$ to facilitate evaluation at large $n$.

$$P_n = \frac{1}{\sqrt{2\pi n}} \exp \left[ \frac{D}{b} \ln \left( \frac{d-b}{d} \right) + \sum_{i=1}^{n} \ln \left( \frac{e}{nd} (D + (i-1)b) \right) \right] \tag{133}$$

*Equation 133* is an analytic expression describing the steady-state spacer abundance distribution that results from our simulations. To compare this prediction with our simulations, we assume that $n_V^i$ on average is equal to $n_V/m$, where $n_V$ is the predicted total phage population size and $m$ is the number of large phage clones approximated by the predicted bacterial diversity.

*Appendix 3—figure 2* compares the analytic distribution to the steady-state spacer clone size distribution from our simulations at several values of the spacer acquisition probability $\eta$. The theoretical prediction captures the qualitative impact of increasing $\eta$ fairly well: as $\eta$ increases, the clone size distribution gains a more pronounced peak.

The discrepancy between the theoretical prediction and simulation data at high $\eta$ results in part from the predicted large phage clone size being larger than the measured large phage clone size in simulations. This can happen when the predicted number of clones $m$ is smaller than the simulation result. To assess whether this influenced our prediction, we also used maximum likelihood estimation to calculate the value of $n_V^i$ that gave the best fit between *Equation 133* and the data; for the two largest values of $\eta$, this does return a smaller value of $n_V^i$ and hence a distribution peak further to the left.

The previous two calculations assumed that the phage clone size is single-valued and constant in time. In reality, however, the phage clone size is both broadly distributed (*Figure 24*) and changing in time (*Figure 3—figure supplement 1*). We relax the first assumption by calculating an average bacterial distribution using the observed distribution of phage clones: we solve *Equation 133* at each observed large phage clone size, then average across the distribution of clone sizes to calculate $P(n_B^i) = P(n_B^i|n_V^i)P(n_V^i)$. (The large phage clone distribution is given by *Equation 28*.) This average distribution more accurately predicts the presence of small bacterial clones at high $\eta$, but it actually behaves worse than the single-$n_V^i$ solution at small $\eta$. This is related to the deviation of the number of large phage clones from the number of bacterial clones at small $\eta$ (*Figure 22*): at small $\eta$, bacteria do not acquire spacers as readily and so phage clones experience clonal interference largely without bacterial influence. The average distribution then predicts more small bacterial clones than there are because the large phage clone distribution includes intermediate phage clone sizes that the bacteria do not end up acquiring spacers from. Essentially, the bacteria 'see' fewer phage clones than the theory predicts, so the observed bacteria clone size distribution has fewer smaller clones than predicted.

*Equation 133* can be approximated for large $n$ as a gamma distribution:

$$P_n \approx \frac{\left(1 - \frac{b}{d}\right)^{\frac{D}{b}}}{\Gamma\left(\frac{D}{b}\right)} e^{-\ln(d/b)n} \left( \frac{1}{n} \right)^{1 - \frac{D}{b}} \tag{134}$$

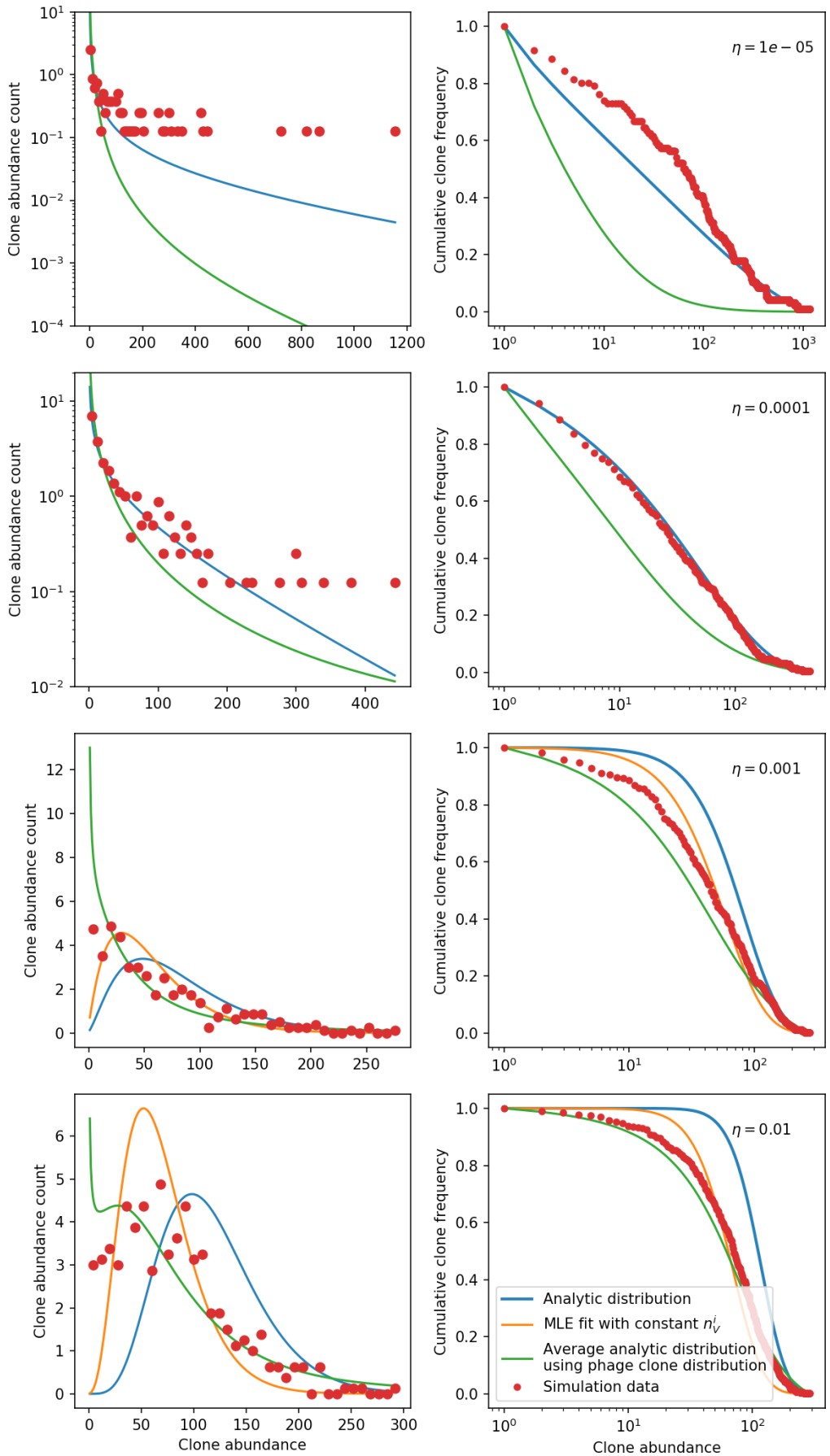

**Appendix 3—figure 2.** Clone size histograms (left) and cumulative distributions (right) for four different values of the spacer acquisition probability $\eta$. In all simulations, $C_0 = 10^4$, $e = 0.95$, and $\mu = 3 \times 10^{-6}$. We sample 30 evenly spaced time points between 2000 and 10,000 bacterial generations and combine the clone sizes at each of the sampled points to create the clone size distributions plotted. Solid lines show three different theoretical solutions. The solid blue line is given by **Equation 133** with all population quantities predicted from solving the system of **Equations 13**–17 with $m$ given by **Equation 32** and $n_V^i = n_V/m$. The solid orange line is given by **Equation 133**, with the value $n_V^i$ determined by maximum likelihood estimation (MLE) to give the best fit to the data. For the two largest values of $\eta$, the value of $n_V^i$ returned by the MLE fit is smaller than the theoretical value of $n_V^{i*}$, while for the two smallest values of $\eta$ the MLE value of $n_V^i$ is larger. For large enough values of $n_V^i$, the bacteria clone death rate $d$ is smaller than the birth rate $b$ which violates the assumptions used to derive **Equation 133**. This happens for the MLE fit at the two smallest $\eta$ values and hence no MLE solution is plotted. The solid green line is an average distribution calculated by solving **Equation 133** at each observed large phage clone size and averaging across the distribution of clone sizes; i.e. $P(n_B^i) = P(n_B^i|n_V^i)P(n_V^i)$. The large phage clone distribution is given by **Equation 28**.

This is a gamma distribution with shape parameter $\frac{D}{b}$ and rate parameter $\ln(d/b)$. Note that $\left(1 - \frac{b}{d}\right)^{\frac{D}{b}} \approx \ln(d/b)^{\frac{D}{b}}$, consistent with the canonical form of the gamma distribution. The shape parameter $D/b$ describes the relative balance between spacer acquisition and growth: if $D/b > 1$, then spacer acquisition is the dominant means by which bacterial clones grow. This often also means that the clone size distribution has a peak at clone size $gt_1$ (provided $d \gtrsim b$). Specifically, the mode of the distribution is greater than 1 if $\frac{D-b}{b} > \frac{d-b}{b}$. The rate parameter describes the decay constant of the exponential distribution resulting if the shape parameter equals 1.

## Bacteria clone extinction

When a matching phage clone exists in the population, bacteria with a particular spacer have a fitness advantage if they encounter that phage. Once the matching phage goes extinct, bacteria tend to quickly go extinct as well; in fact, bacteria often go extinct *before* their matching phage clone (**Appendix 3—figure 3**). To understand why this might be, we derive a theoretical prediction for the mean time to extinction under the assumption that bacterial clones evolve neutrally once they become large. This prediction does describe the extinction time distribution well in some regimes, but it underestimates the time to extinction at large total population sizes. If the neutral assumption is valid, this means that bacterial clones go extinct stochastically and are not necessarily driven to extinction by the extinction of their matching phage clone. This is true at small-to-medium total population sizes ($C_0 \leq 10^4$). On the other hand, when population sizes are large, bacteria clones go extinct more slowly than neutral theory would predict, likely because they are still being challenged by a matching phage and are also able to acquire spacers from that phage clone.

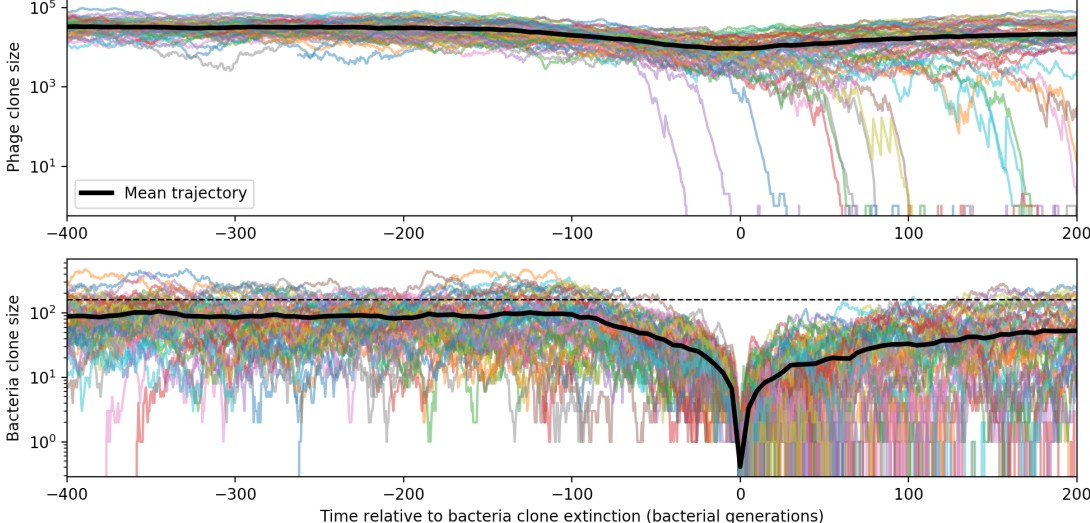

**Appendix 3—figure 3.** Bacteria and phage clone trajectories aligned to the time at which bacteria trajectories go extinct. Bacteria trajectories are included if they reach size $n_B^{i*}$ given by **Equation 25** and all corresponding phage trajectories are plotted. In this simulation $C_0 = 3 \times 10^4$, $e = 0.8$, $\eta = 10^{-3}$, and $\mu = 10^{-6}$.

We calculate the time to extinction using the backward master equation corresponding to *Equation 123*. The backward equation is an equation for the time to extinction $T_n$ from a given state. Instead of working with frequencies, we write this in terms of the number of bacteria belonging to a clone, $n$. Here $b = gC$, $d = F + r + \alpha p_V(n_V - en_V^i)$, and $D = \alpha \eta n_B^0 n_V^i(1 - p_V)$.

$$T_n - \Delta t = bn\Delta t T_{n+1} + D\Delta t T_{n+1} + dn\Delta t T_{n-1} + (1 - bn\Delta t - D\Delta t - dn\Delta t)T_n \quad (135)$$

Notice that the rate terms in the backward equation depend only on $n$, not on $n - 1$ or $n + 1$ like in the forward master equation. The forward equation is a sum of all the ways in which the system could end up at state $n$ at time $t$ from where it could have been at time $t - \Delta t$, so those rates depend on the other states. The backward equation goes in the other direction, looking backwards: it is a sum of all the ways in which the system *could have been* in state $n$ at time $T$ now that time $\Delta t$ has elapsed.

Rearranging *Equation 135*, we arrive at *Equation 136*.

$$-1 = (bn + D)T_{n+1} + dnT_{n-1} - (bn + dn + D)T_n \quad (136)$$

For boundary conditions, we have $T(n = 0) = 0$ (time to extinction is 0 when already extinct) and $\frac{dT}{dn}|_{n=n_B^s} = 0$ (reflecting boundary at $n = n_B^s$). The reflecting boundary is harder to justify because in reality $n_B^s$ is a flexible upper limit on clone size, but in steady state when $n_B^s$ is approximately constant, it is true that no single clone will grow larger than $n_B^s$.

To solve *Equation 136*, we expand about $n$ and keep terms up to 2nd order to get the Fokker–Planck equation:

$$-1 = \frac{dT}{dn}(bn + D - dn) + \frac{1}{2}\frac{d^2T}{dn^2}(bn + D + dn) \quad (137)$$

To get an approximate solution, we drop the drift term, assuming that when clones are large their net growth rate is approximately zero so $bn + D - dn \approx 0$ (this is the same as setting $\dot{n}_B^i = 0$ in Equation 22). This gives the following differential equation:

$$-1 \approx \frac{1}{2}\frac{d^2T}{dn^2}(bn + D + dn) \quad (138)$$

The solution to *Equation 138* with the boundary conditions described above is

$$T(n) = \frac{2}{(b+d)^2}\left[D\ln D - (D + (b+d)n)\ln(D + (b+d)n) + (b+d)n(1 + \ln(D + (b+d)n_B^s))\right] \quad (139)$$

*Equation 139* with $n = n_B^{i*}$ is plotted in *Appendix 3—figure 4*, *Appendix 3—figures 5–7* and *Appendix 3—figure 10*. We also solved the full Fokker–Planck equation numerically without assuming the drift term is 0 (*Equation 137*). In this numerical solution, we change the value of $n_V^i$ for different values of $n = n_B^i$ to reflect the fact that at small bacteria clone sizes phage clones also tend to be smaller; we use the numerical solutions for average $n_v^i(t)$ and $n_B^i(t)$ shown as dashed lines in *Figure 3—figure supplement 1*. This solution is shown in *Appendix 3—figure 3*.

## Approximate time to extinction

Since we are interested in the time to extinction once bacterial clones reach a large size, we can substitute $n = n_B^{i*}$ in *Equation 139* to gain insight into the time to extinction for bacteria. The average clone size $n_B^{i*} = \frac{n_B^s}{m}$.

$$T(n_B^{i*}) = \frac{2}{(b+d)^2}\left[\frac{n_B^s(b+d)}{m}(1 + \ln(n_B^s(b+d) + D)) - \left(\frac{n_B^s(b+d)}{m} + D\right)\ln\left(\frac{n_B^s(b+d)}{m} + D\right) + D\ln(D)\right] \quad (140)$$

We can substitute values for $b + d$ using the steady-state deterministic solution for $n_B^s$ and assuming $n_V^i = n_V/m$:

$$b + d = gC + F + r + \alpha p_V n_V(1 - \frac{e}{m}) = 2F + 2r + 2\alpha p_V n_V(1 - \frac{e}{m}) - \alpha(1 - p_V)\eta n_V\frac{1 - \nu}{\nu} \quad (141)$$

To approximate the extinction time expression, we decompose $n_B$, $n_V$, and $\nu$ into series expansions in $e/m$, i.e. $n_V \approx n_{V0} + \frac{e}{m}n_{V1} + \frac{e}{m}^2 n_{V2}$. We substitute these expressions into *Equation 140* and perform an overall series expansion in $e/m$. The zeroth-order term is shown here and plotted in *Appendix 3—figure 3B*, and the solution to first order is plotted in *Appendix 3—figure 3A*.

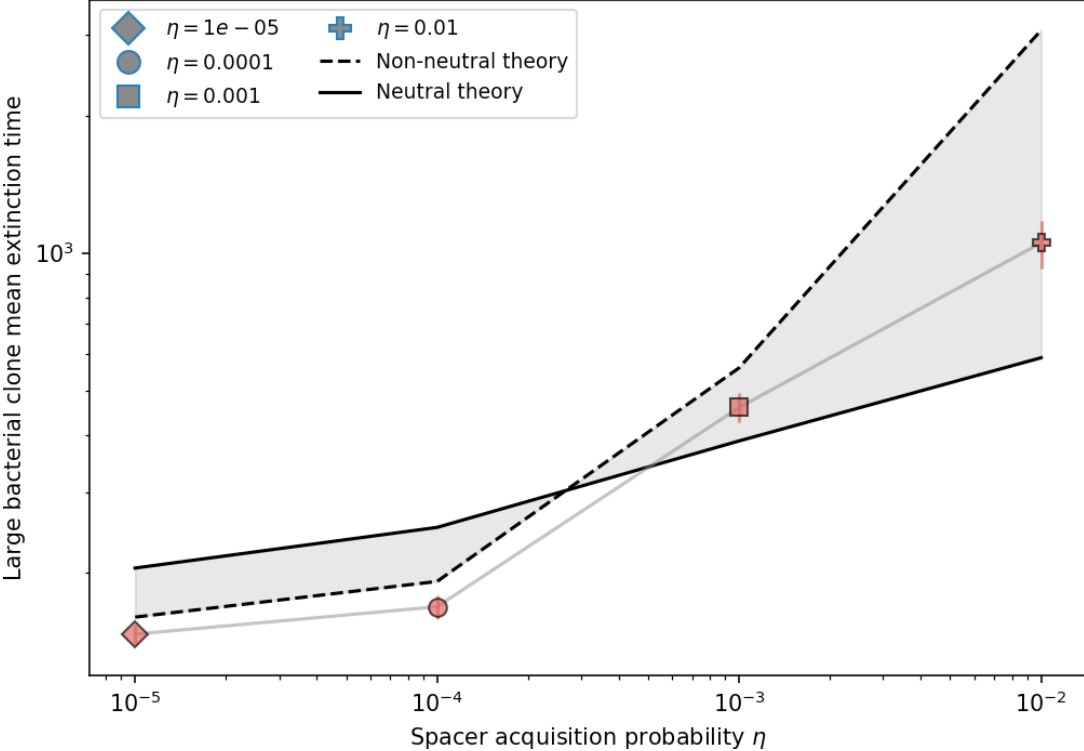

**Appendix 3—figure 4.** Mean time to extinction for bacterial clones after reaching size $n_B^{i\,*}$ as a function of $\eta$ for $C_0 = 10^4$, $e = 0.95$, and $\mu = 3 \times 10^{-6}$. The solid line is given by **Equation 139** with $n = n_B^{i\,*}$, and the dashed line is given by numerically solving **Equation 137**.

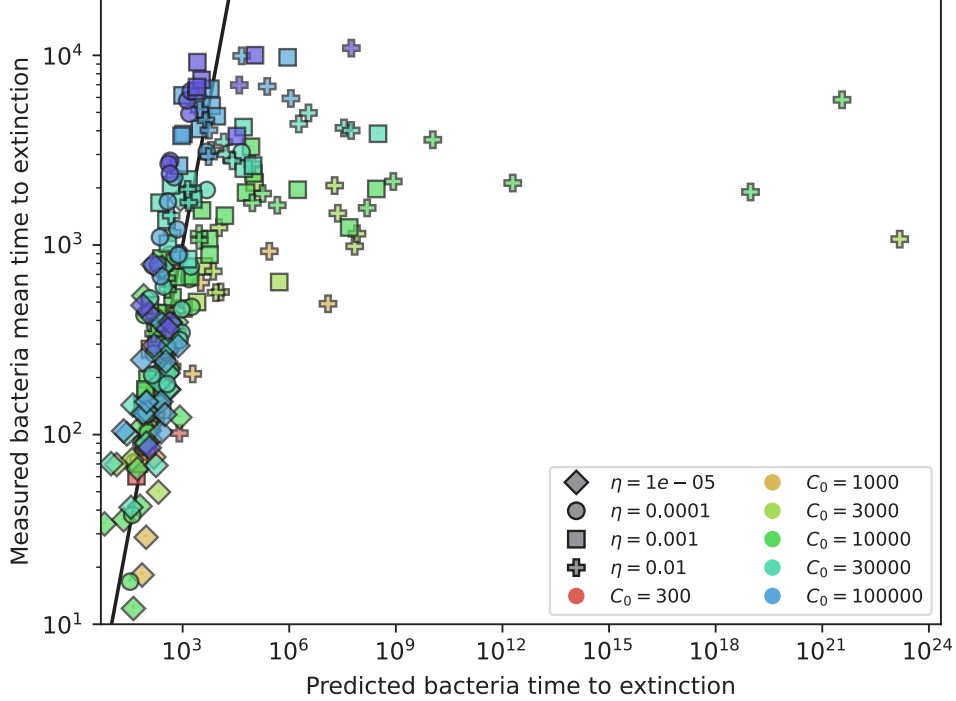

**Appendix 3—figure 5.** Predicted bacteria clone time to extinction with drift. Measured vs. predicted mean time to extinction for bacterial clones after reaching size $n_{B^i}^*$. The predicted time to extinction is the solution with drift, given by numerically solving **Equation 167**.

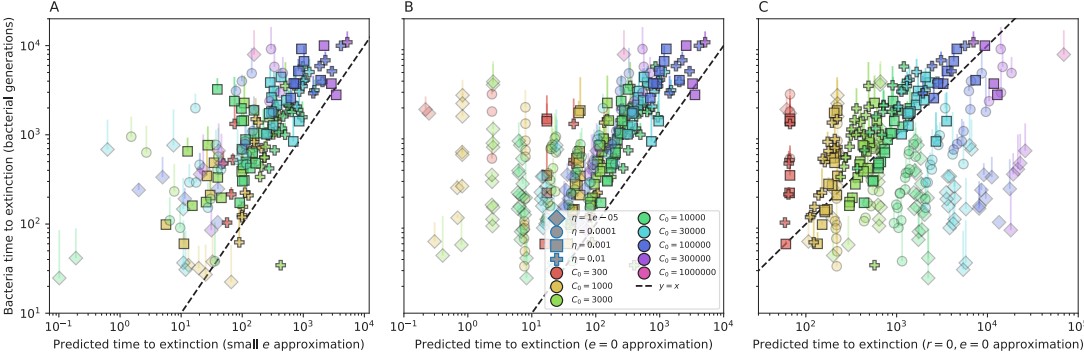

**Appendix 3—figure 6.** Approximations for bacteria clone extinction. Measured mean time to extinction for large bacterial clones vs. three successively more aggressive analytic approximations for the mean time to extinction. (A) The predicted time to extinction is given by taking a series expansion in $e/m$ and keeping terms to 1st order. (B) A series expansion in $e/m$ and keeping the 0th order term ($e = 0$) only. (C) Series expansion in $e/m$ and $r$ and keeping the 0th order term only (*Equation 143*).

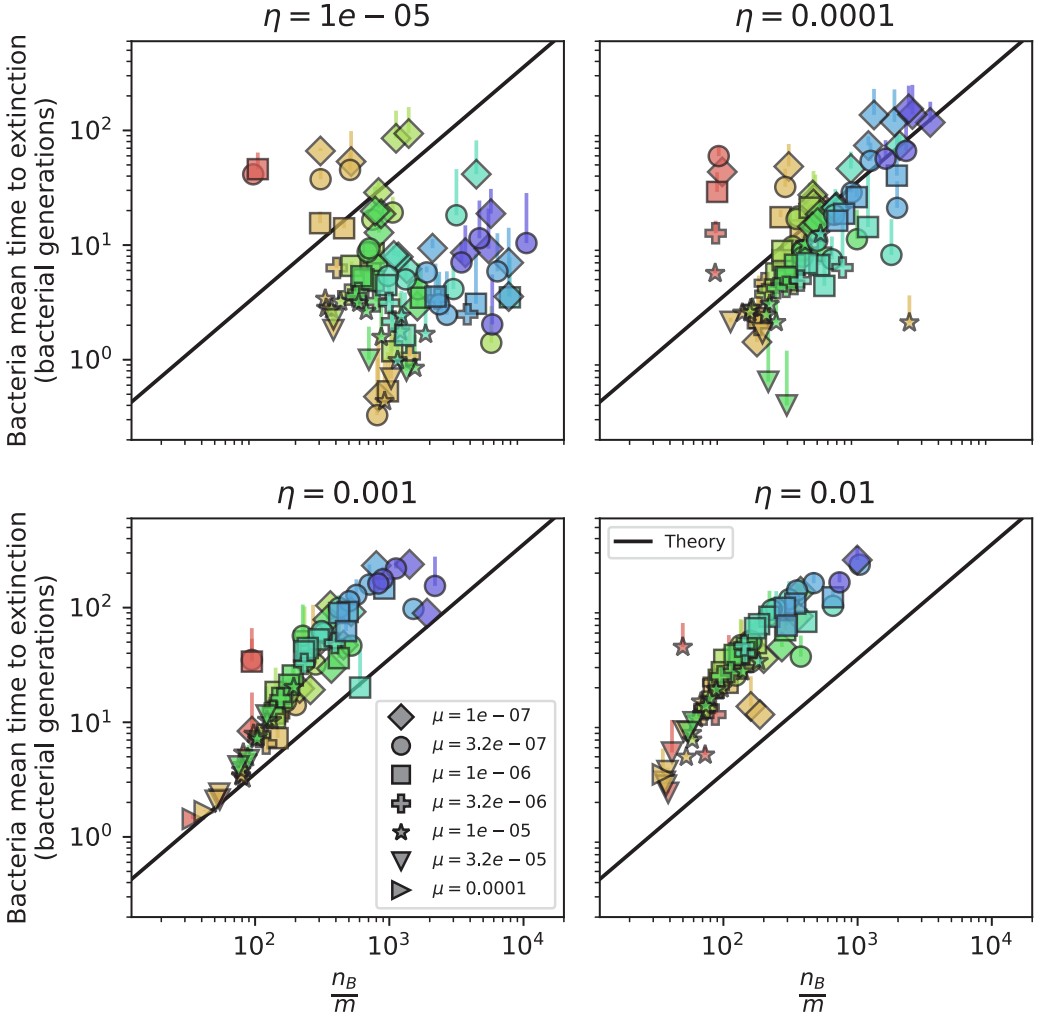

**Appendix 3—figure 7.** Approximations for bacteria clone extinction as a function of mean clone size. Bacteria extinction time as a function of mean clone size. Large bacteria clone mean time to extinction as a function of $\frac{n_B}{m}$, where $m$ is measured from simulations and $n_B$ is the total bacteria population size. This parameter combination describes the trend in both phage and bacteria extinction reasonably well at large values of (bottom panels). The solid line is given by *Equation 145* without the $1 + \ln m$ term.

$$T \approx - \left[ \left( 2C_0^2 f^2 g^2 \eta (\alpha(Bp_V - 1) + g)^2 \left( -\frac{\alpha(f-1)(Bp_V - 1)}{fg} - 1 \right) \right) \right.$$

$$\left( -r \ln \left( -\frac{C_0^2 fg^2 \eta r(\alpha + f(\alpha(Bp_V - 1) + g) + \alpha(-B)p_V)}{\alpha m(Bp_V - 1)\left(g\left(\alpha BC_0(f-1)\eta p_V^2 + p_V(r - \alpha(B+1)C_0(f-1)\eta) + \alpha C_0(f-1)\eta\right) + \alpha p_V r(Bp_V - 1) + C_0 f\eta g^2(p_V - 1)\right)} \right) \right)$$

$$- ((g(2\alpha C_0(p_V - 1)(Bp_V - 1) + (3p_V - 4)r) + \alpha(3p_V - 4)r(Bp_V - 1))$$

$$\ln \left( \frac{C_0^2 fg^2 \eta (\alpha + f(\alpha(Bp_V - 1) + g) + \alpha(-B)p_V)(g(2\alpha C_0(p_V - 1)(Bp_V - 1) + (3p_V - 4)r) + \alpha(3p_V - 4)r(Bp_V - 1))}{\alpha m(Bp_V - 1)(\alpha(Bp_V - 1) + g)\left(g\left(\alpha BC_0(f-1)\eta p_V^2 + p_V(r - \alpha(B+1)C_0(f-1)\eta) + \alpha C_0(f-1)\eta\right) + \alpha p_V r(Bp_V - 1) + C_0 f\eta g^2(p_V - 1)\right)} \right) \qquad (142)$$

$$- (p_V - 1)(g(2\alpha C_0(Bp_V - 1) + 3r) + 3\alpha r(Bp_V - 1))$$

$$\left( \ln \left( \frac{C_0^2 fg^2 \eta (\alpha + f(-\alpha + \alpha Bp_V + g) + \alpha(-B)p_V)(g(2\alpha C_0 m(p_V - 1)(Bp_V - 1) + r(3m(p_V - 1) - 1)) + \alpha r(Bp_V - 1)(3m(p_V - 1) - 1))}{\alpha m(Bp_V - 1)(\alpha(Bp_V - 1) + g)\left(g\left(\alpha BC_0(f-1)\eta p_V^2 + p_V(r - \alpha(B+1)C_0(f-1)\eta) + \alpha C_0(f-1)\eta\right) + \alpha p_V r(Bp_V - 1) + C_0 f\eta g^2(p_V - 1)\right)} \right) + 1 \right) \right)$$

$$/(\alpha(Bp_V - 1) + g)) \div \left( \alpha m(Bp_V - 1)(g(2\alpha C_0(Bp_V - 1) + 3r) + 3\alpha r(Bp_V - 1))^2 \left( C_0 \eta (p_V - 1)(\alpha + f(\alpha(Bp_V - 1) + g) + \alpha(-B)p_V) + \frac{p_V r(\alpha(Bp_V - 1) + g)}{g} \right) \right) \right]$$

This expression remains unwieldy, so we also do a series expansion in $r$ and keep the zeroth-order term (equivalent to setting $r = 0$). Spacer loss is not the dominant death rate for spacer-containing clones; the rate of spacer loss we use is an order of magnitude lower than the rate of cell death due to outflow ($R = 0.04, f = 0.3$). This expression becomes very simple:

$$T(n_B^{i\,*}) \approx \frac{f\left(1 + \ln m\right)(\alpha(Bp_V - 1) + g)}{\alpha^2 m(Bp_V - 1)^2} \qquad (143)$$

We can write *Equation 143* in terms of total population sizes without CRISPR ($\tilde{n}_B$ and $\tilde{n}_V$) to gain insight.

$$T(n_B^{i\,*}) \approx \frac{\tilde{n}_B}{F + p_V \alpha \tilde{n}_V} \frac{\left(1 + \ln m\right)}{m} \qquad (144)$$

If average immunity is low, then $F + \alpha p_V \tilde{n}_V \approx gC$ (from the mean-field equation for $n_B$).

$$T(n_B^{i\,*}) \approx \frac{\tilde{n}_B}{g\tilde{C}} \frac{\left(1 + \ln m\right)}{m} = \frac{\tilde{n}_B}{b} \frac{\left(1 + \ln m\right)}{m} \qquad (145)$$

*Equation 143* is compared to the measured time to extinction in *Appendix 3—figure 3C*. It is not a good approximation for low $\eta$, but at the two highest values of $\eta$ it captures the trend reasonably well. The dependence of bacterial time to extinction on clone size (*Equation 145*, dropping the $1 + \ln m$ term) is shown in *Appendix 3—figure 3*.

Interestingly the dependence on $m$ in this approximate expression is identical to the dependence on $m$ in the phage extinction approximation (*Equation 67*). Furthermore, there is also a proportionality to the mean clone size in the form $n_B/m$ in the bacteria extinction equation (since $n_B^s = n_B$ if $r = 0$, which is why this approximation works best at high $\eta$ when $\nu$ is closer to 1). This is reasonable considering that the dynamics of bacteria and phage are ultimately matched to each other at steady state. This expression still depends on $m$, which is an emergent quantity dependent on all other parameters as well. We apply the approximation for $m$ from 'Approximation for $m$' for more insight.

$$T_{\text{bacteria}} \approx \frac{1}{m} \frac{f(\alpha(Bp_V - 1) + g)}{\alpha^2 (Bp_V - 1)^2} \approx \frac{f}{gC_0(1 - f)} \left[ \frac{r}{4\alpha Be\eta(1 - p_V)\mu L} \right]^{\frac{1}{3}} \qquad (146)$$

This simplified expression for time to extinction gives insight into the main drivers of extinction. Perhaps counterintuitively, the bacteria clone time to extinction decreases as the growth rate $g\tilde{C}$ increases — a higher growth rate means faster dynamics overall and lower time to extinction. Note, however, that a higher growth rate (for instance, with larger $C_0$) also leads to larger $n_B$.

Bacteria extinction depends inversely on $\alpha$, $B$, $e$, $\mu L$, and $\eta(1 - p_V)$, with the time to extinction decreasing with all parameters except the spacer loss rate $r$. The bacteria time to extinction increases with increasing flow rate $f$.

## Phage clone dynamics

In this section, we calculate the probability of establishment for new phage mutants and the time to extinction for large phage clones. The phage establishment probability is a core component of

our calculation of diversity in 'Phages drive stable emergent sequence diversity' and is explored on its own in 'What determines the fitness and establishment of new mutants?'. The phage time to extinction is also related to the speed of phage evolution in *Figure 6*, 'Dynamics are determined by diversity'.

To solve for the dynamics of individual phage clones, we can write a one-dimensional master equation just for $n_V^i$ (*Equation 147*). Here we neglect mutations; we assume that mutations are rare and a burst always contributes to the clonal population being tracked. This master equation is a birth-death master equation with bursts – it is in the form of a classic birth-death master equation except that the population grows with a jump of size $B - 1$. $P_n = P_{n_V^i}(t|N_0)$ is the probability of having $n$ phages of type $i$ at time $t$ given $N_0$ phages of type $i$ at $t = 0$.

$$
\begin{aligned}
\frac{dP_n}{dt} = \quad & (n + 1)P_{n+1}[F + \alpha(1 - p_V)n_B + \alpha p_V en_B^i] \\
& + (n - B + 1)P_{n-B+1}[\alpha p_V n_B - \alpha p_V en_B^i] \\
& - nP_n[\alpha n_B + F]
\end{aligned}
\tag{147}
$$

## Phage clone probability of extinction

*Equation 159* is the basis for our prediction of phage establishment probability (*Equation 114*).

We can find the probability of extinction $P_0$ for new phage mutants under the assumption that $n_B$ and $n_B^i$ are constant using *Equation 147*. We solve this equation using a generating function approach: $G(z, t) = \sum_n z^n P_n(t)$. Multiplying *Equation 147* by $\sum_n z^n$, we get the corresponding generating function partial differential equation:

$$
\frac{1}{\beta + \delta} \partial_t G(z, t) = \partial_z G(z, t) \left( 1 - p + pz^B - z \right)
\tag{148}
$$

Here $p = \frac{\beta}{\beta+\delta}$ where $\beta$ and $\delta$ are the birth and death rates for phage mutants: $\beta = \alpha p_V n_B - \alpha p_V en_B^i$ and $\delta = F + \alpha n_B(1 - p_V) + \alpha p_V en_B^i$. We distinguish two cases of constant $n_B^i$, valid at early times after phage mutation, and $n_B^i = n_B^{i}{}^*$, the deterministic steady-state value of $n_B^i$ given by *Equation 25*, valid at long times after phage mutation. When we are assuming $n_B^i = 0$, we use $\beta = \beta_0$ and $\delta = \delta_0$.

We solve *Equation 148* using the method of characteristics:

$$
\partial_x G(z(x), t(x)) = \frac{\partial G}{\partial z} \frac{\partial z}{\partial x} + \frac{\partial G}{\partial t} \frac{\partial t}{\partial x}
\tag{149}
$$

This gives the following characteristic equations:

$$
\frac{\partial t}{\partial x} = \frac{1}{\beta + \delta}
\tag{150}
$$

$$
\frac{\partial z}{\partial x} = -(1 - p + pz^B - z)
\tag{151}
$$

Integrating these two equations, we get $t = \frac{x}{\beta+\delta}$, and

$$
x = -\int_{z(0)}^{z(x)} \frac{dw}{1 - p + pw^B - w}
\tag{152}
$$

The initial phage clone size is $N_0$. The initial condition is

$$
G(z(0), x = 0) = z(0)^{N_0} = G(z(x), t(x))
$$

The last equality is from the method of characteristics: the solution for $G$ at the initial condition gives the full solution which is constant as parameterized by $x$.

The probability of extinction is given by setting $z = 0$ in $G(z, t)$.

$$
P_0(t) = G(z(t) = 0, t) = z(0)^{N_0}|_{z(t)=0} = \zeta^{N_0}
$$

where $\zeta$ solves **Equation 153**, which we obtain by setting $z(x) = 0$ in **Equation 152** and replacing $z(0)$ with $\zeta$, since $G(z,t)|_{z=0} = z(0)^{N_0}|_{z=0} = P_0(t)$.

$$x = \int_0^{\zeta} \frac{dw}{1 - p + pw^B - w} \tag{153}$$

## Long-time approximation for $P_0(t)$

To approximate $P_0(t)$ at long times, we notice that since extinction probability becomes large at $t = \infty$, then the large-$t$ limit corresponds to $\zeta \to 1$.

The denominator of **Equation 153** will be smallest when $w$ is near 1, so the largest contribution to the integral will come from $w$ near 1. Expanding the denominator to second order with $w = 1 - \epsilon$, where $\epsilon$ is small:

$$1 - p + pw^B - w \approx 1 - p + p(1 - \epsilon B + \frac{1}{2}\epsilon^2 B(B-1)) - 1 + \epsilon = \epsilon - p\epsilon B + \frac{p}{2}\epsilon^2 B(B-1)$$

Changing back to the variable $w$ and substituting back into **Equation 153**, the integral becomes

$$x \approx \int_0^{\zeta} \frac{dw}{1 - w - pB(1-w) + \frac{p}{2}(1-w)^2 B(B-1)} \tag{154}$$

which when evaluated gives

$$\zeta = \frac{(2 - 3Bp + B^2p)(1 - e^{st})}{2 - Bp(3 - e^{st}) + B^2p(1 - e^{st})} \tag{155}$$

$$\zeta = \frac{(B(\delta + s) - 2s)(e^{st} - 1)}{2s + B(e^{st} - 1)(\delta + s)} \tag{156}$$

where $t$ is in units of minutes and we introduce the variable $s = \beta(B-1) - \delta = \delta\left(\frac{p}{1-p}(B-1) - 1\right)$ (and $p = \frac{s+\delta}{s+B\delta}$). The parameter $s$ is the average growth rate of phage clones. When phage mutants first appear and $n_B^i = 0$, this corresponds to $s > 0$ and we write $s = s_0$. As $n_B^i \to n_B^{i\,*}$, $s \to 0$.

The $s = 0$ limit of 155 is

$$\zeta = \frac{B\delta t}{2 + B\delta t} \tag{157}$$

The long-time limit of 156 is

$$\zeta = \frac{2 - 3Bp + B^2p}{Bp(B-1)} = 1 - \frac{2s}{B(s+\delta)} \tag{158}$$

For a phage clone that begins at size $N_0 = 1$ and for which there are no matching bacterial clones ($n_B^i = 0$), **Equation 159** gives the probability of that clone going extinct.

$$P_0 = 1 - \frac{2s_0}{B(s_0 + \delta_0)} \tag{159}$$

If $B = 2$ (a return to a simple birth-death equation where the population grows and dies by increments of 1), **Equation 156** becomes

$$\zeta = \frac{\delta(e^{st} - 1)}{e^{st}(\delta + s) - \delta} \tag{160}$$

This is the expected probability of extinction for a simple birth-death process with selection given in **Desai and Fisher, 2007**. (Note that time can be rescaled so that $\delta = 1$.)

The long-time limit of 160 is

$$\frac{1-p}{p} = \frac{\delta}{s+\delta} = \frac{\delta}{\beta} \tag{161}$$

This is the expected limit for the extinction probability for a simple birth-death process with selection. As $s \rightarrow 0$, the probability of extinction goes to 1.

Note that the probability of extinction for $B > 2$ (**Equation 158**) is larger than for $B = 2$ (**Equation 161**). Somewhat paradoxically, this implies that one effect of the burst size $B > 2$ is to increase the probability of extinction for new phage mutants. This happens because bursts are infrequent events; on average, $B$ death events must occur for every one birth event, making death a more common stochastic outcome. The extinction probability for $B > 2$ is $\frac{1}{B}(\frac{2}{B-1} + \frac{\beta(B-2)}{\delta})$ times greater than for $B = 2$. Expanding in $1/B$ for large $B$, this is $\frac{\beta}{\delta}(B - 3) + \mathcal{O}(1/B)$.

## Small-time approximation for $P_0(t)$

When $t$ is small, $P_0$ will be close to 0, which means $\zeta$ will be close to 0 and therefore we can approximate $w$ as being near 0.

The denominator simplifies to $1 - p - w$ for small $w$, and we get

$$x \approx \int_0^\zeta \frac{dw}{1 - p - w} \tag{162}$$

$$\zeta = (1 - p)(1 - e^{-x}) = (1 - p)\left(1 - e^{-\beta_0 t/p}\right) = \frac{\delta_0}{\beta_0 + \delta_0}\left(1 - e^{-(\beta_0 + \delta_0)t}\right) \tag{163}$$

## Long-time approximation for $P_0(t)$ with no selection

In the above derivations of the probability of phage clone extinction, we generally assumed that phages had a selective advantage because there were no bacteria with matching spacers at the time a phage clone arose by mutation. If instead a phage clone does not have a selective advantage, this corresponds to $s = \beta(B - 1) - \delta \approx 0$, or $p \approx 1/B$ is exactly $1/B$ where $n_B$ is given by the mean-field steady state with no CRISPR; in other words, phage clones do not have a selective advantage if the population is in steady state and there is no CRISPR immunity to distinguish clones from each other.

Expanding the denominator with $w = 1 - \epsilon$ as before:

$$1 - p + pw^B - w \approx \epsilon - p\epsilon B + \frac{p}{2}\epsilon^2 B(B - 1)$$

When $p = 1/B$, the first two terms cancel out. We get the integral

$$x \approx \frac{2}{pB(B - 1)} \int_0^\zeta \frac{dw}{(w - 1)^2} \tag{164}$$

which gives

$$\zeta = \frac{(-B + B^2)px}{2 - Bpx + B^2px} = \frac{(B - 1)Bpx}{2 + (B - 1)Bpx} = \frac{(B - 1)B\beta t}{2 + (B - 1)B\beta t} \tag{165}$$

**Equation 165** is equivalent to **Equation 157**, assuming $\beta(B - 1) = \delta$. This is true when $s = 0$, since $s = 0$ corresponds to $(B - 1)\beta = \delta$.

## Neutral time to extinction from backward master equation

The phage time to extinction calculated here is compared to the speed of phage evolution in **Figure 6**, 'Dynamics are determined by diversity'.

We can calculate the time to extinction using the backward master equation corresponding to **Equation 147**. The backward equation is an equation for the time to extinction $T_n$ from a given state. Instead of working with frequencies, we write this in terms of the number of phages belonging to a clone, $n$. Here $\beta = \alpha p_V n_B - \alpha p_V e n_B^i$ and $\delta = F + \alpha n_B(1 - p_V) + \alpha p_V e n_B^i$.

$$T_n - \Delta t = \beta n \Delta t T_{n+B-1} + \delta n \Delta t T_{n-1} + (1 - \beta n \Delta t - \delta n \Delta t)T_n \tag{166}$$

Rearranging *Equation 166*, we arrive at *Equation 167*.

$$-1 = \beta n T_{n+B-1} + \delta n T_{n-1} - (\beta n + \delta n) T_n \tag{167}$$

For boundary conditions, we have $T(n = 0) = 0$ (time to extinction is 0 when already extinct) and $\frac{dT}{dn}|_{n=n_V} = 0$ (reflecting boundary at $n = n_V$). As with bacteria clones, we assume here that the upper limit on phage clone size is constant at $n_V$ at steady state.

To solve *Equation 167*, we expand about $n$ and keep terms up to second order to get the Fokker–Planck equation:

$$-1 = \frac{dT}{dn}\left(\beta n(B-1) - \delta n\right) + \frac{1}{2}\frac{d^2 T}{dn^2}\left(\beta n(B-1)^2 + \delta n\right) \tag{168}$$

At steady state and for large clones, $\beta(B-1) \approx \delta$, which means clones are approximately neutral and we can drop the drift term:

$$-1 \approx \frac{1}{2}\frac{d^2 T}{dn^2}\left(\beta n(B-1)^2 + \delta n\right) \tag{169}$$

This is a straightforward second-order ordinary differential equation, which has the solution

$$T(n) = c_1 + nc_2 - \frac{2n\left(-1 + \ln[n(\beta(B-1)^2 + \delta)]\right)}{\beta(B-1)^2 + \delta} \tag{170}$$

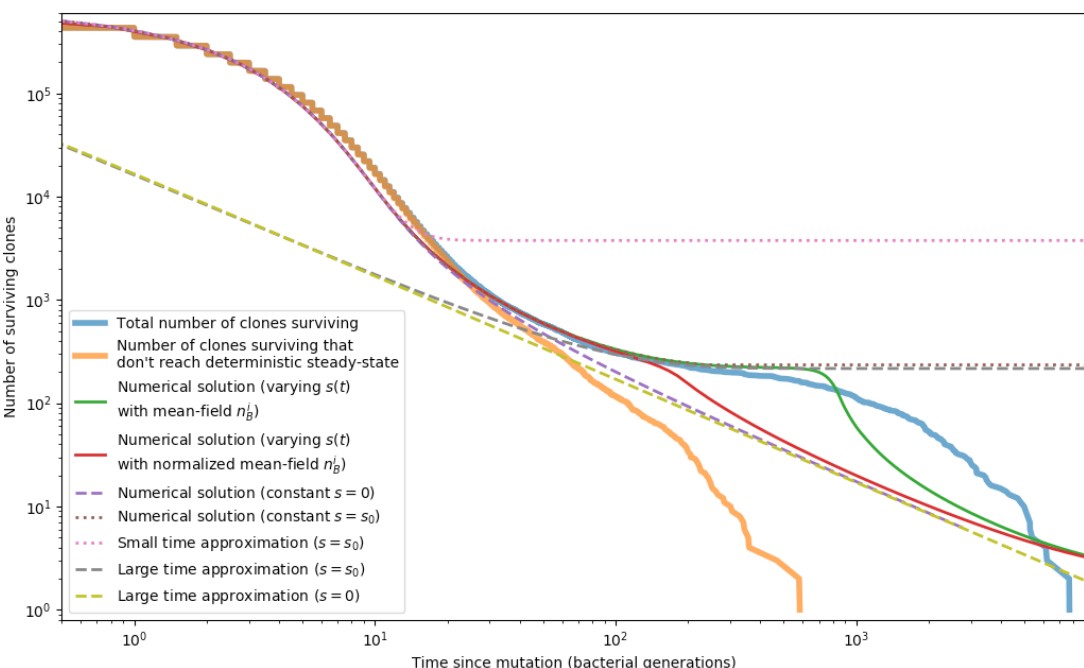

**Appendix 3—figure 8.** Phage clone extinction times and theoretical predictions for a simulation with parameters $C_0 = 10^4, \eta = 0.001, e = 0.95, \mu = 10^{-5}$. Time zero for each trajectory is the time at which that clone arose by mutation. All simulation trajectories are plotted in blue, and a subset of trajectories that do not reach a size of $n_V^{i*} = 16170$ as given by *Equation 26* are plotted in orange. Trajectories that do not become established by this definition go extinct more quickly than ones that do. All other curves are theoretical predictions for the extinction time. The green and red solid lines and purple and brown dashed lines show a numerical solution to *Equation 153* with different values for $s$. The remaining dashed lines are a small time approximation given by *Equation 163*, and a large time approximation given by *Equation 156* with either $s = s_0$ or $s = 0$. All of these predictions agree with the simulation data in different regimes; none accurately captures the entire timecourse of extinctions.

Using the boundary condition $T(0) = 0$, the constant $c_1 = 0$. Using the second boundary condition, $c_2 = \frac{2\ln[n_V(\beta(B-1)^2+\delta)]}{\beta(B-1)^2+\delta}$, and the full solution is

$$T(n) = \frac{2n}{\beta(B-1)^2+\delta}\left(1 - \ln\frac{n}{n_V}\right) \tag{171}$$

Once phage clones become large (by reaching size $n_V^{i\,*}$ as given by **Equation 26**), they behave neutrally and **Equation 171** agrees well with observed extinction times in simulations. We set $n = n_V^{i\,*}$ and use the deterministic steady-state clone size for $n_B^i$ given by **Equation 25** in $\beta$ and $\delta$. **Appendix 3—figure 9** compares observed large clone extinction times in a simulation (blue) to the prediction given by **Equation 171** (orange dashed). Predicted and measured times to extinction for many simulations with **Equation 171** are shown in **Appendix 3—figure 10**.

If $n_V^{i\,*} \approx \frac{n_V}{m}$, which it is at steady state, we can rewrite **Equation 171** as:

$$T(n) = \frac{2n}{\beta(B-1)^2+\delta}\left(1 + \ln m\right) \tag{172}$$

The factor $1 + \ln m$ comes from the integral of the second-order differential equation and from the reflecting boundary assumption, and interestingly it would happen even if $B = 2$ (no burst): integrating the Fokker–Planck equation gives a factor of $1 - \ln n$; if we start from the steady-state value of $n \approx n_V/m$, this becomes $1 + \ln m - \ln n_V$. Applying the reflecting boundary at $n = n_V$ cancels out the $\ln n_V$ term. Very simply: the integral we are solving is $\frac{-1}{n} \propto \frac{d^2T}{dn^2}$, and this gives $n(1 - \ln n)$ up to some constants. This simplified expression for time to extinction gives insight into the main drivers

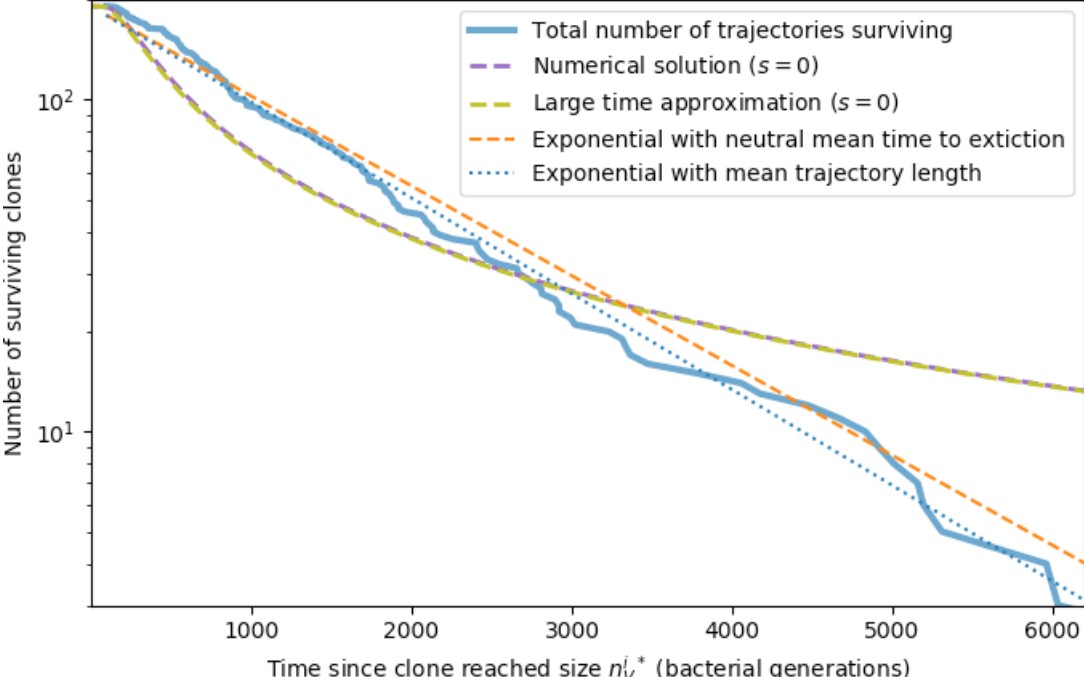

**Appendix 3—figure 9.** Large clone extinction times from a simulation with parameters $C_0 = 10^4, \eta = 0.001, e = 0.95, \mu = 10^{-5}$. Trajectories are counted as large if the phage clone size passes $n_V^{i\,*} = 16170$, the theoretical deterministic mean phage clone size for these parameters given by **Equation 26**. Once a trajectory reaches $n_V^{i\,*}$, we count that point as time zero to measure the extinction time of large clones. The numerical solution is given by solving **Equation 153** with $s = 0$ ($p = 1/B$). The large time approximation is given by **Equation 157**. The orange dashed line gives an exponential decay prediction with the mean time to extinction given by **Equation 171** with $n = n_V^{i\,*}$.

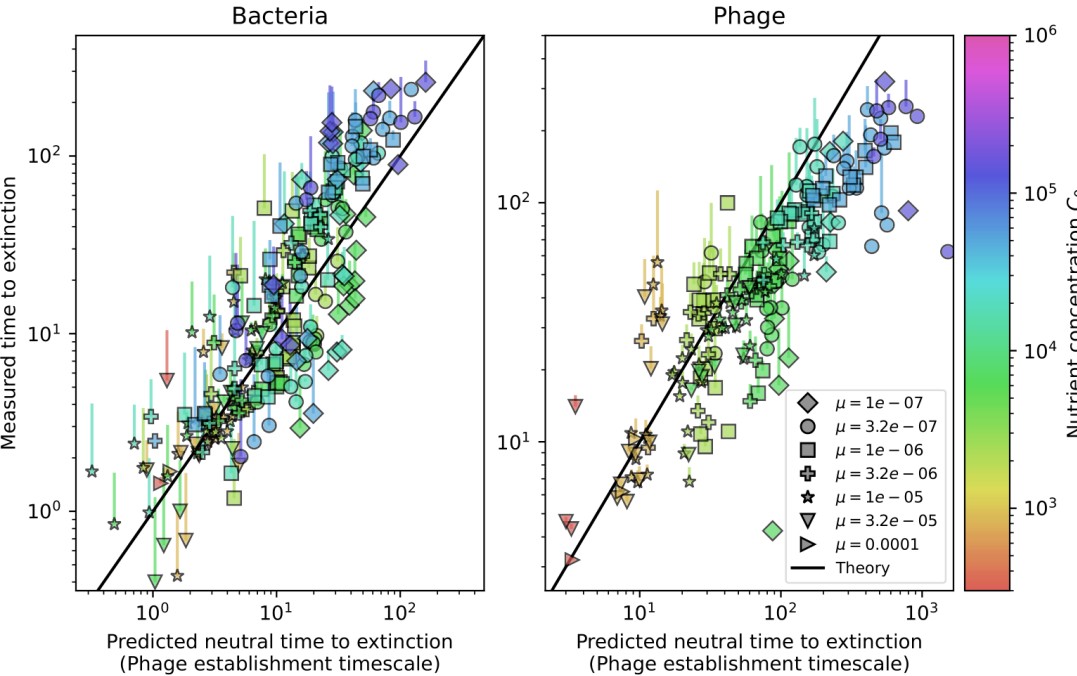

**Appendix 3—figure 10.** Mean time to extinction for large bacteria (left) and phage (right) clones as a function of the neutral fitness mean time to extinction prediction. The solid black lines describe approximate analytic expressions for bacteria and phage time to extinction using a neutral fitness assumption. For bacteria, the black line solves *Equation 139* using the mean bacteria and phage clone sizes. For phages, the black line solves *Equation 171* using the mean phage clone size in place of $n$.

of extinction. The phage time to extinction decreases as the outflow rate $f$ increases — if phages die at a higher rate, they go extinct more quickly. Interestingly, a larger burst size $B$ decreases the time to extinction for large phage clones, consistent with an overall increase in the size of fluctuations as the burst size increases.

This expression still depends on $m$, which is an emergent quantity dependent on all other parameters as well. We apply the approximation for $m$ from 'Approximation for $m$ for more insight.

$$T_{\text{phage}} \approx \frac{1}{m} \frac{2gC_0(1-f)}{f\alpha Bp_V} \approx \frac{1}{f}\left[\frac{2r}{\alpha Be\eta(1-p_V)\mu L}\right]^{\frac{1}{3}} \tag{173}$$

Comparing with *Equation 146* for bacteria, both phage and bacteria extinction depend inversely on $\alpha$, $B$, $e$, $\mu L$, and $\eta(1-p_V)$, with the time to extinction decreasing with all parameters except the spacer loss rate $r$. The bacteria time to extinction increases with increasing flow rate $f$, while the phage time to extinction decreases with $f$.

The clone size dependence of phage extinction is shown in *Appendix 3—figure 11*, comparing measured extinction times to the prediction of *Equation 172* without the $1 + \ln m$ term. This captures the dominant trend in extinction time behaviour.

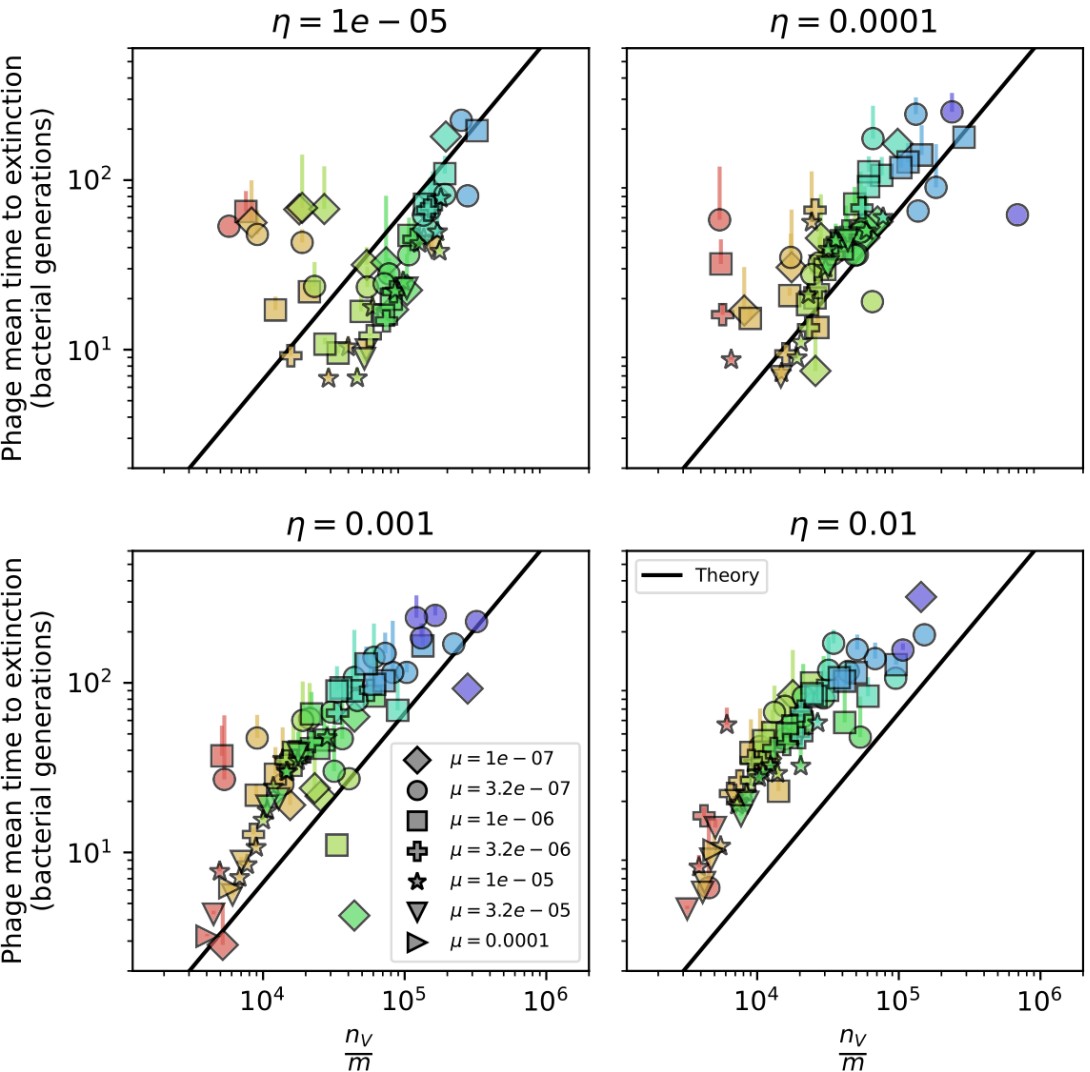

**Appendix 3—figure 11.** Phage extinction time as a function of mean clone size. Large phage clone mean time to extinction as a function of $\frac{n_V}{m}$, where $m$ is measured from simulations and $n_V$ is the total phage population size. The solid line is given by **Equation 172** without the $1 + \ln m$ term.

