## [Editor Report]

In this important work, the authors develop a theory for the coevolutionary dynamics of bacteria and phages, where the major evolutionary pressure comes from CRISPR-Cas adaptive immunity in bacteria. Through extensive stochastic numerical simulations and analytical calculations, the article presents a compelling analysis of the emergent properties of immune interactions, in the regime of a single proto-spacer and a single spacer. Some of the trends highlighted by the model are recovered from experimental data. The main results concern how diversity in both phage and bacteria population is linked and is shaped by immunity, and should be of broad interest in immunology.

---

## [Decision Letter]

**Decision letter after peer review:**

Thank you for submitting your article "Dynamics of immune memory and learning in bacterial communities" for consideration by *eLife*. Your article has been reviewed by 3 peer reviewers, and the evaluation has been overseen by a Reviewing Editor and Aleksandra Walczak as the Senior Editor. The following individual involved in review of your submission has agreed to reveal their identity: Barbara Bravi (Reviewer #1).

Essential revisions:

1) Please clarify why/when the regime of a single spacer per bacterium and a single protospacer per phage is relevant. Please also perform an explicit comparison with the multiple-protospacer regime, which is biologically highly relevant (see Reviewer #3's main point), and include the associated results in the Results section.

2) To increase readability, please make the Discussion section more concise. Please also add an initial sentence to each section of the Supplementary Material that outlines what is done in it and how it connects to the main text. Please also consider moving Results section 3.4 to the Supplementary Material.

*Reviewer #1 (Recommendations for the authors):*

I think the Discussion section is too long, compromising the focus on a few key messages. The authors might consider some streamlining of the text. For example, to make the point about the fact that studying CRISPR can give insight into vertebrate adaptive dynamics, a shorter section would suffice.

General remarks:

- There are 3 main experimental findings listed in lines 63-66 (a-c). I think the authors should clarify what of these findings were hard-coded in the model and what were recovered or rationalized by their analysis.

- In several points, the authors comment on the validity of certain trends across a range of parameters, but in the figures several parameters are fixed and then only η (or μ) is varied. I think the authors should refer to the supplementary figures where they show the variation wrt other parameters or have a sentence where they justify why certain parameters are fixed while mainly η is varied. As far as I am concerned, seeing how the behaviours change for lower e is interesting too.

*Reviewer #3 (Recommendations for the authors):*

My main concern is the single protospacer parameter mentioned in the public review. Of the parameters mentioned in lines 553-559 (burst lag, spatial structure, autoimmunity, etc…), multiple protospacers, and in reality hundreds of possible protospacers, has the highest likelihood to significantly change the bacteria and phage dynamics. I wonder if the “counter-intuitive result” in line 561 is due to this discrepancy. The authors explore multiple protospacers with a “toy model” in the discussion, but I believe this should be more fully explored in the main Results section, similar to what was done for cross-reactivity.

This is particularly important because as soon as a phage infects, there are hundreds of potential protospacers. The likelihood that two cells will acquire the same spacer is therefore very, very small and therefore, if a phage escapes one spacer by mutation, it will still be targeted by the vast majority of cells. This is very different from the model the authors use for the bulk of their paper where a phage simultaneously escapes all spacers. I would like to see the analyses repeated with this multiple protospacer scenario which is biologically very relevant.

---

## [Author Response]

Essential revisions:1) Please clarify why/when the regime of a single spacer per bacterium and a single protospacer per phage is relevant. Please also perform an explicit comparison with the multiple-protospacer regime, which is biologically highly relevant (see Reviewer #3's main point), and include the associated results in the Results section.

Based on Reviewer #3’s comments, we have added a section to the Results (now 3.5) that extensively develops our toy model exploring the impact of multiple protospacers and spacers per organism and have shortened the related section in the Discussion.

2) To increase readability, please make the Discussion section more concise. Please also add an initial sentence to each section of the Supplementary Material that outlines what is done in it and how it connects to the main text. Please also consider moving Results section 3.4 to the Supplementary Material.

We have combined the Results section on clone extinction with Appendix C to reduce the length of our manuscript, have shortened the Discussion, and have added an initial sentence to each sentence of the Supplementary Material that summarizes its purpose and links to the main text. In addition, to reduce length, we removed Methods sections not referred to in the main results – “Quasi-steady-state clone size solution” and “Clone size stability analysis.”

Reviewer #1 (Recommendations for the authors):I think the Discussion section is too long, compromising the focus on a few key messages. The authors might consider some streamlining of the text. For example, to make the point about the fact that studying CRISPR can give insight into vertebrate adaptive dynamics, a shorter section would suffice.

We feel that the comprehensiveness of the Discussion is one of its strong points, but we agree that it is long – we have shortened both the sections on vertebrate adaptive immunity and average immunity.

General remarks:- There are 3 main experimental findings listed in lines 63-66 (a-c). I think the authors should clarify what of these findings were hard-coded in the model and what were recovered or rationalized by their analysis.

None of these features were explicitly hard-coded in the model; we address each as a finding of our work. To make this clear, we have rephrased some of our findings in the last paragraph of the introduction (beginning line 189 in updated draft) to parallel the structure of lines 63-66 with the following text: “We recover and reinterpret experimentally observed features: (a) we find that high diversity is not beneficial for bacteria when phage and bacterial diversity is strongly coupled, (b) we show that bacterial immunity can either track new phage mutations rapidly or keep a memory for a long time, but not both, and (c) we find emergent diversity resulting from selection for phage mutations that evade CRISPR targeting, linking diversity to the dynamical quantities of establishment and extinction.”

- In several points, the authors comment on the validity of certain trends across a range of parameters, but in the figures several parameters are fixed and then only η (or μ) is varied. I think the authors should refer to the supplementary figures where they show the variation wrt other parameters or have a sentence where they justify why certain parameters are fixed while mainly η is varied. As far as I am concerned, seeing how the behaviours change for lower e is interesting too.

In most plots that include many simulation data points, all four major parameters (*e*, *µ*, *η*, and *C*_0_) are varied, but due to labelling constraints we chose two variables to highlight, usually *C*_0_ and *η –* this applies to Figures 2B, 3C, and 6B-C. For Figure 2B and we plotted all points separately in panels by parameter in figure supplements, and we have added a reference to these figure supplements in the main text. In Figures 2A-B, 7D, and 7A-B only *η* is varied; we have added figure supplements that show each plot varied with *e* and *µ* as well (Figure 2 supplements 3 and 4, Figure 6 supplements 3 and 4, and Figure 7 supplements 11 and 12).

Reviewer #3 (Recommendations for the authors):My main concern is the single protospacer parameter mentioned in the public review. Of the parameters mentioned in lines 553-559 (burst lag, spatial structure, autoimmunity, etc…), multiple protospacers, and in reality hundreds of possible protospacers, has the highest likelihood to significantly change the bacteria and phage dynamics. I wonder if the “counter-intuitive result” in line 561 is due to this discrepancy. The authors explore multiple protospacers with a “toy model” in the discussion, but I believe this should be more fully explored in the main Results section, similar to what was done for cross-reactivity.This is particularly important because as soon as a phage infects, there are hundreds of potential protospacers. The likelihood that two cells will acquire the same spacer is therefore very, very small and therefore, if a phage escapes one spacer by mutation, it will still be targeted by the vast majority of cells. This is very different from the model the authors use for the bulk of their paper where a phage simultaneously escapes all spacers. I would like to see the analyses repeated with this multiple protospacer scenario which is biologically very relevant.

We share the view of Reviewer 3 that the number of protospacers and spacers for both bacteria and phages is important for understanding dynamics in this system. We have made significant changes to reflect this, adding a section to the results titled ‘Pathogen and host diversity must be considered together’. First, we note that our results for exponential cross-reactivity are a good approximation for a situation with multiple protospacers and few spacers per bacterium – as phages gain mutations in individual protospacers, their fitness increases gradually (Section 3.5, line 449). In this section we also describe a toy model that explores the relationship between diversity and average immunity when phages and bacteria have multiple spacers and protospacers. We have included more scenarios than in our previous submission to elucidate different realistic types of protospacer diversity, and in all cases we find that average immunity either remains constant or decreases as diversity increases. We emphasize, however, that this model is not a full population dynamics simulation of the multiple-protospacer scenario. Such a simulation is a large undertaking that we believe is better-suited for a follow-up study to this already extensive work.